# Brain charts for the human lifespan

Over the past few decades, neuroimaging has become a ubiquitous tool in basic research and clinical studies of the human brain. However, no reference standards currently exist to quantify individual differences in neuroimaging metrics over time, in contrast to growth charts for anthropometric traits such as height and weight[1]. Here we assemble an interactive open resource to benchmark brain morphology derived from any current or future sample of MRI data (http://www.brainchart.io/). With the goal of basing these reference charts on the largest and most inclusive dataset available, acknowledging limitations due to known biases of MRI studies relative to the diversity of the global population, we aggregated 123,984 MRI scans, across more than 100 primary studies, from 101,457 human participants between 115 days post-conception to 100 years of age. MRI metrics were quantified by centile scores, relative to non-linear trajectories[2] of brain structural changes, and rates of change, over the lifespan. Brain charts identified previously unreported neurodevelopmental milestones[3], showed high stability of individuals across longitudinal assessments, and demonstrated robustness to technical and methodological differences between primary studies. Centile scores showed increased heritability compared with non-centiled MRI phenotypes, and provided a standardized measure of atypical brain structure that revealed patterns of neuroanatomical variation across neurological and psychiatric disorders. In summary, brain charts are an essential step towards robust quantification of individual variation benchmarked to normative trajectories in multiple, commonly used neuroimaging phenotypes.

The simple framework of growth charts to quantify age-related change was first published in the late eighteenth century[1] and remains a cornerstone of paediatric healthcare—an enduring example of the utility of standardized norms to benchmark individual trajectories of development. However, growth charts are currently available only for a small set of anthropometric variables, such as height, weight and head circumference, and only for the first decade of life. There are no analogous charts available for quantification of age-related changes in the human brain, although it is known to go through a prolonged and complex maturational program from pregnancy to the third decade[4], followed by progressive senescence from approximately the sixth decade[5]. The lack of tools for standardized assessment of brain development and ageing is particularly relevant to research studies of psychiatric disorders, which are increasingly recognized as a consequence of atypical brain development[6], and neurodegenerative diseases that cause pathological brain changes in the context of normative senescence[7]. Preterm birth and neurogenetic disorders are also associated with marked abnormalities of brain structure[8,9] that persist into adult life[9,10] and are associated with learning disabilities and mental health disorders. Mental illness and dementia collectively represent the single biggest global health burden[11], highlighting the urgent need for normative brain charts as an anchor point for standardized quantification of brain structure over the lifespan[12].

Such standards for human brain measurement have not yet materialized from decades of neuroimaging research, probably owing to the challenges of integrating MRI data across multiple, methodologically diverse studies targeting distinct developmental epochs and clinical conditions[13]. For example, the perinatal period is rarely incorporated in analysis of age-related brain changes, despite evidence that early

biophysical and molecular processes powerfully influence life-long neurodevelopmental trajectories[14,15] and vulnerability to psychiatric disorders[3]. Primary case–control studies are usually focused on a single disorder despite evidence of trans-diagnostically shared risk factors and pathogenic mechanisms, especially in psychiatry[16,17]. Harmonization of MRI data across primary studies to address these and other deficiencies in the extant literature is challenged by methodological and technical heterogeneity. Compared with relatively simple anthropometric measurements such as height or weight, brain morphometrics are known to be highly sensitive to variation in scanner platforms and sequences, data quality control, pre-processing and statistical analysis[18], thus severely limiting the generalizability of trajectories estimated from any individual study[19]. Collaborative initiatives spurring collection of large-scale datasets[20,21], recent advances in neuroimaging data processing[22,23] and proven statistical frameworks for modelling biological growth curves[2,24,25] provide the building blocks for a more comprehensive and generalizable approach to age-normed quantification of MRI phenotypes over the entire lifespan (see Supplementary Information 1 for details and consideration of previous work focused on the related but distinct objective of inferring brain age from MRI data). Here, we demonstrate that these convergent advances now enable the generation of brain charts that (1) robustly define normative processes of sex-stratified, age-related change in multiple MRI-derived phenotypes; (2) identify previously unreported brain growth milestones; (3) increase sensitivity to detect genetic and early life environmental effects on brain structure; and (4) provide standardized effect sizes to quantify neuroanatomical atypicality of brain scans collected across multiple clinical disorders. We do not claim to have yet reached the ultimate goal of quantitatively precise diagnosis of MRI scans from

A list of authors and their affiliations appears online. ✉e-mail: rb643@medschl.cam.ac.uk; jakob.seidlitz@pennmedicine.upenn.edu

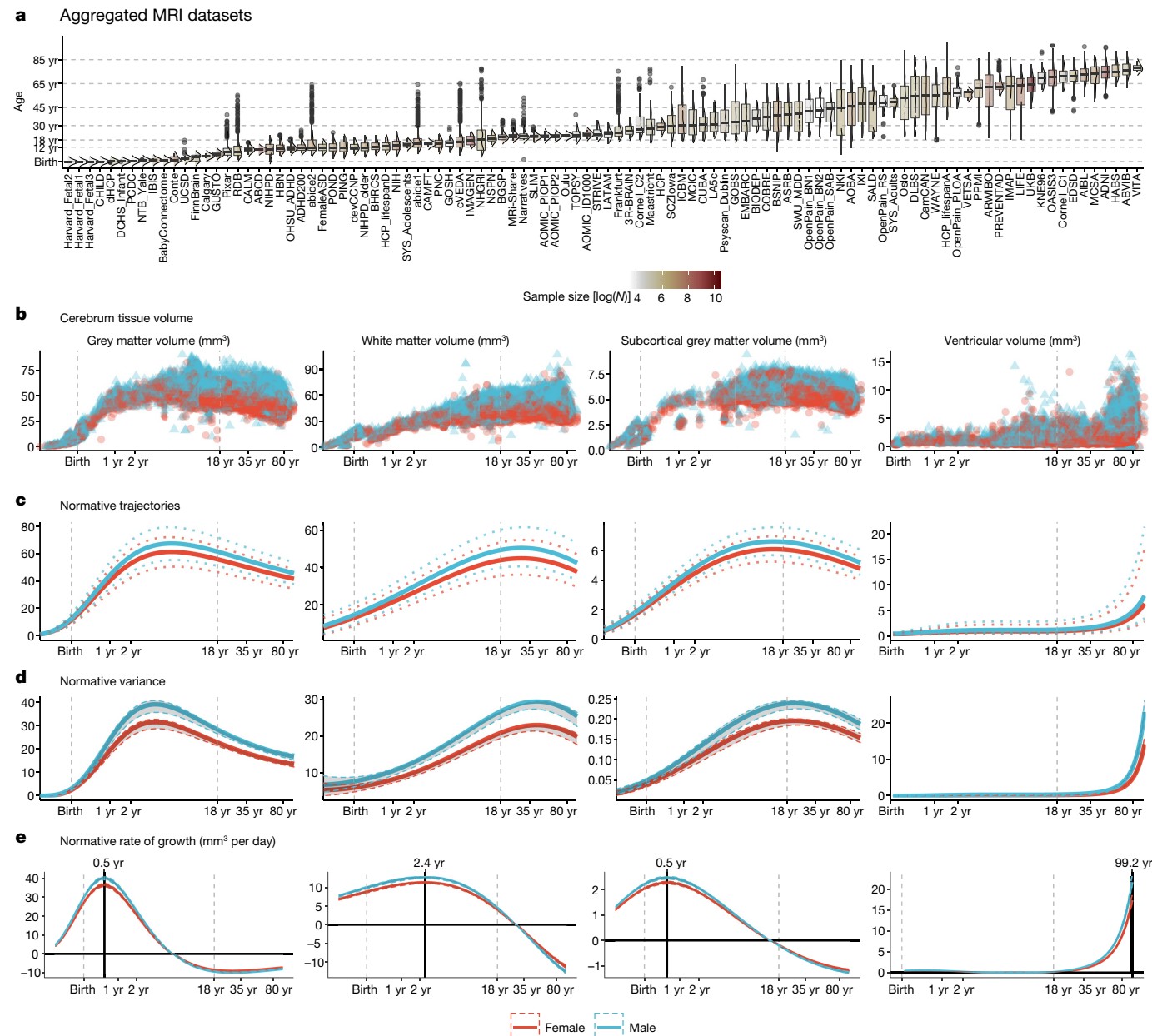

**Fig. 1 | Human brain charts. a**, MRI data were aggregated from over 100 primary studies comprising 123,984 scans that collectively spanned the age range from mid-gestation to 100 postnatal years. Box–violin plots show the age distribution for each study coloured by its relative sample size (log-scaled using the natural logarithm for visualization purposes). **b**, Non-centiled, 'raw' bilateral cerebrum tissue volumes for grey matter, white matter, subcortical grey matter and ventricles are plotted for each cross-sectional control scan as a function of age (log-scaled); points are coloured by sex. **c**, Normative brain-volume trajectories were estimated using GAMLSS, accounting for site- and study-specific batch effects, and stratified by sex (female, red; male, blue). All four cerebrum tissue volumes demonstrated distinct, non-linear trajectories of their medians (with 2.5% and 97.5% centiles denoted as dotted lines) as a function of age over the lifespan. Demographics for each cross-sectional sample of healthy controls

included in the reference dataset for normative GAMLSS modelling of each MRI phenotype are detailed in Supplementary Table 1.2–1.8. **d**, Trajectories of median between-subject variability and 95% confidence intervals for four cerebrum tissue volumes were estimated by sex-stratified bootstrapping (see Supplementary Information 3 for details). **e**, Rates of volumetric change across the lifespan for each tissue volume, stratified by sex, were estimated by the first derivatives of the median volumetric trajectories. For solid (parenchymal) tissue volumes, the horizontal line ($y = 0$) indicates when the volume at which each tissue stops growing and starts shrinking and the solid vertical line indicates the age of maximum growth of each tissue. See Supplementary Table 2.1 for all neurodevelopmental milestones and their confidence intervals. Note that $y$ axes in **b**–**e** are scaled in units of 10,000 mm³ (10 ml).

individual patients in clinical practice. However, the present work proves the principle that building normative charts to benchmark individual differences in brain structure is already achievable at global scale and over the entire life-course; and provides a suite of open science resources for the neuroimaging research community to accelerate further progress in the direction of standardized quantitative assessment of MRI data.

## Mapping normative brain growth

We created brain charts for the human lifespan using generalized additive models for location, scale and shape[2,24] (GAMLSS), a robust and flexible framework for modelling non-linear growth trajectories recommended by the World Health Organization[24]. GAMLSS and related statistical frameworks have previously been applied to developmental

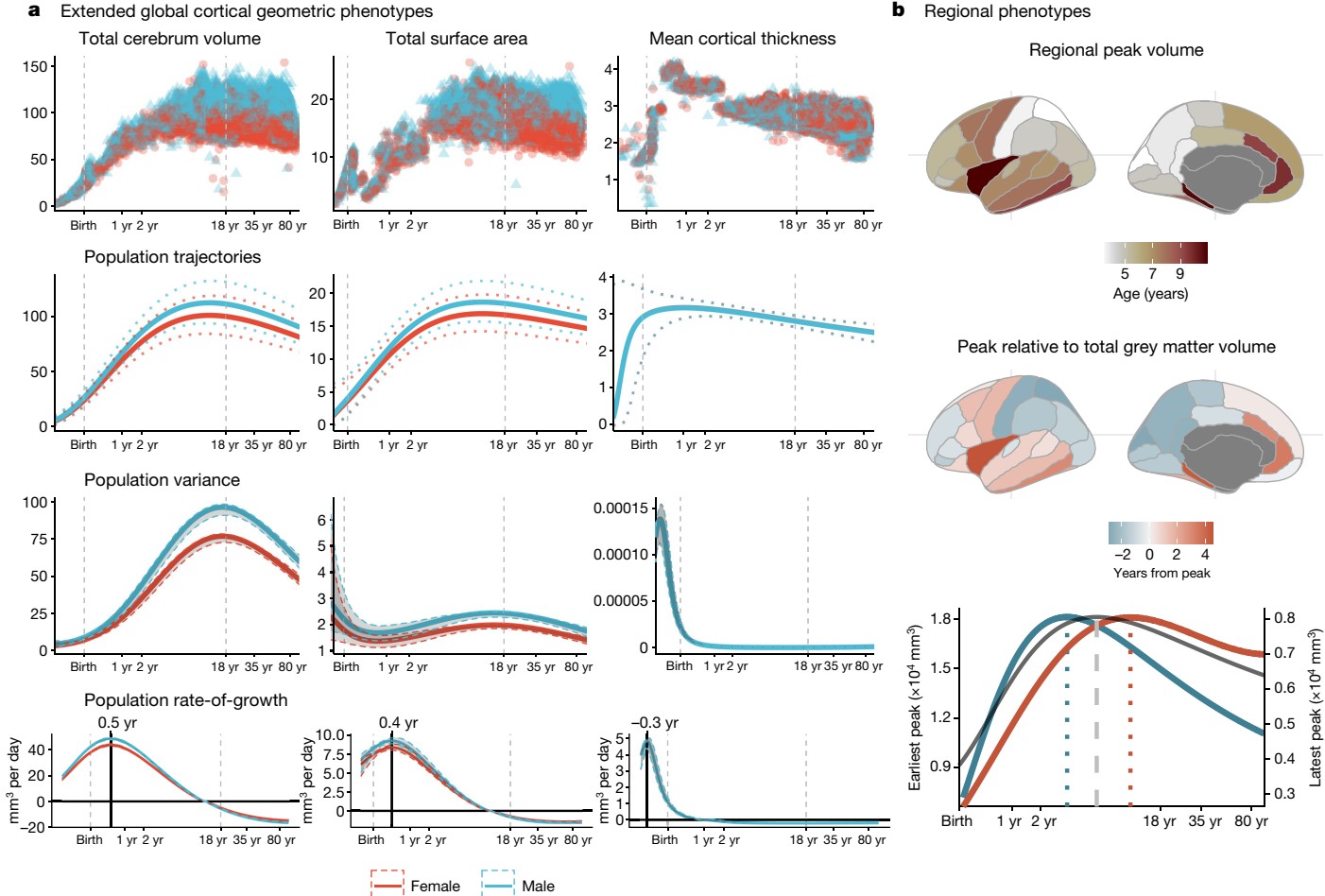

**Fig. 2 | Extended global and regional cortical morphometric phenotypes.**
**a**, Trajectories for total cerebrum volume (TCV), total surface area and mean cortical thickness. For each global cortical MRI phenotype, the following sex-stratified results are shown as a function of age over the lifespan. From top to bottom: raw, non-centiled data; population trajectories of the median (with 2.5% and 97.5% centiles (dotted lines)); between-subject variance (with 95% confidence intervals); and rate of growth (the first derivatives of the median trajectory and 95% confidence intervals). All trajectories are plotted as a function of log-scaled age (x axis) and y axes are scaled in units of the corresponding MRI metrics (10,000 mm³ for TCV, 10,000 mm² for surface area and mm for cortical thickness). **b**, Regional variability of cortical volume trajectories for 34 bilateral brain regions, as defined by the Desikan–Killiany parcellation[47], averaged across sex (see Supplementary Information 7,8 for

details). Since models were generated from bilateral averages of each cortical region, the cortical maps are plotted on the left hemisphere purely for visualization purposes. Top, a cortical map of age at peak regional volume (range 2–10 years). Middle, a cortical map of age at peak regional volume relative to age at peak GMV (5.9 years), highlighting regions that peak earlier (blue) or later (red) than GMV. Bottom, illustrative trajectories for the earliest peaking region (superior parietal lobe, blue line) and the latest peaking region (insula, red line), showing the range of regional variability relative to the GMV trajectory (grey line). Regional volume peaks are denoted as dotted vertical lines either side of the global peak, denoted as a dashed vertical line, in the bottom panel. The left y axis on the bottom panel refers to the earliest peak (blue line); the right y axis refers to the latest peak (red line).

modelling of brain structural and functional MRI phenotypes in open datasets[19,26–31]. Our approach to GAMLSS modelling leveraged the greater scale of data available to optimize model selection empirically, to estimate non-linear age-related trends (in median and variance) stratified by sex over the entire lifespan, and to account for site- or study-specific 'batch effects' on MRI phenotypes in terms of multiple random effect parameters. Specifically, GAMLSS models were fitted to structural MRI data from control subjects for the four main tissue volumes of the cerebrum (total cortical grey matter volume (GMV), total white matter volume (WMV), total subcortical grey matter volume (sGMV) and total ventricular cerebrospinal fluid volume (ventricles or CSF)). Supplementary Tables 1.1–1.8 present details on acquisition, processing and demographics of the dataset; see Methods, 'Model generation and specification' and Supplementary Information 1 for further details regarding GAMLSS model specification and estimation; image quality control, which used a combination of expert visual curation and automated metrics of image quality (Supplementary Information 2);

model stability and robustness (Supplementary Information 3, 4); phenotypic validation against non-imaging metrics (Supplementary Information 3 and 5.2); inter-study harmonization (Supplementary Information 5); and assessment of cohort effects (Supplementary Information 6). See Supplementary Information 19 for details on all primary studies contributing to the reference dataset, including multiple publicly available open MRI datasets[32–42].

Lifespan curves (Fig. 1, Supplementary Table 2.1) showed an initial strong increase in GMV from mid-gestation onwards, peaking at 5.9 years (95% bootstrap confidence interval (CI) 5.8–6.1), followed by a near-linear decrease. This peak was observed 2 to 3 years later than previous reports relying on smaller, more age-restricted samples[43,44]. WMV also increased rapidly from mid-gestation to early childhood, peaking at 28.7 years (95% bootstrap CI 28.1–29.2), with subsequent accelerated decline in WMV after 50 years. Subcortical GMV showed an intermediate growth pattern compared with GMV and WMV, peaking in adolescence at 14.4 years (95% bootstrap

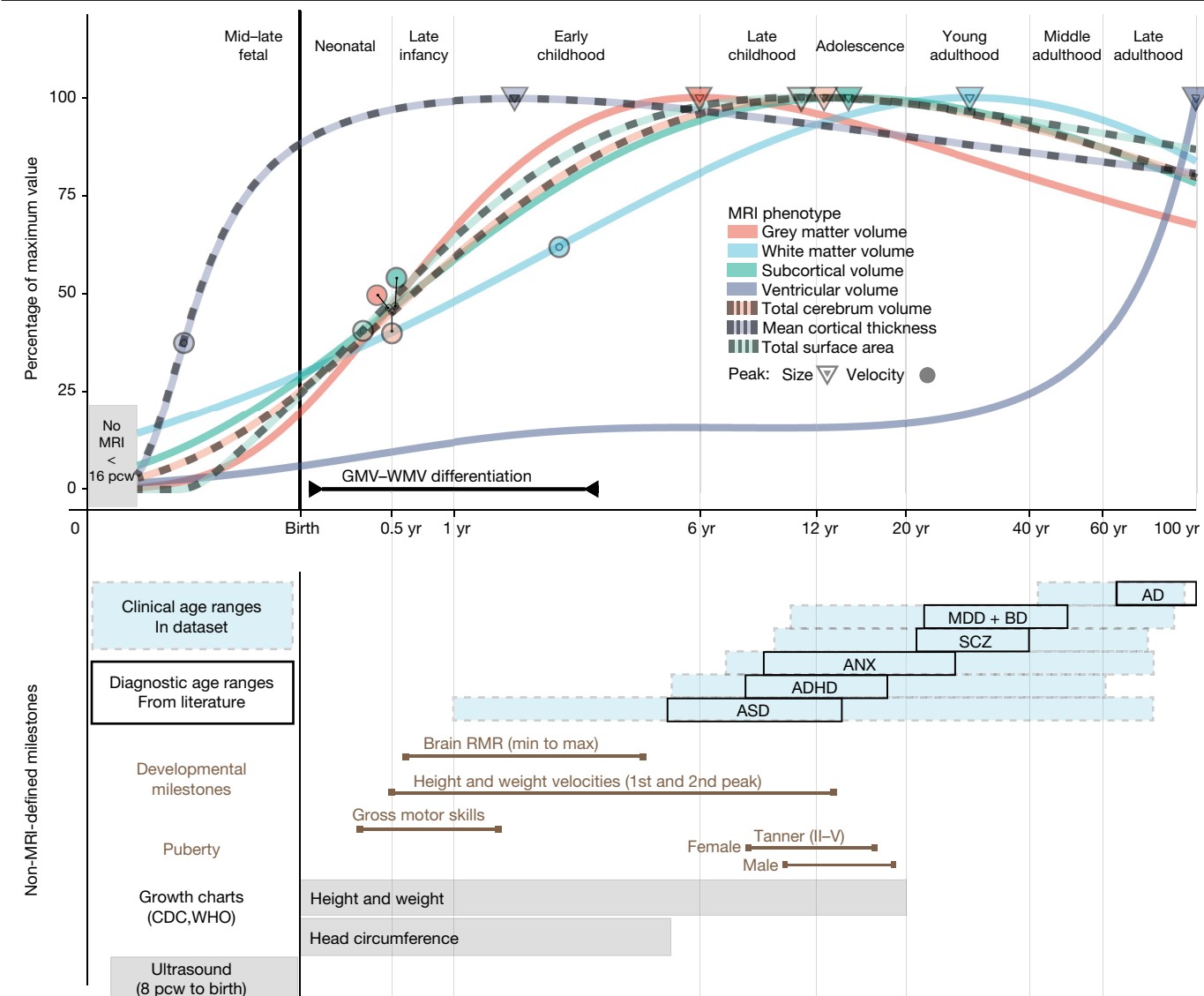

**Fig. 3 | Neurodevelopmental milestones.** Top, a graphical summary of the normative trajectories of the median (50th centile) for each global MRI phenotype, and key developmental milestones, as a function of age (log-scaled). Circles depict the peak rate of growth milestones for each phenotype (defined by the maxima of the first derivatives of the median trajectories (Fig. 1e)). Triangles depict the peak volume of each phenotype (defined by the maxima of the median trajectories); the definition of GMV:WMV differentiation is detailed in Supplementary Information 9.1. Bottom, a graphical summary of additional MRI and non-MRI developmental stages and milestones. From top to bottom: blue shaded boxes denote the age range of incidence for each of the major clinical disorders represented in the MRI dataset; black boxes denote the age at which these conditions are generally diagnosed as derived from literature[73] (Methods); brown lines

represent the normative intervals for developmental milestones derived from non-MRI data, based on previous literature and averaged across males and females (Methods); grey bars depict age ranges for existing (World Health Organization (WHO) and Centers for Disease Control and Prevention (CDC)) growth charts of anthropometric and ultrasonographic variables[24]. Across both panels, light grey vertical lines delimit lifespan epochs (labelled above the top panel) previously defined by neurobiological criteria[63]. Tanner refers to the Tanner scale of physical development. AD, Alzheimer's disease; ADHD, attention deficit hyperactivity disorder; ASD, autism spectrum disorder (including high-risk individuals with confirmed diagnosis at a later age); ANX, anxiety or phobic disorders; BD, bipolar disorder; MDD, major depressive disorder; RMR, resting metabolic rate; SCZ, schizophrenia.

CI 14.0–14.7). Both the WMV and sGMV peaks are consistent with previous neuroimaging and postmortem reports[45,46]. By contrast, CSF showed an increase until age 2, followed by a plateau until age 30, and then a slow linear increase that became exponential in the sixth decade of life. Age-related variance (Fig. 1d), explicitly estimated by GAMLSS, formally quantifies developmental changes in between-subject variability. There was an early developmental increase in GMV variability that peaked at 4 years, whereas subcortical volume variability peaked in late adolescence. WMV variability peaked during the fourth decade of life, and CSF was maximally variable at the end of the human lifespan.

## Extended neuroimaging phenotypes

To extend the scope of brain charts beyond the four cerebrum tissue volumes, we generalized the same GAMLSS modelling approach to estimate normative trajectories for additional MRI phenotypes including other morphometric properties at a global scale (mean cortical thickness and total surface area) and regional volume at each of 34 cortical areas[47] (Fig. 2, Supplementary Information 7–9, Supplementary Tables 1, 2). We found, as expected, that total surface area closely tracked the development of total cerebrum volume (TCV) across the lifespan (Fig. 2a), with both metrics peaking

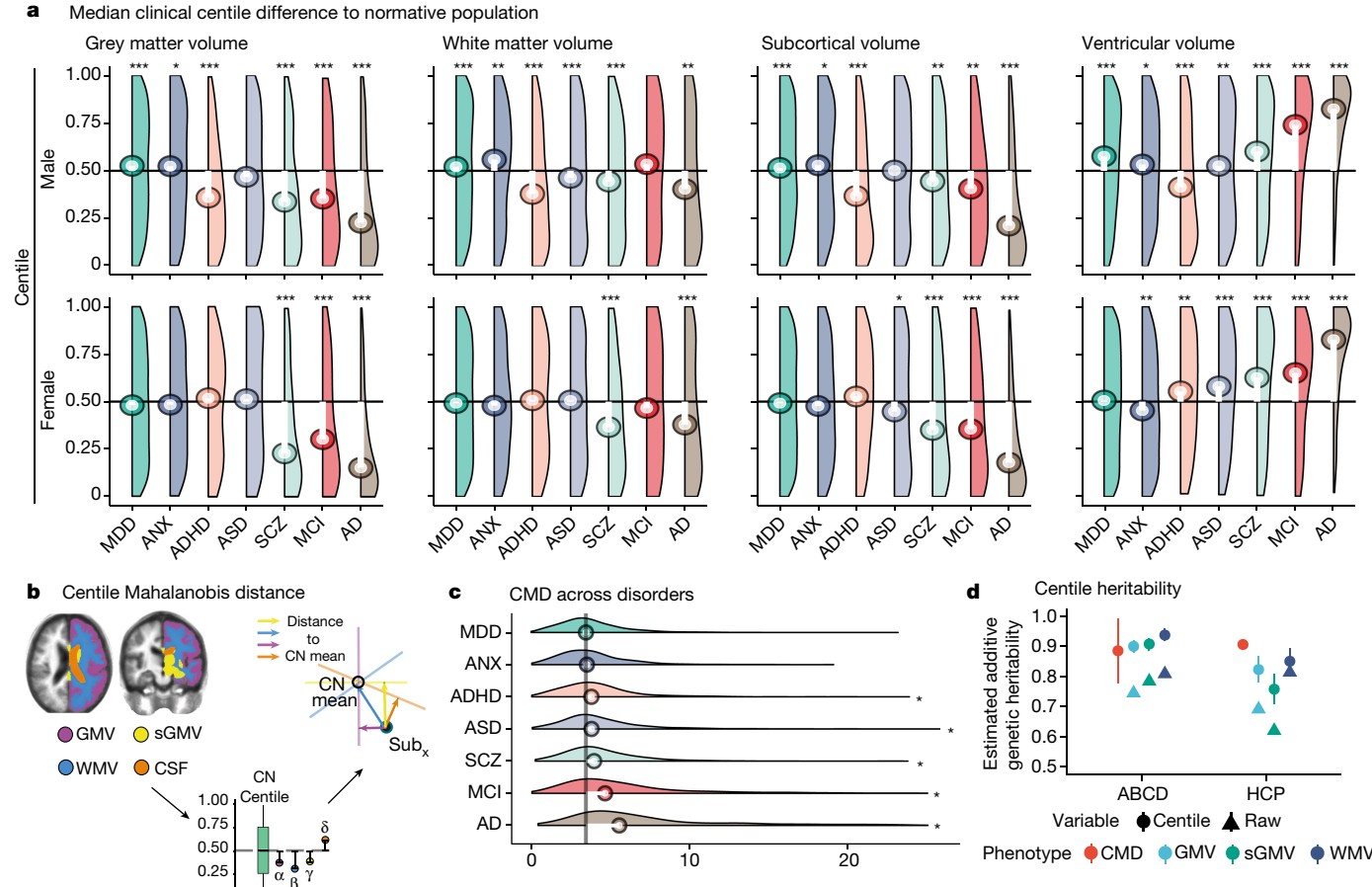

**a**  Median clinical centile difference to normative population

**b** Centile Mahalanobis distance

**c** CMD across disorders

**d** Centile heritability

Variable: ● Centile ▲ Raw
Phenotype: ● CMD ● GMV ● sGMV ● WMV

**Fig. 4 | Case–control differences and heritability of centile scores. a**, Centile score distributions for each diagnostic category of clinical cases relative to the control group median (depicted as a horizontal black line). The median deviation of centile scores in each diagnostic category is overlaid as a lollipop plot (white lines with circles corresponding to the median centile score for each group of cases). Pairwise tests for significance were based on Monte Carlo resampling (10,000 permutations) and *P* values were adjusted for multiple comparisons using the Benjamini–Hochberg false discovery rate (FDR) correction across all possible case–control differences. Only significant differences from the control group (CN) median (with corrected *P* < 0.001) are highlighted with an asterisk. For a complete overview of all pairwise comparisons, see Supplementary Information 10, Supplementary Table 3. Groups are ordered by their multivariate distance from the CN group (see **c** and Supplementary Information 10.3). **b**, The CMD is a summary metric that quantifies the aggregate atypicality of an individual scan in terms of all global MRI phenotypes. The schematic shows segmentation of four cerebrum tissue

volumes, followed by estimation of univariate centile scores, leading to the orthogonal projection of a single participant's scan (Sub$_x$) onto the four respective principal components of the CN (coloured axes and arrows). The CMD for Sub$_x$ is then the sum of its distances from the CN group mean on all four dimensions of the multivariate space. **c**, Probability density plots of CMD across disorders. Vertical black line depicts the median CMD of the control group. Asterisks indicate an FDR-corrected significant difference from the CN group (*P* < 0.001). **d**, Heritability of raw volumetric phenotypes and their centile scores across two twin studies (Adolescent Brain Cognitive Development (ABCD) and Human Connectome Project (HCP)); Supplementary Information 19), see Supplementary Information 13 for a full overview of statistics for each individual feature in each dataset. Data are mean ± s.e.m. (although some confidence intervals are too narrow to be seen). MCI, mild cognitive impairment. See Fig. 3 for other diagnostic abbreviations. FDR-corrected significance: **P* < 0.05, ***P* < 0.01, ****P* < 0.001.

at approximately 11–12 years of age (surface area peak at 10.97 years (95% bootstrap CI 10.42–11.51); TCV peak at 12.5 years (95% bootstrap CI 12.14–12.89). By contrast, cortical thickness peaked distinctively early at 1.7 years (95% bootstrap CI 1.3–2.1), which reconciles previous observations that cortical thickness increases during the perinatal period[48] and declines during later development[49] (Supplementary Information 7).

We also found evidence for regional variability in volumetric neurodevelopmental trajectories. Compared with peak GMV at 5.9 years, the age of peak regional grey matter volume varied considerably–from approximately 2 to 10 years–across 34 cortical areas. Primary sensory regions reached peak volume earliest and showed faster post-peak declines, whereas fronto-temporal association cortical areas peaked later and showed slower post-peak declines (Fig. 2b, Supplementary Information 8.2). Notably, this spatial pattern recapitulated a gradient

from sensory-to-association cortex that has been previously associated with multiple aspects of brain structure and function[50].

## Developmental milestones

Neuroimaging milestones are defined by inflection points of the tissue-specific volumetric trajectories (Fig. 3, Methods, 'Defining developmental milestones'). Among the total tissue volumes, only GMV peaked before the typical age at onset of puberty[51], with sGMV peaking mid-puberty and WMV peaking in young adulthood (Fig. 3). The rate of growth (velocity) peaked in infancy and early childhood for GMV (5.08 months (95% bootstrap CI 4.85–5.22)), sGMV (5.65 months (95% bootstrap CI 5.75–5.83)) and WMV (2.4 years (95% bootstrap CI 2.2–2.6)). TCV velocity peaked between the maximum velocity for GMV and WMV at approximately 7 months. Two major milestones of TCV

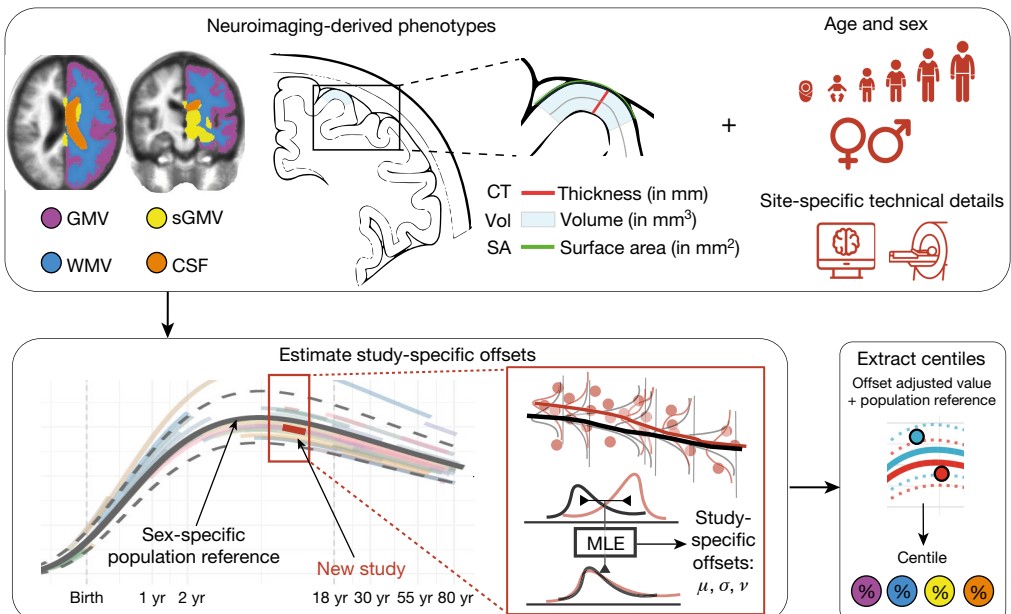

**Fig. 5 | Schematic overview of brain charts, highlighting methods for out-of-sample centile scoring.** Top, brain phenotypes were measured in a reference dataset of MRI scans. GAMLSS modelling was used to estimate the relationship between (global) MRI phenotypes and age, stratified by sex, and controlling for technical and other sources of variation between scanning sites and primary studies. Bottom, the normative trajectory of the median and confidence interval for each phenotype was plotted as a population reference curve. Out-of-sample data from a new MRI study were aligned to the corresponding epoch of the normative trajectory, using maximum likelihood to estimate the study specific offsets (random effects) for three moments of the underlying statistical distributions: mean ($\mu$), variance ($\sigma$), and skewness ($v$) in an age- and sex-specific manner. Centile scores of each phenotype could then be estimated for each scan in the new study, on the same scale as the reference population curve, while accounting for study-specific 'batch effects' on technical or other sources of variation (see Supplementary Information 1.8 for details). MLE, maximum likelihood estimation.

and sGMV (peak velocity and size) (Fig. 3) coincided with the early neonatal and adolescent peaks of height and weight velocity[52,53]. The velocity of mean cortical thickness peaked even earlier, in the prenatal period at −0.38 years (95% bootstrap CI −0.4 to −0.34) (relative to birth), corresponding approximately to mid-gestation. This early peak in cortical thickness velocity has not been reported previously—to our knowledge—in part owing to challenges in acquiring adequate and consistent signal from typical MRI sequences in the perinatal period[54]. Similarly, normative trajectories revealed an early period of GMV:WMV differentiation, beginning in the first month after birth with the switch from WMV to GMV as the proportionally dominant tissue compartment, and ending when the absolute difference of GMV and WMV peaked around 3 years (Supplementary Information 9). This epoch of GMV:WMV differentiation, which may reflect underlying changes in myelination and synaptic proliferation[4,55-58], has not been demarcated in previous studies[45,59]. It was probably identified in this study owing to the substantial amount of early developmental MRI data available for analysis in the aggregated dataset (in total across all primary studies, $N = 2,571$ and $N = 1,484$ participants aged less than 2 years were available for analysis of cerebrum tissue volumes and extended global MRI phenotypes, respectively). The period of GMV:WMV differentiation encompasses dynamic changes in brain metabolites[60] (0–3 months), resting metabolic rate[61] (RMR) (minimum = 7 months, maximum = 4.2 years), the typical period of acquisition of motor capabilities and other early paediatric milestones[62], and the most rapid change in TCV (Fig. 3).

## Individualized centile scores

We computed individualized centile scores that benchmarked each individual scan in the context of normative age-related trends (Methods, 'Centile scores and case–control differences' and Supplementary Information 1–6 for further details). This approach is conceptually similar to quantile rank mapping, as previously reported[26,28,29], where the typicality or atypicality of each phenotype in each scan is quantified by its score on the distribution of phenotypic parameters in the normative or reference sample of scans, with more atypical phenotypes having more extreme centile (or quantile) scores. The clinical diversity of the aggregated dataset enabled us to comprehensively investigate case–control differences in individually specific centile scores across a range of conditions. Relative to the control group (CN), there were highly significant differences in centile scores across large ($N > 500$) groups of cases diagnosed with multiple disorders (Fig. 4a, Supplementary Information 10), with effect sizes ranging from medium (0.2 < Cohen's $d$ < 0.8) to large (Cohen's $d$ > 0.8) (see Supplementary Tables 3, 4 for all false discovery rate (FDR)-corrected $P$ values and effect sizes). Clinical case–control differences in cortical thickness and surface area generally followed the same trend as volume differences (Supplementary Information 10). Alzheimer's disease showed the greatest overall difference, with a maximum difference localized to grey matter volume in biologically female patients (median centile score = 14%, 36 percentage points difference from CN median, corresponding to Cohen's $d$ = 0.88; Fig. 4a). In addition, we generated a cumulative deviation metric, the centile Mahalanobis distance (CMD), to summarize a comparative assessment of brain morphology across all global MRI phenotypes relative to the CN group (Fig. 4b, Supplementary Information 1.6). Notably, schizophrenia ranked third overall behind Alzheimer's disease and mild cognitive impairment (MCI) on the basis of CMD (Fig. 4c). Assessment across diagnostic groups, based on profiles of the multiple centile scores for each MRI phenotype and for CMD, highlighted shared and distinct patterns across clinical conditions (Supplementary Information 10, 11). However, when examining cross-disorder similarity of multivariate centile scores, hierarchical clustering yielded three clusters broadly comprising neurodegenerative, mood and anxiety, and neurodevelopmental disorders (Supplementary Information 11).

Across all major epochs of the lifespan[63], the CMD was consistently greater in cases relative to controls, irrespective of diagnostic category.

The largest case–control differences across epochs occurred in late adulthood when risk for dementia increases and in adolescence, which is well-recognized as a period of increased incidence of mental health disorders (Supplementary Information 10.3). In five primary studies covering the lifespan, average centile scores across global tissues were related to two metrics of premature birth (gestational age at birth: $t = 13.164$, $P < 2 \times 10^{-16}$; birth weight: $t = 36.395$, $P < 2 \times 10^{-16}$; Supplementary Information 12), such that greater gestational age and birth weight were associated with higher average centile scores. Centile scores also showed increased twin-based heritability in two independent studies (total $N = 913$ twin pairs) compared with non-centiled phenotypes (average increase of 11.8 percentage points in narrow sense heritability ($h^2$) across phenotypes; Fig. 4d, Supplementary Information 13). In summary, centile normalization of brain metrics reproducibly detected case–control differences and genetic effects on brain structure, as well as long-term sequelae of adverse birth outcomes even in the adult brain[10].

## Longitudinal centile changes

Owing to the relative paucity of longitudinal imaging data (about 10% of the reference dataset), normative models were estimated from cross-sectional data collected at a single time point. However, the generalizability of cross-sectional models to longitudinal assessment is important for future research. Within-subject variability of centile scores derived from longitudinally repeated scans, measured with the interquartile range (IQR) (Methods, 'Longitudinal stability', Supplementary Information 1.7), was low across both clinical and CN groups (all median IQR < 0.05 centile points), indicating that centile scoring of brain structure was generally stable over time, although there was also some evidence of between-study and cross-disorder differences in within-subject variability (Supplementary Information 14). Notably, individuals who changed diagnostic categories–for example, those who progressed from mild cognitive impairment to Alzheimer's disease over the course of repeated scanning–showed small but significant increases in within-subject variability of centile scores (Supplementary Information 14, Supplementary Tables 5, 6). Within-subject variability was also slightly higher in samples from younger individuals (Supplementary Information 14), which could reflect increased noise due to the technical or data quality challenges associated with scanning younger individuals, but is also consistent with the evidence of increased variability in earlier development observed across other anthropometric traits[64].

## Centile scoring of new MRI data

A key challenge for brain charts is the accurate centile scoring of out-of-sample MRI data, not represented in the reference dataset used to estimate normative trajectories. We therefore carefully evaluated the reliability and validity of brain charts for centile scoring of such 'new' scans. For each new MRI study, we used maximum likelihood to estimate study-specific statistical offsets from the age-appropriate epoch of the normative trajectory; we then estimated centile scores for each individual in the new study benchmarked against the offset trajectory (Fig. 5, Methods, 'Data-sharing and out-of-sample estimation', Supplementary Information 1.8). Extensive jack-knife and leave-one-study-out analyses indicated that a study size of $N > 100$ scans was sufficient for stable and unbiased estimation of out-of-sample centile scores (Supplementary Information 4). This study size limit is in line with the size of many contemporary brain MRI research studies. However, these results do not immediately support the use of brain charts to generate centile scores from smaller-scale research studies, or from an individual patient's scan in clinical practice–this remains a goal for future work. Out-of-sample centile scores proved highly reliable in multiple test–retest datasets and were robust to variations in image processing pipelines (Supplementary Information 4).

## Discussion

We have aggregated the largest neuroimaging dataset to date to modernize the concept of growth charts for mapping typical and atypical human brain development and ageing. The approximately 100-year age range enabled the delineation of milestones and critical periods in maturation of the human brain, revealing an early growth epoch across its constituent tissue classes—beginning before 17 post-conception weeks, when the brain is at approximately 10% of its maximum size, and ending by age 3, when the brain is at approximately 80% of the maximum size. Individual centile scores benchmarked by normative neurodevelopmental trajectories were significantly associated with neuropsychiatric disorders as well as with dimensional phenotypes (Supplementary Information 5.2, 12). Furthermore, imaging–genetics studies[65] may benefit from the increased heritability of centile scores compared with raw volumetric data (Supplementary Information 13). Perhaps most importantly, GAMLSS modelling enabled harmonization across technically diverse studies (Supplementary Information 5), and thus unlocked the potential value of combining primary MRI studies at scale to generate normative, sex-stratified brain growth charts, and individual centile scores of typicality and atypicality.

The analogy to paediatric growth charts is not meant to imply that brain charts are immediately suitable for benchmarking or quantitative diagnosis of individual patients in clinical practice. Even for traditional anthropometric growth charts (height, weight and BMI), there are still important caveats and nuances concerning their diagnostic interpretation in individual children[66]; similarly, it is expected that considerable further research will be required to validate the clinical diagnostic utility of brain charts. However, the current results bode well for future progress towards digital diagnosis of atypical brain structure and development[67]. By providing an age- and sex-normalized metric, centile scores enable trans-diagnostic comparisons between disorders that emerge at different stages of the lifespan (Supplementary Information 10, 11). The generally high stability of centile scores across longitudinal measurements also enabled assessment of brain changes related to diagnostic transition from mild cognitive impairment to Alzheimer's disease (Supplementary Information 14), which provides one example of how centile scoring could be clinically useful in quantitatively predicting or diagnosing progressive neurodegenerative disorders in the future. Our provision of appropriate normative growth charts and online tools also creates an immediate opportunity to quantify atypical brain structure in clinical research samples, to leverage available legacy neuroimaging datasets, and to enhance ongoing studies.

Several important caveats are worth highlighting. Even this large MRI dataset was biased towards European and North American populations and European ancestry groups within those populations. This bias is unfortunately common in many clinical and scientific references, including anthropometric growth charts and benchmark genetic datasets, representing an inequity that must be addressed by the global scientific community[68]. In the particular case of brain charts, further increasing ethnic, socioeconomic and demographic diversity in MRI research will enable more population-representative normative trajectories[69,70] that can be expected to improve the accuracy and strengthen the interpretation of centile scores in relation to appropriate norms[26]. The available reference data were also not equally distributed across all ages—for example, foetal, neonatal and mid-adulthood (30–40 years of age) epochs were under-represented (Supplementary Information 17–19). Furthermore, although our statistical modelling approach was designed to mitigate study- or site-specific effects on centile scores, it cannot entirely correct for limitations of primary study design, such as ascertainment bias or variability in diagnostic criteria. Our decision to stratify the lifespan models by sex followed the analogous logic of sex-stratified anthropometric growth charts. Males have larger brain-tissue volumes than females in absolute terms (Supplementary

Information 16), but this is not indicative of any difference in clinical or cognitive outcomes. Future work would benefit from more detailed and dimensional self-report variables relating to sex and gender[71]. The use of brain charts also does not circumvent the fundamental requirement for quality control of MRI data. We have shown that GAMLSS modelling of global structural MRI phenotypes is in fact remarkably robust to inclusion of poor-quality scans (Supplementary Information 2), but it should not be assumed that this level of robustness will apply to future brain charts of regional MRI or functional MRI phenotypes; therefore, the importance of quality control remains paramount.

We have focused primarily on global brain phenotypes, which were measurable in the largest achievable sample, aggregated over the widest age range, with the fewest methodological, theoretical and data-sharing constraints. However, we have also provided proof-of-concept brain charts for regional grey matter volumetrics, demonstrating plausible heterochronicity of cortical patterning, and illustrating the potential generalizability of this approach to a diverse range of fine-grained MRI phenotypes (Fig. 2, Supplementary Information 8). As ongoing and future efforts provide increasing amounts of high-quality MRI data, we predict an iterative process of improved brain charts for an increasing number of multimodal[72] neuroimaging phenotypes. Such diversification will require the development, implementation and standardization of additional data quality control procedures[27] to underpin robust brain chart modelling. To facilitate further research using our reference charts, we have provided interactive tools to explore these statistical models and to derive normalized centile scores for new datasets across the lifespan at www.brainchart.io.

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

R. A. I. Bethlehem[1,2,219✉], J. Seidlitz[3,4,5,219✉], S. R. White[6,7,219], J. W. Vogel[3,8], K. M. Anderson[9], C. Adamson[10,11], S. Adler[12], G. S. Alexopoulos[13], E. Anagnostou[14,15], A. Areces-Gonzalez[16,17], D. E. Astle[18], B. Auyeung[1,19], M. Ayub[20,21], J. Bae[22], G. Ball[10,23], S. Baron-Cohen[1,24], R. Beare[10,11], S. A. Bedford[1], V. Benegal[25], F. Beyer[26], J. Blangero[27], M. Blesa Cábez[28], J. P. Boardman[28], M. Borzage[29], J. F. Bosch-Bayard[30,31], N. Bourke[32,33], V. D. Calhoun[34], M. M. Chakravarty[31,35], C. Chen[36], C. Chertavian[5], G. Chetelat[37], Y. S. Chong[38,39], J. H. Cole[40,41], A. Corvin[42], M. Costantino[43,44], E. Courchesne[45,46], F. Crivello[47], V. L. Cropley[48], J. Crosbie[49], N. Crossley[50,51,52], M. Delarue[37], R. Delorme[53,54], S. Desrivieres[55], G. A. Devenyi[56,57], M. A. Di Biase[48,58], R. Dolan[59,60], K. A. Donald[61,62], K. Donohoe[63], K. Dunlop[64], A. D. Edwards[65,66,67], J. T. Elison[68], C. T. Ellis[9,69], J. A. Elman[70], L. Eyler[71,72], D. A. Fair[68], E. Feczko[68], P. C. Fletcher[73,74], P. Fonagy[75,76], C. E. Franz[70], L. Galan-Garcia[77], A. Gholipour[78], J. Giedd[79,80], J. H. Gilmore[81], D. C. Glahn[82,83], I. M. Goodyer[1], P. E. Grant[84], N. A. Groenewold[62,85], F. M. Gunning[86], R. E. Gur[3,5], R. C. Gur[3,5], C. F. Hammill[49,87], O. Hansson[88,89], T. Hedden[90,91], A. Heinz[92], R. N. Henson[6,18], K. Heuer[93,94], J. Hoare[95], B. Holla[96,97], A. J. Holmes[98], R. Holt[1], H. Huang[99,100], K. Im[82,84], J. Ipser[101], C. R. Jack Jr[102], A. P. Jackowski[103,104], T. Jia[105,106,107], K. A. Johnson[83,108,109,110], P. B. Jones[6,74], D. T. Jones[102,111], R. S. Kahn[112], H. Karlsson[113,114], L. Karlsson[113,114], R. Kawashima[115], E. A. Kelley[116], S. Kern[117,118], K. W. Kim[119,120,121,122], M. G. Kitzbichler[2,6], W. S. Kremen[70], F. Lalonde[123], B. Landeau[37], S. Lee[124], J. Lerch[87,125,126], J. D. Lewis[127], J. Li[128], W. Liao[128], C. Liston[129], M. V. Lombardo[1,130], J. Lv[48,131], C. Lynch[64], T. T. Mallard[132], M. Marcelis[133,134], R. D. Markello[135], S. R. Mathias[82], B. Mazoyer[47,136], P. McGuire[51], M. J. Meaney[136,137], A. Mechelli[138], N. Medic[6], B. Misic[135], S. E. Morgan[6,139,140], D. Mothersill[141,142,143], J. Nigg[144], M. Q. W. Ong[145], C. Ortinau[146], R. Ossenkoppele[147,148], M. Ouyang[99], L. Palaniyappan[149], L. Paly[37], P. M. Pan[150,151], C. Pantelis[152,153,154], M. M. Park[155], T. Paus[156,157], Z. Pausova[49,158], D. Paz-Linares[16,159], A. Pichet Binette[160,161], K. Pierce[45], X. Qian[145], J. Qiu[162], A. Qiu[163], A. Raznahan[123], T. Rittman[164], A. Rodrigue[82], C. K. Rollins[165,166], R. Romero-Garcia[6,167], L. Ronan[6], M. D. Rosenberg[168], D. H. Rowitch[169], G. A. Salum[170,171], T. D. Satterthwaite[3,8], H. L. Schaare[172,173], R. J. Schachar[49], A. P. Schultz[83,108,174], G. Schumann[175,176], M. Schöll[177,178,179], D. Sharp[32,180], R. T. Shinohara[36,181], I. Skoog[117,118], C. D. Smyser[182], R. A. Sperling[83,108,109], D. J. Stein[183], A. Stolicyn[184], J. Suckling[6,74], G. Sullivan[28], Y. Taki[115], B. Thyreau[115], R. Toro[94,185], N. Traut[185,186], K. A. Tsvetanov[164,187], N. B. Turk-Browne[9,188], J. J. Tuulari[113,189,190], C. Tzourio[191], É. Vachon-Presseau[192], M. J. Valdes-Sosa[77], P. A. Valdes-Sosa[128,193], S. L. Valk[194,195], T. van Amelsvoort[196], S. N. Vandekar[197,198], L. Vasung[135], L. W. Victoria[86], S. Villeneuve[135,160,161], A. Villringer[26,199], P. E. Vértes[6,140], K. Wagstyl[6], Y. S. Wang[200,201,202,203], V. Warrier[6], V. Warrier[6], L. Westlye[204], M. L. Westwater[6], H. C. Whalley[184], A. V. Witte[26,199,205], N. Yang[200,201,202,203], B. Yeo[206,207,208,209], H. Yun[84], A. Zalesky[48,210], H. J. Zar[85], A. Zettergren[117], J. H. Zhou[145,206,211], H. Ziauddeen[6,74,212], A. Zugman[151,213,214], X. N. Zuo[200,201,202,203,215], 3R-BRAIN*, AIBL, Alzheimer's Disease Neuroimaging Initiative, Alzheimer's Disease Repository Without Borders Investigators, CALM Team, Cam-CAN, CCNP, COBRE, cVEDA, ENIGMA Developmental Brain Age Working Group, Developing Human Connectome Project, FinnBrain, Harvard Aging Brain Study, IMAGEN, KNE96, The Mayo Clinic Study of Aging, NSPN, POND, The PREVENT-AD Research Group, VETSA, E. T. Bullmore[6,220] & A. F. Alexander-Bloch[3,4,5,220]

[1]Autism Research Centre, Department of Psychiatry, University of Cambridge, Cambridge, UK. [2]Brain Mapping Unit, Department of Psychiatry, University of Cambridge, Cambridge, UK. [3]Department of Psychiatry, University of Pennsylvania, Philadelphia, PA, USA. [4]Department of Child and Adolescent Psychiatry and Behavioral Science, The Children's Hospital of Philadelphia, Philadelphia, PA, USA. [5]Lifespan Brain Institute, The Children's Hospital of Philadelphia and Penn Medicine, Philadelphia, PA, USA. [6]Department of Psychiatry, University of Cambridge, Cambridge, UK. [7]MRC Biostatistics Unit, University of Cambridge, Cambridge, UK. [8]Lifespan Informatics & Neuroimaging Center, University of Pennsylvania, Philadelphia, PA, USA. [9]Department of Psychology, Yale University, New Haven, CT, USA. [10]Developmental Imaging, Murdoch Children's Research Institute, Melbourne, Victoria, Australia. [11]Department of Medicine, Monash University, Melbourne, Victoria, Australia. [12]UCL Great Ormond Street Institute for Child Health, London, UK. [13]Weill Cornell Institute of Geriatric Psychiatry, Department of Psychiatry, Weill Cornell Medicine, New York, USA. [14]Department of Pediatrics University of Toronto, Toronto, Canada. [15]Holland Bloorview Kids Rehabilitation Hospital, Toronto, Canada. [16]The Clinical Hospital of Chengdu Brain Science Institute, MOE Key Lab for NeuroInformation, University of Electronic Science and Technology of China, Chengdu, China. [17]University of Pinar del Río "Hermanos Saiz Montes de Oca", Pinar del Río, Cuba. [18]MRC Cognition and Brain Sciences Unit, University of Cambridge, Cambridge, UK. [19]Department of Psychology, School of Philosophy, Psychology and Language Sciences, University of Edinburgh, Edinburgh, UK. [20]Queen's University, Department of Psychiatry, Centre for Neuroscience Studies, Kingston, Ontario, Canada. [21]University College London, Mental Health Neuroscience Research Department, Division of Psychiatry, London, UK. [22]Department of Neuropsychiatry, Seoul National University Bundang Hospital, Seongnam, Korea. [23]Department of Paediatrics, University of Melbourne, Melbourne, Victoria, Australia. [24]Cambridge Lifetime Asperger Syndrome Service (CLASS), Cambridgeshire and Peterborough NHS Foundation Trust, Cambridge, UK. [25]Centre for Addiction Medicine, National Institute of Mental Health and Neurosciences (NIMHANS), Bengaluru, India. [26]Department of Neurology, Max Planck Institute for Human Cognitive and Brain Sciences, Leipzig, Germany. [27]Department of Human Genetics, South Texas Diabetes and Obesity Institute, University of Texas Rio Grande Valley, Edinburg, TX, USA. [28]MRC Centre for Reproductive Health, University of Edinburgh, Edinburgh, UK. [29]Fetal and Neonatal Institute, Division of Neonatology, Children's Hospital Los Angeles, Department of Pediatrics, Keck School of Medicine, University of Southern California, Los Angeles, CA, USA. [30]McGill Centre for Integrative Neuroscience, Ludmer Centre for Neuroinformatics and Mental Health, Montreal Neurological Institute, Montreal, Quebec, Canada. [31]McGill University, Montreal, Quebec, Canada. [32]Department of Brain Sciences, Imperial College London, London, UK. [33]Care Research and Technology Centre, Dementia Research Institute, London, UK. [34]Tri-institutional Center for Translational Research in Neuroimaging and Data Science (TReNDS), Georgia State University, Georgia Institute of Technology, and Emory University, Atlanta, GA, USA. [35]Computational Brain Anatomy (CoBrA) Laboratory, Cerebral Imaging Centre, Douglas Mental Health University Institute, Montreal, Quebec, Canada. [36]Penn Statistics in Imaging and Visualization Center, Department of Biostatistics, Epidemiology, and Informatics, Perelman School of Medicine, University of Pennsylvania, Philadelphia, PA, USA. [37]Normandie Univ, UNICAEN, INSERM, U1237, PhIND "Physiopathology and Imaging of Neurological Disorders", Institut Blood and Brain @ Caen-Normandie, Cyceron, Caen, France.

[38]Singapore Institute for Clinical Sciences, Agency for Science, Technology and Research, Singapore, Singapore. [39]Department of Obstetrics and Gynaecology, Yong Loo Lin School of Medicine, National University of Singapore, Singapore, Singapore. [40]Centre for Medical Image Computing (CMIC), University College London, London, UK. [41]Dementia Research Centre (DRC), University College London, London, UK. [42]Department of Psychiatry, Trinity College, Dublin, Ireland. [43]Cerebral Imaging Centre, Douglas Mental Health University Institute, Verdun, Quebec, Canada. [44]Undergraduate program in Neuroscience, McGill University, Montreal, Quebec, Canada. [45]Department of Neuroscience, University of California, San Diego, San Diego, CA, USA. [46]Autism Center of Excellence, University of California, San Diego, San Diego, CA, USA. [47]Institute of Neurodegenerative Disorders, CNRS UMR5293, CEA, University of Bordeaux, Bordeaux, France. [48]Melbourne Neuropsychiatry Centre, University of Melbourne, Melbourne, Victoria, Australia. [49]The Hospital for Sick Children, Toronto, Ontario, Canada. [50]Department of Psychiatry, School of Medicine, Pontificia Universidad Católica de Chile, Santiago, Chile. [51]Department of Psychosis Studies, Institute of Psychiatry, Psychology and Neuroscience, King's College London, London, UK. [52]Instituto Milenio Intelligent Healthcare Engineering, Santiago, Chile. [53]Child and Adolescent Psychiatry Department, Robert Debré University Hospital, AP-HP, Paris, France. [54]Human Genetics and Cognitive Functions, Institut Pasteur, Paris, France. [55]Social, Genetic and Developmental Psychiatry Centre, Institute of Psychiatry, Psychology and Neuroscience, King's College London, London, UK. [56]Cerebral Imaging Centre, McGill Department of Psychiatry, Douglas Mental Health University Institute, Montreal, QC, Canada. [57]Department of Psychiatry, McGill University, Montreal, QC, Canada. [58]Department of Psychiatry, Brigham and Women's Hospital, Harvard Medical School, Boston, MA, USA. [59]Max Planck UCL Centre for Computational Psychiatry and Ageing Research, University College London, London, UK. [60]Wellcome Centre for Human Neuroimaging, London, UK. [61]Division of Developmental Paediatrics, Department of Paediatrics and Child Health, Red Cross War Memorial Children's Hospital, Cape Town, South Africa. [62]Neuroscience Institute, University of Cape Town, Cape Town, South Africa. [63]Center for Neuroimaging, Cognition & Genomics (NICOG), School of Psychology, National University of Ireland Galway, Galway, Ireland. [64]Weil Family Brain and Mind Research Institute, Department of Psychiatry, Weill Cornell Medicine, New York, NY, USA. [65]Centre for the Developing Brain, King's College London, London, UK. [66]Evelina London Children's Hospital, London, UK. [67]MRC Centre for Neurodevelopmental Disorders, London, UK. [68]Institute of Child Development, Department of Pediatrics, Masonic Institute for the Developing Brain, University of Minnesota, Minneapolis, MN, USA. [69]Haskins Laboratories, New Haven, CT, USA. [70]Department of Psychiatry, Center for Behavior Genetics of Aging, University of California, San Diego, La Jolla, CA, USA. [71]Desert-Pacific Mental Illness Research Education and Clinical Center, VA San Diego Healthcare, San Diego, CA, USA. [72]Department of Psychiatry, University of California San Diego, Los Angeles, CA, USA. [73]Department of Psychiatry, University of Cambridge, and Wellcome Trust MRC Institute of Metabolic Science, Cambridge Biomedical Campus, Cambridge, UK. [74]Cambridgeshire and Peterborough NHS Foundation Trust, Cambridge, UK. [75]Department of Clinical, Educational and Health Psychology, University College London, London, UK. [76]Anna Freud National Centre for Children and Families, London, UK. [77]Cuban Center for Neuroscience, La Habana, Cuba. [78]Computational Radiology Laboratory, Boston Children's Hospital, Boston, MA, USA. [79]Department of Child and Adolescent Psychiatry, University of California, San Diego, San Diego, CA, USA. [80]Department of Psychiatry, University of California San Diego, San Diego, CA, USA. [81]Department of Psychiatry, University of North Carolina, Chapel Hill, NC, USA. [82]Department of Psychiatry, Boston Children's Hospital and Harvard Medical School, Boston, MA, USA. [83]Harvard Medical School, Boston, MA, USA. [84]Division of Newborn Medicine and Neuroradiology, Fetal Neonatal Neuroimaging and Developmental Science Center, Boston Children's Hospital, Harvard Medical School, Boston, MA, USA. [85]Department of Paediatrics and Child Health, Red Cross War Memorial Children's Hospital, SA-MRC Unit on Child & Adolescent Health, University of Cape Town, Cape Town, South Africa. [86]Weill Cornell Institute of Geriatric Psychiatry, Department of Psychiatry, Weill Cornell Medicine, New York, NY, USA. [87]Mouse Imaging Centre, Toronto, Ontario, Canada. [88]Clinical Memory Research Unit, Department of Clinical Sciences Malmö, Lund University, Malmö, Sweden. [89]Memory Clinic, Skåne University Hospital, Malmö, Sweden. [90]Department of Neurology, Icahn School of Medicine at Mount Sinai, New York, NY, USA. [91]Athinoula A. Martinos Center for Biomedical Imaging, Department of Radiology, Massachusetts General Hospital, Harvard Medical School, Boston, MA, USA. [92]Charité – Universitätsmedizin Berlin, corporate member of Freie Universität Berlin and Humboldt-Universität zu Berlin, Department of Psychiatry and Psychotherapy, Charité Campus Mitte, Berlin, Germany. [93]Department of Neuropsychology, Max Planck Institute for Human Cognitive and Brain Sciences, Leipzig, Germany. [94]Université de Paris, Paris, France. [95]Department of Psychiatry, University of Cape Town, Cape Town, South Africa. [96]Department of Integrative Medicine, NIMHANS, Bengaluru, India. [97]Accelerator Program for Discovery in Brain disorders using Stem cells (ADBS), Department of Psychiatry, NIMHANS, Bengaluru, India. [98]Departments of Psychology and Psychiatry, Yale University, New Haven, CT, USA. [99]Radiology Research, Children's Hospital of Philadelphia, Philadelphia, PA, USA. [100]The Department of Radiology, Perelman School of Medicine, University of Pennsylvania, Philadelphia, PA, USA. [101]Department of Psychiatry and Mental Health, Clinical Neuroscience Institute, University of Cape Town, Cape Town, South Africa. [102]Department of Radiology, Mayo Clinic, Rochester, MN, USA. [103]Department of Psychiatry, Universidade Federal de São Paulo, São Paulo, Brazil. [104]National Institute of Developmental Psychiatry, Beijing, China. [105]Institute of Science and Technology for Brain-Inspired Intelligence, Fudan University, Shanghai, China. [106]Key Laboratory of Computational Neuroscience and BrainInspired Intelligence (Fudan University), Ministry of Education, Shanghai, China. [107]Centre for Population Neuroscience and Precision Medicine (PONS), Institute of Psychiatry, Psychology and Neuroscience, SGDP Centre, King's College London, London, UK. [108]Harvard Aging Brain Study, Department of Neurology, Massachusetts General Hospital, Boston, MA, USA. [109]Center for Alzheimer Research and Treatment, Department of Neurology, Brigham and Women's Hospital, Boston, MA, USA. [110]Department of Radiology, Massachusetts General Hospital, Boston, MA, USA. [111]Department of Neurology, Mayo Clinic, Rochester, MN, USA. [112]Department of Psychiatry, Icahn School of Medicine, Mount Sinai, NY, USA. [113]Department of Clinical Medicine, Department of Psychiatry and Turku Brain and Mind Center, FinnBrain Birth Cohort Study, University of Turku and Turku University Hospital, Turku, Finland. [114]Centre for Population Health Research, Turku University Hospital and University of Turku, Turku, Finland. [115]Institute of Development, Aging and Cancer, Tohoku University, Seiryocho, Aobaku, Sendai, Japan.

[116]Queen's University, Departments of Psychology and Psychiatry, Centre for Neuroscience Studies, Kingston, Ontario, Canada. [117]Neuropsychiatric Epidemiology Unit, Department of Psychiatry and Neurochemistry, Institute of Neuroscience and Physiology, the Sahlgrenska Academy, Centre for Ageing and Health (AGECAP) at the University of Gothenburg, Gothenburg, Sweden. [118]Region Västra Götaland, Sahlgrenska University Hospital, Psychiatry, Cognition and Old Age Psychiatry Clinic, Gothenburg, Sweden. [119]Department of Brain and Cognitive Sciences, Seoul National University College of Natural Sciences, Seoul, South Korea. [120]Department of Neuropsychiatry, Seoul National University Bundang Hospital, Seongnam, South Korea. [121]Department of Psychiatry, Seoul National University College of Medicine, Seoul, South Korea. [122]Institute of Human Behavioral Medicine, SNU-MRC, Seoul, South Korea. [123]Section on Developmental Neurogenomics, Human Genetics Branch, National Institute of Mental Health, Bethesda, MD, USA. [124]Department of Brain & Cognitive Sciences, Seoul National University College of Natural Sciences, Seoul, South Korea. [125]Department of Medical Biophysics, University of Toronto, Toronto, Ontario, Canada. [126]Wellcome Centre for Integrative Neuroimaging, FMRIB, Nuffield Department of Clinical Neuroscience, University of Oxford, Oxford, UK. [127]Montreal Neurological Institute, McGill University, Montreal, Quebec, Canada. [128]The Clinical Hospital of Chengdu Brain Science Institute, University of Electronic Science and Technology of China, Chengdu, China. [129]Department of Psychiatry and Brain and Mind Research Institute, Weill Cornell Medicine, New York, NY, USA. [130]Laboratory for Autism and Neurodevelopmental Disorders, Center for Neuroscience and Cognitive Systems @UniTn, Istituto Italiano di Tecnologia, Rovereto, Italy. [131]School of Biomedical Engineering and Brain and Mind Centre, The University of Sydney, Sydney, New South Wales, Australia. [132]Department of Psychology, University of Texas, Austin, TX, USA. [133]Department of Psychiatry and Neuropsychology, School of Mental Health and Neuroscience, EURON, Maastricht University Medical Centre, Maastricht, The Netherlands. [134]Institute for Mental Health Care Eindhoven (GGzE), Eindhoven, The Netherlands. [135]McConnell Brain Imaging Centre, Montreal Neurological Institute, McGill University, Montreal, Quebec, Canada. [136]Ludmer Centre for Neuroinformatics and Mental Health, Douglas Mental Health University Institute, Montreal, Quebec, Canada. [137]Singapore Institute for Clinical Sciences, Singapore, Singapore. [138]Bordeaux University Hospital, Bordeaux, France. [139]Department of Computer Science and Technology, University of Cambridge, Cambridge, UK. [140]The Alan Turing Institute, London, UK. [141]Department of Psychology, School of Business, National College of Ireland, Dublin, Ireland. [142]School of Psychology and Center for Neuroimaging and Cognitive Genomics, National University of Ireland Galway, Galway, Ireland. [143]Department of Psychiatry, Trinity College Dublin, Dublin, Ireland. [144]Department of Psychiatry, School of Medicine, Oregon Health and Science University, Portland, OR, USA. [145]Center for Sleep and Cognition, Yong Loo Lin School of Medicine, National University of Singapore, Singapore, Singapore. [146]Department of Pediatrics, Washington University in St Louis, St Louis, MO, USA. [147]Alzheimer Center Amsterdam, Department of Neurology, Amsterdam Neuroscience, Vrije Universiteit Amsterdam, Amsterdam UMC, Amsterdam, The Netherlands. [148]Lund University, Clinical Memory Research Unit, Lund, Sweden. [149]Robarts Research Institute and The Brain and Mind Institute, University of Western Ontario, London, Ontario, Canada. [150]Department of Psychiatry, Federal University of Sao Poalo (UNIFESP), Sao Poalo, Brazil. [151]National Institute of Developmental Psychiatry for Children and Adolescents (INPD), Sao Poalo, Brazil. [152]Melbourne Neuropsychiatry Centre, Department of Psychiatry, The University of Melbourne and Melbourne Health, Carlton South, Victoria, Australia. [153]Melbourne School of Engineering, The University of Melbourne, Parkville, Victoria, Australia. [154]Florey Institute of Neuroscience and Mental Health, Parkville, Victoria, Australia. [155]Department of Psychiatry, Schulich School of Medicine and Dentistry, Western University, London, Ontario, Canada. [156]Department of Psychiatry, Faculty of Medicine and Centre Hospitalier Universitaire Sainte-Justine, University of Montreal, Montreal, Quebec, Canada. [157]Departments of Psychiatry and Psychology, University of Toronto, Toronto, Ontario, Canada. [158]Departments of Physiology and Nutritional Sciences, University of Toronto, Toronto, Ontario, Canada. [159]Cuban Neuroscience Center, Havana, Cuba. [160]Department of Psychiatry, Faculty of Medicine, McGill University, Montreal, Quebec, Canada. [161]Douglas Mental Health University Institute, Montreal, Quebec, Canada. [162]School of Psychology, Southwest University, Chongqing, China. [163]Department of Biomedical Engineering, The N.1 Institute for Health, National University of Singapore, Singapore, Singapore. [164]Department of Clinical Neurosciences, University of Cambridge, Cambridge, UK. [165]Department of Neurology, Harvard Medical School, Boston, MA, USA. [166]Department of Neurology, Boston Children's Hospital, Boston, MA, USA. [167]Instituto de Biomedicina de Sevilla (IBiS) HUVR/CSIC/Universidad de Sevilla, Dpto. de Fisiología Médica y Biofísica, Seville, Spain. [168]Department of Psychology and Neuroscience Institute, University of Chicago, Chicago, IL, USA. [169]Department of Paediatrics and Wellcome-MRC Cambridge Stem Cell Institute, University of Cambridge, Cambridge, UK. [170]Department of Psychiatry, Universidade Federal do Rio Grande do Sul (UFRGS), Hospital de Clinicas de Porto Alegre, Porto Alegre, Brazil. [171]National Institute of Developmental Psychiatry (INPD), São Paulo, Brazil. [172]Otto Hahn Group Cognitive Neurogenetics, Max Planck Institute for Human Cognitive and Brain Sciences, Leipzig, Germany. [173]Institute of Neuroscience and Medicine (INM-7: Brain and Behaviour), Research Centre Juelich, Juelich, Germany. [174]Athinoula A. Martinos Center for Biomedical Imaging, Department of Radiology, Massachusetts General Hospital, Charlestown, MA, USA. [175]Centre for Population Neuroscience and Stratified Medicine (PONS), Institute for Science and Technology for Brain-inspired Intelligence, Fudan University, Shanghai, China. [176]PONS-Centre, Charite Mental Health, Dept of Psychiatry and Psychotherapy, Charite Campus Mitte, Berlin, Germany. [177]Wallenberg Centre for Molecular and Translational Medicine, University of Gothenburg, Gothenburg, Sweden. [178]Department of Psychiatry and Neurochemistry, University of Gothenburg, Gothenburg, Sweden. [179]Dementia Research Centre, Queen's Square Institute of Neurology, University College London, London, UK. [180]Care Research and Technology Centre, UK Dementia Research Institute, London, UK. [181]Center for Biomedical Image Computing and Analytics, Department of Radiology, Perelman School of Medicine, University of Pennsylvania, Philadelphia, PA, USA. [182]Departments of Neurology, Pediatrics, and Radiology, Washington University School of Medicine, St Louis, MO, USA. [183]SA MRC Unit on Risk and Resilience in Mental Disorders, Dept of Psychiatry and Neuroscience Institute, University of Cape Town, Cape Town, South Africa. [184]Division of Psychiatry, Centre for Clinical Brain Sciences, University of Edinburgh, Edinburgh, UK. [185]Department of Neuroscience, Institut Pasteur, Paris, France. [186]Center for Research and Interdisciplinarity (CRI), Université Paris Descartes, Paris, France. [187]Department of Psychology, University of Cambridge, Cambridge, UK. [188]Wu Tsai Institute, Yale University, New Haven, CT, USA. [189]Department of Clinical Medicine, University of Turku, Turku, Finland. [190]Turku Collegium for Science, Medicine and Technology, University of Turku, Turku, Finland. [191]Univ. Bordeaux, Inserm, Bordeaux Population Health Research Center, U1219, CHU Bordeaux, Bordeaux, France. [192]Faculty of Dental Medicine and Oral Health Sciences, McGill University, Montreal, Quebec, Canada. [193]Alan Edwards Centre for Research on Pain (AECRP), McGill University, Montreal, Quebec, Canada. [194]Institute for Neuroscience and Medicine 7, Forschungszentrum Jülich, Jülich, Germany. [195]Max Planck Institute for Human Cognitive and Brain Sciences, Leipzig, Germany. [196]Department of Psychiatry and Neurosychology, Maastricht University, Maastricht, The Netherlands. [197]Department of Biostatistics, Vanderbilt University, Nashville, TN, USA. [198]Department of Biostatistics, Vanderbilt University Medical Center, Nashville, TN, USA. [199]Clinic for Cognitive Neurology, University of Leipzig Medical Center, Leipzig, Germany. [200]State Key Laboratory of Cognitive Neuroscience and Learning, Beijing Normal University, Beijing, China. [201]Developmental Population Neuroscience Research Center, IDG/McGovern Institute for Brain Research, Beijing Normal University, Beijing, China. [202]National Basic Science Data Center, Beijing, China. [203]Research Center for Lifespan Development of Brain and Mind, Institute of Psychology, Chinese Academy of Sciences, Beijing, China. [204]Division of Clinical Geriatrics, Center for Alzheimer Research, Department of Neurobiology, Care Sciences and Society, Karolinska Institutet, Stockholm, Sweden. [205]Faculty of Medicine, CRC 1052 'Obesity Mechanisms', University of Leipzig, Leipzig, Germany. [206]Department of Electrical and Computer Engineering, National University of Singapore, Singapore, Singapore. [207]Centre for Sleep and Cognition and Centre for Translational MR Research, Yong Loo Lin School of Medicine, National University of Singapore, Singapore, Singapore. [208]N.1 Institute for Health & Institute for Digital Medicine, National University of Singapore, Singapore, Singapore. [209]Integrative Sciences and Engineering Programme (ISEP), National University of Singapore, Singapore, Singapore. [210]Department of Biomedical Engineering, University of Melbourne, Melbourne, Victoria, Australia. [211]Center for Translational Magnetic Resonance Research, Yong Loo Lin School of Medicine, National University of Singapore, Singapore, Singapore. [212]Wellcome Trust-MRC Institute of Metabolic Science, University of Cambridge, Cambridge, UK. [213]National Institute of Mental Health (NIMH), National Institutes of Health (NIH), Bethesda, MD, USA. [214]Department of Psychiatry, Escola Paulista de Medicina, São Paulo, Brazil. [215]Key Laboratory of Brain and Education, School of Education Science, Nanning Normal University, Nanning, China. [216]Memory Disorders Clinic, Austin Health, Melbourne, Victoria, Australia. [217]University Hospitals and University of Geneva, Geneva, Switzerland. [218]IRCCS Fatebenefratelli, The National Centre for Alzheimer's and Mental Diseases, Brescia, Italy. [219]These authors contributed equally: R. A. I. Bethlehem, J. Seidlitz, S. R. White. [220]These authors jointly supervised: E. T. Bullmore, A. F. Alexander-Bloch. *Lists of authors and their affiliations appear at the end of the paper. ✉e-mail: rb643@medschl.cam.ac.uk; jakob.seidlitz@pennmedicine.upenn.edu

**3R-BRAIN**

X. N. Zuo[200,201,202,203,215]

**AIBL***

C. Rowe[216]

**Alzheimer's Disease Neuroimaging Initiative***

C. R. Jack Jr[102]

**Alzheimer's Disease Repository Without Borders Investigators***

G. B. Frisoni[217,218]

**CALM Team***

D. E. Astle[18]

**Cam-CAN***

R. N. Henson[6,18]

**CCNP***

Y. S. Wang[200,201,202,203], N. Yang[200,201,202,203] & X. N. Zuo[200,201,202,203,215]

**COBRE***

V. D. Calhoun[34]

**cVEDA***

B. Holla[96,97]

**ENIGMA Developmental Brain Age Working Group***

J. H. Cole[40,41], N. Bourke[32,33], H. C. Whalley[184], D. C. Glahn[82,83], J. Seidlitz[3,4,5,219], R. A. I. Bethlehem[1,2,219] & A. F. Alexander-Bloch[3,4,5,220]

**Developing Human Connectome Project***

A. D. Edwards[65,66,67]

**FinnBrain***

H. Karlsson[113,114], L. Karlsson[113,114], J. D. Lewis[127] & J. J. Tuulari[113,189,190]

**Harvard Aging Brain Study***

K. A. Johnson[83,108,109,110], Reisa Sperling[83,108,109], Aaron Schultz[83,108,174] & Trey Hedden[90,91]

**IMAGEN***

S. Desrivieres[55], A. Heinz[92], T. Jia[105,106,107] & G. Schumann[175,176]

**KNE96***

J. Bae[22], K. W. Kim[119,120,121,122] & S. Lee[124]

**The Mayo Clinic Study of Aging***

C. R. Jack Jr[102] & D. T. Jones[102,111]

**NSPN***

E. T. Bullmore[6,220], R. Dolan[59,60], P. Fonagy[75,76], I. M. Goodyer[6] & P. B. Jones[6,74]

**POND***

E. Anagnostou[14,15], M. Ayub[20,21], J. Crosbie[49], C. F. Hammill[49,87], E. A. Kelley[116], J. Lerch[87,125,126] & R. J. Schachar[49]

**The PREVENT-AD Research Group***

A. Pichet Binette[160,161] & S. Villeneuve[135,160,161]

**VETSA***

J. A. Elman[70], C. E. Franz[70] & W. S. Kremen[70]

## Methods

### Ethics

The research was reviewed by the Cambridge Psychology Research Ethics Committee (PRE.2020.104) and The Children's Hospital of Philadelphia's Institutional Review Board (IRB 20-017874) and deemed not to require PRE or IRB oversight as it consists of secondary analysis of de-identified primary datasets. Informed consent of participants (or their guardians) in primary studies is referenced in Supplementary Information 19 and Supplementary Table 1.

### Model generation and specification

To accurately and comprehensively establish standardized brain reference charts across the lifespan, it is crucial to leverage multiple independent and diverse datasets, especially those spanning prenatal and early postnatal life. Here we sought to chart normative brain development and ageing across the largest age-span and largest aggregated neuroimaging dataset to date using a robust and scalable methodological framework[2,24]. We used GAMLSS[2] to estimate cross-sectional normative age-related trends from 100 studies, comprising a reference dataset of more than 100,000 scans (see Supplementary Tables 1.1–1.7 for full demographic information and Supplementary Information 19 for dataset descriptions). We optimised GAMLSS model specification and parameterization to estimate non-linear normative growth curves, their confidence intervals and first derivatives, separately for males and females, allowing for random effects on the mean and higher order moments of the outcome distributions.

The reliability of the models was assessed and endorsed by cross-validation and bootstrap resampling procedures (Supplementary Information 3). We leveraged these normative trajectories to benchmark individual scans by centile scores, which were then investigated as age-normed and sex-stratified measures of diagnostic and longitudinal atypicalities of brain structure across the lifespan.

The GAMLSS approach allowed not only modelling of age-related changes in brain phenotypes but also age related-changes in the variability of phenotypes, and in the form of both linear and nonlinear changes over time, thereby overcoming potential limitations of conventional additive models that only allow additive means to be modelled[2]. In addition, study-specific offsets (mean and variance) for each brain phenotype were also modelled as random effects. These modelling criteria are particularly important in the context of establishing growth reference charts as recommended by the World Health Organization[24], as it is reasonable to assume the distribution of higher order moments (for example, variance) changes with age, sex, site/study and pre-processing pipeline, and it is impossible to circumvent some of these issues by collecting standardized data longitudinally for individuals spanning the approximately 100-year age range. Furthermore, recent studies suggest that changes in between-subject variability might intersect with vulnerability for developing a mental health condition[74]. The use of data spanning the entire age range is also critical, as data from partial age-windows can bias estimation of growth charts when extrapolated to the whole lifespan. In short, using a sex-stratified approach[24], age, preprocessing pipeline and study were each included in the GAMLSS model estimation of first order ($\mu$) and second order ($\sigma$) distribution parameters of a generalized gamma distribution using fractional polynomials to model nonlinear trends. See Supplementary Information for more details regarding GAMLSS model specification and estimation (Supplementary Information 1), image quality control (Supplementary Information 2), model stability and robustness (Supplementary Information 3, 4), phenotypic validation against non-imaging metrics (Supplementary Information 3, 5.2), inter-study harmonization (Supplementary Information 5) and assessment of cohort effects (Supplementary Information 6).

More formally, the GAMLSS framework can be specified in the following way:

$$Y \sim F(\mu, \sigma, \nu, \tau) \tag{1}$$

$$g_\mu(\mu) = X_\mu \beta_\mu + Z_\mu \gamma_\mu + \sum_i s_{\mu,i}(x_i)$$

$$g_\sigma(\sigma) = X_\sigma \beta_\sigma + Z_\sigma \gamma_\sigma + \sum_i s_{\sigma,i}(x_i)$$

$$g_\nu(\nu) = X_\nu \beta_\nu + Z_\nu \gamma_\nu + \sum_i s_{\nu,i}(x_i)$$

$$g_\tau(\tau) = X_\tau \beta_\tau + Z_\tau \gamma_\tau + \sum_i s_{\tau,i}(x_i)$$

Here, the outcome vector, $Y$, follows a probability distribution $F$ parameterized by up to four parameters, ($\mu, \sigma, \nu, \tau$). The four parameters, depending on the parameterization of the probability density function, may correspond to the mean, variance, skewness, and kurtosis—that is, the first four moments. However, for many distributions there is not a direct one-to-one correspondence. Each component is linked to a linear equation through a link-function, $g_.()$, and each component equation may include three types of terms: fixed effects, $\beta$ (with design matrix $X$); random effects, $\gamma$ (with design matrix $Z$); and non-parametric smoothing functions, $s_{.,i}$ applied to the $i$th covariate for each parameter. The nature of the outcome distribution determines the appropriate link functions and which components are used. In principle any outcome distribution can be used, from well-behaved continuous and discrete outcomes, through to mixtures and truncations.

Here we have used fractional polynomials as a flexible, but not unduly complex, approach to modelling age-related changes in MRI phenotypes. Although non-parametric smoothers are more flexible, they can become unstable and infeasible, especially in the presence of random effects. Hence, the fractional polynomials enter the model within the $X$ terms, with associated coefficients in $\beta$. The GAMLSS framework includes the ability to estimate the most appropriate powers of fractional polynomial expansion within the iterative fitting algorithm, searching across the standard set of powers, $p \in \{-2, -1, -0.5, 0, 0.5, 1, 2, 3\}$, where the design matrix includes the covariate (in this case, age) raised to the power, namely, $x^p$. Fractional polynomials naturally extend to higher-orders, for example a second-order fractional polynomial of the form, $x^{p_1} + x^{p_2}$ (see Supplementary Information 1.3 for further details).

There are several options for including random effects within the GAMLSS framework depending on the desired covariance structures. We consider the simplest case, including a factor-level (or group-level) random intercept, where the observations are grouped by the study covariate. The random effects are drawn from a normal distribution with zero mean and variance to be estimated, $\gamma \sim N(0, \delta^2)$. The ability to include random effects is fundamental to accounting for co-dependence between observations. It is therefore possible to take advantage of the flexibility of 'standard' GAMLSS, as typically used to develop growth charts[24,62,75], while accounting for co-dependence between observations using random effects. The typical applications of GAMLSS assume independent and identically distributed outcomes; however, in this context it is essential to account for within-study covariance implying the observations are no longer independent.

The resulting models were evaluated using several sensitivity analyses and validation approaches. These models of whole-brain and regional morphometric development were robust to variations in image quality, and cross-validated by non-imaging metrics. However, we expect that several sources of variance, including but not limited to

MRI data quality and variability of acquisition protocols, may become increasingly important as brain charting methods are applied to more innovative and/or anatomically fine-grained MRI phenotypes. It will be important for future work to remain vigilant about the potential impact of data quality and other sources of noise on robustness and generalizability of both normative trajectories and the centile scores derived from them.

Based on the model selection criteria, detailed in Supplementary Information 1, the final models for normative trajectories of all MRI phenotypes were specified as illustrated below for GMV:

$$\text{GMV} \sim \text{Generalizsed Gamma}(\mu, \sigma, v) \text{ with}$$
$$\log(\mu) = \alpha_\mu + \alpha_{\mu,\text{sex}}(\text{sex}) + \alpha_{\mu,\text{ver}}(\text{ver}) + \beta_{\mu,1}(\text{age})^{-2} + \beta_{\mu,2}(\text{age})^{-2}$$
$$+ \beta_{\mu,3}(\text{age})^{-2} \log(\text{age})^2 + \gamma_{\mu,\text{study}} \tag{2}$$
$$\log(\sigma) = \alpha_\sigma + \alpha_{\sigma,\text{sex}}(\text{sex}) + \beta_{\sigma,1}(\text{age})^{-2} + \beta_{\sigma,2}(\text{age})^3 + \gamma_{\sigma\text{study}}$$
$$v = \alpha_v$$

For each component of the generalized gamma distribution, $\alpha$ terms correspond to fixed effects of the intercept, sex (female or male), and software version used for pre-processing (five categories); $\beta$ terms correspond to the fixed effects of age, modelled as fractional polynomial functions with the number of terms reflecting the order of the fractional polynomials; and $\gamma$ terms correspond to the study-level random effects. Note that we have explicitly included the link-functions for each component of the generalized gamma, namely the natural logarithm for $\mu$ and $\sigma$ (since these parameters must be positive) and the identity for $v$.

Similarly for the other global MRI phenotypes:

$$\text{WMV} \sim \text{Generalised Gamma}(\mu, \sigma, v) \text{ with}$$
$$\log(\mu) = \alpha_\mu + \alpha_{\mu,\text{sex}}(\text{sex}) + \alpha_{\mu,\text{ver}}(\text{ver}) + \beta_{\mu,1}(\text{age})^{-2} + \beta_{\mu,2}(\text{age})^3$$
$$+ \beta_{\mu,3}(\text{age})^3 \log(\text{age}) + \gamma_{\mu,\text{study}} \tag{3}$$
$$\log(\sigma) = \alpha_\sigma + \alpha_{\sigma,\text{sex}}(\text{sex}) + \beta_{\sigma,1}(\text{age})^{-2} + \beta_{\sigma,2}(\text{age})^3 + \gamma_{\sigma,\text{study}}$$
$$v = \alpha_v,$$

$$\text{sGMV} \sim \text{Generalised Gamma}(\mu, \sigma, v) \text{ with}$$
$$\log(\mu) = \alpha_\mu + \alpha_{\mu,\text{sex}}(\text{sex}) + \alpha_{\mu,\text{ver}}(\text{ver}) + \beta_{\mu,1}(\text{age})^{-2} + \beta_{\mu,2}(\text{age})^{-2}$$
$$\log(\text{age}) + \beta_{\mu,3}(\text{age})^3 + \gamma_{\mu,\text{study}}$$
$$\log(\sigma) = \alpha_\sigma + \alpha_{\sigma,\text{sex}}(\text{sex}) + \beta_{\sigma,1}(\text{age})^{-2} + \beta_{\sigma,2}(\text{age})^{-2} \log(\text{age}) \tag{4}$$
$$+ \gamma_{\sigma,\text{study}}$$
$$v = \alpha_v,$$

$$\text{Ventricles} \sim \text{Generalized Gamma}(\mu, \sigma, v) \text{ with}$$
$$\log(\mu) = \alpha_\mu + \alpha_{\mu,\text{sex}}(\text{sex}) + \alpha_{\mu,\text{ver}}(\text{ver}) + \beta_{\mu,1}(\text{age})^3 + \beta_{\mu,2}(\text{age})^3$$
$$\log(\text{age}) + \beta_{\mu,3}(\text{age})^3 \log(\text{age})^2 + \gamma_{\mu,\text{study}}$$
$$\log(\sigma) = \alpha_\sigma + \alpha_{\sigma,\text{sex}}(\text{sex}) + \beta_{\sigma,1}(\text{age})^{-2} + \beta_{\sigma,2}(\text{age})^{-2}\log(\text{age}) \tag{5}$$
$$+ \beta_{\sigma,3}(\text{age})^{-2} \log(\text{age})^2$$
$$v = \alpha_v,$$

$$\text{TCV} \sim \text{Generalized Gamma}(\mu, \sigma, v) \text{ with}$$
$$\log(\mu) = \alpha_\mu + \alpha_{\mu,\text{sex}}(\text{sex}) + \alpha_{\mu,\text{ver}}(\text{ver}) + \beta_{\mu,1}(\text{age})^{-2}$$
$$+ \beta_{\mu,2}(\text{age})^{-2} \log(\text{age}) + \beta_{\mu,3}(\text{age})^3 + \gamma_{\mu,\text{study}} \tag{6}$$
$$\log(\sigma) = \alpha_\sigma + \alpha_{\sigma,\text{sex}}(\text{sex}) + \beta_{\sigma,1}(\text{age})^{-2} + \beta_{\sigma,2}(\text{age})^{-2}$$
$$\log(\text{age}) + \beta_{\sigma,3}(\text{age})^{-2} \log(\text{age})^2 + \gamma_{\sigma,\text{study}}$$
$$v = \alpha_v$$

$$\text{SA} \sim \text{Generalised Gamma}(\mu, \sigma, v) \text{ with}$$
$$\log(\mu) = \alpha_\mu + \alpha_{\mu,\text{sex}}(\text{sex}) + \alpha_{\mu,\text{ver}}(\text{ver}) + \beta_{\mu,1}(\text{age})^{-2}$$
$$+ \beta_{\mu,2}(\text{age})^{-2} \log(\text{age}) + \beta_{\mu,3}(\text{age})^{-2} \log(\text{age})^2 + \gamma_{\mu,\text{study}} \tag{7}$$
$$\log(\sigma) = \alpha_\sigma + \alpha_{\sigma,\text{sex}}(\text{sex}) + \beta_{\sigma,1}(\text{age})^{-2} + \beta_{\sigma,2}(\text{age})^{-2} \log(\text{age})$$
$$+ \beta_{\sigma,3}(\text{age})^{-2} \log(\text{age})^2 + \gamma_{\sigma,\text{study}}$$
$$v = \alpha_v,$$

$$\text{CT} \sim \text{Generalized Gamma}(\mu, \sigma, v) \text{ with}$$
$$\log(\mu) = \alpha_\mu + \alpha_{\mu,\text{sex}}(\text{sex}) + \alpha_{\mu,\text{ver}}(\text{ver}) + \beta_{\mu,1}(\text{age})^{-2}$$
$$+ \beta_{\mu,2}(\text{age})^{-2} \log(\text{age}) + \gamma_{\mu,\text{study}} \tag{8}$$
$$\log(\sigma) = \alpha_\sigma + \alpha_{\sigma,\text{sex}}(\text{sex}) + \beta_{\sigma,1}(\text{age})^{-1} + \beta_{\sigma,2}(\text{age})^{0.5} + \gamma_{\sigma,\text{study}}$$
$$v = \alpha_v.$$

No smoothing terms were used in any GAMLSS models implemented in this study, although the fractional polynomials can be regarded as effectively a parametric form of smoothing. Reliably estimating higher order moments requires increasing amounts of data, hence none of our models specified any age-related fixed-effects or random effects in the $v$ term. However, $\alpha_v$ was found to be important in terms of model fit and hence we have used a generalized gamma distribution (Supplementary Information 1).

## Defining developmental milestones

GAMLSS modelling also allowed us to leverage the aggregated life-spanning neuroimaging dataset to derive developmental milestones (that is, peaks of trajectories) and compare them to existing literature. The cerebrum tissue classes from 100 studies (Fig. 1, Supplementary Tables 1.1–1.7, Supplementary Information 18) showed clear, predominantly age-related trends, even prior to any modelling. Comparing these models with multiple non-MRI metrics of brain size demonstrated high correspondence across the lifespan (Supplementary Information 3). Peaks were determined based on the GAMLSS model output (50th centile) for each of the tissue classes and TCV, for both total tissue volumes and rates of change or growth (velocity). A similar series of methodological steps was performed for the set of extended global and regional cortical morphometric phenotypes (Fig. 2, Supplementary Information 7, 8). To further contextualize the neuroimaging trajectories, diagnostic age ranges from previous literature[73,76] (blue boxes in Fig. 3) were compared with empirical age ranges of patients with a given diagnosis across the aggregated neuroimaging dataset (black boxes in Fig. 3). Note that age of diagnosis is significantly later than age of symptom onset for many disorders[73]. Developmental milestones were also compared to published work for brain resting metabolic rate[61], from its minimum in infancy to its maximum in early childhood; anthropometric variables (height and weight), which reach a first peak in velocity during infancy and a second peak in velocity in adolescence[52]; typical acquisition of the six gross motor capabilities[62]; and pubertal age ranges as defined based on previous reports[51,53].

## Centile scores and case–control differences

These normative trajectories of brain development and aging also enabled each individual scan to be quantified in terms of its relative distance from the median of the age-normed and sex-stratified distributions provided by the reference model[67,77] (Fig. 4, Supplementary Information 10, 11). Individual centile scores were estimated relative to the reference curves, in a way that is conceptually similar to traditional anthropometric growth charts (Supplementary Information 1). These centiles represent a novel set of population- and age-standardized clinical phenotypes, providing the capacity for cross-phenotype, cross-study and cross-disorder comparison. A single multivariate metric (CMD, Supplementary Information 1.6) was estimated by combining

centile scores on multiple MRI phenotypes for each individual (Fig. 4c). Case–control differences in centile scores were analysed with a bootstrapped (500 bootstraps) non-parametric generalization of Welch's one-way ANOVA. Pairwise, sex stratified, post-hoc comparisons were conducted using non-parametric Monte Carlo permutation tests (10,000 permutations) and thresholded at a Benjamini–Hochberg FDR of $q < 0.05$.

### Longitudinal stability

To use centile scores in a diagnostically meaningful or predictive way, they need to be stable across multiple measuring points. To assess this intra-individual stability, we calculated the subject-specific IQR of centiles across timepoints for the datasets that included longitudinal scans ($N = 9,306$, 41 unique studies). Exploratory longitudinal clinical analyses were restricted to clinical groups that had at least 50 subjects with longitudinal data to allow for robust group-wise estimates of longitudinal variability. In addition, there was a subset of individuals with documented clinical progression over the course of longitudinal scans, for instance from mild cognitive impairment to Alzheimer's disease, where we expected an associated change in centile scored brain structure. To test this hypothesis, we assessed whether these individuals showed longitudinal variation of centile scores (as assessed with IQR) with a direction of change consistent with their clinical progression. See Supplementary Information 14 for further details about the longitudinal stability of centile scores.

### Data sharing and out-of-sample estimation

We have provided an interactive tool (www.brainchart.io) and made our code and models openly available (https://github.com/brainchart/Lifespan). The tool allows the user to visualize the underlying demographics of the primary studies and to explore the normative brain charts in a much more detailed fashion than static images allow. It also provides the opportunity for interactive exploration of case–control differences in centile scores across many diagnostic categories that is beyond the scope of this paper. Perhaps most significantly, the brain chart interactive tool includes an out-of-sample estimator of model parameters for new MRI data that enables the user to compute centile scores for their own datasets without the computational or data-sharing hurdles involved in adding that data to the reference dataset used to estimate normative charts (Fig. 5). Bias and reliability of out-of-sample centile scoring was extensively assessed and endorsed by resampling and cross-validation studies for 'new' studies comprising at least 100 scans. Although already based on the largest and most comprehensive neuroimaging dataset to date, and supporting analyses of out-of-sample data, these normative brain charts will continue to be updated as additional data are made available for aggregation with the reference dataset. See Supplementary Information 1.8, 4 for further details about out-of-sample estimation.

### Reporting summary

Further information on research design is available in the Nature Research Reporting Summary linked to this paper.

### Data availability

Model parameters and out-of-sample centile scores are available at www.brainchart.io and on https://github.com/brainchart/Lifespan. Summary statistics are available in the Supplementary Tables (Supplementary Tables 1–8). Links to open datasets are also listed on https://github.com/brainchart/Lifespan. Availability of other MRI datasets aggregated here is through application procedures individually managed at the discretion of each primary study, with additional information provided in Supplementary Table 1.1 and Supplementary Information 19.

### Code availability

All code is available at https://github.com/brainchart/Lifespan.

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

**Acknowledgements** R.A.I.B. was supported by a British Academy Postdoctoral fellowship and by the Autism Research Trust. J. Seidlitz was supported by NIMH T32MH019112-29 and K08MH120564. S.R.W. was funded by UKRI Medical Research Council MC_UU_00002/2 and was supported by the NIHR Cambridge Biomedical Research Centre (BRC-1215-20014). E.T.B. was supported by an NIHR Senior Investigator award and the Wellcome Trust collaborative award for the Neuroscience in Psychiatry Network. A.F.A.-B. was supported by NIMH K08MH120564. Data were curated and analysed using a computational facility funded by an MRC research infrastructure award (MR/M009041/1) to the School of Clinical Medicine, University of Cambridge and supported by the mental health theme of the NIHR Cambridge Biomedical Research Centre. The views expressed are those of the authors and not necessarily those of the NIH, NHS, the NIHR or the Department of Health and Social Care. We acknowledge the invaluable contribution to this effort made by several openly shared MRI datasets: OpenNeuro (https://openneuro.org/), the Healthy Brain Network (https://healthybrainnetwork.org/), UK BioBank (https://www.ukbiobank.ac.uk/), ABCD (https://abcdstudy.org/), the Laboratory of NeuroImaging (https://loni.usc.edu/), data made available through the Open Science Framework (https://osf.io/), COINS (http://coins.mrn.org/dx), the Developing Human Connectome Project (http://www.developingconnectome.org/), the Human Connectome Project (http://www.humanconnectomeproject.org/), the OpenPain project (https://www.openpain.org), the International Neuroimaging Datasharing Initiative (INDI) (https://fcon_1000.projects.nitrc.org/), and the NIMH Data Archive (https://nda.nih.gov/). See Supplementary Information 21 for further notes on the usage of open MRI data and data sharing. Data used in this article were provided by the brain consortium for reliability, reproducibility and replicability (3R-BRAIN) (https://github.com/zuoxinian/3R-BRAIN). Data used in the preparation of this article was obtained from the Australian Imaging Biomarkers and Lifestyle flagship study of ageing (AIBL) funded by the Commonwealth Scientific and Industrial Research Organisation (CSIRO) which was made available at the ADNI database (https://adni.loni.usc.edu/aibl-australian-imaging-biomarkers-and-lifestyle-study-of-ageing-18-month-data-now-released/). The AIBL researchers contributed data but did not participate in analysis or writing of this report. AIBL researchers are listed at https://www.aibl.csiro.au. Data used in preparation of this article were obtained from the Alzheimer's Disease Neuroimaging Initiative (ADNI) database (https://adni.loni.usc.edu/). The investigators within the ADNI contributed to the design and implementation of ADNI and/or provided data but did not participate in analysis or writing of this report. A complete listing of ADNI investigators can be found at https://adni.loni.usc.edu/wp-content/uploads/how_to_apply/ADNI_Acknowledgement_List.pdf. More information on the ARWIBO consortium can be found at https://www.arwibo.it/. More information on CALM team members can be found at https://calm.mrc-cbu.cam.ac.uk/team/ and in the Supplementary Information. Further information about the Cam-CAN corporate authorship membership can be found at https://www.cam-can.org/index.php?content=corpauth#12. Data used in this article were obtained from the developmental component 'Growing Up in China' of the Chinese Color Nest Project (http://deepneuro.bnu.edu.cn/?p=163). Data were downloaded from the COllaborative Informatics and Neuroimaging Suite Data Exchange tool (COINS) (https://coins.trendscenter.org/) and data collection was performed at the Mind Research Network. Data used in the preparation of this article were obtained from the IConsortium on Vulnerability to Externalizing Disorders and Addictions (c-VEDA), India (https://cveda-project.org/). Details of The ENIGMA Developmental Brain Age working group can be found at https://github.com/ENIGMA-Developmental-BrainAge/main. Data used in the preparation of this article were obtained from the Harvard Aging Brain Study (HABS P01AG036694) (https://habs.mgh.harvard.edu). Data used in the preparation of this article were obtained from the IMAGEN consortium (https://imagen-europe.com/). Data used in this article were obtained from the Korean Longitudinal Study on Cognitive Aging and Dementia (KLOSCAD) (https://recode.re.kr). A full list of NSPN consortium members can be found at https://www.nspn.org.uk/nspn-team/. The POND network (https://pond-network.ca/) is a Canadian translational network in neurodevelopmental disorders, primarily funded by the Ontario Brain Institute.

**Author contributions** R.A.I.B., J.S., S.R.W., E.T.B. and A.F.A.-B. designed the study, conducted analyses, wrote and edited the manuscript. J.V. and K.M.A. helped to design the study and contributed to data analysis. All other authors made substantial contributions to the conception or design of the work, the acquisition, analysis or interpretation of data, the creation of new software used in the work, or drafted or substantively revised the Article.

**Competing interests** E.T.B. serves on the scientific advisory board of Sosei Heptares and as a consultant for GlaxoSmithKline, Boehringer Ingelheim and Monument Therapeutics. G.S.A. has served on advisory boards of Eisai and Janssen and in speakers bureaus of Allergan, Takeda and Lundbeck. K.M.A. is an employee of Neumora Therapeutics. P.B.J. has consulted for MSD. L. Palaniyappan reports personal fees from Janssen Canada for participating in an Advisory Board (2019) and Continuous Professional Development events (2017–2020), Otsuka Canada for Continuous Professional Development events (2017–2020), SPMM Course Limited,

UK for preparing educational materials for psychiatrists and trainees (2010 onwards), Canadian Psychiatric Association for Continuous Professional Development events (2018–2019); book royalties from Oxford University Press (2009 onwards); institution-paid investigator-initiated educational grants with no personal remunerations from Janssen Canada, Sunovion and Otsuka Canada (2016–2019); travel support to attend a study investigator's meeting organized by Boehringer-Ingelheim (2017); travel support from Magstim Limited (UK) to speak at an academic meeting (2014); none of these activities are related to this work. T.R. has received honoraria from Oxford Biomedica. A.P.S. has consulted for Janssen, Biogen, Qynapse, and NervGen. R.T.S. has received consulting income from Octave Bioscience and compensation for scientific review duties from the American Medical Association, the US Department of Defense, the Emerson Collective, and the National Institutes of Health. R.A.S. has consulted for Janssen, AC Immune, NervGen and Genentech. D.J.S. has received research grants and/or consultancy honoraria from Discovery Vitality, Johnson & Johnson, Lundbeck, Sanofi, Servier, Takeda and Vistagen. J. Suckling has consulted for GW Pharmaceuticals, Claritas HealthTech, Fundacion La Caixa and Fondazione Cariplo. All other authors declare no competing interests.

**Additional information**
**Correspondence and requests for materials** should be addressed to R. A. I. Bethlehem or J. Seidlitz.

# Reporting Summary

## Statistics

For all statistical analyses, confirm that the following items are present in the figure legend, table legend, main text, or Methods section.

| n/a | Confirmed | |
|---|---|---|
| ☐ | ☒ | The exact sample size (*n*) for each experimental group/condition, given as a discrete number and unit of measurement |
| ☐ | ☒ | A statement on whether measurements were taken from distinct samples or whether the same sample was measured repeatedly |
| ☐ | ☒ | The statistical test(s) used AND whether they are one- or two-sided *Only common tests should be described solely by name; describe more complex techniques in the Methods section.* |
| ☐ | ☒ | A description of all covariates tested |
| ☐ | ☒ | A description of any assumptions or corrections, such as tests of normality and adjustment for multiple comparisons |
| ☐ | ☒ | A full description of the statistical parameters including central tendency (e.g. means) or other basic estimates (e.g. regression coefficient) AND variation (e.g. standard deviation) or associated estimates of uncertainty (e.g. confidence intervals) |
| ☐ | ☒ | For null hypothesis testing, the test statistic (e.g. *F*, *t*, *r*) with confidence intervals, effect sizes, degrees of freedom and *P* value noted *Give P values as exact values whenever suitable.* |
| ☐ | ☒ | For Bayesian analysis, information on the choice of priors and Markov chain Monte Carlo settings |
| ☐ | ☒ | For hierarchical and complex designs, identification of the appropriate level for tests and full reporting of outcomes |
| ☐ | ☒ | Estimates of effect sizes (e.g. Cohen's *d*, Pearson's *r*), indicating how they were calculated |

*Our web collection on statistics for biologists contains articles on many of the points above.*

## Software and code

Policy information about availability of computer code

| Data collection | No software was used in data collection |
|---|---|
| Data analysis | Data was analysed using a combination of open source R code (v4.1.2) and custom R code made available on https://github.com/ucam-department-of-psychiatry/Lifespan. With respect to all visualisation and statistics represented in graphical format, unless otherwise stated these were generated in R GNU v4.1.2 using the "ggplot" package. Where boxplots are used they indicate the median and lower and upper hinges correspond to the first and third quartiles (the 25th and 75th percentiles). The upper whisker extends from the hinge to the largest value no further than 1.5 * IQR from the hinge (where IQR is the inter-quartile range, or distance between the first and third quartiles). The lower whisker extends from the hinge to the smallest value at most 1.5 * IQR of the hinge. Data beyond the end of the whiskers are called "outlying" points and are plotted individually. Density plots were generated with the 'geom_flat_violin' option from the "raincloudplots" package. Estimation of densities and the resulting number of peaks were done using the default settings of the 'density()' function in the base R "stats" package using a Gaussian smoothing kernel which defaults to 0.9 times the minimum of the standard deviation and the interquartile range divided by 1.34 times the sample size to the negative one-fifth power (Silverman's 'rule of thumb'); unless the quartiles coincide, when a positive result will be guaranteed. Clustering heatmaps were generated using the "ComplexHeatmap" package. Crosshair plots depict the median and standard deviations. Plots depicting linear associations were generated with ggplot's 'geom_point()' function and where linear relations are reported include shaded regions indicating the 95% confidence intervals of that linear relation. Linear regression was performed using the "lm" function in the base "stats" package, as well as the "lmerTest" package for mixed-effects modelling. Student's T-tests were performed using the "t.test" function in the base "stats" package (two-sided, unless otherwise reported). The "ggstatsplot" package was used for the model generalisability analyses to report robust correlation values. Cohen's d effect sizes were calculated using the "effsize" package. A description of the FreeSurfer version and processing pipeline can be found in SI18 (mainly FreeSurfer 6.0.1 unless stated otherwise). |

For manuscripts utilizing custom algorithms or software that are central to the research but not yet described in published literature, software must be made available to editors and reviewers. We strongly encourage code deposition in a community repository (e.g. GitHub). See the Nature Portfolio guidelines for submitting code & software for further information.

## Data

Policy information about [availability of data](availability of data)

All manuscripts must include a [data availability statement](data availability statement). This statement should provide the following information, where applicable:

- Accession codes, unique identifiers, or web links for publicly available datasets
- A description of any restrictions on data availability
- For clinical datasets or third party data, please ensure that the statement adheres to our [policy](policy)

Model parameters and out-of-sample centile scores are available at www.brainchart.io and on https://github.com/brainchart/Lifespan. Summary statistics are available in the Supplementary Tables (ST1-8). Links to open and semi-open datasets are also listed on https://github.com/brainchart/Lifespan. Availability of other MRI datasets aggregated here is through application procedures individually managed at the discretion of each primary study, with additional information provided in ST1.1 and SI19.

# Field-specific reporting

Please select the one below that is the best fit for your research. If you are not sure, read the appropriate sections before making your selection.

☒ Life sciences    ☐ Behavioural & social sciences    ☐ Ecological, evolutionary & environmental sciences

For a reference copy of the document with all sections, see [nature.com/documents/nr-reporting-summary-flat.pdf](nature.com/documents/nr-reporting-summary-flat.pdf)

# Life sciences study design

All studies must disclose on these points even when the disclosure is negative.

| | |
|---|---|
| Sample size | No a-priori sample size was calculated, but we used the largest sample of neuroimaging data reported to date and conduct multiple sensitivity analyses in addition to the built ML optimisation of our models to ensure data was robust. |
| Data exclusions | Exclusion criteria for each dataset at input stage was determined by collecting sites and studies and are listed in the supplementary materials (SI19) where each dataset is described and where relevant. Missing demographic data or failure in image processing (either due to technical problems with the data or other artefacts) was a secondary reason for exclusion. |
| Replication | Reproducibility of findings was ensured by extensive sensitivity and bootstrapping analysis, simulation of model parameters, evaluation of optimal model parameters, validation using iterative leave-one-out analysis, and validation against known growth charts derived from other modalities. |
| Randomization | For our bootstrapping we used random sampling maintaining dataset ratios as described in the supplementary methods. For pairwise comparisons between control and clinical cohorts we used permutation tests that randomly reshuffle case and control labels to generate 10,000 null distributions. |
| Blinding | Blinding was not possible, but also not applicable for establishing growth trajectories, furthermore all analyses were conducted in a data driven manner |

# Reporting for specific materials, systems and methods

We require information from authors about some types of materials, experimental systems and methods used in many studies. Here, indicate whether each material, system or method listed is relevant to your study. If you are not sure if a list item applies to your research, read the appropriate section before selecting a response.

### Materials & experimental systems

| n/a | Involved in the study |
|---|---|
| ☒ | ☐ Antibodies |
| ☒ | ☐ Eukaryotic cell lines |
| ☒ | ☐ Palaeontology and archaeology |
| ☒ | ☐ Animals and other organisms |
| ☐ | ☒ Human research participants |
| ☒ | ☐ Clinical data |
| ☒ | ☐ Dual use research of concern |

### Methods

| n/a | Involved in the study |
|---|---|
| ☒ | ☐ ChIP-seq |
| ☒ | ☐ Flow cytometry |
| ☐ | ☒ MRI-based neuroimaging |

# Human research participants

Policy information about studies involving human research participants

| | |
|---|---|
| Population characteristics | Population characteristics are listed in supplementary tables 1.1-1.48. For the analysis age and sex were included in our models. Diagnosis was provided for each dataset individually and procedures for obtaining these were described in the description of each individual dataset. |
| Recruitment | All analyses in the present manuscript were based on existing data. Recruitment for each existing dataset is described in the supplementary description for each dataset see SI19. |
| Ethics oversight | The project received IRB exemption from CHOP and ethical approval from the Psychology Research Ethics Committee at the University of Cambridge. All contributing datasets already contained their own respective ethical oversight and therefore both committees concluded no additional ethical approval was required. The following statement has been added to the methods section: <br> The research was reviewed by the Cambridge Psychology Research Ethics Committee (PRE.2020.104) and The Children's Hospital of Philadelphia's Institutional Review Board (IRB 20-017874) and deemed not to require PRE or IRB oversight as it consists of secondary analysis of de-identified primary datasets. Informed consent of participants (or their guardians) in primary studies is referenced in supplementary information [SI] 19 and supplementary table [ST] 1. |

Note that full information on the approval of the study protocol must also be provided in the manuscript.

# Magnetic resonance imaging

## Experimental design

| | |
|---|---|
| Design type | Structural MRI |
| Design specifications | No specific experimental setup was used |
| Behavioral performance measures | No behavioural measures are included |

## Acquisition

| | |
|---|---|
| Imaging type(s) | Structural, mainly T1 and/or T2 weighted imaging, variations of each dataset are listed in detail in supplementary table 1.1 |
| Field strength | Varying (description in each dataset description and supplementary table 1.1 under the column "Field Strength") |
| Sequence & imaging parameters | Varying (description in each dataset description and supplementary table 1.1) |
| Area of acquisition | Whole brain |
| Diffusion MRI | ☐ Used    ☒ Not used |

## Preprocessing

| | |
|---|---|
| Preprocessing software | Varying (description in each dataset description) but mainly based on Freesurfer recon-all |
| Normalization | Varying (description in each dataset description) but mainly based on Freesurfer recon-all |
| Normalization template | Varying (description in each dataset description) but mainly based on Freesurfer recon-all (e.g. fsaverage) |
| Noise and artifact removal | Varying (description in each dataset description) but mainly based on Freesurfer recon-all |
| Volume censoring | None |

## Statistical modeling & inference

| | |
|---|---|
| Model type and settings | We used generalised additive models for location scale and shape (GAMLSS) to estimate cross-sectional normative age-related trends. |
| Effect(s) tested | We modelled growth trajectories and generated individual centile scores from these growth charts |
| Specify type of analysis: | ☒ Whole brain    ☐ ROI-based    ☐ Both |
| Statistic type for inference <br> (See Eklund et al. 2016) | Not applicable |
| Correction | For any pairwise comparisons we used Monte-Carlo permutation tests and report all Benjamini-Hochberg FDR corrected values in addition to Cohens d effect sizes |

## Models & analysis

| n/a | Involved in the study |
|---|---|
| ☒ | ☐ Functional and/or effective connectivity |
| ☒ | ☐ Graph analysis |
| ☐ | ☒ Multivariate modeling or predictive analysis |

Multivariate modeling and predictive analysis | We used generalised additive models for location scale and shape (GAMLSS) to estimate cross-sectional normative age-related trends. Including study, sex and processing pipeline as random effects in higher order polynomial models.

