## [Peer Review File · Nature]

Manuscript Title: Brain charts for the human lifespan

Reviewer Comments & Author Rebuttals

Reviewer Reports on the Initial Version:

Referees' comments:

Referee #1 (Remarks to the Author):

Bethlehem, Seidlitz and White provide a lifespan analysis of a 120k+ MRI datasets, for which they should be commended. Clearly this work took a huge amount of effort to bring the many datasets included together, in service of the field's mission to achieve growth charts for the brain. The authors leveraged GAMLSS to generate centiles for basic brain volume measures, as well as to address batch effects. In many ways, the work attempts to fulfill the vision (e.g., pointing to associations with various developmental milestones, noting diagnostic differences), has several impressive demonstrations of validity built in, and provides a Shiny app to facilitate others in extending their work.

A challenge in reviewing the present work is that while very impressive, there are a few key concerns that dramatically reduce enthusiasm for publication, at least in its current form (and admittedly, may take a lot of work to address):

1. The authors consciously chose to disregard meaningful quality assessment and consideration of data quality. As a work that is intended to be a model for the field, and to yield a resource upon which others can build – this is a major red flag. While data quality may not impact the median values for trajectories, it is hard to believe it will not impact the variance. One could argue that the rudimentary anatomical measures employed may be more robust to motion and other artifacts – that could be possible but would need to be proven (consider using the dataset employed by Ai et al., Franco, 2021 to do this if desired). The authors could have used Euler number, or MRIQC measures; or BRAINDR for visual inspection – the option to do nothing of the sort – is concerning. Do we really want to see this practice adopted in the field? Would the AD datasets found to be extreme outliers be difficult to flag by eye? Cleaning up the data could improve our ability to detect psychiatric disorders; and can avoid associations between data quality and psychiatric diagnosis, which could be confounding and are well known in the structural and functional MRI literatures.
2. The authors attempt to account for batch effects using GAMLSS, which is reasonable. Still, it might be good show that an alternative approach (e.g., ComBat) could give converging results, or let one look explore batch effects more directly. Though, bigger picture – these samples are inherently quite diverse in their recruitment and resultant composition. GAMLSS is not a cure-all, and batch effect corrections can under/over correct. Any correction would be inherently limited with respect representativeness in several ways, which is ok, though merits more attention as an issue.
4. It is unclear how uniform the preprocessing for data was, and what specific steps were carried – which is concerning, unless I missed it. Did the authors bring the data to a common space for

calculation of measures? Did they need to for the measures obtained? (if alignment could be avoided, that could minimize introduction of potential biases related to registration error - which is heavily impacted by data quality). There is a lot of detail that is crucial to know to fairly evaluate the work, and it is not easily obtainable.

Figure 1 should show # of datasets per site – the current depictions are not sufficient, and can be viewed as misleading (one can glean from distributions, but that is not trivial to meaningfully/accurately do).

4. A smaller note - how are the authors accounting for variations in diagnostic practices?

A final note is authorship – what was the criteria for data contributors, if they did not actively participate in the actual data analysis / manuscript? If such situations exist, is there an inadvertent benefit to those who are closed with their datasets? At a minimum, please consider grouping all those that only contributed data under a single collaborative name on the author line...then link that to the individuals in pubmed. This would better recognize the contributions of those who did the bulk of this analysis.

Referee #2 (Remarks to the Author):

Using data combined across several primary studies, this article presents cross-sectional lifespan age trends for individual differences in a small number of gross brain morphology “tissue classes” (e.g. total brain volume, white matter volume, subcortical grey matter volume, and ventricular volume) as measured by structural MRI. Rather than simply providing the mean age trends in each tissue class, the authors provide information about how the dispersion changes with age. Additionally, because mean growth is nonlinear, the authors are able to calculate ages at peak velocity for each class. This descriptive exercise is pitched as “brain charts” to “reference standards against which to anchor measures of individual differences in brain morphology,” much in the same way that “growth charts” are used. They provide an online tool for calculating individual however, there are several unsatisfying, if not concerning, aspects to this packaging.

First, is that there is no way to be confident in the use of these reference norms given that scanner to scanner variation can lead to tremendous mean differences in inferred volumes. Without first calibrating an MRI relative to the norms, nearly everyone scanned in a particular MRI might be classified as abnormal. Even if the norms were only used to make comparisons across samples rather than individuals, it would not be clear if the samples actually differed in mean volumes or scanners were simply miscalibrated.

Second, the contributing samples are not necessarily population representative. This makes calculation of centile scores exceedingly hard to interpret.

Third, norms may become outdated over time. In cognitive testing this is known as the Flynn Effect, and it forces test companies to renorm their tests ever ~10 years. It is unclear how up to date these norms are, or whether they will remain up to date. This isn't an argument against the norms, but it does underscore the need for front work on this topic.

Fourth, is that the tissue classes are extremely gross. Normally, MRI research capitalizes on the fine grain spatial nature of the imaging, e.g. in the form of Region of Interest (ROI) analyses. This is important because abnormal growth or atrophy in different regions can have tremendously different implications for whether, and what, functions are clinically affected. Clinicians regularly make these appraisals in individual evaluations. The current submission would seem to add greater quantification and precision (in contrast to clinical judgement) to such work, but allowing for the calculation of centile scores. But in order to calculate these scores, anatomical specificity is nearly entirely sacrificed. For example, it would be impossible to distinguish various forms of neurocognitive disorders of aging (e.g. frontotemporal dementia vs. Alzheimer's disease) without the spatial information that is being thrown out.

Fifth, calculating centile scores may not be of much use without a clear understanding of the functional implications of different scores. Having the location on the distribution is not enough without understanding the clinical and functional correlates of those locations. As per the above point, this may be a difficult endeavor at the low level of spatial resolution provided.

Sixth, the cross-sectional nature of the data and potential differences in protocols across individuals or cohorts of different ages, prevents strong inferences regarding development and aging. Are there age differences in motion that may bias estimates? Are there period, or cohort effects in the data that would suggest that these trends plotted are not indices of "velocity" of growth or shrinkage within person, but instead differences associated with year of birth or historical time of imaging assessment?

Finally, it would be very interesting to conduct a correlational analysis of the different "tissue classes." Are they highly correlated? An excellent model for how such an analysis might be conducted is <https://doi.org/10.1038/ncomms13629> (see especially Figs. 7-8).

Referee #3 (Remarks to the Author):

The manuscript "Brain charts for the human lifespan" by Bethlehem et al. is an impressive and ambitious effort on several fronts. They aggregate a massive amount of structural neuroimaging scans (>122k scans, across >100k individuals) across 96 individual studies and then use sophisticated modeling (GAMLSS) to derive lifespan curves of 4 morphometric phenotypes (gray matter, white matter, subcortical gray matter, and ventricular volume) from 115 days post-conception to 100 years old. That endeavor alone is worthy of publication in a high-profile journal. The resulting age curves largely agree with existing literature, and thus for the most part aren't particularly novel in themselves, although they note some deviations from prior reports. Using centile scoring as a means of normalizing across studies, they then investigate the impact of various clinical diagnoses. While interesting, and certainly suggesting venues for further study, that aspect of the paper is entirely focused on group comparisons, and relies heavily on p-values, which will be driven to significance by the large sample sizes. Reporting of the effect sizes involved would be helpful and would provide at least an indirect sense of the potential of centiled brain charts within the context of "personalised" or "precision" medicine. But the truly groundbreaking potential of aggregating such a large data set, combined with the flexible modeling approach, is the proposed ability to estimate centile scores

meaningfully and reliably in “out-of-sample” data, in the context of *non-harmonized* MR data that would typically be subject to a host of interpretational challenges (e.g., different pulse sequences, protocols, scanner strength and vendor, etc). Indeed, the manuscript is the planned reference for an interactive online resource (www.brainchart.io) that would allow researchers to extract centile scores for new datasets. In this regard, I feel that the manuscript, as currently constituted, falls short of convincingly demonstrating that the challenge of out-of-sample estimation with highly variable MR data has been solved. If this concern can be addressed, I feel that the approach and online resource proposed in this manuscript has intriguing potential and would warrant the visibility that publication in Nature would provide.

1. Currently, the out-of-sample validation with “real-world” is limited to just 4 datasets. The results of that particular analysis are impressive (Fig. 4B and S1.7.2). But the use of only 4 datasets means that the out-of-sample validation is an impoverished sampling of the universe of possible MR studies, given the wide variety of ways in which MR studies can differ. Given that the authors have already amassed a much broader sampling of that universe (96 studies contributing to the model), it isn't clear to me why they didn't assess the generalizability of the GAMLSS + centiles modeling approach across that full study universe by conducting the same out-of-sample analysis using a “leave-one-study-out” (LOSO) modeling approach applied to every available study (i.e., treat every available study as if it is a part of the out-of-sample validation analysis). Indeed, a LOSO analysis is mentioned in Supplement (SI) Section 2.2.1, but that's only in the context of showing the variability in the resulting overall lifespan trajectories. It seems to me that what's needed additionally is a way to assess the likelihood that the estimated centiles for a given study may not be well fit, and correspondingly the study parameters potentially influencing that poor fit (per item (2) below), since that's what an individual investigator interested in applying the model to one particular new study needs to be able to evaluate.

2. Relatedly, the manuscript mentions that “biological” and “technical” covariates are included as part of the fixed effects modeling, but the actual covariates used in the modeling do not appear to be listed anywhere. Also, no analysis is provided (even in the SI) of the estimated effect of these covariates, which seems important for understanding the inner workings of the estimation. Last, understanding the space spanned by the covariates seems important for making assessments about the generalizability of the model to new out-of-sample data. In that regard, I think an SI table that lists the covariate values for each study, as well as other possibly relevant technical scanning details (e.g., imaging parameters) seems like a valuable and important addition.

3. Multiple studies have demonstrated that thickness and surface area are more pertinent measures than cortical GMV, given that cortical GMV is determined entirely by thickness and area, but thickness and area are themselves under independent genetic control. Given that, what is the rationale for not including mean cortical thickness and total surface area as part of the phenotypes investigated, both of which are readily available since all the data was processed through FreeSurfer, and would just need to be modeled?

4. Notwithstanding the reference to “personalized” or “precision” medicine’ in the 2nd sentence of the introduction, most of the results are organized around group differences. There is clearly much value in the ability to make group comparisons in an appropriately normalized fashion across MR

studies collected with disparate imaging protocols. However, it feels like some discussion is warranted, in the main text, of whether the results in the manuscript provide any direct support for the notion (or aspiration) that the centralized brain chart outputs have value for individualized prediction.

5. The group comparisons throughout the manuscript are primarily structured in the language of statistical significance (p -values), with minimal presentation of effect sizes. Effect sizes should be provided whenever a viable effect size measure exists, so as to provide information on the magnitude of the effect independent of the sample size.

6. The quality and added value of the Supplemental material is uneven. Some of it adds considerable value, but some of it is also of marginal quality or seems unnecessary. A number of the supplemental figures have poorly labelled axes or titles. In general, the captions of the SI figures need to be expanded to provide more clarity/details on what is being shown, and the captions or associated text can do a better job of explaining the purpose of the analysis and conclusion to be drawn within the SI. An appreciable number of specific comments/examples related to this point are provided below.

7. The font sizes in some of the figures tend toward being too small.

Other more minor or specific comments and suggestions follow:

8. It would be helpful if the SI included some discussion of model convergence and how that is assessed within GAMLSS – p. 9 of the SI mentions model instability and lack of convergence but no details are provided on how that was assessed.

9. Order of presented phenotypes should be the same across all figures, both in the main text and SI. E.g., Figure 1 is ordered as GMV, WMV, sGMV, Ventricles, so that order should be maintained across all figures, both in the main text (e.g., Figure 4B differs) and Supplement (which uses a variety of orders – e.g., Fig S1.2, S2.2.3).

10. Even though the code is available in GitHub, it would be helpful to include brief code snippets of the GAMLSS modeling in R within the SI as a mechanism to concisely, but technically, explain some of the modeling. This will help knowledgeable individuals quickly see exactly what was done, without needing to slog through a (potentially complicated) code base.

11. Abstract mentions 122123 scans from 100071 individuals. Figure 1 caption says 120685 scans. Summing the N column in SI Table 1.1 yields 121163. Summing the 'total.cn' column (cross-sectional N?) in that same table yields 92081. Why the differences? And if Figure 1 is based on the cross-sectional data only, shouldn't the number of scans be closer to the total number of individuals rather than greater than 120k?

12. The convention of denoting panels as "A|" rather than "A." or "A)" seems odd to me, and leads to situations (in the Supplement) where it visually appears to be "AI" (A-eye, rather than A-bar).

13. It seems like Figure 1B should use some sort of density plot, rather than simply plotting symbols

on top of each other. If necessary, the attempt to color code individual studies can be dropped (in that figure and other SI figures) as it's impossible to map a color to a given study anyway.

14. Figure 1C: The 95% centile boundaries are supposed to be “dotted” but mostly appear to be solid lines.

15. Figure 2: In the lower half of the figure, the “top grey section” isn't very clear. Why are the “empirical age-range (dark grey)” ranges so disparate from the “diagnostic age ranges (black outlines)”, and more broadly, what is the point being made by that distinction? Also, it's inherently confusing to have a ‘key’ with the text “From literature” with a black outline but the same interior color of gray that represents the overall age range of the current study (which itself seems rather unnecessary to include, as the overall age range simply spans the same range as the top portion of the figure).

16. Fig 3: Panel B: Doesn't really provide much intuition as to how the CMD is calculated, or what it represents. Panel D: Why are error bars only present for some of the data points? Are they too small to be seen? If so, that should be stated. Also, are the bars STD's or SEM's? A similar comment applies to the error bars elsewhere (e.g., Fig S9.2).

17. Are all the probability density plots throughout the manuscript computed in a similar manner (e.g., same kernel approach or degree of smoothing) using the same analytical tool? A description of the specifics of the density plot construction in the SI methods seems warranted given the prominence of density plots throughout the manuscript.

18. I would suggest breaking the section header “Longitudinal centile changes and novel data” in the main text into two distinct section sub-headings, as the two cover completely different topics.

19. Main text, p. 8 (“Longitudinal centile changes and novel data” section): It is unclear if the quantification in the first paragraph (“all median <5%” and “~5% median difference”) reflects a percent difference (and if so, relative to what), or a percentage *point* difference (i.e., a 0.05 difference in centile values).

20. Fig 4C – too little detail to quickly grasp what was done. What is a cloned NSPN?

21. Minor grammatical/syntactical errors in Supplement. (e.g., “each terms of the generalized gamma distribution”; lack of space between symbols/equations and following text). Other little errors in various places. The SI needs a careful proof-read before re-submission by a someone with an eye for these issues.

22. Inconsistent formatting of “i.e.” and “e.g.” – both should always be followed by a comma.

23. Fig S1.1: Is “relative AIC” based on a *ratio* or *difference* to the lowest AIC value? If a ratio (which is what “relative” inherently implies) then seems odd that there is such a pronounced difference with the reference model. E.g., For GMV, all models except for “Generalized Beta type 2” had an AIC that was ~ 3000 times greater (or more) than the “Generalized Gamma”.

24. Fig S1.2: The y-axis labels are completely cryptic. Some 'key' is needed in the caption for understanding the naming convention. Also, caption should mention that the model being investigated is the generalized gamma. Additionally, the order of the phenotypes in Fig. S1.2 should match those in S1.1, and the figure titles should be simplified. Last, why the switch to BIC as a criterion vs. AIC?
25. SI Section 1.3: The brief description of the "Model simulations" is generic and inadequate to understand what exactly was simulated.
26. Fig S1.3.[1-2]: Not clear what is being shown, or the point of these figures. Cryptic titles. Tiny font sizes.
27. SI Section 1.4: What is the value/purpose of computing centile normalized z-scores rather than simply using the centile estimate itself? Per the text on p. 13, the latter accounts for study random-effects, while the normalized z-scores do not. Isn't it a good thing to account for the study random-effects and thus wouldn't the centile scores be preferable to the normalized z-scores?
28. Fig S1.4: What is the point of this figure? What do the x and y-axes represent? Figure doesn't appear to be referenced anywhere in the text. Also, another example where is it impossible to map the colors to a specific study.
29. Main text refers to "SI1.6" for description of the CMD, but that definition is actually in SI1.5. Related to this, the main text says the CMD was somehow computed relative to the "CN median", but SI1.5 says that the (usual) mean was used.
30. SI Section 1.6: I would argue its debatable whether interquartile range is "well defined for two [or more] observations", which forms the vast majority of the longitudinal samples in the study. (A number of on-line calculators require at least 3 values, and even with 3 values the notion of "IQR" seems sketchy. The appropriateness of IQR with such a small number of values seem to merit additional justification.
31. Fig S1.6.[1-2]: These results will be highly dependent on the specifics of the simulation, but per above, sufficient details on the simulation aren't provided to interpret these figures properly. Another example of cryptic axis labels and captions that are insufficient to understand the purpose of the figure.
32. SI Section 1.7: Presumably, the 'F' in the equations on p. 19 represents the fixed effects (and not the 'F' of the CDF defined on p. 12)? Also, it's not defined what it means to make a "clone" of a study in the simulation.
33. Fig S2.1.1: Some context for how to interpret a "detrended transformed Owen's plot" would be helpful (more obscure than Q-Q plots).
34. SI Section 2.2.2. Were the stratified bootstrap samples generated by sampling relative to the

proportion of the strata in the original data? If so, wouldn't the results be primarily sampling the variability of the UK-Biobank and ABCD data since the bootstraps would always be dominated by data selected from ABCD and UK-Biobank?

35. Fig S2.2.3: Why are the studies ordered in reverse alphabetical order, rather than alphabetical order, which would be more intuitive?

36. SI Section 2.3: The following statement is a bit imprecise: "whereas in the reference prediction curves the freesurfer contribution is equivalent to the grand-mean across all versions (across all studies), meaning the reference prediction curves do not represent any specific freesurfer version". Namely, the grand-mean would be weighted by the proportion of given FS versions, and thus depending on those proportions, might be close to a specific FS version. Indeed, SI Table 1.3 shows that the vast majority of cases were processed with FS 6.0 (either T1 only, or T1+T2), so the reference prediction curve would be strongly weighted to FS 6.0 (to the extent that FS version has a meaningful impact – see Item (2)).

37. Fig S2.4: In panel (B), why is "Model derived TBV" shown on the x-axis, whereas in all the other panels, the "model derived" value is shown on the y-axis?

38. Fig S4.1.1: The precision on the x-axis for the "Late midfetal" and "Late fetal" windows is insufficient to ascertain the actual time window being plotted (i.e., evidenced by the fact that multiple ticks display the same x-axis value).

39. Fig S4.1.2: Why, in a number of the panels, is the solid line seemingly outside of the dashed lines representing the 95% CI?

40. Fig S5: I would suggest using 'Centile' rather than 'Quantile' for the y-axis label, consistent with the terminology in the caption, and the use of 'centile' throughout the manuscript. (Not only does the caption not use the term 'quantile', but 'quantile' isn't used a single time throughout the main text or SI text). Also, averaging across phenotypes is not strictly the same as computing a centile score for the single summary TCV measure – thus I would suggest avoiding the imprecise claim that it is "akin to computing a centile score for TCV". (If you want the true centile score for TCV, compute them directly). Similarly, labels in Fig S9.4 use the term quantile rather than centile.

41. In the main text, SI7 and SI8 are referenced before first mention of SI4-6. It would be preferable if the SI material is numbered and ordered such that it can be introduced sequentially within the flow of the main text.

42. SI Section 7.1: What symptomology variables and criteria were used for assigning the ABCD and UK-Biobank data into clinical ("non-CN") cohorts?

43. Fig S7.1 and S9.3.1: Preferably, the colors used would be matched to those in Figure 3 for consistency in presentation.

44. Fig S7.2.[1-3]: Very complex, with minimal guidance as to what is being shown.

45. Fig S7.4, panel B: Would be more intuitive if larger absolute mean differences were shown in the “hot” color (yellow), rather than the “cool” color (blue).

46. Fig S7.5.[1-3]: Not clear what point is being made with the inclusion of these figures.

47. Fig S8.[1-2]: Needs a more detailed caption explaining what is being shown. e.g., What do “years” and “total” represent? Also, why is no clustering shown for S8.1 – was k=1 the optimal clustering?

48. Fig S9.1: Second column is labelled “Age range (log-transformed)”. Age-range of what? What are the units (day, weeks, years)?

49. SI Section 9.3 (p. 49): Text refers to “Fig. S9.1.3.3”. Should be “Fig. S9.3.3”.

50. Given that Infant FreeSurfer is the sole means by which data was obtained for individual less than 2 years old, some additional discussion of the validity of Infant FreeSurfer, and the confidence in the those values, seems warranted. The authors already comment that the values it generated for subcortical GMV didn’t seem continuous with those generated by FreeSurfer for 2 years and older. Was there any evidence (milder) of similar concerns for the other 3 phenotypes?

51. It would be helpful if the SI Methods detailed exactly which FS measures were used for the definition of each of the 4 studied phenotypes.

52. It’s “FreeSurfer”. Not “Freesurfer” or “freesurfer”. The inconsistent formatting is careless and could be seen as disrespectful to its creators/developers/maintainers.

53. Overall, the dataset descriptions in SI are quite inconsistent in what is covered. e.g., Not all of them even mention the number of individuals. Any information that can be conveniently provided in tabular format should be removed and placed in an SI spreadsheet (e.g., # individuals, scanner platforms, T2 availability, imaging parameters, FS processing version). This spreadsheet should also make clear whether the FS processing was done de novo for the current study, or whether the current study used FS data already generated/provided by the study itself. The text descriptions of the studies can then be limited to a brief overview of the study, as well as any particularly salient points that cannot conveniently be captured in the proposed spreadsheet.

54. The “OpenPain” study has a very long description in the SI relative to the other studies. Is there a particular reason that it merits such considerable detail relative to the other studies? (It reads like an unedited cut-and-paste from other documentation).

55. Issues related to the SI Tables:

a. The meaning of some of the variables in the SI tables is not clear. (e.g., ‘total.cn’, ‘percentage.cn’ in SI Table 1; ‘V1’, ‘V2’ in SI Table 5). A key/dictionary of some sort is necessary, for at least some of the variables.

b. Table1.5: First row has no label for ‘dx’.

- c. Table1.6: Identical to Table1.5
- d. Inconsistent naming schema of the individual tabs within a table (e.g., “Table1.1” vs. “2_1”). Also, given the Excel character limit on tab name length, make sure that the name of each tab is clearly interpretable.
- e. Please remove the annoying Excel “warning” (and associated green triangle) about “Number stored as text” by converting the cells from text to number.
- f. SI Section 10 (“Sex differences”, p. 52) mentions a “SI table 2.9”, which doesn’t appear to exist.
- g. Are SI Tables 4 and 7 mentioned anywhere in the main text or SI text? If so, I couldn’t find the references to them.
- h. All the tabs in SI Table 6 are identical.

Author Rebuttals to Initial Comments:

Bethlehem, Seidlitz, White et al: Response to Reviewers

We would like to thank all the reviewers for their extensive reviews of our manuscript and supplementary materials. Their valuable comments and suggestions have stimulated us to further refine and validate our work, and we believe these revisions have significantly strengthened the impact and quality of the paper. Specifically, we now include new analyses explicitly designed to address concerns related to quality control, between-site harmonisation, and stability and validity of out-of-sample centile estimation. Additionally, we now provide lifespan trajectories for an extended suite of MRI phenotypes, including total cerebrum volume, mean cortical thickness and total surface area, as well as regional cortical volumes.

We note that we have excerpted relevant changes to the main text or supplementary information in response to each reviewer comment, for ease of reference, although this entails some repetition of excerpted material (green text) in the cases that different reviewers have made related points.

[BLACK BOLD] - ORIGINAL COMMENT

[BLUE] - RESPONSE TO COMMENT

[HIGHLIGHTED] - NEW TEXT AND FIGURE CHANGES

REFEREE #1:	3
REF 1/1:	3
REF 1/2:	9
REF 1/3:	15
REF 1/3:	17
REF 1/4:	19
REF 1/5:	20
REFEREE #2:	21
REF 2/1:	21
REF 2/2:	33
REF 2/3:	34
REF 2/4:	38
REF 2/5:	47
REF 2/6:	52
REF 2/7:	63
REFEREE #3:	66
REF 3/1:	67
REF 3/2:	75
REF 3/3:	79
REF 3/4:	88
REF 3/5:	89
REF 3/6:	89
REF 3/7:	90
REF 3/8:	90
REF 3/9:	90
REF 3/10:	91
REF 3/11:	91
REF 3/12:	91
REF 3/13:	92
REF 3/14:	93
REF 3/15:	93

REF 3/16:.....	94
REF 3/17:.....	96
REF 3/18:.....	98
REF 3/19:.....	98
REF 3/20:.....	98
REF 3/21:.....	100
REF 3/22:.....	100
REF 3/23:.....	100
REF 3/24:.....	101
REF 3/25:.....	102
REF 3/26:.....	104
REF 3/27:.....	104
REF 3/28:.....	105
REF 3/29:.....	105
REF 3/30:.....	106
REF 3/31:.....	107
REF 3/32:.....	108
REF 3/33:.....	109
REF 3/34:.....	109
REF 3/35:.....	110
REF 3/36:.....	110
REF 3/37:.....	111
REF 3/38:.....	111
REF 3/39:.....	113
REF 3/40:.....	114
REF 3/41:.....	115
REF 3/42:.....	115
REF 3/43:.....	116
REF 3/44:.....	119
REF 3/45:.....	120
REF 3/46:.....	121
REF 3/47:.....	123
REF 3/48:.....	124
REF 3/49:.....	126
REF 3/50:.....	126
REF 3/51:.....	127
REF 3/52:.....	128
REF 3/53:.....	128
REF 3/54:.....	128
REF 3/55:.....	128
REFERENCES CITED IN RESPONSE TO REVIEWERS	130

Referee #1:

Bethlehem, Seidlitz and White provide a lifespan analysis of a 120k+ MRI datasets, for which they should be commended. Clearly this work took a huge amount of effort to bring the many datasets included together, in service of the field's mission to achieve growth charts for the brain. The authors leveraged GAMLSS to generate centiles for basic brain volume measures, as well as to address batch effects. In many ways, the work attempts to fulfill the vision (e.g., pointing to associations with various developmental milestones, noting diagnostic differences), has several impressive demonstrations of validity built in, and provides a Shiny app to facilitate others in extending their work.

A challenge in reviewing the present work is that while very impressive, there are a few key concerns that dramatically reduce enthusiasm for publication, at least in its current form (and admittedly, may take a lot of work to address):

We thank the reviewer for their positive appraisal of our work and we are grateful for the detailed feedback focused on a few key concerns. As anticipated by the reviewer, addressing these issues rigorously and comprehensively has entailed major additional analyses, which have now been included in the paper as described in more detail below.

Ref 1/1:

The authors consciously chose to disregard meaningful quality assessment and consideration of data quality. As a work that is intended to be a model for the field, and to yield a resource upon which others can build – this is a major red flag. While data quality may not impact the median values for trajectories, it is hard to believe it will not impact the variance. One could argue that the rudimentary anatomical measures employed may be more robust to motion and other artifacts – that could be possible but would need to be proven (consider using the dataset employed by Ai et al., Franco, 2021 to do this if desired). The authors could have used Euler number, or MRIQC measures; or BRAINDR for visual inspection – the option to do nothing of the sort – is concerning. Do we really want to see this practice adopted in the field? Would the AD datasets found to be extreme outliers be difficult to flag by eye? Cleaning up the data could improve our ability to detect psychiatric disorders; and can avoid associations between data quality and psychiatric diagnosis, which could be confounding and are well known in the structural and functional MRI literatures.

The reviewer raises an excellent point. In response, we have added an extensive supplemental section on quality control, adopting all the reviewer's suggestions to assess the robustness of our procedures and results to image quality. These additional analyses are fully reported in new sections of Supplementary Information (SI) and are summarised briefly below:

1. Euler index filtering – see **SI2.1** including **Fig. S2.1**
2. Expert visual quality control – see **SI2.2** including **Figs. S2.2.1** and **S2.2.2**
3. Image quality and out-of-sample centile scoring – see **SI2.3**

1) **Euler index filtering:** First, we re-ran GAMLSS modelling of the reference dataset after excluding all scans with higher Euler indices, using a threshold that has previously been reported

to have high sensitivity and specificity for low quality scans (Euler index [EI] > 217; ¹). We found that excluding lower quality scans (approximately 5% of the total dataset) did not alter the principal results on the entire, Euler-unfiltered reference dataset. Specifically both mean (μ) and variance (σ) components of the GAMLSS model were unaffected by filtering out lower quality scans. Furthermore, we found that the whole brain MRI phenotypes estimated from the filtered dataset, excluding high EI scans, were very highly correlated with the phenotypes estimated from the total dataset, including high EI scans ($r > 0.99$ for both Pearson's and Spearman's correlations for median trajectories and 97.5% and 2.5% centile lines for all 4 cerebrum tissue volumes). Second, we conducted several supplementary analyses of the relationships between Euler indices and centile scores. We found minimal (i.e., significant but explaining little variance) relationships between the EI measure of scan quality and individual centile scores for any of the 4 primary brain tissue volumes (i) across the total (unfiltered) dataset (**Fig. S2.1**); (ii) across the EI-filtered dataset (**Fig. S2.1**); or (iii) within each of the clinical groups (AD, MCI, schizophrenia, ADHD, ASD, MDD), with or without exclusion of scans with EI > 217.

2) Expert visual quality control: Recognising that the Euler index is but one metric of image quality, we also investigated the robustness of our results to image quality defined by expert visual inspection of scans. We visually assessed and rated image quality on a 6-point scale for a subset of 9,704 images in the reference dataset. We also used image quality scores previously defined by the primary study teams for fetal MRI studies and for the ABCD cohort. In general, we found that a large majority of scans in these datasets had good image quality defined by visual inspection; the results of normative curve modeling by GAMLSS model fitting to all scans were highly correlated with the results of model fitting after exclusion of the minority of poor quality scans; and individual centile scores were more variable for scans with the worst image quality scores.

3) Image quality and out-of-sample centile scoring: Recognising that image quality would likely be most influential for out-of-sample centile scoring of scans that were not included in the reference dataset, we analysed N=72 scans from an open test-retest dataset ² which had been quantitatively QC'd (by 5 independent raters using *Braindr*⁴) but had not previously been included in our analysis (<https://anisha.pizza/braindr-results/#/>). We found no substantial correlation between Braindr-derived image quality scores and out-of-sample centile scores for any of the 4 cerebrum tissue volumes; however, within-subject reliability of out-of-sample centile scores over repeated MRI scans was improved by a prospective motion correction procedure implemented at the time of MRI data acquisition. We note that this improved consistency of centile scoring was likely driven by improved consistency of the un-centiled phenotype, i.e., the "raw" volumetric data as estimated by FreeSurfer were also more consistent between scans acquired with prospective motion correction.

In short, we have demonstrated by multiple complementary sensitivity analyses that our principal results, and additional out-of-sample results for new data not previously analysed, are remarkably robust to image quality. We conclude that our results are not confounded by uncontrolled image quality issues but proper QC procedures should, of course, be implemented on all scans before they are submitted for out-of-sample centile scoring by GAMLSS modeling.

<<The following changes have been made to the main text>>

In Mapping normative growth:

Models were fitted to structural MRI data from control subjects for the four main tissue volumes of the cerebrum (total cortical grey matter volume [GMV] and total white matter volume [WMV], total subcortical grey matter volume [sGMV], and total ventricular cerebrospinal fluid volume [Ventricles or CSF]). See **Online Methods, Supplementary Table [ST] 1.1-1.7** for details on acquisition, processing and demographics of the dataset. See **Supplementary Information [SI]** for details regarding GAMLSS model specification and estimation (**SI1**), image quality control (**SI2**), model stability and robustness (**SI3-4**), phenotypic validation against non-imaging metrics (**SI3 & SI5.2**), inter-study harmonisation (**SI5**) and assessment of cohort effects (**SI6**).

In Discussion:

We have focused primarily on charts of global brain phenotypes, which were measurable in the largest aggregated sample over the widest age range, with the fewest methodological, theoretical and data sharing constraints. However, we have also provided proof-of-concept brain charts for regional grey matter volumetrics, demonstrating plausible heterochronicity of cortical patterning, and illustrating the generalisability of this approach to a more diverse range of fine-grained MRI phenotypes. As ongoing and future efforts provide increasing amounts of high-quality MRI data, we predict an iterative process of improved brain charts for the human lifespan, potentially representing multi-modal MRI phenotypes and enabling out-of-sample centile scoring of smaller samples or individual scans. In the hope of facilitating progress in this direction, we have provided interactive tools to explore these statistical models and to derive normalised centile scores for new datasets across the lifespan at www.brainchart.io

<<The following changes have been made to the Supplementary Information>>

2. Quality control

While developmental and ageing trajectories of cerebrum tissue volumes were expected to be relatively robust to data quality issues²², controlling the quality of data is an important step in any neuroimaging analysis pipeline. We conducted several complementary analyses to evaluate the robustness of our procedures and results to variable image quality defined by the Euler Index (EI)²³ and other quality control (QC) metrics.

2.1 Euler Index filtering

First, we examined the effect of image quality on estimated brain phenotypes and GAMLSS model parameterisation using EI, an automated, quantitative measure of data quality in scans processed by FreeSurfer (~95% of the reference dataset)^{23,24}. Although cerebrum tissue volumes were estimated prior to cortical surface reconstruction (see **SI18 “Data processing”**), EI has previously been used as a measure of the quality of “raw”, unprocessed scans²³. Thus for the large majority of studies where EI was available (N=101,708 total scans on N=82,023 unique subjects), we excluded all unprocessed scans with EI > 217 (a threshold previously used to define poor quality images²³) and all scans where the runtime for FreeSurfer’s *recon-all* function exceeded 20 hours. We found that developmental trajectories estimated for all 4 cerebrum tissue volumes in this EI-filtered dataset were highly correlated with their trajectories estimated on the basis of the full dataset (all $R^2 > 0.999$ for parametric [Pearson’s] and non-parametric [Spearman’s] correlations between EI-filtered vs EI-unfiltered median trajectories and lower and upper centile lines). Identical parameterisation of fractional polynomials for each random effect was identified by the same model selection procedure (**SI1.3**) in both EI-filtered and EI-unfiltered datasets. Importantly,

EI-filtered and unfiltered datasets also showed a high degree of overlap in subsequently estimated model parameters (correlation of study-specific mean (μ) components > 0.99 ; correlation of study-specific variance (σ) components > 0.95). The only exceptions to this generally high level of consistency were the GUSTO and EDSD cohorts where excluding scans with EI > 217 substantially reduced the number of scans (by $>30\%$) with commensurate reduction in the estimated sigma parameters for these studies.

Second, we examined the relationships between image quality measured by EI and individual centile scores of each brain phenotype. Both for the full dataset and the EI-filtered subset of higher quality scans, we found no significant associations between EI and individual centile scores (**Fig. S2.1**).

Fig. S2.1.2 Associations between centile scores and MRI scan quality defined by EI. Top panel depicts the relation between Euler indices (EI)²³ and centile scores for each of 4 cerebrum tissue volumes estimated by GAMLSS using all available data, regardless of EI. Bottom panel depicts the same set of relationships between centile scores and EI for scans with acceptable image quality defined by EI < 217 (~95% of total scans). In both cases, the Spearman correlations between EI and estimated centiles were negligible (GMV, $\rho=0.02$; WMV, $\rho=-0.07$; sGMV, $\rho=0.01$; Ventricles, $\rho=0.05$). The models fitted to the EI-filtered and unfiltered datasets were also identical in terms of their parameterisation, i.e., data driven selection of the number of fractional polynomials as per **SI1.3**, and subsequent study-specific component weights, suggesting that model specification was robust to the presence of the poorer quality data.

To assess whether there were any age-related differences in motion that could influence model estimation, we evaluated the linear effect of age (in years) on EI in healthy controls in the reference dataset used to estimate normative lifespan trajectories. Using linear regression stratified by sex and accounting for study-specific random effects, we found no evidence for an age-related bias in image quality as assessed with EI ($t = -1.244$, $P = 0.213$). **Fig. S2.1.2** shows the median and standard deviation of age and EI and highlights the top 10 studies with the highest median EI.

Fig. S2.1.2 Age-related variation in image quality measured by the Euler index in female (left panel) and male (right panel) control subjects. Median age (in years) and median EI are shown per study with cross-hairs indicating the standard deviations for age and EI per study. In red the top ten studies with the highest median EI are highlighted. There is no significant relationship between image quality and age at scanning.

2.2 Expert visual quality control

Recognising that EI is but one metric of image quality, and mainly based on the capacity of FreeSurfer to correctly process the scans, we also visually rated image quality for a subset of 9,704 raw scans. Visual inspection rated the response of expert assessors to the following questions: is the brain fully covered by the scan; is there visible noise (due to aliasing, motion etc.), blurriness, or ringing; is there acceptable tissue contrast and image orientation? Based on these criteria, each raw scan was expertly classified on a 6-point scale as perfect (1), very good (2), good (3), bad (4), very bad (5) or unacceptable (6). Only 3% of scans (N=374) were assigned to the two worst quality categories (5 and 6). We analysed centile scores for each of the 4 cerebrum tissue volumes in each of these 6 classes of visually curated image quality (**Fig. S2.2.1**). Centile scores for all 4 phenotypes were consistent across the top 4 classes of image quality but significantly variable for the minority of scans with very bad or unacceptable image quality. However, when we excluded these scans from re-analysis of this expertly QC'ed dataset, we found that the median trajectories and 95% confidence intervals for all 4 brain phenotypes were very highly correlated between the results of model fitting to all 9,704 scans and the results of fitting to the 9,380 scans assigned to the top 4 quality classes (all $R^2 > 0.999$ for both Pearson's and Spearman's correlations for all 4 phenotypes).

Fig S2.2.1. Centile scores for images categorized by expert visual quality assessment of 9,704 unprocessed scans. A small subset (~3%) of the raw data were assigned to the two worst categories of

data quality (QC class 5 or 6) and differed significantly from the other QC classes of data in terms of centile scores for cortical grey matter volume, white matter volume, and subcortical grey matter volume. Bars are coloured by log-scaled sample size.

For foetal and some other primary studies where MRI data were not reconstructed with FreeSurfer, and the EI was therefore not available, scan quality had previously been assessed by expert visual curation as part of primary study procedures (Table **ST1.1** lists the QC steps for each combination of dataset, sex, site and processing pipeline). We re-analysed data from these studies stratified by their prior QC ratings. For example, the Harvard foetal cohort conducted independent visual inspection of image reconstruction quality and classified each of the images as 'great', 'good' or 'bad'. Only the best two categories were included in analyses. We found no significant difference in centile scores for each of the 4 phenotypes between 'great' and 'good' images (GMV, $P=0.58$; WMV, $P=0.34$; sGMV, $P=0.14$; CSF was not available for these foetal scans).

Similarly, the ABCD study provided expert visual counts of artefacts identified by their inspection of FreeSurfer-processed data. For the ABCD data (N=9,056) included in our reference dataset, the majority of images had been rated as containing zero artefacts; a small subset (<0.5%) of scans had been rated as containing one or more artefacts. As shown in **Fig. S2.2.2**, there was some variability of centile scores in the small number of scans with high artefact scores, but there was no significant group level difference in centile scores for any of the four cerebrum tissue volumes between scans with zero artefacts and scans with one or more artefacts (ANOVA, $P>0.05$).

Fig. S2.2.2. Centile scores for ABCD scans previously assigned artefact scores by expert visual QC. The majority (>99%) of ABCD scans included in the aggregated dataset had zero artefacts; for scans with more than one artefact detected there was some variability in estimated centile scores. Bars are coloured by log-scaled sample size.

2.3 Image quality and out-of-sample centile scoring

Recognising that image quality would likely be most influential for out-of-sample centile scoring of scans that were not included in the reference dataset, we analysed N=72 scans from an open test-retest dataset² which had been quantitatively QC'd (by 5 independent raters using *BrainDr*⁴) but had not previously been included in our analysis (<https://anisha.pizza/braindr-results/#/>). We found that *BrainDr* QC scores were not substantially correlated with centile scores for each of the 4 cerebrum tissue volumes (Pearson's r ; GMV=0.034, WMV=0.002, sGMV=0.007, Ventricles=0.004). In the same dataset, we did find that prospective motion correction²⁵

somewhat improved the within-subject correlations of *Braindr*-derived QC scores and GMV centile scores (which changed from $r=0.91$ for prospectively uncorrected data to $r=0.98$ for prospectively corrected data). We note that these beneficial effects of prospective motion correction on test-retest reliability of centile scores derived by OoS analysis using our model are consistent with comparable improvements in test-retest reliability of FreeSurfer-derived phenotypes, as previously reported²².

In short, we have demonstrated by multiple complementary QC studies that our principal results, and additional out-of-sample results for new data not previously analysed, are remarkably robust to image quality. Only a small minority of scans in the aggregated dataset had low image quality; however, tissue volume centile scores derived from the worst quality scans were more variable than centile scores derived from quantitatively or manually QC'd scans. We conclude that our results are not confounded by uncontrolled image quality issues but proper QC procedures should, of course, be implemented on all scans before they are submitted for OoS centile scoring on the basis of our model and aggregated reference dataset.

Ref 1/2:

The authors attempt to account for batch effects using GAMLSS, which is reasonable. Still, it might be good show that an alternative approach (e.g., ComBat) could give converging results, or let one look explore batch effects more directly. Though, bigger picture – these samples are inherently quite diverse in their recruitment and resultant composition. GAMLSS is not a cure-all, and batch effect corrections can under/over correct. Any correction would be inherently limited with respect representativeness in several ways, which is ok, though merits more attention as an issue.

As the reviewer correctly notes, “batch effects” or between-site heterogeneity of scanning procedures are indeed a challenge for combining results or data across multi-site or multi-study neuroimaging datasets. Our principal approach to this challenge has been to use GAMLSS to model study-specific distributions, which is “reasonable”, as the reviewer noted. However, we also agree with the reviewer that ComBAT has emerged as a widely-used tool for dealing with batch effects in various kinds of data and could also be applicable in the current context. We have therefore added an extensive new section (**SI5 “Batch correction and site harmonisation”**) to the supplemental information that addresses these issues in two main ways:

1. **Modeling of between-site heterogeneity by GAMLSS: conceptual considerations in comparison to ComBAT batch correction** – see **SI5.1**. Here we provide a more detailed account of the methodological commonalities and differences between GAMLSS and ComBAT batch-correction, justifying in principle our preferred option of GAMLSS, and reporting new data on GAMLSS modeling of site- versus study-specific differences. The normative trajectories modeled on the basis of site-specific random effects were virtually identical to those reported in the main text on the basis of study-specific random effects (all Pearson’s and Spearman’s correlation coefficients > 0.99).
2. **Modeling of between-site heterogeneity by GAMLSS: empirical evaluation compared to ComBAT** – see **SI5.2** including **Figs. S5.2.1-5.2.5**. Here we report the results of extensive new sensitivity analyses, systematically comparing the results of batch correction by ComBAT to the results obtained by GAMLSS modeling of multi-site data

collected as part of the ABCD study. We estimated normalised values for each MRI phenotype in each scan from each of the ABCD sites using our principal pipeline; and we likewise estimated normalised MRI phenotypes for each study after ComBAT pre-processing of the raw images. Both approaches showed that while some variation between sites is retained, this is no longer a significant factor.

In short, we aim to have clarified and justified, conceptually and empirically, the rationale for using GAMLSS, rather than ComBAT, to correct for site- and study-level batch effects in MRI data.

<<The following changes have been made to the Supplementary Information>>

5. Batch correction and site harmonisation

5.1 Modeling of between-site heterogeneity by GAMLSS: conceptual considerations in comparison to ComBAT batch-correction

Batch effects, or heterogeneities between sites or primary studies, are a challenging issue for estimating generalisable results from multi-site or multi-study neuroimaging data. In recent years, methods such as ComBAT^{14,39} have been translated from their primary application for whole genome transcription (microarray) analysis to achieve harmonisation of MRI data acquired across multiple sites. For our principal analysis, however, we preferred to use GAMLSS, a conceptually similar mathematical framework, to account for between-site or between-study heterogeneity. We made this choice *a priori* for several reasons. Firstly, GAMLSS explicitly includes the possibility of accounting for non-linear age effects (including age-related changes to higher order moments such as variance) during the harmonisation process. Adaptations of traditional ComBAT harmonisation have recently been developed that also allow the inclusion of non-linear age-trends as well as longitudinal, within-subject effects^{40,41}; but these refinements of ComBAT remain somewhat restricted to batch correction of the mean and are not trivial to extend to batch correction of higher order moments, such as the variation across sites. Secondly, we chose to use GAMLSS because it is flexible with regards to the underlying distribution of the data that is to be harmonised; thirdly, because GAMLSS is the WHO-recommended statistical framework for growth chart modelling¹⁸; and finally because GAMLSS allows a flexible modelling capacity that would facilitate scaling of this framework to growth charting of additional MRI phenotypes in the future.

Conceptually, normalised centiles derived from the GAMLSS model (see **SI1.5**) are analogous to normalised scores derived from ComBAT. Specifically, multiple groups of observations have an induced co-dependence, arising in the context of our analysis from common study-specific factors, which leads to a common measurement bias. The aim of both ComBAT and GAMLSS is to correct that common measurement bias. However, whereas ComBAT is derived from a conjugate Bayesian approach and hence restricted to a Gaussian distribution of phenotypes, GAMLSS uses a frequentist, iterative maximum likelihood approach that allows a range of distributions including those with non-zero third and fourth statistical moments (the Gaussian distribution by definition has third and fourth moments equal to zero). Flexibility in the distribution

is important, especially for potentially highly skewed measures (with non-zero third moments), and to allow distributions that conform with the distributions of the measurements. ComBAT assumes that these distributions are naturally Gaussian or can be rendered approximately Gaussian by a simple (e.g., log) transformation. However, even if working with Gaussian measurements, the mean and variance may require non-constant terms to account for heteroskedasticity, and the resulting models are dependent on non-intuitive transformations for Gaussianisation.

In the context of the present study, we used the Bayesian information criterion (BIC) to assess the goodness-of-fit of GAMLSS models making different assumptions about the form of the phenotypic distributions. We found that not only was the Gaussian a suboptimal distribution, but that the optimal choice was the generalised gamma distribution, which includes a third order moment. Although we found no evidence of an age-related change in the third order moment, it was different from unity and hence there was evidence of skewness (otherwise we could reduce it to the gamma distribution, which is the simplified form of the generalised gamma). The (generalised) gamma distribution is also defined only on the positive real line, negating the need to perform any transformations (apart from multiplicative scaling for computational stability), meaning the fitted model coefficients are on the same scale as the original phenotype.

The GAMLSS and ComBAT approaches to batch correction differ substantially in a few other ways. Whereas GAMLSS directly uses centiles and medians of the phenotypic distribution, ComBAT uses the mean and variance. Hence, when comparing these methods, we cannot expect exactly the same results, even if we enforce a Gaussian outcome distribution within GAMLSS. Another substantial difference between the GAMLSS and ComBAT approaches is that GAMLSS requires a substantial amount of data. Even with the number of observations available for our analysis, it has been necessary to use restricted forms, i.e., fractional polynomials, for the normative lifespan trajectories rather than more flexible forms, e.g. splines. Furthermore, ComBAT is defined on a multivariate (Gaussian) phenotype distribution, whereas we used GAMLSS to model multiple univariate phenotypes. (GAMLSS does have some capability to model multivariate distributions, but this area is currently under-developed.) Therefore ComBAT is able to adjust for batch effects with fewer observations on the assumption that the batch effect is shared across multiple phenotypes. Running ComBAT in a univariate mode would be most directly equivalent to the GAMLSS approach but this is not how it is used in the wider literature. This implies that multivariate normalisation by ComBAT is to some extent dependent upon the set of phenotypes included; if a new phenotype is included the ComBAT correction for batch effects would need to be re-run.

In short, there are pros and cons to both harmonisation strategies: ComBAT is better suited for smaller datasets, Normalised distributions and multivariate phenotypes; whereas GAMLSS is better suited for large datasets, non-Gaussian distributions and univariate phenotypes. We preferred GAMLSS on the grounds of its greater scalability and flexibility to match the distributional properties of the reference data and the scope of this project.

While we principally modeled lifespan brain trajectories with primary study (not scanning site) as “the batch” to be corrected by GAMLSS or ComBAT, we also modelled trajectories treating both study and site as batch effects. The results were nearly identical for study-batch corrected or study-and-site batch corrected trajectories (all $r^2 > 0.99$ for both parametric [Pearson’s] and non-parametric [Spearman’s] correlations). This near-perfect agreement is likely due in part to the

partitioning of variation. The study and study-site random-effects covariance structures are both dominated by the sigma-component, i.e., phenotype variance. Essentially once we increase the resolution of batch effects to study-and-site specific random-effects, we have reduced the sample size to estimate each random-effect and hence this uncertainty is unable to compete with the raw observation noise (captured by the sigma-component). In an ideal scenario one would use a site within study nested random-effects structure. However the co-dependence of variation in processing pipelines, MRI acquisition parameters, lifespan coverage, and small site-specific sample sizes, combined with the inherent observation noise, means such a covariance specification is unlikely to be viable with the currently available data (also, GAMLSS does not currently support nested covariance structures).

5.2 Modeling of between-site heterogeneity by GAMLSS: empirical evaluation compared to ComBAT

To empirically evaluate the capacity of GAMLSS to account for batch effects or between-site variation, we analysed the well-known multi-site ABCD study⁴² and compared the results of between-site harmonisation by GAMLSS to the results of a standard ComBAT harmonisation pipeline. Compared to the raw ABCD imaging data, which show clear effects of site across all MRI phenotypes, both ComBAT and GAMLSS efficiently removed these batch effects in the normalised (site-corrected) data, but both harmonisation pipelines retained a high degree of variation at the level of individual scans (Fig. S5.2.1-5.2.2).

Fig. S5.2.1. Raw volumetric data and centile scores for male subjects from the ABCD cohort. The top row shows raw volumetric data across the 22 sites included in ABCD, the middle row shows centile normalised data by GAMLSS and the bottom row shows data normalised using ComBAT. ANOVA P-values refer to one-way analyses of variance across sites for each individual phenotype. Bars are coloured by site. ComBAT and GAMLSS are both able to substantially mitigate batch effects in multi-site MRI data.

Fig. S5.2.2. Raw volumetric data and centile scores for female subjects from the ABCD cohort. The top row shows raw volumetric data across the 22 sites included in ABCD, the middle row shows centile normalised data by GAMLSS and the bottom row shows data normalised using ComBAT. ANOVA P-values refer to one-way analyses of variance across sites for each individual phenotype. ComBAT and GAMLSS are both able to substantially mitigate batch effects in multi-site MRI data.

To further assess whether batch-corrected MRI data derived from both ComBAT and GAMLSS pipelines would generate convergent results in subsequent analyses, we estimated the correlations between total cerebrum volume (TCV) and fluid intelligence or birth weight, after TCV was estimated in data that had been batch-corrected by either GAMLSS or ComBAT. Both these psychological and biological factors have previously been shown to be correlated with TCV^{43–45}. We were able to replicate these significant associations with TCV after both GAMLSS and ComBAT batch correction; and batch-corrected data from both pipelines were more consistently associated with fluid intelligence or birth weight than the raw (uncorrected) data from multiple sites within the ABCD cohort (**Fig. S5.4-5.5**).

Fig. S5.2.3. Comparing effects of GAMLSS versus ComBAT batch correction on estimation of total cerebrum volume. TCV was estimated for $N=10,583$ participants in the ABCD multi-site study after MRI

data had been batch-corrected for between-site differences by ComBAT (y-axis) or GAMLSS (x-axis). Estimated TCV was highly correlated ($r > 0.99$) downstream of these two batch correction procedures. Scans are point-coloured according to site.

Fig. S5.2.4. Associations between total cerebrum volume (TCV) and birth weight (top) or fluid intelligence (bottom) after batch correction by GAMLSS (left), by ComBAT (middle), or without batch correction (raw, right). Linear relationships for each of the 22 sites in the ABCD study are in coloured solid lines; dashed lines signify overall model fit across sites; fluid intelligence was assessed using the NIH Toolbox⁴⁶. These results show that predicted relationships between TCV and both birth weight and fluid intelligence are more convincingly replicated in these $N=10,583$ scans from the ABCD multi-site study when the MRI data have been batch-corrected by either GAMLSS or ComBAT compared to when the MRI data have been analysed without correction of between-site differences.

Fig. S5.2.5. Consistency of behavioural (fluid intelligence) and biological (birth weight) associations with total cerebrum volume (TCV) estimated at 22 MRI acquisition sites in the ABCD cohort, after batch correction for site effects by GAMLSS (left column) or ComBAT (right column). Regression coefficients and standard errors from linear regression models of TCV on birth weight or fluid intelligence are plotted using point-ranges for each site. Meta-analytic coefficients and errors, combining all primary

study effects, are shown in black at the top of each column. Coefficients (triangles) are scaled based on sample size at each site within the ABCD study.

Ref 1/3:

It is unclear how uniform the preprocessing for data was, and what specific steps were carried out – which is concerning, unless I missed it. Did the authors bring the data to a common space for calculation of measures? Did they need to for the measures obtained? (if alignment could be avoided, that could minimize introduction of potential biases related to registration error - which is heavily impacted by data quality). There is a lot of detail that is crucial to know to fairly evaluate the work, and it is not easily obtainable.

We thank the reviewer for allowing us the opportunity to expand our methodological description. In response to the request for more accessible detail on pre-processing methods, we now provide:

1. Additional detail on exactly how each individual dataset was quality controlled and pre-processed – see **SI2 “Quality control”** and **SI18 “Data processing”** including a new **Table ST1.1** providing study specific details on QC, acquisition and processing. A large majority (95%) of primary studies used a version of FreeSurfer for pre-processing. For these studies, total tissue volumes were extracted from the resulting *aseg.stats* files, which are generated prior to cortical surface reconstruction and are thus less impacted by data quality.
2. Additional analysis of reliability of out-of-sample centile scoring across multiple versions of FreeSurfer. For empirical evaluation of the influence of different pre-processing strategies, we estimated the intra-class correlations between centile scores derived from a single study after the data had been pre-processed by multiple different pipelines. These results indicated that centiles estimated after pre-processing with FreeSurfer version 5.3 or any later versions of FreeSurfer were highly consistent. This is particularly encouraging because, as shown in **Table ST1.1**, none of the primary studies used an earlier version of FreeSurfer than 5.3. See **SI 4.4 “Reliability of out-of-sample centile scoring across multiple versions of FreeSurfer”** including new **Fig. S4.4**.

<<The following changes have been made to the Supplemental Information>>

18. Data processing

If T1- and T2/FLAIR-weighted raw data were available, as they were for approximately 95% of scans), these data were processed with FreeSurfer 6.0.1²⁴ using the combined T1-T2 recon-all pipeline for improved grey-white matter boundary estimation. If only raw T1-weighted data were available, and subjects were aged over 2 years, data were processed with FreeSurfer 6.0.1 using the standard recon-all pipeline. If subjects were aged 0–2 years, data were processed with Infant FreeSurfer v1⁹⁴. **ST1.1** lists the number of subjects per site per processing pipeline alongside their respective MRI acquisition and quality control protocols. We noticed that Infant FreeSurfer estimated total subcortical grey matter volume (sGMV) differently from other pipelines included in this dataset, while other cerebrum tissue volumes were estimated consistently across pipelines. We therefore excluded scans processed with Infant FreeSurfer from growth curve estimation for subcortical GMV. All four cerebrum tissue volumes were extracted from the *aseg.stats* files that

are generated in the first stage of the recon-all process: 'Total cortical gray matter volume' for GMV; 'Total cortical/cerebral (FreeSurfer version dependent) white matter volume' for WMV; 'Subcortical gray matter volume' for sGMV (inclusive of thalamus, caudate nucleus, putamen, pallidum, hippocampus, amygdala, and nucleus accumbens area; <https://freesurfer.net/fswiki/SubcorticalSegmentation>); and the difference between 'BrainSegVol' and 'BrainSegVolNotVent' for Ventricular volume. The first processing stage of recon-all includes: non-uniformity correction, projection to Talairach space, intensity normalisation, skull-stripping, automatic tissue and subcortical segmentation. Surface interpolation, tessellation and registration are done at the second and third stages of the recon-all pipeline (i.e., after aseg.stats files are created) and all these later stage processes involve projection to a standard stereotactic (fsaverage) space. Regional volume, thickness, and surface area was estimated for each of 34 bilaterally averaged cortical regions defined by the Desikan-Killiany⁴⁸ parcellation template following the final stages of the recon-all pipeline and using the hemisphere-specific apars.stats files generated by FreeSurfer.

4.4. Reliability of out-of-sample centile scoring across multiple versions of FreeSurfer

Knowing that a large majority (~95%) of primary studies in the reference dataset used one of a series of versions of FreeSurfer for image analysis, we also evaluated the impact of these incrementally different image analysis pipelines on reliability of OoS centile scores. To do this we re-analysed a single dataset³⁶ repeatedly using 4 different versions of FreeSurfer (5.1, 5.3, 6.01, and 7.1). Each version of the processed dataset was treated as an independent OoS study for GAMLSS modeling and then we estimated ICCs between individual centile scores for each possible pair of FreeSurfer pipelines and for each of four cerebrum tissue volumes. This analysis demonstrated generally high within-subject reliability of OoS centiles across all four pipelines: ICCs for GMV=0.978, WMV=0.972, sGMV=0.816 and Ventricles=0.982 (**Fig. S4.4**). We noted that there was somewhat reduced reliability of subcortical grey matter volume in both raw and centiled data from FreeSurfer version 5.1 in comparison to later FreeSurfer versions. While the reasons for this are unclear, none of the studies included in the principal dataset were processed with FreeSurfer 5.1, or any version of FreeSurfer older than 5.3. Furthermore, we found the highest between-pipeline reliability for both raw volumetric data and centile scores derived from the two most recent versions of FreeSurfer, 6.0.1 and 7.1, suggesting that minor inconsistencies due to FreeSurfer pre-processing are becoming less problematic as this widely used software package incrementally evolves.

Fig. S4.4. Between-pipeline reliability of volumetric data and out-of-sample centile scores for four cerebrum tissue volumes measured in the same set of $N=1,468$ scans re-analysed using 4 different versions of FreeSurfer (5.1, 5.3, 6.01, and 7.1). Top row shows scatterplot matrices representing the correlations between raw volumetric data derived from each possible pair of FreeSurfer pipelines, from left to right: GMV, WMV, sGMV, Ventricles. Bottom row shows scatterplot matrices representing the correlations between out-of-sample centile scores derived from each possible pair of FreeSurfer pipelines, from left to right: GMV, WMV, sGMV, Ventricles. Intra-class correlations of out-of-sample centile scores and uncensored volumetric data, on average over all pairs of four pipelines, were generally high (GMV=0.978, WMV=0.972, sGMV=0.816 and Ventricles=0.982). Although the reliability of sGMV volumetrics and centile scores was somewhat lower due to discrepant measurements by the oldest version of FreeSurfer, v5.1, this version of FreeSurfer was not used to analyse any of the scans included in the reference dataset.

Ref 1/3:

Figure 1 should show # of datasets per site – the current depictions are not sufficient, and can be viewed as misleading (one can glean from distributions, but that is not trivial to meaningfully/accurately do).

We agree that it is important to be clear about the number of scans included in each primary study. Detailed information on each primary study, including sample size, is now provided in supplementary tables **ST1.1-1.8**. We have also updated **Fig. 1** so that the box plot representing each study is now coloured to represent sample size “at first glance”. Given the wide range and 10-fold increase in UK BioBank sample size compared to many other studies, colour is plotted on a log-scale. We tried incorporating exact numbers in the figures but making those legible compromised the quality of the figure and distracted from the main findings depicted, hence we preferred this coloured box-plot format instead.

<<The following changes have been made to the main text>>

A | Aggregated MRI Datasets

B | Cerebrum tissue volumes

C | Normative trajectories

D | Normative variance

E | Normative rate-of-growth

Female Male

Fig. 1. Human brain charts. A | MRI data were aggregated from 100 primary studies comprising 123,984 scans that collectively spanned the age range from late pregnancy to 100 postnatal years. Box-violin plots show age distributions (log-scaled) for each study coloured by its relative sample-size (log-scaled) B | Non-centilied bilateral cerebrum tissue volumes (right to left: grey matter, white matter, subcortical grey matter and ventricles) are plotted for each cross-sectional control scan, point-coloured by sex, as a function of age (log-scaled). C | Normative brain growth curves, analogous to paediatric growth charts, were estimated by generalised additive modelling for location scale and shape (GAMLSS), accounting for site- and study-specific batch effects, and stratified by sex (female/male curves coloured red/blue). All four cerebrum tissue volumes demonstrated distinct, non-linear trajectories of their medians and 95% centile boundaries as a function of age over the life-cycle. Demographics for each cross-sectional sample of healthy controls included in the reference dataset for normative GAMLSS modeling of each MRI phenotype are detailed in **ST1.2-1.7**. D | Trajectories of median between-subject variability and 95% confidence intervals for four cerebrum issue volumes were estimated by sex-stratified bootstrapping (1,000 times; see **SI3** for details). E | Rates of volumetric change across the lifespan for each tissue volume, stratified by sex, were estimated by the first derivatives of the median volumetric trajectories. For solid (parenchymal) tissue volumes, the solid horizontal line ($y=0$) indicates when the volume of each tissue stops growing and starts shrinking; the solid vertical line indicates the age of maximum growth of each tissue. See **ST2.1** for all

neurodevelopmental milestones and their confidence intervals. Note that y-axes in panels B-E are scaled in units of 10,000 mm³ (10ml).

Ref 1/4:

A smaller note - how are the authors accounting for variations in diagnostic practices?

There is indeed variation in diagnostic practices between primary clinical studies, e.g., due to varying diagnostic eligibility criteria and/or questionnaire assessment tools used to ascertain “caseness”. Although this source of study-level variation is evident across the primary clinical studies in the aggregated dataset, we can account for its impact primarily by explicitly modeling **all** study-specific factors (including diagnostic differences in clinical studies) as random effects on three moments of the generalised gamma distributions of the MRI phenotypes. As such, each study is adjusted with its own unique random effect parameterization that would encompass any deviation due to different assessment of caseness. The rationale and justification for this approach to “batch correction” is detailed in **SI5 “Batch correction and site harmonisation”** and throughout **SI1 “Modelling lifespan trajectories of brain maturation”**.

We additionally provide new detail relevant to this point, as follows:

- We note that normative growth charts are estimated from reference data exclusive of primary clinical studies of formally diagnosed cases of disorder, thus eliminating any potential impact of variation in diagnostic practice on normative brain growth charts.
- We show that allowing random effects on study-specific distributions protects the normative growth curves from being unduly influenced by between-study variability. It also ensures centiles estimated at subject level (for both cases and controls) are appropriately adjusted in relation to the study from which they were drawn: **SI1.1-1.7 “Modelling lifespan trajectories”**.
- We conducted a comprehensive set of leave-one-study-out (LOSO) analyses to ensure normative curves were not unduly biased by any single study and to identify which studies were most idiosyncratically influential on the statistics of the reference dataset: **SI3.2 “Model sensitivity analyses”**.

<<The following changes have been made to Supplementary Information>>

In *SI1 Modelling lifespan trajectories*

Furthermore, there could be variability between studies in the standards used for diagnosis of disorders, and/or for ascertainment of healthy controls, and the clinically diagnosed cases are spread across several studies with potentially different diagnostic standards. We note that this issue is unlikely to impact on the normative brain charts which were estimated on the basis of healthy control data only. However, it is a potential source of study-specific differences that could bias centile scoring if not corrected. To that end, we require study-specific estimates from our model so that we can use the parameters obtained when modelling the normative curves to also adjust for study random effects in the individuals that were not included in the normative reference (i.e., cases). This requirement excludes many non-parametric outcome distribution approaches and conditional inference methods, e.g., generalised estimating equations (GEEs)¹³, would be unsuitable for this purpose since they explicitly avoid or side-step estimating study-specific

random-effects. Although many of these methods reduce bias under model mis-specification and can effectively harmonise data across studies, we require the feature they integrate out. Similarly this excludes approaches like the recent application of ComBAT¹⁴ in neuroimaging from its origins in the genomics literature (see **SI5** for an in-depth comparison between the two approaches). The principal way in which we have accounted for between-study - differences is by using the GAMLSS modeling framework to estimate study-specific random effects on the three moments of the statistical distributions of the MRI phenotypes To demonstrate the robustness to study-specific variability of growth curves and the individual centiles derived from them, we conducted 'leave-one-study-out' (LOSO) analyses whereby the growth curves and centiles were repeatedly estimated after exclusion of each individual study (see **SI 3.2 "Model sensitivity analyses"**). These analyses confirm that trajectories from the total dataset are in general highly conserved after exclusion of each individual study, suggesting that study-specific differences do not materially influence model parameters.

Ref 1/5:

A final note is authorship – what was the criteria for data contributors, if they did not actively participate in the actually data analysis / manuscript? If such situations exist, is there an inadvertent benefit to those who are closed with their datasets? At a minimum, please consider grouping all those that only contributed data under a single collaborative name on the author line...then link that to the individuals in pubmed. This would better recognize the contributions of those who did the bulk of this analysis.

We appreciate this important point and recognize that the authorship list is longer than usual. The study would not have been possible without an inclusive "team science" approach. As required by the *Nature* authorship guidelines, all listed authors were actively involved in data collection, aggregation or processing; in addition, all authors read and commented on the manuscript, and revised the manuscript and supplementary materials. The contributors who designed and conducted the bulk of the analysis are designated as joint first (RB, JS, SW) and joint senior (EB, AA-B) authors. We note that in several cases, consortia were named rather than individual co-authors when this was considered to be more appropriate. We would be more than happy to discuss this issue further, if appropriate.

Referee #2:

Using data combined across several primary studies, this article presents cross-sectional lifespan age trends for individual differences in a small number of gross brain morphology “tissue classes” (e.g. total brain volume, white matter volume, subcortical grey matter volume, and ventricular volume) as measured by structural MRI. Rather than simply providing the mean age trends in each tissue class, the authors provide information about how the dispersion changes with age. Additionally, because mean growth is nonlinear, the authors are able to calculate ages at peak velocity for each class. This descriptive exercise is pitched as “brain charts” to “reference standards against which to anchor measures of individual differences in brain morphology,” much in the same way that “growth charts” are used. They provide an online tool for calculating individual however, there are several unsatisfying, if not concerning, aspects to this packaging.

Ref 2/1:

There is no way to be confident in the use of these reference norms given that scanner to scanner variation can lead to tremendous mean differences in inferred volumes. Without first calibrating an MRI relative to the norms, nearly everyone scanned in a particular MRI might be classified as abnormal. Even if the norms were only used to make comparisons across samples rather than individuals, it would not be clear of the samples actually differed in mean volumes or scanners were simply miscalibrated.

The reviewer raises a key point concerning the importance of adequately correcting the data for different sites or studies when estimating normative trajectories and individual centile scores. To address the question of between-site variation, or “batch effects” in the MRI data -- and to show that the statistical approach we have adopted does indeed satisfactorily account for these potentially troublesome effects -- we have now made the following changes to the paper:

1. We have provided an extensive conceptual discussion of between-site harmonisation, including a detailed comparison with ComBAT, which is a well-known alternative method for between-site harmonisation: **SI5.1 “Modeling of between-site heterogeneity by GAMLSS: conceptual considerations in comparison to ComBAT batch-correction”**.
2. We have quantitatively compared GAMLSS and ComBAT methods for correction of between-study effects, and for correction of between-site differences in a multi-site study, the results of which are discussed in **SI5.2 “Modeling of between-site heterogeneity by GAMLSS: empirical evaluation compared to ComBAT”**, including 5 new supplementary **Figs. S5.2.1-S5.2.5**
3. We have conducted an extensive evaluation of the stability of our methods for out-of-sample centile scoring, which is now included in a new supplementary section: **SI4 “Out-of-sample centile scoring: bias, stability and reliability”**.

<<The following changes have been made to the Supplemental Information>>

5. Batch correction and site harmonisation

5.1 Modeling of between-site heterogeneity by GAMLSS: conceptual considerations in comparison to ComBAT batch-correction

Batch effects, or heterogeneities between sites or primary studies, are a challenging issue for estimating generalisable results from multi-site or multi-study neuroimaging data. In recent years, methods such as ComBAT^{14,39} have been translated from their primary application for whole genome transcription (microarray) analysis to achieve harmonisation of MRI data acquired across multiple sites. For our principal analysis, however, we preferred to use GAMLSS, a conceptually similar mathematical framework, to account for between-site or between-study heterogeneity. We made this choice *a priori* for several reasons. Firstly, GAMLSS explicitly includes the possibility of accounting for non-linear age effects (including age-related changes to higher order moments such as variance) during the harmonisation process. Adaptations of traditional ComBAT harmonisation have recently been developed that also allow the inclusion of non-linear age-trends as well as longitudinal, within-subject effects^{40,41}; but these refinements of ComBAT remain somewhat restricted to batch correction of the mean and are not trivial to extend to batch correction of higher order moments, such as the variation across sites. Secondly, we chose to use GAMLSS because it is flexible with regards to the underlying distribution of the data that is to be harmonised; thirdly, because GAMLSS is the WHO-recommended statistical framework for growth chart modelling¹⁸; and finally because GAMLSS allows a flexible modelling capacity that would facilitate scaling of this framework to growth charting of additional MRI phenotypes in the future.

Conceptually, normalised centiles derived from the GAMLSS model (see **SI1.5**) are analogous to normalised scores derived from ComBAT. Specifically, multiple groups of observations have an induced co-dependence, arising in the context of our analysis from common study-specific factors, which leads to a common measurement bias. The aim of both ComBAT and GAMLSS is to correct that common measurement bias. However, whereas ComBAT is derived from a conjugate Bayesian approach and hence restricted to a Gaussian distribution of phenotypes, GAMLSS uses a frequentist, iterative maximum likelihood approach that allows a range of distributions including those with non-zero third and fourth statistical moments (the Gaussian distribution by definition has third and fourth moments equal to zero). Flexibility in the distribution is important, especially for potentially highly skewed measures (with non-zero third moments), and to allow distributions that conform with the distributions of the measurements. ComBAT assumes that these distributions are naturally Gaussian or can be rendered approximately Gaussian by a simple (e.g., log) transformation. However, even if working with Gaussian measurements, the mean and variance may require non-constant terms to account for heteroskedasticity, and the resulting models are dependent on non-intuitive transformations for Gaussianisation.

In the context of the present study, we used the Bayesian information criterion (BIC) to assess the goodness-of-fit of GAMLSS models making different assumptions about the form of the phenotypic distributions. We found that not only was the Gaussian a suboptimal distribution, but that the optimal choice was the generalised gamma distribution, which includes a third order moment. Although we found no evidence of an age-related change in the third order moment, it was different from unity and hence there was evidence of skewness (otherwise we could reduce

it to the gamma distribution, which is the simplified form of the generalised gamma). The (generalised) gamma distribution is also defined only on the positive real line, negating the need to perform any transformations (apart from multiplicative scaling for computational stability), meaning the fitted model coefficients are on the same scale as the original phenotype.

The GAMLSS and ComBAT approaches to batch correction differ substantially in a few other ways. Whereas GAMLSS directly uses centiles and medians of the phenotypic distribution, ComBAT uses the mean and variance. Hence, when comparing these methods, we cannot expect exactly the same results, even if we enforce a Gaussian outcome distribution within GAMLSS. Another substantial difference between the GAMLSS and ComBAT approaches is that GAMLSS requires a substantial amount of data. Even with the number of observations available for our analysis, it has been necessary to use restricted forms, i.e., fractional polynomials, for the normative lifespan trajectories rather than more flexible forms, e.g, splines. Furthermore, ComBAT is defined on a multivariate (Gaussian) phenotype distribution, whereas we used GAMLSS to model multiple univariate phenotypes. (GAMLSS does have some capability to model multivariate distributions, but this area is currently under-developed.) Therefore ComBAT is able to adjust for batch effects with fewer observations on the assumption that the batch effect is shared across multiple phenotypes. Running ComBAT in a univariate mode would be most directly equivalent to the GAMLSS approach but this is not how it is used in the wider literature. This implies that multivariate normalisation by ComBAT is to some extent dependent upon the set of phenotypes included; if a new phenotype is included the ComBAT correction for batch effects would need to be re-run.

In short, there are pros and cons to both harmonisation strategies: ComBAT is better suited for smaller datasets, Normalised distributions and multivariate phenotypes; whereas GAMLSS is better suited for large datasets, non-Gaussian distributions and univariate phenotypes. We preferred GAMLSS on the grounds of its greater scalability and flexibility to match the distributional properties of the reference data and the scope of this project.

While we principally modeled lifespan brain trajectories with primary study (not scanning site) as “the batch” to be corrected by GAMLSS or ComBAT, we also modelled trajectories treating both study and site as batch effects. The results were nearly identical for study-batch corrected or study-and-site batch corrected trajectories (all $r^2 > 0.99$ for both parametric [Pearson’s] and non-parametric [Spearman’s] correlations). This near-perfect agreement is likely due in part to the partitioning of variation. The study and study-site random-effects covariance structures are both dominated by the sigma-component, i.e., phenotype variance. Essentially once we increase the resolution of batch effects to study-and-site specific random-effects, we have reduced the sample size to estimate each random-effect and hence this uncertainty is unable to compete with the raw observation noise (captured by the sigma-component). In an ideal scenario one would use a site within study nested random-effects structure. However the co-dependence of variation in processing pipelines, MRI acquisition parameters, lifespan coverage, and small site-specific sample sizes, combined with the inherent observation noise, means such a covariance specification is unlikely to be viable with the currently available data (also, GAMLSS does not currently support nested covariance structures).

5.2 Modeling of between-site heterogeneity by GAMLSS: empirical evaluation compared to ComBAT

To empirically evaluate the capacity of GAMLSS to account for batch effects or between-site variation, we analysed the well-known multi-site ABCD study⁴² and compared the results of between-site harmonisation by GAMLSS to the results of a standard ComBAT harmonisation pipeline. Compared to the raw ABCD imaging data, which show clear effects of site across all MRI phenotypes, both ComBAT and GAMLSS efficiently removed these batch effects in the normalised (site-corrected) data, but both harmonisation pipelines retained a high degree of variation at the level of individual scans (Fig. S5.2.1-5.2.2).

Fig. S5.2.1. Raw volumetric data and centile scores for male subjects from the ABCD cohort. The top row shows raw volumetric data across the 22 sites included in ABCD, the middle row shows centile normalised data by GAMLSS and the bottom row shows data normalised using ComBAT. ANOVA P-values refer to one-way analyses of variance across sites for each individual phenotype. Bars are coloured by site. ComBAT and GAMLSS are both able to substantially mitigate batch effects in multi-site MRI data.

Fig. S5.2.2. Raw volumetric data and centile scores for female subjects from the ABCD cohort. The top row shows raw volumetric data across the 22 sites included in ABCD, the middle row shows centile normalised data by GAMLSS and the bottom row shows data normalised using ComBAT. ANOVA P-values

refer to one-way analyses of variance across sites for each individual phenotype. ComBAT and GAMLSS are both able to substantially mitigate batch effects in multi-site MRI data.

To further assess whether batch-corrected MRI data derived from both ComBAT and GAMLSS pipelines would generate convergent results in subsequent analyses, we estimated the correlations between total cerebrum volume (TCV) and fluid intelligence or birth weight, after TCV was estimated in data that had been batch-corrected by either GAMLSS or ComBAT. Both these psychological and biological factors have previously been shown to be correlated with TCV^{43–45}. We were able to replicate these significant associations with TCV after both GAMLSS and ComBAT batch correction; and batch-corrected data from both pipelines were more consistently associated with fluid intelligence or birth weight than the raw (uncorrected) data from multiple sites within the ABCD cohort (Fig. S5.4-5.5).

Fig. S5.2.3. Comparing effects of GAMLSS versus ComBAT batch correction on estimation of total cerebrum volume. TCV was estimated for $N=10,583$ participants in the ABCD multi-site study after MRI data had been batch-corrected for between-site differences by ComBAT (y-axis) or GAMLSS (x-axis). Estimated TCV was highly correlated ($r > 0.99$) downstream of these two batch correction procedures. Scans are point-coloured according to site.

Fig. S5.2.4. Associations between total cerebrum volume (TCV) and birth weight (top) or fluid intelligence (bottom) after batch correction by GAMLSS (left), by ComBAT (middle), or without batch correction (raw, right). Linear relationships for each of the 22 sites in the ABCD study are in coloured solid lines; dashed lines signify overall model fit across sites; fluid intelligence was assessed using the NIH Toolbox⁴⁶. These results show that predicted relationships between TCV and both birth weight and fluid intelligence are more convincingly replicated in these $N=10,583$ scans from the ABCD multi-site study when the MRI data have been batch-corrected by either GAMLSS or ComBAT compared to when the MRI data have been analysed without correction of between-site differences.

Fig. S5.2.5. Consistency of behavioural (fluid intelligence) and biological (birth weight) associations with total cerebrum volume (TCV) estimated at 22 MRI acquisition sites in the ABCD cohort, after batch correction for site effects by GAMLSS (left column) or ComBAT (right column). Regression coefficients and standard errors from linear regression models of TCV on birth weight or fluid intelligence are plotted using point-ranges for each site. Meta-analytic coefficients and errors, combining all primary study effects, are shown in black at the top of each column. Coefficients (triangles) are scaled based on sample size at each site within the ABCD study.

4. Out-of-sample centile scoring: bias, stability and reliability

4.1. Bias of out-of-sample centile scores: leave-one-study-out analyses for 100 studies

To further evaluate the robustness and consistency of centile scoring of OoS MRI data that were not included in the reference dataset used to estimate population trajectories, we performed a comprehensive series of leave-one-study-out (LOSO) analyses. For each one of the 100 studies in the reference dataset, we removed the study from the reference dataset, re-fitted the GAMLSS model to the remaining dataset of 99 studies, computed the OoS centile scores for the excluded study, and compared the OoS centile scores to the in-sample centile scores computed for the same study from the complete dataset including all 100 studies. Supplementary tables **ST7.1-7.4** list the correlations between OoS and in-sample centile scores for all 4 cerebrum tissue volumes in each of 100 primary studies. Overall, we found very high levels of correlation (Pearson's $r \sim 0.99$) for almost all studies, indicating that centile scores can be estimated accurately for most studies even if they were not included in the reference dataset used to define population norms. Correlations between OoS and in-sample centile scores were lower than $r = 0.99$ for only 3 out of 100 studies in the reference dataset: namely, the FinnBrain ($r = 0.93$), UCSD ($r = 0.96$) and NIHPD ($r = 0.95$) studies. These studies were characterised by relatively small sample size, foetal or early postnatal age range of participants, or idiosyncratic processing pipelines.

4.2. Stability of out-of-sample centile scoring: bootstrapped LOSO analyses for 100 studies

In addition, we tested the reliability of OoS centile scores for each individual participant by bootstrapping. Specifically, for each LOSO sample, bootstrapped model parameters were generated (see **SI3.2.2 “Bootstrap analysis”**), resulting in 1,000 bootstrapped models with maximum likelihood estimated parameters for each bootstrap iteration of each left-out study. From this we obtained a bootstrapped distribution of out-of-sample centile scores for each individual subject in each individual iteration of left-out studies, thus providing a stability assessment in the form of the standard deviation of individual OoS centile scores across 1,000 bootstrap iterations. Across the datasets included in the model, we found that the average standard deviation of (bootstrapped) OoS centiles was 0.014, which is well below the level of within-subject longitudinal variation (see **Fig. S4.2.1** and **SI14 “Longitudinal centiles”**). Furthermore, we found increased standard deviation of OoS centile scores for datasets with comparatively small sample sizes (e.g., the OpenPain cohorts, Cambridge foetal Testosterone and CHILD studies; see **Fig. S4.2.2**). OoS centile scores were also more variable for datasets that had a more unique combination of age range, acquisition and processing pipelines (e.g., FinnBrain, IBIS and HBN; see **Fig. S4.2.2**). These observations reinforce the recommendation -- see main text, ‘**Out-of-sample centile scoring of “new” MRI data**’ -- that OoS centile scoring is reliable for studies comprising $N > 100$ scans. It was also notable that the reliability of OoS centile scores was weakly correlated with data quality as quantified by the Euler index (EI). So studies with higher EI²³, indicating poorer image quality, tended to have higher variability of bootstrapped OoS centile scores (Pearson's r for all 4 cerebrum tissue volumes: GMV=0.05, WMV =0.11, sGMV =0.14, and Ventricular volume = 0.13). These results were not substantially different when the whole set of analyses was repeated without including scans with EI > 217. We conclude that OoS estimation of centile scores is generally reliable at the level of individual scans, and (as expected) reliability is greater for higher quality scans.

Fig. S4.2.1. Stability of out-of-sample centile scores for four cerebrum tissue volumes when each of 100 studies was excluded from the reference dataset before bootstrapping. The standard deviation of bootstrapped centile scores (y-axis) is plotted for each study (x-axis) for each phenotype, from top to bottom panels: total cortical grey matter volume, total cortical white matter volume, subcortical grey matter volume, and ventricular volume. Each study- and phenotype-specific boxplot is coloured according to log sample size. For each study, we estimated the normative model leaving that study out of the reference dataset and repeated this procedure after iteratively bootstrapping the reference dataset 1,000 times. This procedure allowed us to summarise the reliability of the out-of-sample estimates of centile scores in terms of the standard deviation of the 1,000 centile scores generated for each bootstrapped resampling of the reference dataset. Studies are ordered by median standard deviation of out-of-sample centile scores (small to large) indicating that scans are reliably assigned centile scores with the out-of-sample approach.

Fig. S4.2.2. Stability of out-of-sample centile scores as a function of age and sample size. The standard deviation (SD) of bootstrapped centile scores for four cerebrum tissue volumes (y-axis) is plotted against mean age of study participants (top row) or sample size (bottom row). Studies with the most unstable OoS centile scores ($SD > 0.05$) are highlighted in red and labelled (see **ST1.1** for study details).

4.3. Test-retest reliability of out-of-sample centile scoring

We also assessed the reliability of OoS centile scoring in three independent datasets that acquired multiple MRI scans within a single session or two closely spaced sessions^{22,34–36}. We analyzed each scan as a novel OoS dataset, then compared the consistency of centile scores across different scans of the same subject. We similarly compared the consistency of the uncensored volumetric data and found that the out-of-sample centile scores were as consistent between scans in the same session as the “raw” volumetric data generated by FreeSurfer.

First, we analysed test-retest reliability using the multimodal MRI reproducibility resource³⁴, which provides two sessions of MRI data for multiple modalities. This dataset comprising 21 subjects was specifically designed for assessment of test-retest reliability as all subjects were scanned in two sessions separated by a one-hour break and the whole cohort was completed within a two week period. We analyzed each session of 21 scans as an independent OoS study (**Fig. 5**) and then estimated intra-class correlation coefficients (ICCs) to assess the between-session or test-retest reliability of individual centile scores for four cerebrum tissue volumes³⁷. All ICCs were ~ 0.99 (**Fig. S4.3.1**).

Fig. S4.3.1. Test-retest reliability of out-of-sample centile scores for cerebrum tissue volumes. MRI data were collected in two separate scanning sessions from $N=21$ participants and each session was analysed as an independent out-of-sample study using GAMLSS. Scatterplots represent OoS centile scores for session 1 (y-axis) versus OoS centile scores for session 2 (x-axis) for each brain tissue volume, from left to right: GMV, WMV, sGMV, Ventricular CSF. Data points represent individual subject centile scores. Test-retest reliability was consistently very high (all ICCs > 0.99) for all cerebrum tissue volumes.

Second, we analysed the test-retest reliability of OoS centile scoring using MRI data on $N=72$ participants in the Healthy Brain Network (HBN) cohort²², which was not originally included in the reference dataset. The HBN cohort was designed to assess the influence of an alternate MRI data acquisition protocol, which included prospective motion correction²⁵ to improve quality and reliability of MRI. The study protocol included 2 sessions of scanning using a conventional MPRAGE sequence for T1-weighted data acquisition and another 2 sessions of scanning using an innovative, prospectively motion-corrected sequence, VNaV, for T1-weighted imaging²⁵. For all 72 individuals each session of each sequence was analysed as an OoS study (Fig. 5; S11.8 “Out-of-sample estimation”) and then we estimated ICCs as a measure of the test-retest reliability of individual centile scores for each brain tissue volume derived from each sequence (MPRAGE or VNaV). Test-retest reliability was uniformly high (ICCs > 0.95) for all OoS centile scores on all cerebrum tissue volumes estimated from both MPRAGE and VNaV sequences (Fig. S4.3.2). Reliability was incrementally higher for OoS centile scores derived from the VNaV sequence, under-scoring the importance of high quality data especially for OoS analysis of datasets with $N<100$. However, we note that this increased reliability of centile scoring was most likely driven by a comparably increased consistency of the raw volumes estimated by FreeSurfer (as also noted in the original paper describing the impact of prospective motion correction²²).

Fig. S4.3.2. Test-retest reliability of out-of-sample centile scores for cerebrum tissue volumes measured twice in the same N=72 participants using two T1-weighted sequences, MPRAGE and VNaV. Top row shows out-of-sample centile scores for session 1 (y-axis) versus out-of-sample centile scores for session 2 (x-axis) for cerebrum tissue volumes estimated from MPRAGE data, from left to right: GMV, WMV, sGMV, Ventricles. Bottom row shows out-of-sample centile scores for session 1 (y-axis) versus out-of-sample centile scores for session 2 (x-axis) for cerebrum tissue volumes estimated from VNaV data, from left to right: GMV, WMV, sGMV, Ventricles. In all plots, data points represent individual subject centile scores. Test-retest reliability was uniformly high (all ICCs > 0.95) and generally somewhat higher for volumetrics derived from prospectively motion-corrected data (VNaV).

Third, we assessed the test-retest reliability of OoS centile scoring using the Vietnam Era Twin Study of Ageing (VETSA) study cohort³⁵. VETSA is a longitudinal study following 1,200 twins from the Vietnam Era Twin Registry, which includes two technically identical MPRAGE acquisitions within the first (baseline) scanning session. Both these scans were processed with FreeSurfer 6.0.1 for all participants, then the two sets of scans were each analysed as an independent OoS study, and ICCs were estimated to assess the test-retest reliability of individual centile scores on all four cerebrum tissue volumes. Test-retest reliability of OoS centile scores was uniformly very high (all ICCs > 0.98) across all phenotypes, comparable to the high reliability of the uncensored volumetric data generated by FreeSurfer 6.0.1 (all ICCs > 0.95), and in line with the constraints on reliability expected from technical sources of noise³⁸ (Fig. S4.3.3).

Fig. S4.3.3. Test-retest reliability of out-of-sample centile scores for cerebrum tissue volumes measured twice in the same 1,200 participants (600 twin pairs). Scatterplots show out-of-sample centile scores for scan 1 (y-axis) versus out-of-sample centile scores for scan 2 (x-axis) for cerebrum tissue volumes estimated from MPRAGE data, from left to right: GMV, WMV, sGMV, Ventricles. Data points represent individual subject centile scores. Reliability was uniformly high across all phenotypes (ICCs > 0.95) and comparable to reliability of uncensored volumetric measurements from the same set of scans (data not shown).

4.4. Reliability of out-of-sample centile scoring across multiple versions of FreeSurfer

Knowing that a large majority (~95%) of primary studies in the reference dataset used one of a series of versions of FreeSurfer for image analysis, we also evaluated the impact of these incrementally different image analysis pipelines on reliability of OoS centile scores. To do this we

re-analysed a single dataset³⁶ repeatedly using 4 different versions of FreeSurfer (5.1, 5.3, 6.01, and 7.1). Each version of the processed dataset was treated as an independent OoS study for GAMLSS modeling and then we estimated ICCs between individual centile scores for each possible pair of FreeSurfer pipelines and for each of four cerebrum tissue volumes. This analysis demonstrated generally high within-subject reliability of OoS centiles across all four pipelines: ICCs for GMV=0.978, WMV=0.972, sGMV=0.816 and Ventricles=0.982 (**Fig. S4.4**). We noted that there was somewhat reduced reliability of subcortical grey matter volume in both raw and centiled data from FreeSurfer version 5.1 in comparison to later FreeSurfer versions. While the reasons for this are unclear, none of the studies included in the principal dataset were processed with FreeSurfer 5.1, or any version of FreeSurfer older than 5.3. Furthermore, we found the highest between-pipeline reliability for both raw volumetric data and centile scores derived from the two most recent versions of FreeSurfer, 6.0.1 and 7.1, suggesting that minor inconsistencies due to FreeSurfer pre-processing are becoming less problematic as this widely used software package incrementally evolves.

Fig. S4.4. Between-pipeline reliability of volumetric data and out-of-sample centile scores for four cerebrum tissue volumes measured in the same set of $N=1,468$ scans re-analysed using 4 different versions of FreeSurfer (5.1, 5.3, 6.01, and 7.1). Top row shows scatterplot matrices representing the correlations between raw volumetric data derived from each possible pair of FreeSurfer pipelines, from left to right: GMV, WMV, sGMV, Ventricles. Bottom row shows scatterplot matrices representing the correlations between out-of-sample centile scores derived from each possible pair of FreeSurfer pipelines, from left to right: GMV, WMV, sGMV, Ventricles. Intra-class correlations of out-of-sample centile scores and uncensored volumetric data, on average over all pairs of four pipelines, were generally high (GMV=0.978, WMV=0.972, sGMV=0.816 and Ventricles=0.982). Although the reliability of sGMV volumetrics and centile scores was somewhat lower due to discrepant measurements by the oldest version of FreeSurfer, v5.1, this version of FreeSurfer was not used to analyse any of the scans included in the reference dataset.

4.5. Effects of sample size on reliability of out-of-sample centile scores

To further assess the validity of the OoS estimates we generated ‘clones’ of existing datasets. Clones are resampled copies of studies included in the reference dataset used to estimate the study specific GAMLSS parameters, that are then treated as if they were “new” studies using the

methods for out-of-sample centile scoring. This allows us to compare the OoS estimates to a relative truth, i.e., from the original, non-cloned version of the study included in the reference dataset, we know what the GAMLSS parameters ‘truly’ are, and we have an estimation of their ‘true’ uncertainty from the bootstrap resampling distributions. Thus for a given study dataset, D_m , we generate a cloned copy D_1 , and if our approach is unbiased we expect the out-of-sample parameter estimates for D_1 to be equal to the in-sample parameters estimated for D_m , i.e., $\gamma_{,m}$ (representing the set of random effects estimated by in-sample analysis of the original study treated as part of the reference dataset) should approximate $\gamma_{,1}$ (representing the set of random effects estimated by OoS analysis of the cloned study treated as a new dataset): see **SI1.8 “Out-of-sample estimation”** and **Fig. S4.5**.

In other words, we validated the OoS estimation by simulating a “new” study with the same underlying distribution as one of the studies included in the reference dataset. Hence, we expect the OoS random-effect estimates for this ‘clone’ to agree with the in-sample random-effect estimates. More formally, we are comparing $\gamma = MLE_{\beta,\gamma}(D)$ and $\gamma^{clone} = MLE_{\gamma}(D_{clone}|\beta(D))$, where the clone is contained within the data, i.e., $D \cap D_{clone} = D_{clone}$; see **SI1.8 “Out-of-sample estimation”** for further details on OoS MLE estimation. As illustrated in **Fig. S4.5**, these simulations indicated good performance for the OoS approach for “new” study sizes greater than $N=100$ scans.

Fig. S4.5. Out-of-sample estimates of cloned study random-effect parameters compared to in-sample estimates of random-effect parameters in the original or non-cloned study. The plot shows random-effects estimated using the out-of-sample approach across a range of possible sample sizes for a “new” study, generated by taking subsets of the same cloned study with uncertainty intervals derived from the bootstrap replicates. The purple horizontal lines are the equivalent in-sample estimates of the random-effects parameters. We see that the out of sample estimates are somewhat unreliable below $N=100$ subjects, but with larger samples the out-of-sample estimates from the cloned data converge with the in-sample estimates from the original data for both μ -component and σ -component random effects.

Ref 2/2:

Second, the contributing samples are not necessarily population representative. This makes calculation of centile scores exceedingly hard to interpret.

The reference dataset on the basis of which we have modeled normative brain growth trajectories represents the most comprehensive aggregation of primary neuroimaging datasets published to date, representing >100,000 research participants from 30+ different countries. However, we agree with the reviewer that even such a large and diverse dataset is not fully population-representative. That will require larger amounts of primary MRI data to be collected in an epidemiologically-principled way to reflect socio-demographic and other factors which might

moderate brain growth trajectories and contribute to individual differences in centile scores. However, it is important to emphasise that the GAMLSS modeling framework is flexible and scalable, and the principles of brain growth charting (and the on-line tool for their practical implementation) can be generalised to incorporate and account for more population-representative reference datasets in future. Further notes on representativeness and diversity have been expanded in the discussion.

<<The following changes have been made to the the main text>>

Presently, even the current large and diverse dataset is not fully representative of the global population at all ages. For example, foetal, neonatal and mid-adulthood (30-40y) epochs were under-represented (**SI17-19**); and, as is also common in existing genetic datasets, ethnicity and geography were heavily biased towards European and North American populations. While our statistical modeling approach was designed to mitigate study- or site-specific bias in centile scores, further increasing diversity in MRI research will enable more population-representative normative trajectories^{46,47} that can be expected to improve the accuracy and strengthen the interpretation of centile scores in relation to demographically appropriate norms.

Ref 2/3:

Third, norms may become out-dated over time. In cognitive testing this is known as the Flynn Effect, and it forces test companies to renorm their tests ever ~10 years. It is unclear how up to date these norms are, or whether they will remain up to date. This isn't an argument against the norms, but it does underscore the need for front work on this topic.

As is the case for traditional growth charts, the reviewer is correct in noting that reference norms change over time and as the population changes. The main motivation behind the present work was to generate a comprehensive, flexible and scalable modelling framework that can adapt to new and updated data, including changing population demographics. As such we completely agree with the reviewer that “front work” is needed. We have done this in the form of providing a flexible modelling approach and an interactive, easily updated online tool. Dissemination of the online tool and our current statistical models is intended to widely engage the community in continuously on-going development and updating of the underlying reference datasets.

We now discuss this motivation more explicitly in the context of potential cohort effects: **SI6. “Cohort effects”** including new **Figs. S6.1 and S6.2**. In this new section of Supplementary Information, we also report additional analyses of potential cohort effects on centile scores in the NIH dataset, which collected MRI data over a 20 year period and multiple scanner upgrades. These results show no evidence of significant cohort effects on centile scores estimated from data acquired at different historical times.

Knowing that one technical factor that changes over time is the software used for brain MRI image analysis, we also investigated the effects of a series of different versions of the widely-used FreeSurfer software package that have been released over the 20-year period in which most of the primary datasets were collected and analysed; see **SI4.4 “Reliability of out-of-sample centile scoring across multiple versions of FreeSurfer”** including new **Fig. S4.4**. We found that out-of-sample centile scoring was robust to different versions of FreeSurfer, indicating that this potentially influential source of historical drifts in reference volumetric data distributions was not materially important.

<<The following changes have been made to the Supplemental Information>>

4.4. Reliability of out-of-sample centile scoring across multiple versions of FreeSurfer

Knowing that a large majority (~95%) of primary studies in the reference dataset used one of a series of versions of FreeSurfer for image analysis, we also evaluated the impact of these incrementally different image analysis pipelines on reliability of OoS centile scores. To do this we re-analysed a single dataset³⁶ repeatedly using 4 different versions of FreeSurfer (5.1, 5.3, 6.01, and 7.1). Each version of the processed dataset was treated as an independent OoS study for GAMLSS modeling and then we estimated ICCs between individual centile scores for each possible pair of FreeSurfer pipelines and for each of four cerebrum tissue volumes. This analysis demonstrated generally high within-subject reliability of OoS centiles across all four pipelines: ICCs for GMV=0.978, WMV=0.972, sGMV=0.816 and Ventricles=0.982 (**Fig. S4.4**). We noted that there was somewhat reduced reliability of subcortical grey matter volume in both raw and centiled data from FreeSurfer version 5.1 in comparison to later FreeSurfer versions. While the reasons for this are unclear, none of the studies included in the principal dataset were processed with FreeSurfer 5.1, or any version of FreeSurfer older than 5.3. Furthermore, we found the highest between-pipeline reliability for both raw volumetric data and centile scores derived from the two most recent versions of FreeSurfer, 6.0.1 and 7.1, suggesting that minor inconsistencies due to FreeSurfer pre-processing are becoming less problematic as this widely used software package incrementally evolves.

Fig. S4.4. Between-pipeline reliability of volumetric data and out-of-sample centile scores for four cerebrum tissue volumes measured in the same set of $N=1468$ scans re-analysed using 4 different versions of FreeSurfer (5.1, 5.3, 6.01, and 7.1). Top row shows scatterplot matrices representing the correlation between raw volumetric data derived from each possible pair of FreeSurfer pipelines, from left to right: GMV, WMV, sGMV, Ventricles. Bottom row shows scatterplot matrices representing the correlation between out-of-sample centile scores derived from each possible pair of FreeSurfer pipelines, from left to right: GMV, WMV, sGMV, Ventricles. Intra-class correlations of out-of-sample centile scores and uncensored volumetric data, on average over all pairs of four pipelines, were generally high (GMV=0.978, WMV=0.972,

sGMV=0.816 and Ventricles=0.982); although the reliability of sGMV volumetrics and centile scores was somewhat lower due to discrepant measurements by the oldest version of FreeSurfer, v5.1.

6. Cohort effects

As is the case for traditional growth charts, reference norms for brain charts may change over time, underscoring the need for “front work” on constructing normative reference models that are adaptive to future trends. Our choice of GAMLSS as the preferred modeling framework was in part motivated by its ability to provide a flexible and scalable basis that could support ongoing updates to the reference data. Likewise, our effort to share these models on an interactive web-platform (www.brainchart.io & <https://github.com/ucam-department-of-psychiatry/Lifespan>) was also motivated by the likely need for continuous updates to the reference dataset as and when more MRI data become available.

To assess the potential risk of cohort effects, or population norms shifting over historical time and biasing estimation of centile scores in future, we used a single (NIH) study already included in our aggregated dataset, which collected data from 1991 onwards in a constrained age range (5–25 years; N=1,468 scans). While MRI is a comparatively novel methodology (~30 years), it is possible that there may be systematic cohort effects within studies that have sampled individuals over prolonged periods of time⁴⁷, or between measurements aggregated in different age bins at different times. To quantitatively assess this possibility and the robustness of our procedures and results against such cohort effects, we analysed this NIH study containing longitudinal scans collected over two decades, from 1991 to 2011. We found no evidence for systematic variation of centile scores on any of the 4 cerebrum tissue volumes as a function of year-of-scanning or in relation to changes or upgrades to the scanner platform (**Fig. S6.1-2**).

Thus there was no clear evidence of cohort effects in one of the few large studies to have sustained scanning over a long period of time, and there was no evidence of measurement biases related to technical development of image analysis software that potentially could contribute to cohort effects in large aggregated MRI datasets. However, the ongoing technical development of MRI scanners and image analysis software, as well as the possibility of more general secular trends in brain growth over time, mean that the risk of cohort effects should nonetheless be iteratively re-evaluated as the currently available reference dataset continues to be updated in the future.

Fig. S6.1. Assessment of potential cohort effects based on date of scanning over two decades. The longitudinal study at the National Institutes of Health (NIH) contains $N=1,468$ longitudinal scans ($N=788$ subjects) collected across the age range 5–25 years and over the historical period 1991–2011. Scatterplots represent individual centile scores (y-axis), ordered by date of scanning (x-axis), for each of the four cerebrum tissue volumes (top four rows); and age at scan (y-axis) versus date of scanning (x-axis) (bottom row). Lines represent locally-weighted regression lines (LOESS regression) for qualitative analysis of possibly non-linear cohort effects on brain phenotypes or age at scanning. Filled circles denote baseline scans, empty circles denote follow-up scans in this longitudinal dataset; vertical lines indicate the timing of scanner upgrades over the course of the study (see also Fig. S6.2).

Fig. S6.2. Assessment of potential cohort effects related to scanner upgrades in the NIH longitudinal study. Centile scores for all four cerebrum tissue volumes estimated at baseline (time point 1) or two follow-up assessments (time points 2 and 3) were assigned to one of four epochs partitioned by the timing of upgrades to the 1.5T MRI scanner used for data collection. Box-violin plots show the distribution of centiles, and the range (whiskers) and 25th, 50th, and 75th percentiles of the centile distributions (boxes). Linear mixed effect modeling demonstrated no evidence of a significant effect ($t=-1.577$, $P=0.115$). This analysis was restricted to time points with $N > 100$ subjects.

Ref 2/4:

The tissue classes are extremely gross. Normally, MRI research capitalizes on the fine grain spatial nature of the imaging, e.g. in the form of Region of Interest (ROI) analyses. This is important because abnormal growth or atrophy in different regions can have tremendously different implications for whether, and what, functions are clinically affected. Clinicians regularly make these appraisals in individual evaluations. The current submission would seem to add greater quantification and precision (in contrast to clinical judgement) to such work, by allowing for the calculation of centile scores. But in order to calculate these scores, anatomical specificity is nearly entirely sacrificed. For example, it would be impossible to distinguish various forms of neurocognitive disorders of aging (e.g. frontotemporal dementia vs. Alzheimer's disease) without the spatial information that is being thrown out.

We thank the reviewer for their positive appraisal of the growth chart framework and the precision and quantitative value this may add to clinical assessment of brain scans. We note that, even at the whole brain level of anatomical resolution, these charts support case-control differentiation in terms of significantly atypical mean centile scores for groups of cases with mild cognitive impairment (MCI), Alzheimer's disease (AD) and schizophrenia (Fig. 4). We agree that more fine-grained anatomical resolution of MRI phenotypes could likely add further value to the clinical applications of brain growth charts. To demonstrate the flexibility of our modeling framework to adapt to a wide range of MRI phenotypes, we have provided extensive new analyses of global cortical metrics and regional volume measurements. These new results are now presented in the main text, and more extensively in supplemental information, as proof-of-concept that GAMLSS modeling based on available reference data can resolve normative trajectories of diverse and

relatively fine-grained MRI phenotypes. However, we have preferred not to report immediately the results of case-control comparisons for centile scores on regional volumetrics, on the grounds that to do so comprehensively and definitively would go beyond the scope of this principally normative paper and would be better communicated separately in a future paper distinctly focused on the clinical implications of regional brain growth charts.

- We have reported the normative life-span trajectories of bilateral cortical volumes for each of 34 distinct regions defined by the Desikan-Killiany parcellation template; see changes to main text, including a new section on **Extended MRI phenotypes** a new **Fig. 2** and revised **Fig. 3** (formerly Fig. 2). We also provide further details in **SI8: “Regional specificity”**, including new **Figs. S8.1.1-8.2.2**.

<<The following changes have been made to the main text>>

Extended brain MRI phenotypes

To extend the scope of brain charts beyond the four cerebrum tissue volumes, we used the same GAMLSS modeling approach to estimate normative trajectories for additional MRI phenotypes including other geometric properties at a similar scale (mean cortical thickness and total surface area) and regional volume at each of 34 cortical areas²⁵ (**Fig.2, SI7-9, ST1-2**). We found, as expected, that total surface area closely tracked the development of total cerebrum volume (TCV) across the lifespan (**Fig.2A**), with both metrics peaking at ~11-12 years (SA 10.97_{CI-Bootstrap:10.42-11.51}; TCV 12.5_{CI-Bootstrap:12.14-12.89}). In contrast, cortical thickness peaked distinctly early at 1.7_{CI-Bootstrap:1.3-2.1} years, which reconciles prior observations that cortical thickness increases during the perinatal period²⁶ and declines during later development²⁷. We also found evidence for regional variability in volumetric neurodevelopmental trajectories. Compared to GMV’s peak at 5.9 years, the age of peak regional volume varied considerably – from approximately 2 to 10 years – across 34 cortical areas. Primary sensory regions reached peak volume earliest, and fronto-temporal association cortical areas matured later (**Fig.2B; SI8**). In general, earlier maturing ventral-caudal regions also showed faster post-peak declines in cortical volume, and later maturing dorsal-rostral regions showed slower post-peak declines (**Fig.2B; SI8.2**). Notably, this spatial pattern recapitulates a gradient from sensory-to-association cortex that has been previously associated with multiple aspects of brain structure and function²⁸.

Fig 2. Extended global and regional cortical geometric phenotypes. A | Trajectories for total cerebrum volume (TCV; left column), total surface area (SA; middle column), and mean cortical thickness (CT; right column). For each global cortical geometric MRI phenotype, the following sex-stratified results are shown as a function of age over the life-span, from top to bottom rows: raw, non-censored data, population trajectories of the median (with 2.5% and 97% centiles; dotted lines), between-subject variance (and 95% confidence intervals), and rate-of-growth (the first derivatives of the median trajectory and 95% confidence interval). All trajectories are plotted on log-scaled age (x-axis) and y-axes are scaled in units of the corresponding MRI metrics (10,000 mm³ for TCV, 10,000 mm² for SA and mm for CT). B | Regional variability of cortical volume trajectories for 34 bilateral brain regions as defined in the Desikan-Killiany parcellation²⁵, averaged across sex (see also **S17-8** for details). From top to bottom panels: cortical map of age at peak regional volume (range 2-10 years); cortical map of age at peak regional volume relative to age at peak GMV (5.9 years), highlighting regions that peak earlier (blue) or later (red) than GMV; and illustrative trajectories for the earliest peaking region (superior parietal lobe) and the latest peaking region (insula), showing the range of regional variability. Regional volume peaks are denoted as dotted vertical lines either side of the global peak denoted as a dashed vertical line in the bottom panel. Left hand y-axis on the bottom panel refers to the earliest peak, the right hand y-axis refers to the latest peak, and both are in units of 10,000 mm³.

In: *Developmental milestones*

The velocity of mean cortical thickness peaked even earlier, in the prenatal period at -0.38_{CI} . Bootstrap: -0.4 to -0.34 years (relative to birth), corresponding approximately to the second half of pregnancy. This very early peak in cortical thickness velocity has not been reported previously, probably due to the unprecedented aggregation of foetal MRI data allowing precise estimation of early human brain growth in the current study^{23,32}.

This epoch of GMV:WMV differentiation encompasses dynamic changes in brain metabolites³⁸ (0-3 postnatal months), resting metabolic rate (RMR; minimum=7 months, maximum=4.2 years)³⁹, the typical period of acquisition of motor capabilities and other early paediatric milestones⁴⁰, interneuron migration, and the most rapid change in TCV (**Fig.3**).

Fig. 3. Neurodevelopmental milestones. Top panel: A graphical summary of the normative trajectories of the median, i.e., 50th centile, for each global MRI phenotype, and key developmental milestones, as a function of age (log-scaled). Circles depict the peak rate-of-growth milestones for each phenotype (defined by the maxima of the first derivatives of the median trajectories; see Fig. 1E). Triangles depict the peak volume of each phenotype (defined by the maxima of the median trajectories), definition of GMV:WMV differentiation is detailed in S19.1. Bottom panel: A graphical summary of additional MRI and non-MRI developmental stages and milestones. From top to bottom: blue shaded boxes denote the age-range of incidence for each of the major clinical disorders represented in the MRI dataset; black boxes denote the age at which these conditions are generally diagnosed as derived from literature⁴¹ (Online Methods); brown lines represent the normative intervals for developmental milestones derived from non-MRI data, based on previous literature and averaged across males and females (Online Methods); grey bars depict age ranges for existing (WHO and CDC) growth charts of anthropometric and ultrasonographic variables. Across both panels, light grey vertical lines delimit lifespan epochs (labelled above the top panel) previously defined by neurobiological criteria⁴². Abbreviations: resting metabolic rate (RMR), Alzheimer's disease (AD), attention deficit hyperactivity disorder (ADHD), anxiety or phobic disorders (ANX), autism spectrum disorder (ASD, including high-risk individuals with confirmed diagnosis at a later age), major depressive disorder (MDD), bipolar disorder (BD), and schizophrenia (SCZ).

In: Discussion

We have focused primarily on charts of global brain phenotypes, which were measurable in the largest aggregated sample over the widest age range, with the fewest methodological, theoretical and data sharing constraints. However, we have also provided proof-of-concept brain charts for regional grey matter volumetrics, demonstrating plausible heterochronicity of cortical patterning, and illustrating the generalisability of this approach to a more diverse range of fine-grained MRI phenotypes. As ongoing and future efforts provide increasing amounts of high quality MRI data, we predict an iterative process of improved brain charts for the human lifespan, potentially

representing multi-modal MRI phenotypes and enabling out-of-sample centile scoring of smaller samples or individual scans. In the hope of facilitating progress in this direction, we have provided interactive tools to explore these statistical models and to derive normalised centile scores for new datasets across the lifespan at www.brainchart.io

<<The following changes have been made to the Supplemental Information>>

8. Regional cortical volumetric trajectories and milestones

To analyse trajectories and milestones of brain development with finer-grained anatomical resolution, we extracted volumetric information from 34 bilateral regions in the Desikan-Killiany parcellation⁴⁸ for a subset of ~65,000 unique individuals (depending on the region) from birth until 100 years (**ST1.9-1.42**). Since we expected data quality to have a greater impact on the accuracy of regional volumetrics, compared to the minor impact of data quality demonstrated for cerebrum tissue volumes (see **SI2 “Quality control”**), we only included quality controlled scans with EI<217 in these analyses, or scans that had undergone prior visual inspection. We applied exactly the same modelling pipeline to these regional volumetric phenotypes as previously applied to cerebrum tissue volumes. Briefly, we first specified the optimal combination of fractional polynomials in each term of the model using BIC, then fitted the optimal model to the sex-stratified data and to 1,000 bootstrapped resamples of the original data, and finally plotted the trajectories for the median and between-subject variability (with confidence intervals) of each regional volume. This work extends previous work on developmental trajectories of brain regional volumes in several important ways. Most prominently, for the first time, these trajectories encompass the full age-range of the lifespan, including the earliest period of development before postnatal year 2. There is evidently considerable variation between cortical regions in their developmental trajectories, but all regions show peak volume, and peak rate-of-growth of volume, in the first decade, which is compatible with our results for global cortical volume estimated in a larger and more inclusive sample.

8.1. Charting development of regional volumes

Fig. S8.1.1. Raw regional volumetric data across the lifespan for 34 bilateral brain regions as defined in the Desikan-Killiany parcellation⁴⁸ (mm^3). These data are analogous to the raw data depicted in Figs. 1 and 2 for cerebrum tissue volumes and other global cortical metrics (SA and CT), respectively. Demographics for the QC'd sample available for estimation of each regional volume are provided in ST1.9-1.42.

Fig. S8.1.2. Normative trajectories of median regional volumes (and confidence intervals) across the lifespan for 34 bilateral brain regions as defined in the Desikan-Killiany parcellation⁴⁸ (mm^3). Dotted lines indicate the 97.5% and 2.5% centile lines. These trajectories were fitted to the raw data in Fig. S8.1.1 using the same GAMLSS model used for estimation of tissue volume trajectories, as shown in Fig. 1 and Fig. 2 of the main text. Further details on milestones (age at peak volume and age at maximum rate-of-growth of volume) are provided for each region in ST2.2 and S18.2 “Regional volumetric milestones”.

Fig. S8.1.3. Normative trajectories of between-subject variation of regional volumes (and confidence intervals) across the lifespan for 34 bilateral cortical regions as defined in the Desikan-Killiany parcellation⁴⁸ (mm³). Shaded areas represent the 95% confidence interval defined by 1,000 bootstrapped resamples of the original data, as identically done for estimation of between-subject variation in global brain phenotypes (Figs. 1 and 2 in the main text). Further details on milestones (age at peak variation and age at maximum rate-of-growth of variation) are provided for each region in ST2.2.

Fig. S8.1.4. Estimated rates of change in regional volumes across the lifespan (first derivatives of the median trajectories) for 34 bilateral brain regions as defined in the Desikan-Killiany parcellation⁴⁸. Shaded areas represent the 95% confidence interval defined by 1,000 bootstrapped resamples of the original data, as identically done for estimation of rate-of-growth curves for global brain phenotypes (Figs. 1 and 2 in the main text). The numbers displayed at the top of each chart denote age at peak rate-of-growth for each regional volume and the solid horizontal line at $y=0$ indicates the age at

which regional volumes stop growing and start to shrink. Further details on milestones (age at peak volume and age at maximum rate-of-growth of volume) are provided for each region in **ST2.2**.

Fig. S8.1.5. GAMLSS estimated confidence interval for model fits to regional volumes across the lifespan for 34 bilateral brain regions as defined in the Desikan-Kiliany parcellation⁴⁸. Shaded areas represent the 95% confidence interval estimated by 1,000 bootstraps. These results are analogous to the sensitivity analysis depicted in **SI3.2.2** and show that for most regions the confidence intervals are extremely narrow, i.e., it barely extends beyond the thickness of the lines. However, in entorhinal cortex, frontal pole and temporal pole the bootstrapped variability is considerably greater in early development, possibly indicating marginal quality of data or cortical surface reconstruction for these regions in this age range.

8.2. Regional volumetric milestones

We also estimated the developmental milestones of each region in terms of age at peak volume or peak between-subject variation, and age at peak rate-of-growth in volume or between-subject variation. **Fig. S8.2.1** shows the regions ordered by age at peak median volume alongside the bootstrapped confidence intervals of those milestones. The shaded grey bar shows the age at peak total cortical grey matter volume, with the width of the bar indicating the 95% confidence interval for that milestone. In the corresponding figure of the main text (**Fig. 2**), we excluded outlying data points, defined as age at peak volumes more than 2 median absolute deviations away from the median of the regional distribution of age at peak volume. This removed the 3 regions with the highest between-subject variability, especially in early development (entorhinal cortex, temporal and frontal poles). Perhaps unsurprisingly, both the temporal and frontal poles are regions with notoriously questionable signal quality⁴⁹. The entorhinal cortex is the smallest cortical region defined by the Desikan-Kiliany atlas and is often missing in parcellated foetal and neonatal MRI data for that reason. These results further underscore the need for conducting quality control on scanning data prior to estimation of brain charts at regional resolution. Further details on milestones (age at peak volume and age at maximum rate-of-growth of volume) are provided for each region in **ST2.2**.

Fig. S8.2.1. Milestones for development of regional volumes estimated from the first derivatives of the trajectories of median volume and between-subject variation for each of 34 cortical regions defined by the Desikan-Killiany template. Each point-range plot shows, from left to right, the age at peak volume, the age at peak between-subject variation, and the age at maximum rate-of-growth in volume. In each case, median milestones are shown in the context of their 95% confidence intervals, which are not always visible for narrow intervals. The shaded grey area in each panel shows the median and 95% confidence interval for the corresponding milestone for total cortical grey matter volume.

To contextualize the spatial distribution of the regional volume peaks, we compared the age at peak volume to the x-, y-, and z-coordinates of the centroids of each region-of-interest in the Desikan-Killiany cortical parcellation. We observed a relatively wide distribution of age at peak regional volume, centred around the age of peak total cortical grey matter volume (grey dashed line in **Fig. S8.2.2**). Moreover, there was a clear trend for rostral and dorsal regions to have later peak volumes compared to caudal and ventral regions (**Fig. S8.2.2**). Regions in the cingulate and frontal cortices, which span greater distances (especially in rostral-caudal and dorsal-ventral dimensions), had a wider range of age peaks.

Fig. S8.2.2. Relative timing of regional volume peak milestones, highlighting spatial gradients in timing of peak volumes. Scatterplots show the relationship between age of peak volume for each region of the Desikan-Killiany parcellation (x-axes) versus x (left), y (middle), or z (right) coordinates in MNI space (y-axes). Coordinates are based on the left hemisphere, thus the interpretation (from negative to positive) is: x=lateral-to-medial, y=caudal-to-rostral, z=ventral-to-dorsal. Spearman's r was computed for each scatterplot, represented by black lines: x-coordinates were not significantly correlated with age at peak volume, $r=-0.21$, $P=0.26$; y-coordinates were positively correlated with age at peak volume, $r=0.42$, $P=0.02$; and z-coordinates were negatively correlated with age at peak volume, $r=-0.50$, $P=0.004$. Labels represent the most extreme (top two and bottom two) region peaks relative to peak total cortical grey matter volume. Grey dashed lines represent the age at peak total cortical grey matter volume.

Ref 2/5:

Calculating centile scores may not be of much use without a clear understanding of the functional implications of different scores. Having the location on the distribution is not enough without understanding the clinical and functional correlates of those locations. As per the above point, this may be a difficult endeavor at the low level of spatial resolution provided.

This is indeed a very interesting point. We now provide several additional analyses in supplemental information to illustrate the functional implications of centile scores, and to demonstrate clinical and functional correlations with centile scores. Specifically, we show that:

1. Centile scores have clinical significance as evidenced by significant case-control differences in mean centiles across a range of disorders: main text Fig. 3, and S10-11.
2. Low centile scores, e.g., below the 5% centile, on brain tissue volumes and global cortical metrics were generically associated with significantly increased risk (approximately 2-fold increase in odds ratio) for clinical disorder, over all diagnostic classes of disorder: see S11.1 "Sliding window analysis of cross-disorder discriminability", including new Fig. S11.1.
3. Centile scores were robustly associated with birth weight and gestational age: S12 "Associations of birth weight and gestational duration with centile scores on cerebrum tissue volumes".

- Centile scoring showed increased heritability estimates compared to the original raw values: **SI13: “Twin-based heritability of centile scores”**.
- Centile scores measured longitudinally by repeated scanning of the same participant were generally stable in healthy controls but showed clinically meaningful within-subject variation in individuals who transitioned between diagnostic classes, e.g., from being designated a healthy control to being diagnosed as a case of Alzheimer’s disease, over the course of repeated MRI scanning: **SI14** including **Fig. S14.4.1**.

In the revised main text, we also now summarise and review the data related to this important question of functional and clinical correlations with centile scores.

<<The following changes have been made to the main text>>

In *Individualised centile scores in clinical samples*

The clinical diversity of the aggregated dataset allowed us to comprehensively investigate case-control differences in individually-specific centile scores. Relative to the control group (CN), there were highly significant differences in centile scores across large (N>500) diagnostic groups of multiple disorders (**Fig.4A; SI10, ST3**). The pattern of these centile differences varied across tissue types and disorders. Clinical differences in cortical thickness and surface area generally followed the same trend as volume differences (**SI10**).

Fig. 4. Case-control differences and heritability of centile scores. A | Centile score distributions for each diagnostic category of clinical cases relative to the control group median (depicted as a horizontal black line). The median deviation of centile scores in each diagnostic category is overlaid as a lollipop plot (white line with circle corresponding to the median centile score for each group of cases). Pairwise tests for significance were based on Monte-Carlo resampling (10,000 permutations) and P-values were adjusted for multiple comparisons using the Benjamini-Hochberg False Discovery Rate (FDR) correction across all possible case-control differences. Only significant deviations from the control group median (with corrected $P < 0.001$) are highlighted with an asterisk. For a complete overview of all pairwise comparisons, see **SI10**

and **ST3**. Groups are ordered by their multivariate distance from the control group (see panel C and **SI10.3**). B | The centile Mahalanobis distance (CMD) is a summary metric of multivariate deviation that quantifies the aggregate atypicality of an individual scan in terms of all MRI phenotypes. The schematic shows segmentation of four cerebrum tissue volumes, followed by estimation of univariate centile scores, leading to the orthogonal projection of a single subject (Sub_x) onto the four principal components of the control group (CN; coloured axes and arrows): the CMD for Sub_x is then the sum of its distances from the CN group mean on all 4 dimensions of the multivariate space. C | Probability density plots of CMD across disorders. Vertical black line depicts the median CMD of the control group. Asterisks indicate an FDR-corrected significant difference from the CN group ($P < 0.001$). D | Heritability of raw volumetric phenotypes and their centile scores across two twin studies (ABCD and HCP). All dots have error bars for the standard error, but in some cases these are too narrow to be observed. Abbreviations: control (CN), Alzheimer's disease (AD), attention deficit hyperactivity disorder (ADHD), anxiety or phobia (ANX), autism spectrum disorder (ASD), mild cognitive impairment (MCI), major depressive disorder (MDD), schizophrenia (SCZ); grey matter volume (GMV), subcortical grey matter volume (sGMV), white matter volume (WMV), centile Mahalanobis distance (CMD).

In Discussion:

We have aggregated the largest neuroimaging dataset to date to modernise the concept of growth charts for mapping typical and atypical human brain development and ageing. The ~100 year age range enabled the delineation of milestones and critical periods in brain maturation, revealing an early growth epoch across its constituent tissue classes -- starting before 17 post-conception weeks, when the brain is at ~10% of its overall size and ending at ~80% by age 3. Individual centile scores benchmarked by normative neurodevelopmental trajectories were significantly associated with neuropsychiatric disorders as well as with individual differences in birth outcomes and fluid intelligence (**SI5.2** and **SI12**). Furthermore, imaging-genetics studies⁴⁴ may benefit from the increased heritability of centile scores compared to raw volumetric data (**SI13**). Perhaps most importantly, GAMLSS modeling enabled harmonisation across technically diverse studies (**SI5**), and thus leveraged the potential power of aggregating MRI datasets at scale.

The current results also bode well for future progress towards individualised prediction⁴⁵. By providing an age- and sex-normalised metric, centile scores enable trans-diagnostic comparisons between disorders that emerge at different stages of the lifespan (**SI10-11**). The generally high stability of centile scores across longitudinal measurements also enabled assessment of documented changes in diagnosis such as the transition from MCI to AD (**SI14**), which provides one example of how centile scoring could be clinically useful in quantitatively predicting or diagnosing progressive neurodegenerative disorders. The analogy to paediatric growth charts is not meant to imply a predetermined or immediate application for brain charts in a typical clinical setting. However, our provision of appropriate normative growth charts and on-line tools creates an opportunity to quantify atypical brain structure, precisely benchmarked against age- and sex-typical distributions, and thus to enhance diagnostic yield from clinical scans as well as neuroimaging research studies.

<<The following section has been added to the Supplementary Information>>

11.1. Sliding window analyses of cross-disorder discriminability

We computed odds ratios for clinical disorders using a sliding window across the full range of centile scores for cerebrum tissue volumes (window size=0.1, increment size=0.05). Major diagnostic categories (as in **Fig. 4**) were combined to form one group of all diagnosed cases (DX or non-CN) and compared to healthy controls (CN) to estimate the odds ratio of being diagnosed with any clinical disorder. These analyses indicated that lower centile scores, especially <5%, on

cerebrum tissue volumes, cortical surface area and cortical thickness were all significantly over-represented in individuals with neuropsychiatric disorders (**Fig. S11.1**). This means that a lower centile score on any or all of these brain MRI metrics was associated with a higher probability of any clinical disorder. It will be important to discover if low centile scores on brain MRI metrics are predictive of later clinical outcomes, meaning that brain charts could be used in future as paediatric growth charts are used now, to raise levels of clinical concern proportionately, rather than to make a specific diagnosis.

Fig. S11.1. Brain MRI centile scores are related to the probability of any clinical disorder. The odds ratio for clinical disorder (versus healthy control) is plotted on the y-axis of both panels; positive OR indicates greater risk of disorder. Centile scores by GAMLSS modeling are plotted (on the x-axis) for global brain MRI phenotypes: left panel, 4 cerebrum tissue volumes; right panel, total cerebrum volume, cortical surface area, and mean cortical thickness. Odds ratios were computed using a sliding window across centiles (window size=0.1, increment size=0.05). Diagnostic categories in **Fig. 4** were combined (i.e., binarised to make any diagnosis, or 'dx' vs. 'cn') to estimate the odds ratio of being in any clinical cohort. Scans with lower centile scores on all phenotypes, especially centiles <5%, have increased odds ratio for all clinical disorders.

12. Associations of birth weight and gestational duration with centile scores on cerebrum tissue volumes

To examine the effects of early life stress on centile scores, we examined 5 independent samples across the lifespan with self-reported gestational age at birth and/or birth weight (dHCP, neonatal; UNC, neonatal and early infancy/childhood; ABCD, late childhood; NIH, childhood/adolescence/young adulthood; UKB, mid-late adulthood). Average centile scores on all four cerebrum tissue volumes were significantly related to multiple metrics of premature birth across datasets (gestational age at birth, $t = 13.164$, $P < 2e-16$; birth weight, $t = 36.395$, $P < 2e-16$). This corroborates previous work indicating the ability to capture relationships between early life factors such as birth weight and brain volumetrics measured several decades later⁷².

Fig. S12. Relationships between centile scores on cerebrum tissue volumes and birth weight (left panel) and gestational age at birth (right panel) for each of 5 primary studies with relevant data available. Centile-normalised z-scores were computed for each phenotype in each individual study and then averaged across phenotypes to compute a mean centile z-score for each subject. The black dashed lines represent the relationships between mean centile scores and birth weight or gestational age at birth estimated by a linear mixed-effects model: for gestational age at birth, $t = 13.164$, $P < 2e-16$; for birth weight, $t = 36.395$, $P < 2e-16$. The black dotted line in the right panel denotes the commonly-used threshold for defining premature birth at 37 weeks post-conception.

13. Twin-based heritability of centile scores

We examined the heritability of centile scores on cerebrum tissue volumes, leveraging available data of monozygotic (MZ) and dizygotic (DZ) twins in the ABCD cohort of adolescents ($N=297$ MZ, $N=400$ DZ pairs), and in the HCP cohort of adults ($N=138$ MZ, $N=78$ DZ pairs). For both cohorts, zygosity was previously determined based on parental and/or self endorsement, and genetic kinship^{73–75}. Heritability was estimated using Cholesky decomposition, allowing 'ACE' partitioning of the phenotypic variance into additive genetic (A), common environmental (C), and unique environmental (E) components, as implemented in the *umx* R package⁷⁶. As shown in **Fig. 4**, we found greater heritability of centile normalised scores compared to the respective raw, non-centiled volumetric phenotypes.

14.4 Longitudinal centile score changes and diagnostic progression

Similarly to paediatric growth charts, further value from having age-appropriate standardised reference curves will likely come from the ability to more reliably detect atypical longitudinal changes in brain changes within individuals. As an example of this approach, we have tracked centile scores in longitudinal (repeated) cerebrum tissue volumes for a large cohort of older individuals, some of whom transitioned between diagnostic categories during the period of

longitudinal follow-up from CN to MCI (CN → MCI), from CN to AD (CN → AD), or from MCI to AD (MCI → AD). Interestingly, in contrast to the lower within-subject variability (IQR) of cases compared to healthy controls in general, there was a reverse trend of increased within-subject variation in cerebrum tissue volumes (especially GMV and Ventricles) in the subset of cases that changed diagnostic status. Specifically, there was faster than normal decrease of grey matter volume, and faster than normal increase of ventricular CSF volume, among participants who transitioned from CN or MCI to AD over the course of repeated scanning (Fig. S14.4.1 and ST6.1-6.7).

Analysis of within-subject changes in centile scores focused on individuals with the most frequently observed diagnostic transitions, all in the direction of greater severity or disability: from CN to MCI, from CN to AD and from MCI to AD (ST6). The longitudinal change in centile scores occurred in the same direction as predicted by the cross-sectional case-control differences (compare Fig. 4A and SI10 “Clinical applications of centile scores”). We rescaled the longitudinal data to generate a group-level trajectory for each transition (CNI → AD, CN → MCI, and MCI → AD) using linear mixed effects models. As shown in Fig. S14.4.1, all clinical transitions were associated with significantly increased rates of cortical and subcortical grey matter loss, and ventricular CSF volume expansion – both reflected by decreases in centile scores. Because the significant age-related changes expected in healthy older individuals are incorporated into the reference norms, centile scores provide a clear indication of a change in trajectory for individuals with neurodegenerative disease.

Fig. S14.4.1. Longitudinal changes in centile scores are associated with diagnostic transitions between the groups of healthy controls (CN), mild cognitive impairment (MCI), and Alzheimer’s disease (AD). A | Shows the within-subject changes in centile scores for CN→MCI, CN→AD, and MCI→AD, with the dotted black lines showing the median slope for all controls that had longitudinal measurements and the solid black lines showing the median slope for all controls from the datasets that contributed to the diagnostic change group. B | Shows the model fixed effects standardised coefficients (e.g., model fixed effects divided by two standard deviations), to denote the slope differences in longitudinal changes in centile scores between the groups. Asterisks indicate the level of uncorrected significance (* is $P<0.05$, ** is $P<0.01$, *** is $P<0.001$) as tested with a linear mixed model restricted maximum likelihood (REML) fit that included a subject-level random effect. These results show that for both GMV and Ventricular CSF the rate-of-change in centile scores is significantly greater in individuals undergoing a clinically documented transition (from less to more severe diagnostic categories).

Ref 2/6:

The cross-sectional nature of the data and potential differences in protocols across individuals or cohorts of different ages, prevents strong inferences regarding

development and aging. Are there age differences in motion that may bias estimates? Are there period, or cohort effects in the data that would suggest that these trends plotted are not indices of “velocity” of growth or shrinkage within person, but instead differences associated with year of birth or historical time of imaging assessment?

We agree with the reviewer’s general comment that between-study differences, and the largely cross-sectional datasets currently available, are both significant challenges. We have already addressed some of these key points in response to the reviewer’s related first comment (Ref 2/1) and in the following new sections of supplementary information: **SI5.1 “Modeling of between-site heterogeneity by GAMLSS: conceptual considerations in comparison to ComBAT batch-correction”** and **SI5.2 “Modeling of between-site heterogeneity by GAMLSS: empirical evaluation compared to ComBAT”**. We make the following observations more specifically in response to this point:

- We agree it is conceivable that head-motion could impact image quality in an age-related way. The primary studies constituting the reference dataset provide limited data on head motion but we have extensively analysed image quality (quantified by the Euler index; see **SI2 “Quality control”**) and found no significant impact of scan quality on model estimation. In addition, we specifically evaluated whether age impacted scan quality (see **SI2.1 “Euler Index filtering”** and **SFig. 2.1.2**). We found no evidence for a linear relationship between age and Euler Index (EI) ($t = -1.244$, $P = 0.213$) when accounting for the same variables as included in the principal GAMLSS framework, i.e., age and study-specific random effects.
- We extensively evaluated bias and stability of out-of-sample estimation in a new supplementary section (**SI4 “Out-of-sample centile scoring: bias, stability and reliability”**), including an overview of age-related stability in centile estimation. While some early-life studies showed increased variability of centile scores, we note that this is in line with what is expected during a period of increased variability early in life, and that increased variability in general was more influenced by sample-size rather than age (see also **SI4.5 “Effects of sample size on reliability of out-of-sample centile scores”**).
- We have also reported additional analysis highlighting improved test-retest reliability of centile scores on volumetrics from MRI scans that had been collected with prospective correction for head motion: see **SI 4.3** including new **Figs S4.3.1-4.3.3**.
- The possibility of cohort effects is an interesting question especially in the context of MRI being a relatively new and evolving technique with historical changes in scanner specification, or image analysis software, that could influence the statistics of MRI phenotypes. We now provide additional discussion of cohort effects, with new analysis of potential cohort effects indexed by year of scanning, and scanner upgrade intervals, in the long-running NIH longitudinal dataset: see **SI6 “Cohort effects”** including new **Figs. S6.1** and **S6.2** for details. We have also reported new analysis of potential cohort effects due to serial releases of different versions of FreeSurfer image analysis software: see **SI4.4 “Reliability of out-of-sample centile scoring across multiple versions of FreeSurfer”**. Neither of these additional sensitivity analyses raised material concerns about the risk of cohort effects in whole brain tissue volumes.
- The reviewer is correct in noting that the trajectories generated and described in our main text are not ‘indices of “velocity” of growth or shrinkage within person’. Instead, they are the 50% (median) and other centiles of normative growth curves estimated on the basis of all the scans in the reference dataset. But these normative trajectories can provide a

batch-corrected, sex-stratified and age-appropriate benchmark for characterising longitudinal changes in single subjects: see SI14 “Longitudinal centiles”.

<<The following changes have been made to the Supplementary Information>>

In SI2.1 “Euler Index filtering”

To assess whether there were any age-related differences in motion that could influence model estimation we evaluated the linear effect of age (in years) on EI in healthy controls in the reference dataset used to estimate normative lifespan trajectories. Using linear regression stratified by sex and accounting for study-specific random effects, we found no evidence for an age-related bias in image quality as assessed with EI ($t = -1.244$, $P = 0.213$). **Fig. S2.1.2** shows the median and standard deviation of age and EI and highlights the top 10 studies with the highest median EI.

Fig. S2.1.2 Age-related variation in image quality measured by the Euler index in female (left panel) and male (right panel) control subjects. Median age (in years) and median EI are shown per study with cross-hairs indicating the standard deviations for age and EI per study. In red the top ten studies with the highest median EI are highlighted. There is no significant relationship between image quality and age at scanning.

4. Out-of-sample centile scoring: bias, stability and reliability

4.1. Bias of out-of-sample centile scores: leave-one-study-out analyses for 100 studies

To further evaluate the robustness and consistency of centile scoring of OoS MRI data that were not included in the reference dataset used to estimate population trajectories, we performed a comprehensive series of leave-one-study-out (LOSO) analyses. For each one of the 100 studies in the reference dataset, we removed the study from the reference dataset, re-fitted the GAMLSS model to the remaining dataset of 99 studies, computed the OoS centile scores for the excluded study, and compared the OoS centile scores to the in-sample centile scores computed for the same study from the complete dataset including all 100 studies. **ST7.1-7.4** lists the correlations between OoS and in-sample centile scores for all 4 cerebrum tissue volumes in each of 100 primary studies. Overall, we found very high levels of correlation ($r \sim 0.99$) for almost all studies, indicating that centile scores can be estimated accurately for most studies even if they were not included in the reference dataset used to define population norms. Correlations between OoS and in-sample centile scores were lower than $r = 0.99$ for only 3 out of 100 studies in the reference dataset: namely, the FinnBrain ($r = 0.93$), UCSD ($r = 0.96$) and NIHPD ($r = 0.95$) studies. These

studies were characterised by relatively small sample size, foetal or early postnatal age range of participants, or idiosyncratic processing pipelines.

4.2. Stability of out-of-sample centile scoring: bootstrapped LOSO analyses for 100 studies

In addition, we tested the reliability of OoS centile scores for each individual participant by bootstrapping. Specifically, for each LOSO sample, bootstrapped model parameters were generated (see **SI3.2.2 “Bootstrap analysis”**), resulting in 1,000 bootstrapped models with maximum likelihood estimated parameters for each bootstrap iteration of each left-out study. From this we obtained a bootstrapped distribution of out-of-sample centile scores for each individual subject in each individual iteration of left-out studies, thus providing a stability assessment in the form of the standard deviation of individual OoS centile scores across 1,000 bootstrap iterations. Across the datasets included in the model, we found that the average standard deviation of (bootstrapped) OoS centiles was 0.014, which is well below the level of within-subject longitudinal variation (see **Fig. S4.2.1** and **SI14 “Longitudinal centiles”**). Furthermore, we found increased standard deviation of OoS centile scores for datasets with comparatively small sample sizes (e.g., the OpenPain cohorts, Cambridge foetal Testosterone and CHILD studies; see **Fig. S4.2.2**). OoS centile scores were also more variable for datasets that had a more unique combination of age range, acquisition and processing pipelines (e.g., FinnBrain, IBIS and HBN; see **Fig. S4.2.2**). These observations reinforce the recommendation -- see main text, ‘**Out-of-sample centile scoring of “new” MRI data**’ -- that OoS centile scoring is reliable for studies comprising $N > 100$ scans. It was also notable that the reliability of OoS centile scores was weakly correlated with data quality as quantified by the Euler index (EI). So studies with higher EI²³, indicating poorer image quality, tended to have higher variability of bootstrapped OoS centile scores (Pearson’s r for all 4 cerebrum tissue volumes: GMV=0.05, WMV =0.11, sGMV =0.14, and Ventricular volume = 0.13). These results were not substantially different when the whole set of analyses was repeated without including scans with $EI > 217$. We conclude that OoS estimation of centile scores is generally reliable at the level of individual scans, and (as expected) reliability is greater for higher quality scans.

Fig. S4.2.1. Stability of out-of-sample centile scores for four cerebrum tissue volumes when each of 100 studies was excluded from the reference dataset before bootstrapping. The standard deviation of bootstrapped centile scores (y-axis) is plotted for each study (x-axis) for each phenotype, from top to bottom panels: total cortical grey matter volume, total cortical white matter volume, subcortical grey matter volume, and ventricular volume. Each study- and phenotype-specific boxplot is coloured according to log sample size. For each study, we estimated the normative model leaving that study out of the reference dataset and repeated this procedure after iteratively bootstrapping the reference dataset 1,000 times. This procedure allowed us to summarise the reliability of the out-of-sample estimates of centile scores in terms of the standard deviation of the 1,000 centile scores generated for each bootstrapped resampling of the reference dataset. Studies are ordered by median standard deviation of out-of-sample centile scores (small to large) indicating that scans are reliably assigned centile scores with the out-of-sample approach.

Fig. S4.2.2. Stability of out-of-sample centile scores as a function of age and sample size. The standard deviation (SD) of bootstrapped centile scores for four cerebrum tissue volumes (y-axis) is plotted against mean age of study participants (top row) or sample size (bottom row). Studies with the most unstable OoS centile scores ($SD > 0.05$) are highlighted in red and labelled (see **ST1.1** for study details).

4.3. Test-retest reliability of out-of-sample centile scoring

We also assessed the reliability of OoS centile scoring in three independent datasets that acquired multiple MRI scans within a single session^{22,34–36}. We analyzed each scan (conducted within the same scanning session) as a novel OoS dataset, then compared the consistency of centile scores across different runs of the same subject. We similarly compared the consistency of the uncensored volumetric data and found that the out of sample estimation is as consistent as the ability of FreeSurfer to extract consistent values across runs within the same session.

First, we analysed test-retest reliability using the multimodal MRI reproducibility resource³⁴, which provides two sessions of MRI data for multiple modalities. This dataset comprising 21 subjects was specifically designed for assessment of test-retest reliability as all subjects were scanned in two sessions separated by a one-hour break and the whole cohort was completed within a two week period. We analyzed each session of 21 scans as an independent OoS study (**Fig.5**) and then estimated intra-class correlation coefficients (ICCs) to assess the between-session or test-retest reliability of individual centile scores for four cerebrum tissue volumes³⁷. All ICCs were ~ 0.99 (**Fig.S4.3.1**).

Fig. S4.3.1. Test-retest reliability of out-of-sample centile scores for cerebrum tissue volumes. MRI data were collected in two separate scanning sessions from $N=21$ participants and each session was analysed as an independent out-of-sample study using GAMLSS. Scatterplots represent OoS centile scores for session 1 (y-axis) versus out-of-sample centile scores for session 2 (x-axis) for each brain tissue volume, from left to right: GMV, WMV, sGMV, Ventricular CSF. Data points represent individual subject centile scores. Test-retest reliability was consistently very high (all ICCs > 0.99) for all cerebrum tissue volumes.

Second, we analysed the test-retest reliability of OoS centile scoring using MRI data on $N=72$ participants in the Healthy Brain Network (HBN) cohort²², which was not originally included in the reference dataset. The HBN cohort was designed to assess the influence of an alternate MRI data acquisition protocol, which included prospective motion correction²⁵ to improve quality and reliability of MRI. The study protocol included 2 sessions of scanning using a conventional MPRAGE sequence for T1-weighted data acquisition and another 2 sessions of scanning using an innovative, prospectively motion-corrected sequence, VNaV, for T1-weighted imaging²⁵. For all 72 individuals each session of each sequence was analysed as an OoS study (**Fig. 5; S11.8 “Out-of-sample estimation”**) and then we estimated ICCs as a measure of the test-retest reliability of individual centile scores for each brain tissue volume derived from each sequence (MPRAGE or VNaV). Test-retest reliability was uniformly high (ICCs > 0.95) for all OoS centile scores on all cerebrum tissue volumes estimated from both MPRAGE and VNaV sequences (**Fig. S4.3.2**). Reliability was incrementally higher for OoS centile scores derived from the VNaV sequence, under-scoring the importance of high quality data especially for OoS analysis of datasets with $N < 100$. However, we note that this increased reliability of centile scoring was most likely driven by a comparably increased consistency of the raw volumes estimated by FreeSurfer (as also noted in the original paper describing the impact of prospective motion correction²²).

Fig. S4.3.2. Test-retest reliability of out-of-sample centile scores for cerebrum tissue volumes measured twice in the same N=72 participants using two T1-weighted sequences, MPRAGE and VNaV. Top row shows out-of-sample centile scores for session 1 (y-axis) versus out-of-sample centile scores for session 2 (x-axis) for cerebrum tissue volumes estimated from MPRAGE data, from left to right: GMV, WMV, sGMV, Ventricles. Bottom row shows out-of-sample centile scores for session 1 (y-axis) versus out-of-sample centile scores for session 2 (x-axis) for cerebrum tissue volumes estimated from VNaV data, from left to right: GMV, WMV, sGMV, Ventricles. In all plots, data points represent individual subject centile scores. Test-retest reliability was uniformly high (all ICCs > 0.95) and generally somewhat higher for volumetrics derived from prospectively motion-corrected data (VNaV).

Third, we assessed the test-retest reliability of OoS centile scoring using the Vietnam Era Twin Study of Ageing (VETSA) study cohort³⁵. VETSA is a longitudinal study following 1,200 twins from the Vietnam Era Twin Registry, which includes two technically identical MPRAGE acquisitions within the first (baseline) scanning session. Both these scans were processed with FreeSurfer 6.0.1 for all participants, then the two sets of scans were each analysed as an independent OoS study, and ICCs were estimated to assess the test-retest reliability of individual centile scores on all four cerebrum tissue volumes. Test-retest reliability of OoS centile scores was uniformly very high (all ICCs > 0.98) across all phenotypes, comparable to the high reliability of the uncensored volumetric data generated by FreeSurfer 6.0.1 (all ICCs > 0.95), and in line with the constraints on reliability expected from technical sources of noise³⁸ (Fig. S4.3.3).

Fig. S4.3.3. Test-retest reliability of out-of-sample centile scores for cerebrum tissue volumes measured twice in the same 1,200 participants (600 twin pairs). Scatterplots show out-of-sample centile scores for scan 1 (y-axis) versus out-of-sample centile scores for scan 2 (x-axis) for cerebrum tissue volumes estimated from MPRAGE data, from left to right: GMV, WMV, sGMV, Ventricles. Data points represent individual subject centile scores. Reliability was uniformly high across all phenotypes (ICCs > 0.95) and comparable to reliability of uncensored volumetric measurements from the same set of scans (data not shown).

4.4. Reliability of out-of-sample centile scoring across multiple versions of FreeSurfer

Knowing that a large majority (~95%) of primary studies in the reference dataset used one of a series of versions of FreeSurfer for image analysis, we also evaluated the impact of these incrementally different image analysis pipelines on reliability of OoS centile scores. To do this we

re-analysed a single dataset³⁶ repeatedly using 4 different versions of FreeSurfer (5.1, 5.3, 6.01, and 7.1). Each version of the processed dataset was treated as an independent OoS study for GAMLSS modeling and then we estimated ICCs between individual centile scores for each possible pair of FreeSurfer pipelines and for each of four cerebrum tissue volumes. This analysis demonstrated generally high within-subject reliability of OoS centiles across all four pipelines: ICCs for GMV=0.978, WMV=0.972, sGMV=0.816 and Ventricles=0.982 (**Fig. S4.4**). We noted that there was somewhat reduced reliability of subcortical grey matter volume in both raw and centiled data from FreeSurfer version 5.1 in comparison to later FreeSurfer versions. While the reasons for this are unclear, none of the studies included in the principal dataset were processed with FreeSurfer 5.1, or any version of FreeSurfer older than 5.3. Furthermore, we found the highest between-pipeline reliability for both raw volumetric data and centile scores derived from the two most recent versions of FreeSurfer, 6.0.1 and 7.1, suggesting that minor inconsistencies due to FreeSurfer pre-processing are becoming less problematic as this widely used software package incrementally evolves.

Fig. S4.4. Between-pipeline reliability of volumetric data and out-of-sample centile scores for four cerebrum tissue volumes measured in the same set of $N=1468$ scans re-analysed using 4 different versions of FreeSurfer (5.1, 5.3, 6.01, and 7.1). Top row shows scatterplot matrices representing the correlation between raw volumetric data derived from each possible pair of FreeSurfer pipelines, from left to right: GMV, WMV, sGMV, Ventricles. Bottom row shows scatterplot matrices representing the correlation between out-of-sample centile scores derived from each possible pair of FreeSurfer pipelines, from left to right: GMV, WMV, sGMV, Ventricles. Intra-class correlations of out-of-sample centile scores and uncensored volumetric data, on average over all pairs of four pipelines, were generally high (GMV=0.978, WMV=0.972, sGMV=0.816 and Ventricles=0.982); although the reliability of sGMV volumetrics and centile scores was somewhat lower due to discrepant measurements by the oldest version of FreeSurfer, v5.1.

4.5. Effects of sample size on reliability of out-of-sample centile scores

To further assess the validity of the OoS estimates we generated ‘clones’ of existing datasets. Clones are resampled copies of studies included in the reference dataset used to estimate the study specific GAMLSS parameters, that are then treated as if they were “new” studies using the methods for out-of-sample centile scoring. This allows us to compare the OoS estimates to a relative truth, i.e., from the original, non-cloned version of the study included in the reference

dataset, we know what the GAMLSS parameters ‘truly’ are, and we have an estimation of their ‘true’ uncertainty from the bootstrap resampling distributions. Thus for a given study dataset, D_m , we generate a cloned copy D_1 , and if our approach is unbiased we expect the out-of-sample parameter estimates for D_1 to be equal to the in-sample parameters estimated for D_m , i.e., γ_m (representing the set of random effects estimated by in-sample analysis of the original study treated as part of the reference dataset) should approximate $\gamma_{,1}$ (representing the set of random effects estimated by OoS analysis of the cloned study treated as a new dataset): see **S11.8 “Out-of-sample estimation”** and **Fig. S4.5**.

In other words, we validated the OoS estimation by simulating a “new” study with the same underlying distribution as one of the studies included in the reference dataset. Hence, we expect the OoS random-effect estimates for this ‘clone’ to agree with the in-sample random-effect estimates. More formally, we are comparing $\gamma = MLE_{\beta,\gamma}(D)$ and $\gamma^{clone} = MLE_{\gamma}(D_{clone}|\beta(D))$, where the clone is contained within the data, i.e., $D \cap D_{clone} = D_{clone}$; see **S11.8 “Out-of-sample estimation”** for further details on OoS MLE estimation. As illustrated in **Fig. S4.5**, these simulations indicated good performance for the OoS approach for “new” study sizes greater than $N=100$ scans.

Fig. S4.5. Out-of-sample estimates of cloned study random-effect parameters compared to in-sample estimates of random-effect parameters in the original or non-cloned study. The plot shows random-effects estimated using the out-of-sample approach across a range of possible sample sizes for a “new” study, generated by taking subsets of the same cloned study with uncertainty intervals derived from the bootstrap replicates. The purple horizontal lines are the equivalent in-sample estimates of the random-effects parameters. We see that the out of sample estimates are somewhat unreliable below $N=100$ subjects, but with larger samples the out-of-sample estimates from the cloned data converge with the in-sample estimates from the original data for both μ -component and σ -component random effects.

6. Cohort effects

As is the case for traditional growth charts, reference norms for brain charts may change over time, underscoring the need for “front work” on constructing normative reference models that are adaptive to future trends. Our choice of GAMLSS as the preferred modeling framework was in part motivated by its ability to provide a flexible and scalable basis that could support ongoing updates to the reference data. Likewise, our effort to share these models on an interactive web-platform (www.brainchart.io & <https://github.com/ucam-department-of-psychiatry/Lifespan>) was also motivated by the likely need for continuous updates to the reference dataset as and when more MRI data become available.

To assess the potential risk of cohort effects, or population norms shifting over historical time and biasing estimation of centile scores in future, we used a single (NIH) study already included in our aggregated dataset, which collected data from 1991 onwards in a constrained age range (5–25 years; $N=1,468$ scans). While MRI is a comparatively novel methodology (~30 years), it is possible that there may be systematic cohort effects within studies that have sampled individuals over prolonged periods of time⁴⁷, or between measurements aggregated in different age bins at different times. To quantitatively assess this possibility and the robustness of our procedures and results against such cohort effects, we analysed this NIH study containing longitudinal scans collected over two decades, from 1991 to 2011. We found no evidence for systematic variation of centile scores on any of the 4 cerebrum tissue volumes as a function of year-of-scanning or in relation to changes or upgrades to the scanner platform (**Fig. S6.1-2**).

Thus there was no clear evidence of cohort effects in one of the few large studies to have sustained scanning over a long period of time, and there was no evidence of measurement biases related to technical development of image analysis software that potentially could contribute to cohort effects in large aggregated MRI datasets. However, the ongoing technical development of MRI scanners and image analysis software, as well as the possibility of more general secular trends in brain growth over time, mean that the risk of cohort effects should nonetheless be iteratively re-evaluated as the currently available reference dataset continues to be updated in the future.

Fig. S6.1. Assessment of potential cohort effects based on date of scanning over two decades. The longitudinal study at the National Institutes of Health (NIH) contains $N=1,468$ longitudinal scans ($N=788$ subjects) collected across the age range 5–25 years and over the historical period 1991–2011. Scatterplots represent individual centile scores (y-axis), ordered by date of scanning (x-axis), for each of the four cerebrum tissue volumes (top four rows); and age at scan (y-axis) versus date of scanning (x-axis) (bottom row). Lines represent locally-weighted regression lines (LOESS regression) for qualitative analysis of

possibly non-linear cohort effects on brain phenotypes or age at scanning. Filled circles denote baseline scans, empty circles denote follow-up scans in this longitudinal dataset; vertical lines indicate the timing of scanner upgrades over the course of the study (see also **Fig. S6.2**).

Fig. S6.2. Assessment of potential cohort effects related to scanner upgrades in the NIH longitudinal study. Centile scores for all four cerebrum tissue volumes estimated at baseline (time point 1) or two follow-up assessments (time points 2 and 3) were assigned to one of four epochs partitioned by the timing of upgrades to the 1.5T MRI scanner used for data collection. Box-violin plots show the distribution of centiles, and the range (whiskers) and 25th, 50th, and 75th percentiles of the centile distributions (boxes). Linear mixed effect modeling demonstrated no evidence of a significant effect ($t=-1.577$, $P=0.115$). This analysis was restricted to time points with $N > 100$ subjects.

Ref 2/7:

Finally, it would be very interesting to conduct a correlational analysis of the different “tissue classes.” Are they highly correlated? An excellent model for how such an analysis might be conducted is <https://doi.org/10.1038/ncomms13629> (see especially Figs. 7-8).

We thank the reviewer for raising this point. There are indeed interesting associations between the different tissue class volumes, yet we still observe unique lifespan trajectories for each. We now provide an additional supplementary section on the comparison between phenotypes across the lifespan and across datasets: see **SI15 “Correlations between cerebrum tissue volumes”** including new **Figs. S15.1 and S15.2**.

<<The following section has been added to the Supplementary Information>>

15. Interactions between cerebrum tissue volumes

It has been hypothesized that age-varying cellular processes could be captured by neuroimaging milestones, in terms of the growth trajectories of relative volumetric measurements⁸¹. In line with these expectations, we found an initial postnatal increase in GMV relative to WMV, likely due to increased complexity of neuropil including synaptic proliferation^{82,83}. Subsequently, GMV declined

relative to WMV (S19.2 “Grey-white matter differentiation”), likely due to both continued myelination and synaptic pruning⁸⁴. To further explore the patterning of tissue interactions, we performed supplementary analyses to empirically assess the correlations between global tissue classes. **Fig. S15.1** presents these inter-relationships as Pearson’s correlation coefficients between each pair of global brain MRI phenotypes across participants within each study. These results highlight the variability of these relationships across studies (which themselves vary in terms of technical and biological variables – see **Fig. 1A, ST1.1**). However, it is also clear that there are generally high correlations between grey and white matter volumes and surface area (SA). Comparatively, GMV and WMV are less strongly correlated with CT and CSF. Additionally, we substantiated the prior consensus in the literature concerning the orthogonality of CT and SA by finding that these two global metrics were not correlated with each other (**Fig. S15.1**).

Fig. S15.1. Box-plots of Pearson’s correlation between each possible pair of global brain metrics over all studies in the reference dataset. Each datapoint represents a single study; boxes highlight the median and interquartile range of correlation values (across studies) between feature pairs.

Given these findings in the context of each study in our aggregated dataset, we examined the same inter-relationships between phenotypes across age, in line with previous work examining regional correlations of diffusion-weighted imaging phenotypes across age⁸⁵. We used a sliding window approach to apply this framework to global MRI phenotypes, binning segments of the lifespan based on age (each window = 300 days, sliding by 50 days). Pearson’s correlation between phenotypes was then calculated within each bin, and locally-weighted (LOESS) regression was used to fit a nonlinear curve to the age-related changes in each pair-wise phenotypic correlation (**Fig. S15.2**). These results recapitulate some of the findings of the correlational analyses within each primary study, e.g., the GMV/WMV correlation is consistently more strongly positive than the CT/SA correlation. However, there are also some age-related shifts in the strength and/or sign of these phenotypic correlations, especially in late gestation and early postnatal life, that will be interesting to investigate in more detail as additional early-life MRI data become available in future .

Fig. S15.2. Sliding-window analysis of age-related changes in pairwise correlations for all possible pairs of 7 global MRI phenotypes (4 cerebrum tissue volumes and 3 extended global MRI phenotypes) over the course of the lifespan. We used a window size of 300 days, sliding by 50 days. Plotted lines are colour-coded by pairwise correlation and represent the fitted lines and 95% confidence intervals from locally-weighted (LOESS) regression for each correlated pair of phenotypes.

Referee #3:

The manuscript “Brain charts for the human lifespan” by Bethlehem et al. is an impressive and ambitious effort on several fronts. They aggregate a massive amount of structural neuroimaging scans (>122k scans, across >100k individuals) across 96 individual studies and then use sophisticated modeling (GAMLSS) to derive lifespan curves of 4 morphometric phenotypes (gray matter, white matter, subcortical gray matter, and ventricular volume) from 115 days post-conception to 100 years old. That endeavor alone is worthy of publication in a high-profile journal. The resulting age curves largely agree with existing literature, and thus for the most part aren’t particularly novel in themselves, although they note some deviations from prior reports. Using centile scoring as a means of normalizing across studies, they then investigate the impact of various clinical diagnoses.

We thank the reviewer for their positive appraisal of our team’s effort.

While interesting, and certainly suggesting venues for further study, that aspect of the paper is entirely focused on group comparisons, and relies heavily on p-values, which will be driven to significance by the large sample sizes. Reporting of the effect sizes involved would be helpful and would provide at least an indirect sense of the potential of centiled brain charts within the context of “personalised” or “precision” medicine.

This is an excellent point - especially in the context of dealing with large sample sizes. All tables reporting results of statistical tests now include point and interval estimates of Cohen’s *d* as a measure of effect size. Text references to key statistical results now also include effect sizes as well as *P*-values and median differences.

But the truly groundbreaking potential of aggregating such a large data set, combined with the flexible modeling approach, is the proposed ability to estimate centile scores meaningfully and reliably in “out-of-sample” data, in the context of *non-harmonized* MR data that would typically be subject to a host of interpretational challenges (e.g., different pulse sequences, protocols, scanner strength and vendor, etc). Indeed, the manuscript is the planned reference for an interactive online resource (www.brainchart.io) that would allow researchers to extract centile scores for new datasets. In this regard, I feel that the manuscript, as currently constituted, falls short of convincingly demonstrating that the challenge of out-of-sample estimation with highly variable MR data has been solved. If this concern can be addressed, I feel that the approach and online resource proposed in this manuscript has intriguing potential and would warrant the visibility that publication in Nature would provide.

We thank the reviewer for their very positive appraisal of the paper and its potential impact and we agree with their assessment of the most ground-breaking aspects of the work.

Ref 3/1:

Currently, the out-of-sample validation with “real-world” is limited to just 4 datasets. The results of that particular analysis are impressive (Fig. 4B and S1.7.2). But the use of only 4 datasets means that the out-of-sample validation is an impoverished sampling of the universe of possible MR studies, given the wide variety of ways in which MR studies can differ. Given that the authors have already amassed a much broader sampling of that universe (96 studies contributing to the model), it isn’t clear to me why they didn’t assess the generalizability of the GAMLSS + centiles modeling approach across that full study universe by conducting the same out-of-sample analysis using a “leave-one-study-out” (LOSO) modeling approach applied to every available study (i.e., treat every available study as if it is a part of the out-of-sample validation analysis). Indeed, a LOSO analysis is mentioned in Supplement (SI) Section 2.2.1, but that’s only in the context of showing the variability in the resulting overall lifespan trajectories. It seems to me that what’s needed additionally is a way to assess the likelihood that the estimated centiles for a given study may not be well fit, and correspondingly the study parameters potentially influencing that poor fit (per item (2) below), since that’s what an individual investigator interested in applying the model to one particular new study needs to be able to evaluate.

We have adopted the reviewer’s suggestion to convincingly demonstrate that “the challenge of out-of-sample estimation with highly variable MRI data has been solved” by a set of convergent new analyses focused on this key point of out-of-sample (OoS) centile scoring. These new results and discussion are now referenced in the main text and fully reported in the supplemental material: **SI4 “Out-of-sample centile scoring: bias, stability and reliability”**, including 7 new Figs. **S4.2.1, S4.2.2, S4.3.1, S4.3.2, S4.3.3, S4.4 and S4.5**.

- We previously reported leave-one-study-out (LOSO) analyses for 4 primary studies; at the reviewer’s request to explore the “universe” of studies, we have now conducted LOSO analysis for each and every one of 100 primary studies. We now show that bias (difference between in-sample and OoS centile scores) and reliability (standard deviation of bootstrapped centile scores) are excellent for the large majority of primary studies using FreeSurfer to process $N > 100$ scans: see **SI4.1 “Bias of out-of-sample centile scores: leave-one-study-out analyses for 100 studies”** and **SI4.2 “Reliability of out-of-sample centile scoring: bootstrapped LOSO analyses for 100 studies”**. We have estimated intra-class correlations for out-of-sample centile scores on several independent test-retest MRI datasets^{22,34–36} and found that the test-retest reliability of OoS centile scores was on par with the very high test-retest reliability of the non-normalised, un-centiled volumetric data generated by FreeSurfer (all ICCs > 0.9). New results and discussion are now referenced in the main text and reported fully in the supplemental material: see **SI4.3 “Test-retest reliability of out-of-sample centile scoring”**.
- Given the widespread use of FreeSurfer, and its ongoing technical evolution through a series of versions, we additionally investigated consistency of OoS centile scores between multiple versions of FreeSurfer: see: **SI4.4 “Reliability of out-of-sample centile scoring across multiple versions of FreeSurfer”**.
- We have also systematically investigated the influence of sample size of primary studies on reliability of their out-of-sample centile scores: see **SI4.5 “Effects of sample size on reliability of out-of-sample centile scores”**.

- In addition to these extensive additions to Supplementary Information, we have substantially revised **Fig. 4** in the earlier version of the paper, now **Fig. 5** in the main text, to highlight the methods and supporting evidence for out-of-sample centile scoring.

Overall, we consider that these new results considerably strengthen the evidence that out-of-sample centile scoring is unbiased and reliable for the large majority of scans in the available “universe” of MRI studies.

<<The following changes have been made to the main text>>

Out-of-sample centile scoring of “new” MRI data

A key challenge for brain charts is the accurate centile scoring of out-of-sample MRI data, not represented in the normative distribution of trajectories. As such, we carefully evaluated the reliability and validity of brain charts for centile scoring of “new” scans. For each new MRI study, we used maximum likelihood to estimate study-specific statistical offsets from the age-appropriate epoch of the normative trajectory; then we estimated centile scores for each individual in the new study benchmarked against the offset trajectory (**Fig.5; SI1.8**). Extensive jack-knife and leave-one-study-out (LOSO) analyses indicated that a study size of $N > 100$ scans was sufficient for stable and unbiased estimation of out-of-sample centile scores (**SI4**). Furthermore, out-of-sample centile scores proved highly reliable in multiple test-retest datasets and robust to variations in image processing pipelines (**SI4**).

Fig. 5. Schematic overview of brain charts, highlighting methods for out-of-sample centile scoring. Top panel: Brain phenotypes are measured in a reference dataset of MRI scans. GAMLSS modeling is used to estimate the relationship between (global) MRI phenotypes and age, stratified by sex, and controlling for technical and other sources of variation between scanning sites and primary studies. Bottom panel: The normative trajectory of the median and confidence interval for each phenotype is plotted as a population reference curve. Out-of-sample data from a new MRI study are aligned to the corresponding

epoch of the normative trajectory, using maximum likelihood to estimate the study specific offsets (random effects) for three moments of the underlying statistical distributions: mean (μ), variance (σ), and skewness (ν) in an age- and sex-specific manner. Centile scores can then be estimated for each scan in the new study, on the same scale as the reference population curve, while accounting for study-specific “batch effects” on technical or other sources of variation (see **SI1.8** for details).

<<The following changes have been made to the Supplementary Information>>

4. Out-of-sample centile scoring: bias, stability and reliability

4.1. Bias of out-of-sample centile scores: leave-one-study-out analyses for 100 studies

To further evaluate the robustness and consistency of centile scoring of OoS MRI data that were not included in the reference dataset used to estimate population trajectories, we performed a comprehensive series of leave-one-study-out (LOSO) analyses. For each one of the 100 studies in the reference dataset, we removed the study from the reference dataset, re-fitted the GAMLSS model to the remaining dataset of 99 studies, computed the OoS centile scores for the excluded study, and compared the OoS centile scores to the in-sample centile scores computed for the same study from the complete dataset including all 100 studies. Supplementary tables **ST7.1-7.4** list the correlations between OoS and in-sample centile scores for all 4 cerebrum tissue volumes in each of 100 primary studies. Overall, we found very high levels of correlation (Pearson’s $r \sim 0.99$) for almost all studies, indicating that centile scores can be estimated accurately for most studies even if they were not included in the reference dataset used to define population norms. Correlations between OoS and in-sample centile scores were lower than $r = 0.99$ for only 3 out of 100 studies in the reference dataset: namely, the FinnBrain ($r = 0.93$), UCSD ($r = 0.96$) and NIHPD ($r = 0.95$) studies. These studies were characterised by relatively small sample size, foetal or early postnatal age range of participants, or idiosyncratic processing pipelines.

4.2. Stability of out-of-sample centile scoring: bootstrapped LOSO analyses for 100 studies

In addition, we tested the reliability of OoS centile scores for each individual participant by bootstrapping. Specifically, for each LOSO sample, bootstrapped model parameters were generated (see **SI3.2.2 “Bootstrap analysis”**), resulting in 1,000 bootstrapped models with maximum likelihood estimated parameters for each bootstrap iteration of each left-out study. From this we obtained a bootstrapped distribution of out-of-sample centile scores for each individual subject in each individual iteration of left-out studies, thus providing a stability assessment in the form of the standard deviation of individual OoS centile scores across 1,000 bootstrap iterations. Across the datasets included in the model, we found that the average standard deviation of (bootstrapped) OoS centiles was 0.014, which is well below the level of within-subject longitudinal variation (see **Fig. S4.2.1** and **SI14 “Longitudinal centiles”**). Furthermore, we found increased standard deviation of OoS centile scores for datasets with comparatively small sample sizes (e.g., the OpenPain cohorts, Cambridge foetal Testosterone and CHILd studies; see **Fig. S4.2.2**). OoS centile scores were also more variable for datasets that had a more unique combination of age range, acquisition and processing pipelines (e.g., FinnBrain, IBIS and HBN; see **Fig. S4.2.2**). These observations reinforce the recommendation -- see main text, ‘**Out-of-sample centile scoring of “new” MRI data**’ -- that OoS centile scoring is reliable for studies comprising $N > 100$ scans. It was also notable that the reliability of OoS centile

scores was weakly correlated with data quality as quantified by the Euler index (EI). So studies with higher EI²³, indicating poorer image quality, tended to have higher variability of bootstrapped OoS centile scores (Pearson's r for all 4 cerebrum tissue volumes: GMV=0.05, WMV =0.11, sGMV =0.14, and Ventricular volume = 0.13). These results were not substantially different when the whole set of analyses was repeated without including scans with EI > 217. We conclude that OoS estimation of centile scores is generally reliable at the level of individual scans, and (as expected) reliability is greater for higher quality scans.

Fig. S4.2.1. Stability of out-of-sample centile scores for four cerebrum tissue volumes when each of 100 studies was excluded from the reference dataset before bootstrapping. The standard deviation of bootstrapped centile scores (y-axis) is plotted for each study (x-axis) for each phenotype, from top to bottom panels: total cortical grey matter volume, total cortical white matter volume, subcortical grey matter volume, and ventricular volume. Each study- and phenotype-specific boxplot is coloured according to log sample size. For each study, we estimated the normative model leaving that study out of the reference dataset and repeated this procedure after iteratively bootstrapping the reference dataset 1,000 times. This procedure allowed us to summarise the reliability of the out-of-sample estimates of centile scores in terms of the standard deviation of the 1,000 centile scores generated for each bootstrapped resampling of the reference dataset. Studies are ordered by median standard deviation of out-of-sample centile scores (small to large) indicating that scans are reliably assigned centile scores with the out-of-sample approach.

Fig. S4.2.2. Stability of out-of-sample centile scores as a function of age and sample size. The standard deviation (SD) of bootstrapped centile scores for four cerebrum tissue volumes (y-axis) is plotted against mean age of study participants (top row) or sample size (bottom row). Studies with the most unstable OoS centile scores ($SD > 0.05$) are highlighted in red and labelled (see **ST1.1** for study details).

4.3. Test-retest reliability of out-of-sample centile scoring

We also assessed the reliability of OoS centile scoring in three independent datasets that acquired multiple MRI scans within a single session or two closely spaced sessions^{22,34–36}. We analyzed each scan as a novel OoS dataset, then compared the consistency of centile scores across different scans of the same subject. We similarly compared the consistency of the uncensored volumetric data and found that the out-of-sample centile scores were as consistent between scans in the same session as the “raw” volumetric data generated by FreeSurfer.

First, we analysed test-retest reliability using the multimodal MRI reproducibility resource³⁴, which provides two sessions of MRI data for multiple modalities. This dataset comprising 21 subjects was specifically designed for assessment of test-retest reliability as all subjects were scanned in two sessions separated by a one-hour break and the whole cohort was completed within a two week period. We analyzed each session of 21 scans as an independent OoS study (**Fig. 5**) and then estimated intra-class correlation coefficients (ICCs) to assess the between-session or test-retest reliability of individual centile scores for four cerebrum tissue volumes³⁷. All ICCs were ~ 0.99 (**Fig. S4.3.1**).

Fig. S4.3.1. Test-retest reliability of out-of-sample centile scores for cerebrum tissue volumes. MRI data were collected in two separate scanning sessions from $N=21$ participants and each session was analysed as an independent out-of-sample study using GAMLSS. Scatterplots represent OoS centile scores for session 1 (y-axis) versus OoS centile scores for session 2 (x-axis) for each brain tissue volume, from left to right: GMV, WMV, sGMV, Ventricular CSF. Data points represent individual subject centile scores. Test-retest reliability was consistently very high (all ICCs > 0.99) for all cerebrum tissue volumes.

Second, we analysed the test-retest reliability of OoS centile scoring using MRI data on $N=72$ participants in the Healthy Brain Network (HBN) cohort²², which was not originally included in the reference dataset. The HBN cohort was designed to assess the influence of an alternate MRI data acquisition protocol, which included prospective motion correction²⁵ to improve quality and reliability of MRI. The study protocol included 2 sessions of scanning using a conventional MPRAGE sequence for T1-weighted data acquisition and another 2 sessions of scanning using an innovative, prospectively motion-corrected sequence, VNaV, for T1-weighted imaging²⁵. For all 72 individuals each session of each sequence was analysed as an OoS study (Fig. 5; S11.8 “Out-of-sample estimation”) and then we estimated ICCs as a measure of the test-retest reliability of individual centile scores for each brain tissue volume derived from each sequence (MPRAGE or VNaV). Test-retest reliability was uniformly high (ICCs > 0.95) for all OoS centile scores on all cerebrum tissue volumes estimated from both MPRAGE and VNaV sequences (Fig. S4.3.2). Reliability was incrementally higher for OoS centile scores derived from the VNaV sequence, under-scoring the importance of high quality data especially for OoS analysis of datasets with $N < 100$. However, we note that this increased reliability of centile scoring was most likely driven by a comparably increased consistency of the raw volumes estimated by FreeSurfer (as also noted in the original paper describing the impact of prospective motion correction²²).

Fig. S4.3.2. Test-retest reliability of out-of-sample centile scores for cerebrum tissue volumes measured twice in the same N=72 participants using two T1-weighted sequences, MPRAGE and VNav. Top row shows out-of-sample centile scores for session 1 (y-axis) versus out-of-sample centile scores for session 2 (x-axis) for cerebrum tissue volumes estimated from MPRAGE data, from left to right: GMV, WMV, sGMV, Ventricles. Bottom row shows out-of-sample centile scores for session 1 (y-axis) versus out-of-sample centile scores for session 2 (x-axis) for cerebrum tissue volumes estimated from VNav data, from left to right: GMV, WMV, sGMV, Ventricles. In all plots, data points represent individual subject centile scores. Test-retest reliability was uniformly high (all ICCs > 0.95) and generally somewhat higher for volumetrics derived from prospectively motion-corrected data (VNav).

Third, we assessed the test-retest reliability of OoS centile scoring using the Vietnam Era Twin Study of Ageing (VETSA) study cohort³⁵. VETSA is a longitudinal study following 1,200 twins from the Vietnam Era Twin Registry, which includes two technically identical MPRAGE acquisitions within the first (baseline) scanning session. Both these scans were processed with FreeSurfer 6.0.1 for all participants, then the two sets of scans were each analysed as an independent OoS study, and ICCs were estimated to assess the test-retest reliability of individual centile scores on all four cerebrum tissue volumes. Test-retest reliability of OoS centile scores was uniformly very high (all ICCs > 0.98) across all phenotypes, comparable to the high reliability of the uncensored volumetric data generated by FreeSurfer 6.0.1 (all ICCs > 0.95), and in line with the constraints on reliability expected from technical sources of noise³⁸ (Fig. S4.3.3).

Fig. S4.3.3. Test-retest reliability of out-of-sample centile scores for cerebrum tissue volumes measured twice in the same 1,200 participants (600 twin pairs). Scatterplots show out-of-sample centile scores for scan 1 (y-axis) versus out-of-sample centile scores for scan 2 (x-axis) for cerebrum tissue volumes estimated from MPRAGE data, from left to right: GMV, WMV, sGMV, Ventricles. Data points represent individual subject centile scores. Reliability was uniformly high across all phenotypes (ICCs > 0.95) and comparable to reliability of uncensored volumetric measurements from the same set of scans (data not shown).

4.4. Reliability of out-of-sample centile scoring across multiple versions of FreeSurfer

Knowing that a large majority (~95%) of primary studies in the reference dataset used one of a series of versions of FreeSurfer for image analysis, we also evaluated the impact of these incrementally different image analysis pipelines on reliability of OoS centile scores. To do this we re-analysed a single dataset³⁶ repeatedly using 4 different versions of FreeSurfer (5.1, 5.3, 6.01, and 7.1). Each version of the processed dataset was treated as an independent OoS study for GAMLSS modeling and then we estimated ICCs between individual centile scores for each possible pair of FreeSurfer pipelines and for each of four cerebrum tissue volumes. This analysis demonstrated generally high within-subject reliability of OoS centiles across all four pipelines: ICCs for GMV=0.978, WMV=0.972, sGMV=0.816 and Ventricles=0.982 (Fig. S4.4). We noted that there was somewhat reduced reliability of subcortical grey matter volume in both raw and centiled data from FreeSurfer version 5.1 in comparison to later FreeSurfer versions. While the reasons for this are unclear, none of the studies included in the principal dataset were processed with FreeSurfer 5.1, or any version of FreeSurfer older than 5.3. Furthermore, we found the highest between-pipeline reliability for both raw volumetric data and centile scores derived from the two most recent versions of FreeSurfer, 6.0.1 and 7.1, suggesting that minor inconsistencies due to FreeSurfer pre-processing are becoming less problematic as this widely used software package incrementally evolves.

Fig. S4.4. Between-pipeline reliability of volumetric data and out-of-sample centile scores for four cerebrum tissue volumes measured in the same set of $N=1,468$ scans re-analysed using 4 different versions of FreeSurfer (5.1, 5.3, 6.01, and 7.1). Top row shows scatterplot matrices representing the correlations between raw volumetric data derived from each possible pair of FreeSurfer pipelines, from left to right: GMV, WMV, sGMV, Ventricles. Bottom row shows scatterplot matrices representing the correlations between out-of-sample centile scores derived from each possible pair of FreeSurfer pipelines, from left to right: GMV, WMV, sGMV, Ventricles. Intra-class correlations of out-of-sample centile scores and uncensored volumetric data, on average over all pairs of four pipelines, were generally high (GMV=0.978, WMV=0.972, sGMV=0.816 and Ventricles=0.982). Although the reliability of sGMV volumetrics and centile scores was somewhat lower due to discrepant measurements by the oldest version of FreeSurfer, v5.1, this version of FreeSurfer was not used to analyse any of the scans included in the reference dataset.

4.5. Effects of sample size on reliability of out-of-sample centile scores

To further assess the validity of the OoS estimates we generated ‘clones’ of existing datasets. Clones are resampled copies of studies included in the reference dataset used to estimate the study specific GAMLSS parameters, that are then treated as if they were “new” studies using the methods for out-of-sample centile scoring. This allows us to compare the OoS estimates to a relative truth, i.e., from the original, non-cloned version of the study included in the reference dataset, we know what the GAMLSS parameters ‘truly’ are, and we have an estimation of their ‘true’ uncertainty from the bootstrap resampling distributions. Thus for a given study dataset, D_m , we generate a cloned copy D_1 , and if our approach is unbiased we expect the out-of-sample parameter estimates for D_1 to be equal to the in-sample parameters estimated for D_m , i.e., γ_m (representing the set of random effects estimated by in-sample analysis of the original study treated as part of the reference dataset) should approximate $\gamma_{.1}$ (representing the set of random effects estimated by OoS analysis of the cloned study treated as a new dataset): see **SI1.8 “Out-of-sample estimation”** and **Fig. S4.5**.

In other words, we validated the OoS estimation by simulating a “new” study with the same underlying distribution as one of the studies included in the reference dataset. Hence, we expect the OoS random-effect estimates for this ‘clone’ to agree with the in-sample random-effect estimates. More formally, we are comparing $\gamma = MLE_{\beta,\gamma}(D)$ and $\gamma^{clone} = MLE_{\gamma}(D_{clone}|\beta(D))$, where the clone is contained within the data, i.e., $D \cap D_{clone} = D_{clone}$; see **SI1.8 “Out-of-sample estimation”** for further details on OoS MLE estimation. As illustrated in **Fig. S4.5**, these simulations indicated good performance for the OoS approach for “new” study sizes greater than $N=100$ scans.

Fig. S4.5. Out-of-sample estimates of cloned study random-effect parameters compared to in-sample estimates of random-effect parameters in the original or non-cloned study. The plot shows random-effects estimated using the out-of-sample approach across a range of possible sample sizes for a “new” study, generated by taking subsets of the same cloned study with uncertainty intervals derived from the bootstrap replicates. The purple horizontal lines are the equivalent in-sample estimates of the random-effects parameters. We see that the out of sample estimates are somewhat unreliable below $N=100$ subjects, but with larger samples the out-of-sample estimates from the cloned data converge with the in-sample estimates from the original data for both μ -component and σ -component random effects.

Ref 3/2:

Relatedly, the manuscript mentions that “biological” and “technical” covariates are included as part of the fixed effects modeling, but the actual covariates used in the modeling do not appear to be listed anywhere. Also, no analysis is provided (even in the SI) of the estimated effect of these covariates, which seems important for understanding

the inner workings of the estimation. Last, understanding the space spanned by the covariates seems important for making assessments about the generalizability of the model to new out-of-sample data. In that regard, I think an SI table that lists the covariate values for each study, as well as other possibly relevant technical scanning details (e.g., imaging parameters) seems like a valuable and important addition.

We apologize for any lack of clarity about our approach to site-specific technical variability in the original manuscript. In response to this important point, as requested, we have included a new supplemental table listing the technical details of each study: **Table ST1.1**.

Due to variation and complexity of site-specific covariates, e.g., model of scanner, acquisition parameters, analysis software etc., we used random effects to model study- and site-specific differences in the moments of the statistical distributions of MRI phenotypes, thus controlling for a combination of (unspecified) factors that might contribute to between-study or between-site differences in MRI data and the centile scores derived from them. This approach is conceptually similar to other successful harmonization approaches such as the use of ComBAT to adjust for “batch effects” resulting from scanner differences, and our revised manuscript includes a detailed comparison between GAMLSS-based harmonization and ComBAT harmonization: see **SI5 “Batch correction and site harmonisation”**.

We note that in contrast to the typical usage of random effects in neuroimaging studies, the optimally specified GAMLSS model included random effects on three moments of the generalised gamma distributions of the MRI phenotypes (mean, variance and skewness) to more flexibly account for site-specific variability as described in detail in **SI5 “Batch correction and site harmonisation”**. One drawback of this approach is that it is not possible to conduct an analysis of the effect of specific technical covariates, such as scanning parameters. However, we do now report several approaches to quantifying study-specific effects from the GAMLSS model parameterisation: see **SI3.3 “Parameter estimates”** and Fig. **S3.2.3**.

As described in response to the reviewer’s previous point, Ref 3/1, we now also report a much more extensive validation of the out-of-sample approach to accurately estimate site-specific effects in “new” studies that may have been conducted with technical parameters that were not represented in the reference dataset: see **SI4 “Out-of-sample centile scoring: bias, stability and reliability”**.

<<The following changes have been made to the main text>>

Models were fitted to structural MRI data from control subjects for the four main tissue volumes of the cerebrum (total cortical grey matter volume [GMV] and total white matter volume [WMV], total subcortical grey matter volume [sGMV], and total ventricular cerebrospinal fluid volume [Ventricles or CSF]). See **Online Methods, Supplementary Table [ST] 1.1-1.7** for details on acquisition, processing and demographics of the dataset. See **Supplementary Information [SI]** for details regarding GAMLSS model specification and estimation (**SI1**), image quality control (**SI2**), model stability and robustness (**SI3-4**), phenotypic validation against non-imaging metrics (**SI3 & SI5.2**), inter-study harmonisation (**SI5**) and assessment of cohort effects (**SI6**).

<<The following changes have been made to the supplementary information>>

3.2.3 Parameter estimates

From our bootstrapping approach, we can also derive confidence intervals for the models' parameter estimates (e.g., the μ and σ terms) for study-specific random effects. Qualitatively we observed very narrow confidence intervals on the estimated μ term, with some smaller sample foetal studies (e.g., CHILD and Harvard foetal cohorts) showing wider intervals, likely commensurate with the smaller sample size and general lack of reference data in that age range (**Fig. S3.2.3**). While there were generally wider confidence intervals on the σ term offsets, across studies all estimated random effect parameters were well contained within their bootstrapped confidence bounds.

Fig. S3.2.3. Point-range plots of study-specific random effects on the first (μ) and second (σ) moments of the generalised gamma distribution for parenchymal tissue volumes and study-specific random effects on μ only for ventricular CSF volume. Bootstrapped 95% confidence intervals are shown and point estimates (dots) are coloured by the range of the confidence interval. Where not observable, the confidence intervals are smaller than the size of the dots.

Ref 3/3:

Multiple studies have demonstrated that thickness and surface area are more pertinent measures than cortical GMV, given that cortical GMV is determined entirely by thickness and area, but thickness and area are themselves under independent genetic control. Given that, what is the rationale for not including mean cortical thickness and total surface area as part of the phenotypes investigated, both of which are readily available since all the data was processed through FreeSurfer, and would just need to be modeled?

The reviewer makes a very fair point. The GAMLSS modeling framework we introduced in the context of brain tissue volume analysis is indeed generalisable to other MRI phenotypes, including the global cortical phenotypes (surface area and thickness) suggested by the reviewer. Although we were not able to access data on cortical surface area and thickness from all primary studies in the reference dataset, due to data sharing restrictions, we have aggregated and analysed these global cortical metrics on a large subset of the reference dataset ($N_{\text{total}}=105,067$, $N_{\text{unique subjects}}=84,574$ and $N_{\text{unique CN subjects}}=66,225$ for SA and $N_{\text{total}}=105,093$, $N_{\text{unique subjects}}=84,532$ and $N_{\text{unique CN subjects}}=66,181$ for CT). These new results are now summarised in the main text, including a new **Fig. 2** and revised **Fig. 3**, and are summarised in more detail in new sections of supplemental information: **SI7 “Extended global cortical phenotypes”** including 7 new supplementary **Figs. S7.1, S7.2.1, S7.2.2, S7.3.1, S7.3.2, S7.4.1** and **S7.4.2**.

<<The following changes have been made to the main text>>

Extended brain MRI phenotypes

To extend the scope of brain charts beyond the four cerebrum tissue volumes, we used the same GAMLSS modeling approach to estimate normative trajectories for additional MRI phenotypes including other geometric properties at a similar scale (mean cortical thickness and total surface area) and regional volume at each of 34 cortical areas²⁵ (**Fig.2, SI7-9, ST1-2**). We found, as expected, that total surface area closely tracked the development of total cerebrum volume (TCV) across the lifespan (**Fig.2A**), with both metrics peaking at ~11-12 years (SA $10.97_{\text{CI-Bootstrap:10.42-11.51}}$; TCV $12.5_{\text{CI-Bootstrap:12.14-12.89}}$). In contrast, cortical thickness peaked distinctly early at $1.7_{\text{CI-Bootstrap:1.3-2.1}}$ years, which reconciles prior observations that cortical thickness increases during the perinatal period²⁶ and declines during later development²⁷. We also found evidence for regional variability in volumetric neurodevelopmental trajectories. Compared to GMV's peak at 5.9 years, the age of peak regional volume varied considerably – from approximately 2 to 10 years – across 34 cortical areas. Primary sensory regions reached peak volume earliest, and fronto-temporal association cortical areas matured later (**Fig.2B; SI8**). In general, earlier maturing ventral-caudal regions also showed faster post-peak declines in cortical volume, and later maturing dorsal-rostral regions showed slower post-peak declines (**Fig.2B; SI8.2**). Notably, this spatial pattern recapitulates a gradient from sensory-to-association cortex that has been previously associated with multiple aspects of brain structure and function²⁸.

Fig 2. Extended global and regional cortical geometric phenotypes. A | Trajectories for total cerebrum volume (TCV; left column), total surface area (SA; middle column), and mean cortical thickness (CT; right column). For each global cortical geometric MRI phenotype, the following sex-stratified results are shown as a function of age over the life-span, from top to bottom rows: raw, non-centiled data, population trajectories of the median (with 2.5% and 97% centiles; dotted lines), between-subject variance (and 95% confidence intervals), and rate-of-growth (the first derivatives of the median trajectory and 95% confidence interval). All trajectories are plotted on log-scaled age (x-axis) and y-axes are scaled in units of the corresponding MRI metrics (10,000 mm³ for TCV, 10,000 mm² for SA and mm for CT). B | Regional variability of cortical volume trajectories for 34 bilateral brain regions as defined in the Desikan-Killiany parcellation²⁵, averaged across sex (see also **S17-8** for details). From top to bottom panels: cortical map of age at peak regional volume (range 2-10 years); cortical map of age at peak regional volume relative to age at peak GMV (5.9 years), highlighting regions that peak earlier (blue) or later (red) than GMV; and illustrative trajectories for the earliest peaking region (superior parietal lobe) and the latest peaking region (insula), showing the range of regional variability. Regional volume peaks are denoted as dotted vertical lines either side of the global peak denoted as a dashed vertical line in the bottom panel. Left hand y-axis on the bottom panel refers to the earliest peak, the right hand y-axis refers to the latest peak, and both are in units of 10,000 mm³.

Fig. 3. Neurodevelopmental milestones. Top panel: A graphical summary of the normative trajectories of the median, i.e., 50th centile, for each global MRI phenotype, and key developmental milestones, as a function of age (log-scaled). Circles depict the peak rate-of-growth milestones for each phenotype (defined by the maxima of the first derivatives of the median trajectories; see Fig. 1E). Triangles depict the peak volume of each phenotype (defined by the maxima of the median trajectories), definition of GMV:WMV differentiation is detailed in S19.1. Bottom panel: A graphical summary of additional MRI and non-MRI developmental stages and milestones. From top to bottom: blue shaded boxes denote the age-range of incidence for each of the major clinical disorders represented in the MRI dataset; black boxes denote the age at which these conditions are generally diagnosed as derived from literature⁴¹ (**Online Methods**); brown lines represent the normative intervals for developmental milestones derived from non-MRI data, based on previous literature and averaged across males and females (**Online Methods**); grey bars depict age ranges for existing (WHO and CDC) growth charts of anthropometric and ultrasonographic variables. Across both panels, light grey vertical lines delimit lifespan epochs (labelled above the top panel) previously defined by neurobiological criteria⁴². Abbreviations: resting metabolic rate (RMR), Alzheimer's disease (AD), attention deficit hyperactivity disorder (ADHD), anxiety or phobic disorders (ANX), autism spectrum disorder (ASD, including high-risk individuals with confirmed diagnosis at a later age), major depressive disorder (MDD), bipolar disorder (BD), and schizophrenia (SCZ).

In Discussion:

We have focused primarily on charts of global brain phenotypes, which were measurable in the largest aggregated sample over the widest age range, with the fewest methodological, theoretical and data sharing constraints. However, we have also provided proof-of-concept brain charts for regional grey matter volumetrics, demonstrating plausible heterochronicity of cortical patterning, and illustrating the generalisability of this approach to a more diverse range of fine-grained MRI phenotypes. As ongoing and future efforts provide increasing amounts of high quality MRI data, we predict an iterative process of improved brain charts for the human lifespan, potentially

representing multi-modal MRI phenotypes and enabling out-of-sample centile scoring of smaller samples or individual scans. In the hope of facilitating progress in this direction, we have provided interactive tools to explore these statistical models and to derive normalised centile scores for new datasets across the lifespan at www.brainchart.io.

<<The following changes have been made to the Supplemental Information>>

7. Extended global cortical phenotypes

In addition to the principal results based on cerebrum tissue volumes, we also developed brain charts, based on the same GAMLSS modeling strategy, for other global cortical phenotypes including mean cortical thickness (CT) and surface area (SA). We refer to these as ‘cortical geometric phenotypes’, because they are derived from MRI data at a later stage of the widely-used FreeSurfer pipeline, subsequent to cortical surface reconstruction, which is a necessary precondition for measuring cortical geometry. Geometric cortical phenotypes are likely to be useful in addition to, and complementary to, cerebrum tissue volumes that can be derived from MRI data without surface reconstruction and are therefore more robust to estimation in MRI data of marginal image quality. CT and SA were estimated from a subset of the representative dataset for which we had access to quality-controlled, surface-reconstructed MRI data suitable for cortical geometry ($N_{\text{total}}=105,067$, $N_{\text{unique subjects}}=84,574$ and $N_{\text{unique CN subjects}}=66,225$ for SA and $N_{\text{total}}=105,093$, $N_{\text{unique subjects}}=84,532$ and $N_{\text{unique CN subjects}}=66,181$ for CT; see **ST1.6-1.9** for demographic and other details for each study included; see also **SI19 “Primary dataset descriptions”**). Another extended phenotype was total cerebrum volume (TCV)—a composite metric defined as the aggregate volume of GMV and WMV (measurable in $N_{\text{total}}=121,650$ and $N_{\text{unique subjects}}=98,724$). TCV estimated by combining all 4 cerebrum tissue volumes, i.e., inclusive of sGMV and CSF as well as GMV and WMV, was highly similar to $\text{TCV} = \text{GMV} + \text{WMV}$ ($r=0.99$); but a smaller subset of the reference cohort had analysable data for all 4 tissue classes.

7.1. Model optimisation

CT, SA and TCV were all estimated by FreeSurfer 6.01 and analysed using the same GAMLSS modeling strategy (see **SI1-6**) as we originally used for growth charts of cerebrum tissue volumes. For 2 extended phenotypes (TCV and SA), optimal GAMLSS model specification converged on 3rd order polynomial fits for μ and σ and a 2nd order polynomial fit for mean thickness on the μ and σ terms (**Fig. S7.1**). We found that fractional polynomial modelling for ν resulted in model instability, i.e., the GAMLSS model specification process did not converge on an optimal parameterisation, and these terms were therefore not included as fixed-effects of time in the GAMLSS model. On the other hand, model specification endorsed the inclusion of study-specific random effects on both mean and variance (μ and σ terms) of all extended phenotypes.

Fig. S7.1. Optimization of GAMLSS model specification by analysis of the Bayesian information criterion (BIC) for multiple possible models on the generalised gamma distribution For each of three global metrics – TCV, total SA and mean CT – we compared model fit across multiple possible models combining fractional polynomial fixed effects of time and study-specific random effects on statistical moments of MRI phenotypes. Model goodness was quantified by the Bayesian information criterion (BIC) with greater log BIC indicating better-fitting models. Here log BIC is plotted relative to the best-fitting model with lowest BIC for each combination of fractional polynomials and random effects for which the model converged. All BIC values were scaled to the lowest value for the set of models fitted to each cerebrum tissue volume (log-scored difference to the lowest scoring model). For all phenotypes the best-fitting model included 3 fractional polynomials for μ ; and for all but CT the ordering also suggested 3 polynomials for σ . The various models fitted are summarised by y-axis labels denoting the base fractional polynomial configuration (“baseFO”) that are structured as follows: baseFO[a][b][c][x][y][z], where a-c denote the number of fractional polynomials included in the age term on μ , σ , and ν respectively, and x-z denote whether a study random effect was estimated for each of the model components (1 means a study random effect was included, 0 means no study random effect was included).

7.2. Normative trajectories of extended global MRI phenotypes

Following the data-driven determination of the optimal GAMLSS specification of the number of random-effect fractional polynomials on each of the distribution parameters, normative trajectories were generated using the same framework as outlined in **SI1-6** including the same bootstrapping procedure. Briefly, we generated 1,000 bootstrap iterations with stratified (by study and sex) sampling with replacement. The figure below (**Fig. S7.2.1**) shows the mean trajectory across bootstraps with a shaded region indicating the 95% confidence intervals (across the bootstrap replicates). In addition, and analogous to our primary phenotypes, we evaluated the stability of all GAMLSS derived study specific parameters (**Fig. S7.2.2**). Again, we find that smaller studies in specific age ranges tend to have somewhat wider confidence intervals on both mean and variance parameters.

A| Bootstrapped confidence interval (log-scale)

B| Bootstrapped confidence interval (non-log)

Fig. S7.2.1. Normative trajectories of median and bootstrapped confidence intervals for three extended global MRI phenotypes, from left to right: TCV, SA and CT. A | Sex-stratified curves plotted on a log scale. B | Sex-stratified curves plotted on natural scale.

Fig. S7.2.2. Pointrange plot of study-specific estimation of the first (μ) and second (σ) parameters of the generalised gamma fitting (where present in the selected model). Confidence intervals across bootstraps (see above) are shown and dots are coloured by the range of the confidence interval. Where not observable, the confidence intervals are smaller than the size of the dots.

7.3. Quality control of extended global MRI phenotypes

We applied the same quality control procedures for extended global MRI phenotypes as for cerebrum tissue volumes (S12), but excluded individuals with $EI < 217$ (~5%). No large effect of EI on centiles was found, nor did visual classification of a subset of raw images reveal centile

differences across included QC classes—apart from in the 2 worst rated classes of images that constituted less than 5% of the data, exclusion of which did not affect models. We note, however, that especially for phenotypes extracted from the reconstructed surfaces, averaging (as in the case of mean thickness) and summing (as in the case of total surface area) likely mitigated the impact of regional reconstruction inaccuracies driven by bad data quality (see also **SI8** on regional variability).

Fig. S7.3.1. Association between EI and estimated centiles. Spearman correlations between Euler Index (EI) and centiles for extended phenotypes revealed a negligible association between EI estimated image quality and derived centiles.

Fig. S7.3.2. Manual quality control rating from visual inspection of raw data. A small subset (~5%) of the two worst categories of raw data showed significant deviations in their estimated centiles. Excluding this subset from model estimation did not impact the model. Bars are coloured by log-scaled sample size.

7.4. Stability of out-of-sample centile scoring for extended global phenotypes: LOSO analyses

Analogous to the primary four phenotypes (**SI4**) we conducted a LOSO analysis of all studies that included the extended phenotypes. While the overall variability, i.e., standard deviation across bootstrap iterations, across studies was low (<0.05 centiles), a similar pattern of increased variability of OoS estimation emerged whereby smaller studies or those in a narrow age range in a period of rapid change were slightly more variable (**Fig. S7.4.1-2**).

Fig. S7.4.1. Stability of OoS estimates of centile scores on three extended global MRI phenotypes when each study was excluded from the reference dataset before bootstrapping. The standard deviation of bootstrapped centile scores (y-axis) is plotted for each study (x-axis) for each phenotype, from top to bottom panels: total cerebrum volume, mean cortical thickness and total surface area. Each study- and phenotype-specific boxplot is coloured according to log sample size. For each study, we estimated the normative model leaving that study out of the reference dataset and repeated this procedure after iteratively bootstrapping the reference dataset 1,000 times. We estimated the OoS centile scores for each individual in the left-out study, normalised by each of the bootstrapped normative trajectories. This procedure allowed us to summarise the reliability of the OoS estimates of centile scores in terms of the standard deviation of the 1,000 centile scores generated for each bootstrapped resampling of the reference dataset. Studies are ordered by median standard deviation of out-of-sample centile scores (small to large) indicating that scans are reliably assigned centile scores with the out-of-sample approach.

Fig. S7.4.2. Stability of out-of-sample estimates of centile scores on extended global MRI phenotypes across age and sample size. Standard deviation (sd) of individual centile scores for the extended neuroimaging phenotypes were computed across leave-one-study-out lifespan models, and plotted as a function of age (top) and sample size (bottom) for each study.

Ref 3/4:

Notwithstanding the reference to “personalized” or “precision” medicine’ in the 2nd sentence of the introduction, most of the results are organized around group differences. There is clearly much value in the ability to make group comparisons in an appropriately normalized fashion across MR studies collected with disparate imaging protocols. However, it feels like some discussion is warranted, in the main text, of whether the results in the manuscript provide any direct support for the notion (or aspiration) that the centilized brain chart outputs have value for individualized prediction.

We agree this is an important point that needs further discussion. We have now expanded upon this point in the discussion as follows:

<<The following changes have been made to the main text>>

In: Discussion

We have aggregated the largest neuroimaging dataset to date to modernise the concept of growth charts for mapping typical and atypical human brain development and ageing. The ~100 year age range enabled the delineation of milestones and critical periods in brain maturation, revealing an early growth epoch across its constituent tissue classes -- starting before 17 post-conception weeks, when the brain is at ~10% of its overall size and ending at ~80% by age 3. Individual centile scores benchmarked by normative neurodevelopmental trajectories were significantly associated with neuropsychiatric disorders as well as with individual differences in birth outcomes

and fluid intelligence (**SI5.2** and **SI12**). Furthermore, imaging-genetics studies⁴⁴ may benefit from the increased heritability of centile scores compared to raw volumetric data (**SI13**). Perhaps most importantly, GAMLSS modeling enabled harmonisation across technically diverse studies (**SI5**), and thus leveraged the potential power of aggregating MRI datasets at scale.

The current results also bode well for future progress towards individualised prediction⁴⁵. By providing an age- and sex-normalised metric, centile scores enable trans-diagnostic comparisons between disorders that emerge at different stages of the lifespan (**SI10-11**). The generally high stability of centile scores across longitudinal measurements also enabled assessment of documented changes in diagnosis such as the transition from MCI to AD (**SI14**), which provides one example of how centile scoring could be clinically useful in quantitatively predicting or diagnosing progressive neurodegenerative disorders. The analogy to paediatric growth charts is not meant to imply a predetermined or immediate application for brain charts in a typical clinical setting. However, our provision of appropriate normative growth charts and on-line tools creates an opportunity to quantify atypical brain structure, precisely benchmarked against age- and sex-typical distributions, and thus to enhance diagnostic yield from clinical scans as well as neuroimaging research studies.

Ref 3/5:

The group comparisons throughout the manuscript are primarily structured in the language of statistical significance (p-values), with minimal presentation of effect sizes. Effect sizes should be provided whenever a viable effect size measure exists, so as to provide information on the magnitude of the effect independent of the sample size.

We agree that it is important to report effect sizes as well as *P*-values. We have now reported Cohen's *d* effect size estimates (including Hedges confidence intervals for these estimates⁴³) for all case-control comparisons in relevant tables of supplemental material: **SI Tables ST2.1-2.7**. The main figure (formerly **Fig. 3** now **Fig. 4**) already listed the median case-control difference on the same scale as the original centiles to highlight the group level difference. Broadly speaking, all case-control differences referenced in the manuscript were medium ($d > 0.5$) to large ($d > 0.8$) and effect sizes are now reported for all key statistical results highlighted in the main text.

Ref 3/6:

The quality and added value of the Supplemental material is uneven. Some of it adds considerable value, but some of it is also of marginal quality or seems unnecessary. A number of the supplemental figures have poorly labelled axes or titles. In general, the captions of the SI figures need to be expanded to provide more clarity/details on what is being shown, and the captions or associated text can do a better job of explaining the purpose of the analysis and conclusion to be drawn within the SI. An appreciable number of specific comments/examples related to this point are provided below.

We thank the reviewer for their exceptionally detailed and critical appraisal of the supplementary material. We have addressed all the minor points listed below as well as aiming overall to improve and even-out the quality of the supplementary data displays and figure legends. Dataset descriptions are more balanced throughout and redundant figures have been removed (or made available only through the website).

Ref 3/7:

The font sizes in some of the figures tend toward being too small.

The goal of presenting large amounts of data in a legible format has certainly been a challenge in the present project. We have increased the font sizes where possible without interfering with the presentation of the data itself. We also note that interactive versions of most figures are available on www.brainchart.io, where the size of figures and labelling can be arbitrarily expanded by the user. We welcome further editorial guidance concerning optimal legibility of the figures, if needed, but hopefully this issue is much improved in the revision.

Other more minor or specific comments and suggestions follow:

Ref 3/8:

It would be helpful if the SI included some discussion of model convergence and how that is assessed within GAMLSS – p. 9 of the SI mentions model instability and lack of convergence but no details are provided on how that was assessed.

We have added a new section to the Supplementary Information explaining model convergence and the bootstrapping procedures used to assess model (in)stability: see **SI1.2 “Convergence within GAMLSS”**.

<<The following changes have been made to Supplementary Information>>

1.2 Convergence within GAMLSS

Model convergence within GAMLSS, like many iteratively fitting statistical models, is defined in terms of the estimated likelihood staying equivalent across several iterative steps, where equivalence is in terms of a defined convergence threshold. The threshold is with respect to changes in the (log-)likelihood between iterations (we use the default convergence criterion of 0.001)^{7,8}. Instability, or non-convergence, is typically when the GAMLSS model cannot converge on a maximum likelihood estimate and jumps between multiple solutions, whose likelihood values differ by more than the threshold and hence the algorithm never converges.

If the model is over-parameterised there may be multiple solutions that fit the data, which will lead to non-convergence. Equivalently, within the bootstrapping procedure, it is possible for a bootstrap replication to become degenerate, meaning the resampled subset of data causes the model fitting to fail, e.g., the bootstrap replicate of a small study may, by chance, consist of copies of only one subject and have no variability with which to estimate the study random-effects. We employ a stratified bootstrap procedure to limit this issue (see **SI3.2.2 “Bootstrap analysis”**); but given the sample size of some primary studies we experienced a small number (<1%) of model convergence failures across bootstrap replicates. A priori, we deemed the model unstable if more than 5% of bootstrap replicates failed to converge but this situation did not occur for any of the MRI phenotypes.

Ref 3/9:

Order of presented phenotypes should be the same across all figures, both in the main text and SI. E.g., Figure 1 is ordered as GMV, WMV, sGMV, Ventricles, so that order should

be maintained across all figures, both in the main text (e.g., Figure 4B differs) and Supplement (which uses a variety of orders – e.g., Fig S1.2, S2.2.3).

All figures have now been updated so that the cerebrum tissue volumes are consistently presented in the same order.

Ref 3/10:

Even though the code is available in GitHub, it would be helpful to include brief code snippets of the GAMLSS modeling in R within the SI as a mechanism to concisely, but technically, explain some of the modeling. This will help knowledgeable individuals quickly see exactly what was done, without needing to slog through a (potentially complicated) code base.

Given the complex nature of the R code -- including bespoke alterations to the GAMLSS code base for nested model fitting and refitting, and distributed computing on a high performance computing cluster -- we considered that it was possible to provide brief code snippets that were accessible while also accurately reflecting the analyses performed. We have therefore elected to address this comment by improving the accessibility of the GitHub repository. In particular, we now provide a tutorial (<https://github.com/ucam-department-of-psychiatry/Lifespan>) using simulated data on the GitHub, alongside improved descriptions of customised functions contained in the code. We are open to including this tutorial in the supplemental information, but currently we feel that this would negatively impact the readability of an already extensive supplementary document. In a further effort to provide concise, technical summaries of the modeling, we have also added relevant equations to the supplementary information, and systematically cross-referenced these formal representations of the model to more informal descriptions throughout the paper.

Ref 3/11:

Abstract mentions 122123 scans from 100071 individuals. Figure 1 caption says 120685 scans. Summing the N column in SI Table 1.1 yields 121163. Summing the ‘total.cn’ column (cross-sectional N?) in that same table yields 92081. Why the differences? And if Figure 1 is based on the cross-sectional data only, shouldn’t the number of scans be closer to the total number of individuals rather than greater than 120k?

We apologise for any confusion concerning the numbers of scans available for different aspects of the analysis. We have now finalised the reference dataset and double-checked all sample sizes for the various analyses reported throughout the paper and supplementary information. The different rows of **Fig. 1** depict different subsets of the total dataset, i.e., panel A depicts all the primary studies comprising the reference dataset; panels B, C and D depict all the cross sectional data on healthy controls used for GAMLSS modelling, with varying numbers of scans available for normative modeling of different phenotypes. We now provide additional demographic details for these cross-sectional control datasets used for normative modeling of each MRI phenotype: see **ST1.2-ST1.42**.

Ref 3/12:

The convention of denoting panels as “A|” rather than “A.” or “A)” seems odd to me, and leads to situations (in the Supplement) where it visually appears to be “A|” (A-eye, rather than A-bar).

We have added an extra space to avoid this confusion, so panels are denoted “A |” rather than “A|”, to clarify their interpretation as “A-bar” rather than “A-eye”. However, we would be happy to adapt to further editorial guidance concerning conventions for figure panel lettering.

Ref 3/13:

It seems like Figure 1B should use some sort of density plot, rather than simply plotting symbols on top of each other. If necessary, the attempt to color code individual studies can be dropped (in that figure and other SI figures) as its impossible to map a color to a given study anyway.

In an effort to clarify Fig. 1B, we have now changed the colour-coding to denote sex (instead of primary study), which is more aligned with the colour-coding of Figs. 1C, 1D and 1E. We also note that density plots are shown in Fig. 1A. More details on the demographics and other characteristics of each primary study included in the reference dataset are now provided in Table S1.

<<The following changes have been made to the main text>>

A | Aggregated MRI Datasets

B | Cerebrum tissue volumes

C | Normative trajectories

D | Normative variance

E | Normative rate-of-growth

Fig. 1. Human brain charts. A | MRI data were aggregated from 100 primary studies comprising 123,984 scans that collectively spanned the age range from late pregnancy to 100 postnatal years. Box-violin plots show age distributions (log-scaled) for each study coloured by its relative sample-size (log-scaled) B | Non-centilied bilateral cerebrum tissue volumes (right to left: grey matter, white matter, subcortical grey matter and ventricles) are plotted for each cross-sectional control scan, point-coloured by sex, as a function of age (log-scaled). C | Normative brain growth curves, analogous to paediatric growth charts, were estimated by generalised additive modelling for location scale and shape (GAMLSS), accounting for site- and study-specific batch effects, and stratified by sex (female/male curves coloured red/blue). All four cerebrum tissue volumes demonstrated distinct, non-linear trajectories of their medians and 95% centile boundaries as a function of age over the life-cycle. Demographics for each cross-sectional sample of healthy controls included in the reference dataset for normative GAMLSS modeling of each MRI phenotype are detailed in **ST1.2-1.7**. D | Trajectories of median between-subject variability and 95% confidence intervals for four cerebrum issue volumes were estimated by sex-stratified bootstrapping (1,000 times; see **SI3** for details). E | Rates of volumetric change across the lifespan for each tissue volume, stratified by sex, were estimated by the first derivatives of the median volumetric trajectories. For solid (parenchymal) tissue volumes, the solid horizontal line ($y=0$) indicates when the volume of each tissue stops growing and starts shrinking; the solid vertical line indicates the age of maximum growth of each tissue. See **ST2.1** for all neurodevelopmental milestones and their confidence intervals. Note that y-axes in panels B-E are scaled in units of $10,000 \text{ mm}^3$ (10ml).

Ref 3/14:

Figure 1C: The 95% centile boundaries are supposed to be “dotted” but mostly appear to be solid lines.

This graphical typo has been corrected: it was the unfortunate side-effect of log-scaling a rather dense figure and using dashed instead of dotted lines.

Ref 3/15:

Figure 2: In the lower half of the figure, the “top grey section” isn’t very clear. Why are the “empirical age-range (dark grey)” ranges so disparate from the “diagnostic age ranges (black outlines)”, and more broadly, what is the point being made by that distinction? Also, it’s inherently confusing to have a ‘key’ with the text “From literature” with a black outline but the same interior color of gray that represents the overall age range of the current study (which itself seems rather unnecessary to include, as the overall age range simply spans the same range as the top portion of the figure).

This figure has been updated (and is now **Fig. 3** in the main text) to incorporate the reviewer’s suggestions to improve clarity and to represent new results on extended global cortical metrics.

<<The following changes have been made to the main text>>

Fig. 3. Neurodevelopmental milestones. Top panel: A graphical summary of the normative trajectories of the median, i.e., 50th centile, for each global MRI phenotype, and key developmental milestones, as a function of age (log-scaled). Circles depict the peak rate-of-growth milestones for each phenotype (defined by the maxima of the first derivatives of the median trajectories; see Fig. 1E). Triangles depict the peak volume of each phenotype (defined by the maxima of the median trajectories), definition of GMV:WMV differentiation is detailed in S19.1. Bottom panel: A graphical summary of additional MRI and non-MRI developmental stages and milestones. From top to bottom: blue shaded boxes denote the age-range of incidence for each of the major clinical disorders represented in the MRI dataset; black boxes denote the age at which these conditions are generally diagnosed as derived from literature⁴¹ (Online Methods); brown lines represent the normative intervals for developmental milestones derived from non-MRI data, based on previous literature and averaged across males and females (Online Methods); grey bars depict age ranges for existing (WHO and CDC) growth charts of anthropometric and ultrasonographic variables. Across both panels, light grey vertical lines delimit lifespan epochs (labelled above the top panel) previously defined by neurobiological criteria⁴². Abbreviations: resting metabolic rate (RMR), Alzheimer's disease (AD), attention deficit hyperactivity disorder (ADHD), anxiety or phobic disorders (ANX), autism spectrum disorder (ASD, including high-risk individuals with confirmed diagnosis at a later age), major depressive disorder (MDD), bipolar disorder (BD), and schizophrenia (SCZ).

Ref 3/16:

Fig 3: Panel B: Doesn't really provide much intuition as to how the CMD is calculated, or what it represents. Panel D: Why are error bars only present for some of the data points? Are they too small to be seen? If so, that should be stated. Also, are the bars STD's or SEM's? A similar comment applies to the error bars elsewhere (e.g., Fig S9.2).

The CMD is a multivariate-scaled summary across multiple centiled phenotypes. Unlike a simple average of the phenotypes, CMD explicitly quantifies deviation (from 0.5, the center of the centile distribution) based on the principal component space of all constituent centiles, and therefore exploits the intrinsic covariance between the phenotypes. While it is difficult to make a simple figure to illustrate how a multivariate statistic like CMD is calculated, we have edited the figure and caption (**Fig.4**) in the revised manuscript. We have also expanded our supplementary description of how the CMD is calculated; see **SI1.6 “Centile Mahalanobis distance”**. Error bars in panel D are depicted for all features but are indeed too small to be observed relative to the others, this has now been clarified in the Fig. 4 caption.

<<The following changes have been made to the main text>>

Fig. 4. Case-control differences and heritability of centile scores. **A | Centile score distributions** for each diagnostic category of clinical cases relative to the control group median (depicted as a horizontal black line). The median deviation of centile scores in each diagnostic category is overlaid as a lollipop plot (white line with circle corresponding to the median centile score for each group of cases). Pairwise tests for significance were based on Monte-Carlo resampling (10,000 permutations) and *P*-values were adjusted for multiple comparisons using the Benjamini-Hochberg False Discovery Rate (FDR) correction across all possible case-control differences. Only significant deviations from the control group median (with corrected $P < 0.001$) are highlighted with an asterisk. For a complete overview of all pairwise comparisons, see **SI10** and **ST3**. Groups are ordered by their multivariate distance from the control group (see panel C and **SI10.3**). **B | The centile Mahalanobis distance (CMD)** is a summary metric of multivariate deviation that quantifies the aggregate atypicality of an individual scan in terms of all MRI phenotypes. The schematic shows segmentation of four cerebrum tissue volumes, followed by estimation of univariate centile scores, leading to the orthogonal projection of a single subject (Sub_x) onto the four principal components of the control group (CN; coloured axes and arrows): the CMD for Sub_x is then the sum of its distances from the CN group mean on all 4 dimensions of the multivariate space. **C | Probability density plots of CMD across disorders.** Vertical black line depicts the median CMD of the control group. Asterisks indicate an FDR-corrected significant difference from the CN group ($P < 0.001$). **D | Heritability of raw volumetric phenotypes and their centile scores** across two twin studies (ABCD and HCP). All dots have error bars for the standard error, but in some cases these are too narrow to be observed. Abbreviations: control (CN), Alzheimer's disease (AD), attention deficit hyperactivity disorder (ADHD), anxiety or phobia (ANX), autism spectrum disorder (ASD), anxiety or phobic disorders (ANX), mild cognitive impairment (MCI), major depressive disorder (MDD),

schizophrenia (SCZ); grey matter volume (GMV), subcortical grey matter volume (sGMV), white matter volume (WMV), centile Mahalanobis distance (CMD).

<<The following changes have been made to the Supplementary Information>>

1.6 Centile Mahalanobis distance

To create an integrated measure of normative deviation across all centile scores we computed a Mahalanobis distance²⁰ in the 4-dimensional feature space relative to the normative mean across those phenotypes. This centile Mahalanobis distance (CMD), D_M , can be formalised as follows:

$$D_M(x) = \sqrt{(x - \mu)^T S^{-1} (x - \mu)} \quad (1.6.1)$$

where x denotes the set of observations across multiple phenotypes, μ denotes the mean across those observations, and S denotes the covariance matrix across both. The squared Mahalanobis distance is also equivalent to the sum of squares of all non-zero standardised principal components scores (as illustrated in **Fig.4B**). As such, CMD provides an indication of the distance of an individual from the centre of the normative multi-dimensional (multi-phenotype) space, taking into account the potential correlated structure of the dimensions (and thereby being arguably less sensitive to outliers along a single dimension than other possible distance metrics). The scale-invariant nature of CMD also makes it generalisable to centile scores on additional MRI phenotypes as they are included in the future.

Ref 3/17:

Are all the probability density plots throughout the manuscript computed in a similar manner (e.g., same kernel approach or degree of smoothing) using the same analytical tool? A description of the specifics of the density plot construction in the SI methods seems warranted given the prominence of density plots throughout the manuscript.

The density plots throughout the manuscript were all generated with the same smoothing kernel and more details are now provided to describe the relevant methods. For consistency, we have also updated the density plots in the supplementary material to be in the same style and avoid any confusion.

<<The following changes have been made to the Supplementary Information>>

In: *SI10.2 “Multi-modality of centile distributions in clinical disorders”*

To explore the possible or even likely existence of subgroups within the space of centile scores, we assessed the number of peaks in the probability density function. Density plots were generated with the ‘geom_flat_violin’ option from the Raincloud package⁶¹. Estimation of densities and the resulting number of peaks were done using the default settings of the ‘density()’ function in the R stats package⁶² using a Gaussian smoothing kernel^{63,64} which defaults to 0.9 times the minimum of the standard deviation and the interquartile range divided by 1.34 times the sample size to the negative one-fifth power (Silverman's ‘rule of thumb’⁶⁵); unless the quartiles coincide, when a positive result will be guaranteed. The number of peaks was defined as the inflection point on these Gaussian smoothed density curves. Unimodality of smoothed density curves was tested using Hartigan's dip-test⁶⁶ which indicated that none of the distributions were perfectly unimodal (see **ST4.1-4.7**).

Fig. S10.2.1. Probability density plots of centile scores on cerebrum tissue volumes for clinical cohorts with at least N=500 diagnosed cases. Labels underneath each density plot show the estimated number of peaks or modes in the smoothed distribution.

Fig. S10.2.2. Probability density plots of centile scores on extended global MRI phenotypes for clinical cohorts with at least N=500 diagnosed cases. Labels underneath each density plot show the estimated number of peaks in the smoothed distribution.

Ref 3/18:

I would suggest breaking the section header “Longitudinal centile changes and novel data” in the main text into two distinct section sub-headings, as the two cover completely different topics.

We have adopted this suggestion and now report these results under separate section headings in the main text. Please also see revised **SI4 “Out-of-sample centile scoring: bias, stability and reliability”** and **SI14 “Longitudinal centiles”**

Ref 3/19:

Main text, p. 8 (“Longitudinal centile changes and novel data” section): It is unclear if the quantification in the first paragraph (“all median <5%” and “~5% median difference”) reflects a percent difference (and if so, relative to what), or a percentage *point* difference (i.e., a 0.05 difference in centile values).

This referred to a percentage point difference and has now been clarified in the main text.

Ref 3/20:

Fig 4C – too little detail to quickly grasp what was done. What is a cloned NSPN?

We have completely revised Fig. 4 (now **Fig. 5**) in the main text and no longer include the sample size simulation in the main text. We have moved this aspect of our analyses to the supplementary materials where there is space to more fully describe the analyses. In revised **SI4.5 “Effects of sample size on reliability of out-of-sample centile scores”**, we have expanded the explanation of the clone studies, including updated **Fig S4.5**.

<<The following changes were made to the main text>>

In ***Out-of-sample centile scoring of “new” MRI data***

Fig. 5. Schematic overview of brain charts, highlighting methods for out-of-sample centile scoring. Top panel: Brain phenotypes are measured in a reference dataset of MRI scans. GAMLSS modeling is used to estimate the relationship between (global) MRI phenotypes and age, stratified by sex, and controlling for technical and other sources of variation between scanning sites and primary studies. Bottom panel: The normative trajectory of the median and confidence interval for each phenotype is plotted as a population reference curve. Out-of-sample data from a new MRI study are aligned to the corresponding epoch of the normative trajectory, using maximum likelihood to estimate the study specific offsets (random effects) for three moments of the underlying statistical distributions: mean (μ), variance (σ), and skewness (ν) in an age- and sex-specific manner. Centile scores can then be estimated for each scan in the new study, on the same scale as the reference population curve, while accounting for study-specific “batch effects” on technical or other sources of variation (see **SI1.8** for details).

<<The following changes were made to the supplementary information>>

4.5. Effects of sample size on reliability of out-of-sample centile scores

To assess the validity of the OoS estimates we generated ‘clones’ of existing datasets. Clones are resampled copies of studies included in the reference dataset used to estimate the study specific GAMLSS parameters, that are then treated as if they were “new” studies using the methods for out-of-sample centile scoring. This allows us to compare the OoS estimates to a relative truth, i.e., from the original, non-cloned version of the study included in the reference dataset, we know what the GAMLSS parameters ‘truly’ are, and we have an estimation of their ‘true’ uncertainty from the bootstrap resampling distributions. Thus for a given study dataset, D_m , we generate a cloned copy D_1 , and if our approach is unbiased we expect the out-of-sample parameter estimates for D_1 to be equal to the in-sample parameters estimated for D_m , i.e., $\gamma_{,m}$ (representing the set of random effects estimated by in-sample analysis of the original study treated as part of the reference dataset) should approximate $\gamma_{,1}$ (representing the set of random effects estimated by OoS analysis of the cloned study treated as a new dataset; - see **SI1.8 “Out-of-sample estimation”** and **Fig. S4.5**).

In other words, we validated the OoS estimation by simulating a “new” study with the same underlying distribution used for one of the studies included in the reference dataset. Hence, we expect the OoS random-effect estimates for this ‘clone’ to agree with the in-sample random-effect estimates. More formally, we are comparing $\gamma = MLE_{\beta, \gamma}(D)$ and $\gamma^{clone} = MLE_{\gamma}(D_{clone} | \beta(D))$, where the clone is contained within the data, i.e., $D \cap D_{clone} = D_{clone}$; see **SI1.8 “Out-of-sample estimation”** for further details on OoS MLE estimation. As illustrated in **Fig. S4.5**, these simulations indicated good performance for the OoS approach for “new” study sizes greater than $N=100$ scans.

Fig. S4.5. Out-of-sample estimates of cloned study random-effect parameters compared to in-sample estimates of random-effect parameters in the original or non-cloned study. The plot shows random-effects estimated using the out-of-sample approach across a range of possible sample sizes for a “new” study, generated by taking subsets of the same cloned study with uncertainty intervals derived from the bootstrap replicates. The purple horizontal lines are the equivalent in-sample estimates of the random-effects parameters. We see that the out of sample estimates are somewhat unreliable below $N=100$ subjects, but with larger samples the out-of-sample estimates from the cloned data converge with the in-sample estimates from the original data for both μ -component and σ -component random effects.

Ref 3/21:

Minor grammatical/syntactical errors in Supplement. (e.g., “each terms of the generalized gamma distribution”; lack of space between symbols/equations and following text). Other little errors in various places. The SI needs a careful proof-read before re-submission by a someone with an eye for these issues.

All the supplementary materials have been thoroughly edited and checked by all co-authors in an effort to eliminate any grammatical, syntactical or formatting errors in the text or equations.

Ref 3/22:

Inconsistent formatting of “i.e.” and “e.g.” – both should always be followed by a comma.

Thank you. This has now been corrected.

Ref 3/23:

Fig S1.1: Is “relative AIC” based on a *ratio* or *difference* to the lowest AIC value? If a ratio (which is what “relative” inherently implies) then seems odd that there is such a pronounced difference with the reference model. E.g., For GMV, all models except for “Generalized Beta type 2” had an AIC that was ~ 3000 times greater (or more) than the “Generalized Gamma”.

The relative AIC is based on a difference to the lowest AIC, not on a ratio. However, since BIC was the main criterion used for model selection we have replaced this figure to denote the relative BIC for each model instead (see also our later response to Ref 3/24).

<<The following changes have been made to the Supplementary Information>>

Fig. S1.1. Relative Bayesian information criterion (BIC) for each family of distributions of cerebrum tissue volumes evaluated for GAMLSS modeling. Log BIC scores are shown in terms of their difference from the lowest BIC score, corresponding to the best-fitting form of the outcome distribution. All BIC values were scaled to the lowest value for each cerebrum tissue volume. For all phenotypes, a generalised gamma (GG) distribution provided the best fit. Distribution family acronyms are adapted directly from the way they are listed within the GAMLSS package ⁸.

Ref 3/24:

Fig S1.2: The y-axis labels are completely cryptic. Some ‘key’ is needed in the caption for understanding the naming convention. Also, caption should mention that the model being investigated is the generalized gamma. Additionally, the order of the phenotypes in Fig. S1.2 should match those in S1.1, and the figure titles should be simplified. Last, why the switch to BIC as a criterion vs. AIC?

We accept that this figure was not accessible. It has been updated in line with the reviewer’s comments and a key has been added to the figure caption. We now report only model specification results using the BIC: see Fig S1.3.

<<The following changes have been made to the Supplementary Information>>

Fig. S1.3. Optimization of GAMLSS model specification by analysis of the Bayesian information criterion (BIC) for multiple possible models on the generalised gamma distribution Here \log BIC is plotted relative to the best-fitting model with lowest BIC for each combination of fractional polynomials and random effects for which the model converged. All BIC values were scaled to the lowest value for the set of models fitted to each cerebrum tissue volume (log-scored difference to the lowest scoring model). For all phenotypes, a model that included 3 polynomials for μ provided the best fit; and for all phenotypes other than sGMV the best fit also specified 3 polynomials for σ . The various models fitted are summarised by y-axis labels denoting the base fractional polynomial configuration (“baseFO”) that are structured as follows: baseFO[a][b][c][x][y][z], where a-c denote the number of fractional polynomials included in the age term on μ , σ , and ν respectively, and x-z denote whether a study random effect was estimated for each of the model components (1 means a study random effect was included, 0 means no study random effect was included).

Ref 3/25:

SI Section 1.3: The brief description of the “Model simulations” is generic and inadequate to understand what exactly was simulated.

We agree that more detail is needed to explain the simulation studies. The relevant section of supplementary information has been expanded accordingly: see SI1.4 “Model simulations”.

<<The following changes have been made to the Supplementary Information>>

1.4 Model simulations

In order to motivate the specific use of GAMLSS for lifespan modelling as done here, we designed a simulation scenario that matches our use case for a single outcome or MRI phenotype. Specifically, we simulated data from twenty studies across the lifespan. We simulated data on both healthy controls (CN) and diagnosed cases (Dx), some with longitudinal follow-up, as well as study-specific random-effects. We chose the generalised gamma for the true outcome distribution with age and sex fixed-effects, random-effects within the μ -component, and constant σ - and ν -components. The lifespan relationship was quadratic with age. Importantly, the simulated data also included a subject-level random-effect, which is fitted by the GAMLSS model. This allowed us to set the within- and between-subject covariance, which in turn allowed us to assess the utility of the longitudinal centiles (see SI1.7 “Longitudinal centiles”).

Fig. S1.4.1. Simulated data for baseline observations. A | Female and male healthy controls (CN) coloured according to 20 simulated studies, highlighting the coverage of the lifespan and the within- and between-study variability. These simulated observations ($N=13,500$) were used to estimate lifespan curves with GAMLSS in order to motivate the application to real data. B | Healthy controls (CN) and diagnosed (Dx) individuals from each study (black and red respectively) ($n=20,250$). This simulation posits a diverging lifespan trajectory for Dx individuals, such that at the start of the lifespan CN and Dx overlap but gradually separate, which is induced by using different true age-related quadratics. The specific functional form of the CN and Dx curves are $((0.4 - x) * (0.5 - x) + 1.8)$ and $((0.35 - x) * (0.3 - x) + 1.55)$ respectively. (values were scaled for computational stability and visualisation purposes).

The inclusion of individual-level random-effects within the simulation is necessary to induce a dependence between longitudinal observations. While the analysis shown in **Fig. S1.4.1** only uses baseline observations, **Fig. S1.4.2** illustrates the longitudinal follow-up for a subset of individuals across five of the twenty simulated studies for CN and Dx individuals to assess the capacity to model longitudinal trajectories. The simulated dataset also mimics the real world data with an uneven coverage of the lifespan, as shown in comparing **Fig. S1.4.3** to **Fig. 1A**.

Fig. S1.4.2. Simulated data for longitudinal observations. An illustrative sample of individuals (250 from among 1,500 for clarity) with longitudinal follow-up within the simulated data coloured by CN (black) or cases (Dx: red). Within the simulation individuals have relatively stable longitudinal trajectories relative to the between person variation, implying longitudinal centiles will be relatively stable for both CN and Dx.

Fig. S1.4.3. Box-violin plots show age distributions (log-scaled) of twenty simulated studies. The design of the simulation mimics the structure of the observed datasets, with some periods of the lifespan being represented by multiple studies, for example adolescence (studies C, S, P and L), while other periods have sparser coverage with fewer studies.

Ref 3/26:

Fig S1.3.[1-2]: Not clear what is being shown, or the point of these figures. Cryptic titles. Tiny font sizes.

We accept the reviewer's criticism of these figures in the original manuscript. The whole of supplementary section 1.3 (now **SI1.4 "Model simulations"**) has been rewritten and all the figures have been updated to improve their legibility and accessibility, as detailed in our earlier response to Ref 3/25.

Ref 3/27:

SI Section 1.4: What is the value/purpose of computing centile normalized z-scores rather than simply using the centile estimate itself? Per the text on p. 13, the latter accounts for study random-effects, while the normalized z-scores do not. Isn't it a good thing to account for the study random-effects and thus wouldn't the centile scores be preferable to the normalized z-scores?

We agree with the reviewer that centiles are likely preferable in most contexts, and indeed we strongly recommend using centiles as the default metric for any subsequent analysis. Normalised scores, based on the centile deviation of the population level random-effect, are discussed as an optional alternative to centile scores because they are scaled to the same units as the scored phenotype and so may be more interpretable in some contexts. Given that normalised scores use only the population level random-effects, they are only appropriate for scoring scans that were included in the reference dataset, i.e., only healthy controls. We now clarify this point in the section discussing normalised values: see **SI1.5 “Centile normalisation”**.

<<The following changes have been made to the Supplementary Information>>

In **SI1.5 “Centile normalisation”**

These normalised values, w , are on the same scale as the original values, y , having been corrected for the study-specific effects: namely, the μ -component and σ -component study random-effects. However, these corrections are only appropriate for scoring scans that were included in the reference dataset, i.e., healthy controls, and normalised values are therefore not useful for scoring scans from cases of clinical disorder or for out-of-sample scoring of “new” scans. We have included a brief consideration of normalised values, w , for completeness and because they may be more interpretable than centile scores in some contexts, since they are scaled to the same units as the scored phenotypes. However, for most applications (including the case-control comparisons and out-of-sample analyses reported in this paper), we therefore strongly recommend the use of centiles.

Ref 3/28:

Fig S1.4: What is the point of this figure? What do the x and y-axes represent? Figure doesn’t appear to be referenced anywhere in the text. Also, another example where is it impossible to map the colors to a specific study.

This figure was included to show how the GAMLSS model adjusts the centile scores in a study specific manner. Given that these study specific curves are already available through the online tool, and in an effort to mitigate the volume of supplementary figures, we have chosen to remove this figure from the revised supplementary information.

Ref 3/29:

Main text refers to “SI1.6” for description of the CMD, but that definition is actually in SI1.5. Related to this, the main text says the CMD was somehow computed relative to the “CN median”, but SI1.5 says that the (usual) mean was used.

We thank the reviewer for their keen eye for detail. The cross-referencing between the main text and supplementary information has been completely updated and is now correct. The method for CMD estimation has been clarified. In the case of the CN data, the mean and the median centile scores were actually identical (0.5). We now use the mean as it is more conventional, and this did not change any results.

<<The following changes have been made to the Supplementary Information>>

In **SI1.6 Centile Mahalanobis distance**

To create an integrated measure of normative deviation across all centile scores we computed a Mahalanobis distance²⁰ in the 4-dimensional feature space relative to the normative mean across those phenotypes. This centile Mahalanobis distance (CMD), D_M , can be formalised as follows:

$$D_M(x) = \sqrt{(x - \mu)^T S^{-1} (x - \mu)} \quad (1.6.1)$$

where x denotes the set of observations across multiple phenotypes, μ denotes the mean across those observations, and S denotes the covariance matrix across both. The squared Mahalanobis distance is also equivalent to the sum of squares of all non-zero standardised principal components scores (as illustrated in **Fig.4B**). As such, CMD provides an indication of the distance of an individual from the centre of the normative multi-dimensional (multi-phenotype) space, taking into account the potential correlated structure of the dimensions (and thereby being arguably less sensitive to outliers along a single dimension than other possible distance metrics). The scale-invariant nature of CMD also makes it generalisable to centile scores on additional MRI phenotypes as they are included in the future.

Ref 3/30:

SI Section 1.6: I would argue its debatable whether interquartile range is “well defined for two [or more] observations”, which forms the vast majority of the longitudinal samples in the study. (A number of on-line calculators require at least 3 values, and even with 3 values the notion of “IQR” seems sketchy. The appropriateness of IQR with such a small number of values seem to merit additional justification.

We thank the reviewer for encouraging us to further expand on the justification of IQR as a measure of within-subject variability. There are in fact multiple definitions used to operationalise IQR (9 in GNU R alone). We now further clarify the definition we chose and why this one is appropriate for 2 or more observations: see **SI1.7 “Longitudinal centiles”**.

<<The following changes have been made to the Supplementary Information>>

Thus, comparing longitudinal centiles, with varying numbers of observations per individual, is approached via a univariate summary statistic. A univariate summary for variation across observations will assess the stability of the centiles within an individual. The summary must be valid for two or more observations, the minimal longitudinal follow-up period, and be comparable across individuals. The range, i.e., $\max(q_{ij1}, q_{ij2}, \dots, q_{ijm}) - \min(q_{ij1}, q_{ij2}, \dots, q_{ijm})$, would be well defined for two or more observations; however, the range is susceptible to outliers and statistically unstable under small samples. Instead, the interquartile range (IQR) acts as a robust equivalent of the range (in the same way that the trimmed mean is a robust version of the mean). However, unlike the trimmed mean which requires a large enough sample, the IQR is valid for small samples. Unfortunately there is not a single definition of the IQR (there are 9 different definitions available within GNU R), and some versions are not defined for two observations. We use IQR calculated as a continuous value by linear interpolation (within GNU R the default, type 7), which is well defined for two observations.

Ref 3/31:

Fig S1.6.[1-2]: These results will be highly dependent on the specifics of the simulation, but per above, sufficient details on the simulation aren't provided to interpret these figures properly. Another example of cryptic axis labels and captions that are insufficient to understand the purpose of the figure.

We have expanded the explanation of the simulation: see revised S11.4 "Model simulations" and S11.7 "Longitudinal centiles", as well as our earlier response to Ref 3/25. The key aspect is an individual-level random-effect within the simulation in order to incorporate dependence between longitudinal follow-up observations. We have also expanded the figure captions to explain the purpose of the simulation: see revised Figs S1.7.1 and S1.7.2.

<<The following changes have been made to the Supplementary Information>>

Fig. S1.7.1. Comparing baseline centiles between healthy controls (CN) and diagnosed cases (Dx) in simulated data. The CN and Dx simulations follow two distinct lifespan trajectories, both quadratic in shape and starting slightly offset in early life, both peaking in mid-life with growing divergence, and fully diverging in later life. The analysis of simulated data is formally equivalent to the analysis of observational data and the GAMLSS model is fitted to only simulated data of CN baseline scans. The figure shows the distribution of baseline centile scores across the twenty simulated studies (spanning different ranges of the lifespan, in four groups: A-J, K-O, P-R, S-T; see Fig S1.4.3). We note that the Dx centiles are not uniformly distributed between zero and one, but are skewed to the lower end of the distribution as expected from the simulation scenario: namely, that the Dx simulations are always below the fitted CN lifespan trajectory. Further, the skewness of the Dx centiles increases later in the lifespan (compare study J to study A). Conversely, the CN centile distributions are uniformly distributed from zero to one as expected.

Fig. S1.7.2. Comparison of interquartile range (IQR) of observed longitudinal centiles between healthy controls (CN) and cases (Dx) in simulated data. Simulations P-T included longitudinal follow-up data for CN and Dx (non-CN) individuals. As described, the simulated analysis model (fit to the CN baseline observations) is used to derive centile scores for all observations, Dx and longitudinal. Taking the IQR as a summary statistic of within-subject variability of longitudinal centiles, the boxplots for CN simulations highlight the stability of longitudinal centiles over follow-up. For the non-CN simulations, we see an echo of the effect from **Fig S1.7.1**; importantly the collapse of IQR variability towards zero does not imply the Dx centiles are more stable per se but rather the Dx status might coincide with more limited variability by being confined to the tail end of the distribution. This plot confirms that cross-sectional brain charts can be used to benchmark longitudinal measurements.

Ref 3/32:

SI Section 1.7: Presumably, the ‘F’ in the equations on p. 19 represents the fixed effects (and not the ‘F’ of the CDF defined on p. 12)? Also, it’s not defined what it means to make a “clone” of a study in the simulation.

The reviewer is correct. 'F' represents the fitted model and hence the fixed-effects (for the mu-, sigma- and tau-components). We recognise that the mathematical notation within this section was not clear and we have re-written it to avoid duplicating symbols unnecessarily: see **SI1.8 “Out-of-sample estimation”**.

<<The following changes have been made to the Supplementary Information>>

In SI1.8 Out-of-sample estimation

Let $D = \{D_1, D_2, D_3, \dots, D_k\}$ be the combined datasets used to estimate the model parameters, specifically the fixed-effects for each component of the GAMLSS model, $\beta = (\beta_\mu, \beta_\sigma, \beta_\tau, \beta_\nu)$, and the study-specific random-effects for each component, $\gamma = (\gamma_\mu, \gamma_\sigma, \gamma_\nu, \gamma_\tau)$, where each γ contains a parameter for each dataset D_i , i.e., $\gamma_\mu = (\gamma_{\mu,1}, \gamma_{\mu,2}, \gamma_{\mu,3}, \dots, \gamma_{\mu,k})$.

In symbolic terms, we may consider the set of fixed- and random-effects from our model to be obtained from fitting the GAMLSS model,

$$(\beta_D, \gamma_D) = \text{GAMLSS}(D) \quad (1.8.1)$$

where β_D and γ_D are the maximum likelihood estimates of the fixed- and random-effects, respectively, from the GAMLSS model conditional on a given dataset, D . Note that the GAMLSS model includes specification of the functional form, namely the fractional polynomial specification; however, during OoS estimation the fractional polynomial specification of the GAMLSS model is fixed and hence has been omitted here for clarity.

For a “new” dataset, say D_m , we require inference on its study-specific random-effects parameters. However, we condition on the fixed-effects parameters from **Eq 1.8.1**, namely β_D . We can obtain these estimates from a conditional maximum likelihood estimator (MLE).

$$\gamma_{:,m} = (\gamma_{\mu,m}, \gamma_{\sigma,m}, \gamma_{\nu,m}, \gamma_{\tau,m}) = \text{MLE}(D_m | \beta_D). \quad (1.8.2)$$

Combining the OoS estimate of study-specific random-effects with the fixed-effects, we can derive centile scores for the new study in the same way as centile scores are calculated for studies that were included in the reference dataset.

Ref 3/33:

Fig S2.1.1: Some context for how to interpret a “detrended transformed Owen’s plot” would be helpful (more obscure than Q-Q plots).

More context and explanation on this is now provided, although we note that we also provide the more conventional Q-Q plot assessment: see SI3.1 “Model diagnostics”.

<<The following changes have been made to the Supplementary Information>>

In SI3.1 “Model diagnostics”

Detrended transformed Owen’s plots (DTOPs) are an alternative visual approach to assessing the adequacy of a fitted distribution, derived from a non-parametric approach to the data that uses the empirical samples to derive uncertainty intervals. DTOPs have the slight advantage over the traditional Q-Q (quantile-quantile) plots of being more flexible in relation to the form of the distribution and thus provide a way to compare goodness-of-fit across different distributions. Q-Q plots for GAMLSS fits are derived using transformations of the residuals, from the uniform 0–1 scale to the more familiar normal (Gaussian) distribution, hence they are based on a parametric approach. Neither approach alone is definitive for assessing GAMLSS fits, and Stasinopoulos⁷ recommends a variety of approaches including both Q-Q plots and DTOPs.

Ref 3/34:

SI Section 2.2.2. Were the stratified bootstrap samples generated by sampling relative to the proportion of the strata in the original data? If so, wouldn’t the results be primarily sampling the variability of the UK-Biobank and ABCD data since the bootstraps would always be dominated by data selected from ABCD and UK-Biobank?

We have added text to clarify our bootstrap resampling procedures and to justify our use of a stratified bootstrap approach. In addition, we conducted a leave-one-study-out analysis to show that our findings are not in fact unduly dominated by the few large studies: see SI3.2.2 “Bootstrap analysis”.

<<The following changes have been made to the Supplementary Information>>

The bootstrap replicates were stratified by study and sex, which maintains the relative proportions of the original datasets. We have chosen to stratify on sex since it is one of our primary fixed-effects of interest within the GAMLSS model, hence it was important to ensure that the bootstrap resampling was representative of the relative sex proportions within studies. With regard to stratifying by study, there are two inter-linked considerations: between-study differences in sample size and lifespan coverage. Failing to stratify by study sample size could cause a study to be omitted entirely from a bootstrap replicate, or more typically to have a smaller or greater number of observations, meaning the bootstrap intervals would be incoherent for study-level inference. More importantly, the normative trajectories are derived from studies across the lifespan, but each study only partially covers the lifespan; hence failing to stratify by study age-range could alter the bootstrap distribution and lead to incoherent confidence intervals for the

lifespan curves. The foetal and early postnatal periods of the lifespan would be particularly vulnerable to this effect because relatively few studies have covered this age range. Our LOSO analysis showed that the lifespan curves were not in fact unduly affected by the removal of any single study (even large ones, for example ABCD and UK-Biobank).

Ref 3/35:

Fig S2.2.3: Why are the studies ordered in reverse alphabetical order, rather than alphabetical order, which would be more intuitive?

Thanks for picking this up. The studies are now ordered alphabetically and this reorganization has been propagated into **SI7.2 “Extended global cortical phenotypes”** for the extended global MRI phenotypes.

Ref 3/36:

SI Section 2.3: The following statement is a bit imprecise: “whereas in the reference prediction curves the FreeSurfer contribution is equivalent to the grand-mean across all versions (across all studies), meaning the reference prediction curves do not represent any specific FreeSurfer version”. Namely, the grand-mean would be weighted by the proportion of given FS versions, and thus depending on those proportions, might be close to a specific FS version. Indeed, SI Table 1.3 shows that the vast majority of cases were processed with FS 6.0 (either T1 only, or T1+T2), so the reference prediction curve would be strongly weighted to FS 6.0 (to the extent that FS version has a meaningful impact – see Item (2)).

The reviewer is correct. The interpretation of the grand-mean trajectory over all versions of FreeSurfer is influenced by the relative frequency of different versions of FreeSurfer, and we have amended this sentence accordingly. However, we stress that as discussed in **SI1 “Modelling lifespan trajectories of brain maturation”**, our models account for the software version as an additive (on the link-scale) effect, meaning that all study- and individual-level centiles use the appropriate software variable rather than the grand-mean when centiles are estimated.

<<The following changes have been made to the Supplementary Information>>

In *SI3.3 Study specific curves*

The study-specific prediction curves are obtained using the same method as the reference prediction curves described in **SI1.4**, using the mu-, sigma- and nu-component equations (**Eq1.5.1**) to calculate the predicted median (i.e., 50th percentile of the outcome distribution) across age and sex. However, there are two important differences. Firstly, we include a study-specific random-effect (where present) within the prediction calculations (i.e., random-effect terms within the component equations; **Eq1.1-1.2**), whereas in the reference prediction curves these are all set to zero (effectively not included). Secondly, the study-specific predictions are for the most common FreeSurfer version used within that study (if multiple FreeSurfer versions were used), whereas in the reference prediction curves the FreeSurfer contribution is equivalent to the grand-mean across all versions (across all studies), meaning the reference prediction curves correspond to a weighted average of FreeSurfer versions. All study- and individual-level analyses appropriately adjust for the specific version of FreeSurfer used.

Ref 3/37:

Fig S2.4: In panel (B), why is “Model derived TBV” shown on the x-axis, whereas in all the other panels, the “model derived” value is shown on the y-axis?

Thanks for picking up this inconsistency. We have relabelled the axes in panels A, C and D; see revised figure, now **Fig S3.4**.

<<The following changes have been made to the Supplementary Information>>

A| Foetal ultrasound

B| Head circumference

C| 10k-in-a-day

D| Brain weight

Fig. S3.4. Validation of lifespan model-predicted values in independent datasets and modalities. A | Three foetal ultrasound datasets, B | two head circumference reference norms (foetal=INTERGROWTH consortium, postnatal=WHO), C | a brain MRI dataset not included in the present models with only binned ages available, and D | four independent post-mortem brain weight datasets across the postnatal lifespan³³ (GTEx: <https://qtexportal.org/home/>, PsychENCODE: <https://psychencode.synapse.org/>). The neuroimaging models demonstrated high correlations (predicted vs. empirical values) across each of these modalities, thus showing the potential for inter-modal aggregation in future work.

Ref 3/38:

Fig S4.1.1: The precision on the x-axis for the “Late midfetal” and “Late fetal” windows is insufficient to ascertain the actual time window being plotted (i.e., evidenced by the fact that multiple ticks display the same x-axis value).

Thanks for this feedback. We have increased the granularity of the axis labels in the revised figures, now **Figs S9.1.1 and S9.1.2** in **SI9.1 “Trajectories within developmental epochs”**.

<<The following changes have been made to the Supplementary Information>>

9.1 Trajectories within developmental epochs

To clarify the developmental trajectories at different stages across the lifespan below we provide the fitted trajectories on a non-log scale for each of the lifespan windows defined by Kang et al.⁵⁰

Fig. S9.1.1. Normative trajectories of median (and 2.5-97.5% centile boundaries) of cerebrum tissue volumes. As shown in main Fig. 1, but stratified by age-defined developmental windows – from late midfoetal to late adulthood – and plotted on natural scale of age in years (x -axis) to allow further examination of the trajectory shapes over time.

Fig. S9.1.2. Normative trajectories of median (and 2.5-97.5% centile boundaries) of extended global MRI phenotypes stratified by age-defined developmental windows – from late midfoetal to late

adulthood – and plotted on natural scale of age in years (x-axis) to allow further examination of the trajectory shapes over time.

Ref 3/39:

Fig S4.1.2: Why, in a number of the panels, is the solid line seemingly outside of the dashed lines representing the 95% CI?

We thank the reviewer for noticing this discrepancy, which was due to a plotting artefact related to variable line thickness. None of the solid lines actually fall outside the confidence bounds. We observe that for two developmental windows the estimated trajectory for between-subject variability of WMV is indeed close to the boundary of the 95% confidence interval, which likely reflects an increase of skewness of the distribution of WMV variability in that age-range. The figure has now also been updated with the inclusion of additional data: see revised **Figs S9.1.3 and S9.1.4.**

<<The following changes have been made to the Supplementary Information>>

Fig. S9.1.3. Normative trajectories of between-subject variability (and bootstrapped confidence interval) of cerebrum tissue volumes. As shown in main Fig. 1, but stratified by age-defined developmental windows – from late midfoetal to late adulthood – and plotted on natural scale of age in years (x-axis) to allow further examination of the trajectory shapes over time.

Fig. S9.1.4. Normative trajectories of between-subject variability (and bootstrapped confidence interval) of extended global MRI phenotypes stratified by age-defined developmental windows – from late midfoetal to late adulthood – and plotted on natural scale scale of age in years (x-axis) to allow further examination of the trajectory shapes over time.

Ref 3/40:

Fig S5: I would suggest using ‘Centile’ rather than ‘Quantile’ for the y-axis label, consistent with the terminology in the caption, and the use of ‘centile’ throughout the manuscript. (Not only does the caption not use the term ‘quantile’, but ‘quantile’ isn’t used a single time throughout the main text or SI text). Also, averaging across phenotypes is not strictly the same as computing a centile score for the single summary TCV measure – thus I would suggest avoiding the imprecise claim that it is “akin to computing a centile score for TCV”. (If you want the true centile score for TCV, compute them directly). Similarly, labels in Fig S9.4 use the term quantile rather than centile.

We agree it is important to use centile and other terms consistently throughout the paper. We have now corrected the title of this figure and removed the phrase “akin to computing a centile score for TCV” from the legend; see **SI12 “Associations of birth weight and gestational duration with centile scores on cerebrum tissue volumes”** including new **Fig. S12**. We also note that we have now computed centile scores for TCV in the revised manuscript; see **SI7 “Extended global cortical phenotypes”**.

<<The following changes have been made to the Supplementary Information>>

12. Associations of birth weight and gestational duration with centile scores on cerebrum tissue volumes

To examine the effects of early life stress on centile scores, we examined 5 independent samples across the lifespan with self-reported gestational age at birth and/or birth weight (dHCP, neonatal; UNC, neonatal and early infancy/childhood; ABCD, late childhood; NIH, childhood/adolescence/young adulthood; UKB, mid-late adulthood). Average centile scores on all four cerebrum tissue volumes were significantly related to multiple metrics of premature birth

across datasets (gestational age at birth, $t = 13.164$, $P < 2e-16$; birth weight, $t = 36.395$, $P < 2e-16$). This corroborates previous work indicating the ability to capture relationships between early life factors such as birth weight and brain volumetrics measured several decades later⁷².

Fig. S12. Relationships between centile scores on cerebrum tissue volumes and birth weight (left panel) and gestational age at birth (right panel) for each of 5 primary studies with relevant data available. Centile-normalised z-scores were computed for each phenotype in each individual study and then averaged across phenotypes to compute a mean centile z-score for each subject. The black dashed lines represent the relationships between mean centile scores and birth weight or gestational age at birth estimated by a linear mixed-effects model: for gestational age at birth, $t = 13.164$, $P < 2e-16$; for birth weight, $t = 36.395$, $P < 2e-16$. The black dotted line in the right panel denotes the commonly-used threshold for defining premature birth at 37 weeks post-conception.

Ref 3/41:

In the main text, SI7 and SI8 are referenced before first mention of SI4-6. It would be preferable if the SI material is numbered and ordered such that it can be introduced sequentially within the flow of the main text.

We agree. We have restructured the supplementary information after the addition of new material and all SI text, figures and tables are now ordered and numbered in alignment with the sequence of their citations in the main text.

Ref 3/42:

SI Section 7.1: What symptomology variables and criteria were used for assigning the ABCD and UK-Biobank data into clinical (“non-CN”) cohorts?

We have clarified these details in the relevant section on these datasets in the supplemental materials **SI19 “Dataset descriptions”**.

In UKB - UK Biobank:

Individuals were included in the reference dataset as healthy controls (CN) based on the response recorded in data-field 20544 (<https://biobank.ndph.ox.ac.uk/showcase/field.cgi?id=20544>) of the UKB mental health questionnaire, including only individuals who had never had mental health problem diagnosed by a mental health professional.

In ABCD

Individuals were included in the reference model as healthy controls (CN) based on the parental response to the ABCD screening and risk questionnaire (https://nda.nih.gov/data_structure.html?short_name=abcd_screen01) indicating the individual had never been diagnosed with a mental health disorder.

Ref 3/43:

Fig S7.1 and S9.3.1: Preferably, the colors used would be matched to those in Figure 3 for consistency in presentation.

We agree. In the revision, all figures have been updated to conform to the same NPG-style colour template. Note that these specific figures have now been merged into revised **S110 “Clinical applications of centile scores”** where all case-control differences are now discussed for clarity.

<<The following changes have been made to the Supplementary Information>>

Fig. S10.1.1. Case-control and between-disorder comparisons of centile scores on cerebrum volumes. The same as shown in main **Figure 4A** but not limited to comparison with the CN group only. Asterisks indicate significance after FDR correction ($q < 0.001$) as computed using Monte Carlo permutation tests and the Benjamini-Hochberg⁵⁶ procedure to correct for multiple comparisons entailed by all possible pairwise tests. Abbreviations; Control (CN), Alzheimer's Disease (AD), Attention Deficit Hyperactivity Disorder (ADHD), Anxiety or Phobia (ANX), Autism Spectrum Disorder (ASD), anxiety or phobic disorders (ANX), Mild Cognitive Impairment (MCI), Major Depressive Disorder (MDD), Schizophrenia (SCZ); Grey Matter Volume (GMV), Subcortical Grey Matter Volume (sGMV), White Matter Volume (WMV), Ventricular Cerebrospinal Fluid (CSF).

The same case-control analysis was performed for the three extended global MRI phenotypes. **Fig. S10.1.2** shows significant differences relative to the CN group in a similar presentation as represented in **Fig. 4A**. All significant pairwise combinations are visualised in **Fig. S10.1.3** and all statistical pairwise effect-sizes and P -values are provided in **ST3.5-3.7**.

Fig. S10.1.2. Case-control differences of centile scores on extended global MRI phenotypes. Centile distributions for each of the clinical disorders with $N > 500$ cases relative to the CN group median (depicted as a horizontal black line). The top row depicts the male only subset, the bottom the female only subset. The deviation in each clinical group is overlaid as a lollipop plot (white line with circle corresponding to the clinical group median). Pairwise tests for significance were done using Monte Carlo permutation (10,000 permutations) and P -values adjusted using the Benjamini-Hochberg FDR procedure for the multiple comparisons entailed by testing all possible pairs. Only significant differences to CN (corrected $P < 0.001$) are depicted here and highlighted with an asterisk. For a complete overview of all pairwise comparisons, see supplementary tables **ST3.5-3.7**. Abbreviations; control, CN; Alzheimer's disease, AD; attention deficit hyperactivity disorder, ADHD; autism spectrum disorder, ASD; anxiety or phobic disorders (ANX); mild cognitive impairment, MCI; major depressive disorder, MDD; schizophrenia, SCZ; grey matter volume, GMV; subcortical grey matter volume, sGMV; white matter volume, WMV; ventricular cerebrospinal fluid volume, CSF.

Fig. S10.1.3. Case-control and between-disorder comparisons of centile scores on extended global MRI phenotypes. The same as shown in main Fig. 4A and S10.1.1 but not limited to comparison with the CN group only. Asterisks indicate significance after FDR correction ($q < 0.001$) as computed using Monte Carlo permutation tests and using a Benjamini-Hochberg⁵⁶ correction to correct for multiple comparisons accounting for all possible pairwise combinations. Abbreviations; control, CN; Alzheimer's disease, AD; attention deficit hyperactivity disorder, ADHD; autism spectrum disorder, ASD; anxiety or phobic disorders (ANX); mild cognitive impairment, MCI; major depressive disorder, MDD; schizophrenia, SCZ; grey matter volume, GMV; subcortical grey matter volume, sGMV; white matter volume, WMV; ventricular cerebrospinal fluid volume, CSF.

Ref 3/44:

Fig S7.2.[1-3]: Very complex, with minimal guidance as to what is being shown.

These figures were intended to show case-control comparisons for a more lenient inclusion threshold on the number of cases per diagnostic category, to provide some context for the main figure (current Fig.4) which only depicted clinical disorders that had $N > 500$ cases in a given diagnostic category. However, on reflection, we agree with the reviewer that these figures were very difficult to read in printed form. We have therefore now removed Figs S7.2.1 and S7.2.2 from

the Supplementary Information. The data they represented are now provided instead by supplementary tables **ST 3.1-3.28**, which have been updated to include Cohen's *d* effect size estimates, as well as *P*-values, for each comparison. Additionally, we have noted in the supplementary information, that the online tool allows users to explore pairwise comparisons for any possible pair of clinical disorders (regardless of *N*), albeit using parametric pairwise *t*-tests rather than the computationally more intensive resampling-based inference used for the reported analyses

Ref 3/45:

Fig S7.4, panel B: Would be more intuitive if larger absolute mean differences were shown in the “hot” color (yellow), rather than the “cool” color (blue).

The figure has been updated to conform to the same NPG style colour palette as all other figures, and specifically panel B has been adjusted to have a divergent colour scale, with white zero centered; see **Fig. S10.4**.

<<The following changes have been made to the Supplementary Information>>

In 10.3 Case-control differences on CMD

In order to determine whether centiles provided sensitivity to detect case-control differences over all clinical groups at specific developmental epochs, we conducted an exploratory analysis using developmental windows as defined by Kang et al.⁵⁰ Specifically, we re-coded all diagnostic labels to either healthy controls (CN) or diagnosed cases of any disorder (DX), then estimated the centile Mahalanobis distance (CMD; analogous to **Fig. 4**) across the four cerebrum tissue volumes relative to the CN group mean (0.5). Then we ran two-sided Monte Carlo permutation tests (10,000 permutations) on CMD within each developmental window. We found overall case-control differences in CMD across the lifespan (**Fig. S10.3**), indicating that relatively increased CMD - a multivariate marker of atypicality - was associated with DX status. These differences were most strongly pronounced in late adulthood (mean difference, 0.655, $P < 0.001$; Cohen's $d = 0.25$), middle/late childhood (mean difference = 0.493, $P < 0.001$; Cohen's $d = 0.24$), adolescence (mean difference = 0.512, $P < 0.001$; Cohen's $d = 0.24$), young adulthood (mean difference, 0.363, $P < 0.001$; Cohen's $d = 0.17$) and middle adulthood (mean difference, 0.133, $P < 0.001$; Cohen's $d = 0.06$). In foetal, neonatal, and very early childhood, the current dataset was insufficiently powered to determine gross differences on disease status (**Fig. S10.3**, panel B label provides the number of individuals with any kind of diagnostic label).

A | CMD all case-control differences by developmental epoch

B | Significance testing of CMD case-control differences by developmental epoch

Fig. S10.3. Case-control differences, between healthy controls (CN) and all diagnosed cases (DX), for centile Mahalanobis distance (CMD) over all four cerebrum tissue volumes at each developmental window over the life-span. A | The relative distributions of CMD for CN and DX groups in each developmental window. B | The point-range plot of the P-values and their confidence intervals as computed using a Monte Carlo permutation test (10,000 permutations). Labels above each point indicate the number of individuals in the DX group in each developmental window. The red-dotted line shows $P=0.01$

Ref 3/46:

Fig S7.5.[1-3]: Not clear what point is being made with the inclusion of these figures.

The point of including these figures is to clarify the rationale for our preferred choice of centile Mahalanobis distance as a summary metric of an individual's deviation from multiple normative trajectories of multiple univariate phenotypes. Section **S17.5 "Summary centile comparison"** (now **S10.4 "Summary centile comparison"**) explores the different methods for summarising multiple centile scores for each subject. The purpose of this comparison is to distinguish the multivariate statistic of centile Mahalanobis distance (CMD) from the average of the univariate

centile scores for each of the MRI phenotypes. As also noted in response to Ref 3/16, CMD is unlike a simple average because it explicitly quantifies deviation (from 0.5, the center of the centile distribution) based on the principal component space of all constituent centiles, and therefore exploits the intrinsic covariance between the phenotypes. **Fig S10.4.1** plots the distribution of the two summary metrics across subjects, stratified by the binary variable of healthy control (CN) versus diagnosed cases (DX, with any clinical disorder). This demonstrates that i) CMD and mean centile have different distributions across subjects, and ii) these distributions are largely similar between CN and DX groups overall. **Figs S10.5.2-3** represent the two summary metrics for each clinical disorder, revealing the differences between them across diagnostic categories.

<<The following changes have been made to the Supplementary Information>>

10.4 Summary centile comparison

Here we highlight the difference in two summary centile metrics that could be used to characterise (a) typicality across all neuroimaging phenotypes: the mean centile and the centile Mahalanobis distance. The mean centile is simply the average of the centile scores for all 4 cerebrum tissue volumes for a given subject. The centile Mahalanobis distance (CMD) is a summary dispersion metric, which is statistically distinct from the mean (see **Fig. S10.3**, **Fig. S10.4.1** and **S11.6**). Whereas the mean centile score is normally distributed across subjects, CMD is skewed—biased towards lower estimates. Thus, while the mean centile can obscure correlated changes in phenotypes—such as increased CSF with decreased GMV in AD patients—CMD can directly capture this covariation. Overall, both metrics showed relatively similar distributions in diagnosed cases and healthy controls (**Fig. S10.4.1**), with highly varying estimates across diagnostic groups (medians for each category plotted in **Fig. S10.4.2** and **Fig. S10.4.3**).

Fig. S10.4.1. Hex plot showing all-case-control differences (between healthy controls (CN) and all cases regardless of diagnostic category (DX) of centile scores averaged across phenotypes (mean centile) versus the preferred multivariate metric of centile dispersion (CMD: centile Mahalanobis distance). Count refers to the hex-binning percentage of the total dataset within the CN and DX groups. Thus, as each coloured hexagon represents multiple data points (subjects), it is clear that both groups show a skewed distribution for CMD despite a relatively normal distribution for the mean centile (with DX having a preponderance of subjects with low mean centiles, see **Fig. S11.1**).

Ref 3/47:

Fig S8.[1-2]: Needs a more detailed caption explaining what is being shown. e.g., What do “years” and “total” represent? Also, why is no clustering shown for S8.1 – was $k=1$ the optimal clustering?

We thank the reviewer for pointing this out. The optimal clustering (k clusters) was determined by using the gap statistic, which is now added to the captions ($k=1$ for **Fig. S11.2.1** and $k=3$ for **Fig. S11.2.2**). The captions also have been corrected to contain information linking the legend scales and the accompanying annotations on the x-axis of each heatmap. These changes are reflected in the updated and renumbered **S11 “Cross diagnostic analyses”**.

<<The following changes have been made to the Supplementary Information>>

Fig. S11.2.1. Profiles of centile scores for median cerebrum tissue volumes (GMV, WMV, sGMV, CSF), centile scores for between-subject standard deviation of cerebrum tissue volumes (GMVv, WMVs, sGMVv, CSFv), mean age, percentage female, and number of primary studies for 7 clinical disorders with $N > 500$ cases, as per Fig. 3. Legend (x-axis and right annotation): 'years' corresponds to 'age' and represents median age of the diagnostic groups, '% female' corresponds to 'sex' and represents the percentage of female patients in each diagnostic group, 'total' corresponds to 'study' and represents the number of studies containing patients in the respective diagnostic group. Values of each cell represent z-scores of median centiles (row-wise across diagnostic groups) for visualisation. Clustering was determined using the gap statistic ($k=1$). Lowercase 'v' stands for 'variability' and was calculated as the standard deviation (rather than median), and was Z-scored as per the median centiles for each phenotype across diagnostic groups. Abbreviations: Alzheimer's disease, AD; attention deficit hyperactivity disorder, ADHD; autism spectrum disorder, ASD; anxiety or phobic disorders (ANX); mild cognitive impairment, MCI; major depressive disorder, MDD; schizophrenia, SCZ; grey matter volume, GMV; subcortical grey matter volume, sGMV; white matter volume, WMV; ventricular cerebrospinal fluid volume, CSF.

Fig. S11.2.2. Hierarchical clustering of clinical disorder profiles of median and standard deviation of centile scores on cerebrum tissue volumes. Centile profiles using a less-stringent $N < 500$ cutoff for the number of patients with similar diagnoses, as per **Fig.S10.5.2** above. Legend (x-axis and right annotation): 'years' corresponds to 'age' and represents median age of the diagnostic groups, '% female' corresponds to 'sex' and represents the percentage of female patients in each diagnostic group, 'total' corresponds to 'study' and represents the number of studies containing patients in the respective diagnostic group. Values of each cell represent z-scores of median centiles (row-wise across diagnostic groups) for visualisation. Clustering was determined using the gap statistic ($k=3$). Lowercase 'v' stands for 'variability' and was calculated as the standard deviation (rather than median), and was z-scored as per the median centiles for each feature across diagnostic groups. Abbreviations: Alzheimer's disease, AD; attention deficit hyperactivity disorder, ADHD; anorexia nervosa or bulimia nervosa, AN/BN; anxiety or phobia, ANX; autism spectrum disorder, ASD; bipolar disorder, BD; fronto-temporal dementia, FTD; Lewy body dementia, LBD; mild cognitive impairment, MCI; major depressive disorder, MDD; obsessive-compulsive disorder, OCD; Parkinson's disease, PD; schizophrenia, SCZ.

Ref 3/48:

Fig S9.1: Second column is labelled "Age range (log-transformed)". Age-range of what? What are the units (day, weeks, years)?

Thank you for picking this up. This x-axis label refers to the duration of follow-up, between baseline and final follow-up scans, for participants with longitudinal data; the units are log-transformed years. The x-axis labels and figure legends have been revised accordingly for **Figs S14.1.1** and **S14.1.2**.

<<The following changes have been made to the Supplementary Information>>

Grey matter volume

White matter volume

Subcortical grey matter volume

Ventricular volume

Fig. S14.1.1. Overview of possible associations between within-subject variation (interquartile range, IQR) of longitudinal centile scores on cerebrum tissue volumes and factors that could influence longitudinal stability of centile scores. First column shows the IQR in relation to the individual's age (in years) at the time of their baseline scan. Second column shows the length of follow-up (in years, log-transformed) between the baseline scan and the final follow-up scan. Third column shows the IQR in relation to the number of longitudinally repeated scans available per participant.

Total cerebrum volume**Total surface area****Mean cortical thickness**
Fig. S14.1.2. Overview of possible associations between within-subject variation (interquartile range, IQR) of the longitudinal centile scores on extended global MRI phenotypes and likely factors that could influence longitudinal stability of centile scores First column shows the IQR in relation to the individual's age (in years) at the time of their baseline scan. Second column shows the length of follow-up (in years, log-transformed) between the baseline scan and the final follow-up scan. Third column shows the IQR in relation to the number of longitudinally repeated scans available per participant.

Ref 3/49:

SI Section 9.3 (p. 49): Text refers to “Fig. S9.1.3.3”. Should be “Fig. S9.3.3”.

Thanks for picking this up. This has been corrected and, more broadly, all cross-referencing between main text and supplementary information has been updated and harmonized following the inclusion of new analyses, text and figures.

Ref 3/50:

Given that Infant FreeSurfer is the sole means by which data was obtained for individual less than 2 years old, some additional discussion of the validity of Infant FreeSurfer, and the confidence in the those values, seems warranted. The authors already comment that the values it generated for subcortical GMV didn't seem continuous with those generated by FreeSurfer for 2 years and older. Was there any evidence (milder) of similar concerns for the other 3 phenotypes?

Although Infant FreeSurfer was used to process a subset (~37%) of the data on subjects less than 2 years old, the majority of perinatal MRI datasets were processed with different pipelines, including adapted versions of regular FreeSurfer, custom internal processing pipelines, or expert manual segmentation in the case of some fetal datasets. Therefore the reference curves in this age range do not depend solely on the validity of Infant FreeSurfer. Given the diversity of processing pipelines used for this relatively small subset of primary studies we have not commented further on Infant FreeSurfer's estimation of sGMV specifically, and in revised **SI18**

we clarify that there were no similar concerns for estimates of other brain tissue volumes (GMV, WMV, CSF). We have also included a new analysis demonstrating good consistency of brain tissue volume estimation across multiple versions of FreeSurfer for out-of-sample centile score (**SI4.4 “Reliability of out-of-sample centile scoring across multiple versions of FreeSurfer”**); and we have added a new supplementary table (**ST 1.1**) that now provides more detailed information on data processing and acquisition for all primary studies included in the reference dataset.

Ref 3/51:

It would be helpful if the SI Methods detailed exactly which FS measures were used for the definition of each of the 4 studied phenotypes.

The new and expanded data processing section (**SI18. “Data processing”**) now provides further details on what files were used at what stage in the FreeSurfer pipelines and what values were extracted from those files.

<<The following changes have been made to the Supplementary Information>>

18. Data processing

If T1- and T2/FLAIR-weighted raw data were available, as they were for approximately 95% of scans), these data were processed with FreeSurfer 6.0.1²⁴ using the combined T1-T2 recon-all pipeline for improved grey-white matter boundary estimation. If only raw T1-weighted data were available, and subjects were aged over 2 years, data were processed with FreeSurfer 6.0.1 using the standard recon-all pipeline. If subjects were aged 0–2 years, data were processed with Infant FreeSurfer v1⁹⁴. **ST1.1** lists the number of subjects per site per processing pipeline alongside their respective MRI acquisition and quality control protocols. We noticed that Infant FreeSurfer estimated total subcortical grey matter volume (sGMV) differently from other pipelines included in this dataset, while other cerebrum tissue volumes were estimated consistently across pipelines. We therefore excluded scans processed with Infant FreeSurfer from growth curve estimation for subcortical GMV. All four cerebrum tissue volumes were extracted from the aseg.stats files that are generated in the first stage of the recon-all process: 'Total cortical gray matter volume' for GMV; 'Total cortical/cerebral (FreeSurfer version dependent) white matter volume' for WMV; 'Subcortical gray matter volume' for sGMV (inclusive of thalamus, caudate nucleus, putamen, pallidum, hippocampus, amygdala, and nucleus accumbens area; <https://freesurfer.net/fswiki/SubcorticalSegmentation>); and the difference between 'BrainSegVol' and 'BrainSegVolNotVent' for Ventricular volume. The first processing stage of recon-all includes: non-uniformity correction, projection to Talairach space, intensity normalisation, skull-stripping, automatic tissue and subcortical segmentation. Surface interpolation, tessellation and registration are done at the second and third stages of the recon-all pipeline (i.e., after aseg.stats files are created) and all these later stage processes involve projection to a standard stereotactic (fsaverage) space. Regional volume, thickness, and surface area was estimated for each of 34 bilaterally averaged cortical regions defined by the Desikan-Killiany⁴⁸ parcellation template following the final stages of the recon-all pipeline and using the hemisphere-specific apars.stats files generated by FreeSurfer.

Ref 3/52:

It's "FreeSurfer". Not "Freesurfer" or "freesurfer". The inconsistent formatting is careless and could be seen as disrespectful to its creators/developers/maintainers.

We apologize for this inadvertent inconsistency, which has been corrected throughout the manuscript.

Ref 3/53:

Overall, the dataset descriptions in SI are quite inconsistent in what is covered. e.g., Not all of them even mention the number of individuals. Any information that can be conveniently provided in tabular format should be removed and placed in an SI spreadsheet (e.g., # individuals, scanner platforms, T2 availability, imaging parameters, FS processing version). This spreadsheet should also make clear whether the FS processing was done de novo for the current study, or whether the current study used FS data already generated/provided by the study itself. The text descriptions of the studies can then be limited to a brief overview of the study, as well as any particularly salient points that cannot conveniently be captured in the proposed spreadsheet.

This is an excellent suggestion. We now provide details on FreeSurfer processing, MRI acquisition, sample size and many other comparable details for each primary study in a new supplementary table **ST 1.1**. This table also denoted whether data was processed in-house or included in pre-processed format. We have also gone through the supplementary descriptions of the primary datasets to improve consistency and minimise redundancy of each study summary

Ref 3/54:

The "OpenPain" study has a very long description in the SI relative to the other studies. Is there a particular reason that it merits such considerable detail relative to the other studies? (It reads like an unedited cut-and-paste from other documentation).

Dataset descriptions were provided with the MRI data by contributing authors, consortia and sites and were only minimally edited. We have asked the original authors to provide more succinct versions of their description and further shorten long descriptions (i.e., OpenPain). In addition, details that are now provided in table ST1.1 have been removed from the dataset descriptions.

Ref 3/55:

Issues related to the SI Tables:

We thank the reviewer for their extremely conscientious proof-reading of all tables and we apologise for any inconsistencies found. All the suggestions below have been adopted and all (new and old) tables have been double-checked for consistency.

a. The meaning of some of the variables in the SI tables is not clear. (e.g., 'total.cn', 'percentage.cn' in SI Table 1; 'V1', 'V2' in SI Table 5). A key/dictionary of some sort is necessary, for at least some of the variables.

We now provide more descriptive column labels.

b. Table1.5: First row has no label for 'dx'.

This error has been fixed in the revision, and all tables have been checked for consistency.

c. Table1.6: Identical to Table1.5

This error has been fixed in the revision and all tables have been checked for consistency.

d. Inconsistent naming schema of the individual tabs within a table (e.g., “Table1.1” vs. “2_1”). Also, given the Excel character limit on tab name length, make sure that the name of each tab is clearly interpretable.

In response to this comment, all tabs and naming conventions have been made consistent throughout the revised document.

e. Please remove the annoying Excel “warning” (and associated green triangle) about “Number stored as text” by converting the cells from text to number.

Due to the fact that MS Excel uses a different decimal character notation in different versions/regional locations (i.e., comma or period), there are compatibility issues when we stored data as numbers in xlsx format. Most software (R, Python etc.) write out dataframes or tables to xlsx formatting numbers explicitly as text to avoid such formatting issues when opened in different locations. To avoid any compatibility issues and to ensure the tab names are appropriately names (i.e., without character limits), we now instead provide each table as a collection of individual csv files, numbered accordingly, but replacing “.” with “_” in the filename to avoid any issues with different operating systems or analyses software being unable to interpret file formats.

f. SI Section 10 (“Sex differences”, p. 52) mentions a “SI table 2.9”, which doesn’t appear to exist.

Thank you for noticing this inadvertent omission. This table is now included as Supplemental Table 8 in the revised SI.

g. Are SI Tables 4 and 7 mentioned anywhere in the main text or SI text? If so, I couldn’t find the references to them.

Thank you for noticing the absence of a reference to these tables. These tables are now referenced in their respective supplemental section and expanded to include effect size estimates in line with the other tables listing case-control differences.

h. All the tabs in SI Table 6 are identical.

These were all generated automatically depending on the phenotype in question and would have only been different if subjects would have had missing data in a specific phenotype. Since in this case there was no missing data, the additional tabs were redundant and have been removed.

References cited in response to reviewers

*note that these will be numbered differently in the updated manuscript as they will be merged there with existing references.

1. Rosen, A. F. G. *et al.* Quantitative assessment of structural image quality. *Neuroimage* **169**, 407–418 (2018).
2. Ai, L. *et al.* Is it time to switch your T1W sequence? Assessing the impact of prospective motion correction on the reliability and quality of structural imaging. *NeuroImage* vol. 226 117585 (2021).
3. Fischl, B. FreeSurfer. *Neuroimage* **62**, 774–781 (2012).
4. Keshavan, A., Yeatman, J. D. & Rokem, A. Combining Citizen Science and Deep Learning to Amplify Expertise in Neuroimaging. *Front. Neuroinform.* **13**, 29 (2019).
5. Tisdall, M. D. *et al.* Volumetric navigators for prospective motion correction and selective reacquisition in neuroanatomical MRI. *Magn. Reson. Med.* **68**, 389–399 (2012).
6. Johnson, W. E., Li, C. & Rabinovic, A. Adjusting batch effects in microarray expression data using empirical Bayes methods. *Biostatistics* **8**, 118–127 (2007).
7. Fortin, J.-P. *et al.* Harmonization of cortical thickness measurements across scanners and sites. *Neuroimage* **167**, 104–120 (2018).
8. Pomponio, R. *et al.* Harmonization of large MRI datasets for the analysis of brain imaging patterns throughout the lifespan. *Neuroimage* **208**, 116450 (2020).
9. Beer, J. C. *et al.* Longitudinal ComBat: A method for harmonizing longitudinal multi-scanner imaging data. *Neuroimage* **220**, 117129 (2020).
10. Child growth standards. <https://www.who.int/tools/child-growth-standards>.
11. Casey, B. J. *et al.* The Adolescent Brain Cognitive Development (ABCD) study: Imaging acquisition across 21 sites. *Dev. Cogn. Neurosci.* **32**, 43–54 (2018).
12. Pietschnig, J., Penke, L., Wicherts, J. M., Zeiler, M. & Voracek, M. Meta-analysis of associations between human brain volume and intelligence differences: How strong are they and what do they mean? *Neurosci. Biobehav. Rev.* **57**, 411–432 (2015).
13. Wheater, E. *et al.* Birth weight is associated with brain tissue volumes seven decades later

- but not with MRI markers of brain ageing. *Neuroimage Clin* **31**, 102776 (2021).
14. Walhovd, K. B. *et al.* Long-term influence of normal variation in neonatal characteristics on human brain development. *Proc. Natl. Acad. Sci. U. S. A.* **109**, 20089–20094 (2012).
 15. Weintraub, S. *et al.* Cognition assessment using the NIH Toolbox. *Neurology* **80**, S54–64 (2013).
 16. Zöllei, L., Iglesias, J. E., Ou, Y., Grant, P. E. & Fischl, B. Infant FreeSurfer: An automated segmentation and surface extraction pipeline for T1-weighted neuroimaging data of infants 0-2 years. *Neuroimage* **218**, 116946 (2020).
 17. Desikan, R. S. *et al.* An automated labeling system for subdividing the human cerebral cortex on MRI scans into gyral based regions of interest. *Neuroimage* **31**, 968–980 (2006).
 18. Giedd, J. N. *et al.* Child psychiatry branch of the National Institute of Mental Health longitudinal structural magnetic resonance imaging study of human brain development. *Neuropsychopharmacology* **40**, 43–49 (2015).
 19. Dong, H.-M. *et al.* Charting brain growth in tandem with brain templates at school age. *Sci Bull.* **65**, 1924–1934 (2020).
 20. Sharma, E. *et al.* Consortium on Vulnerability to Externalizing Disorders and Addictions (cVEDA): A developmental cohort study protocol. *BMC Psychiatry* **20**, 2 (2020).
 21. Alcohol Research: Current Reviews Editorial Staff. NIH's Adolescent Brain Cognitive Development (ABCD) Study. *Alcohol Res.* **39**, 97 (2018).
 22. Blanchard, R. D., Bunker, J. B. & Wachs, M. Distinguishing aging, period and cohort effects in longitudinal studies of elderly populations. *Socioecon. Plann. Sci.* **11**, 137–146 (1977).
 23. Vasung, L. *et al.* Association between Quantitative MR Markers of Cortical Evolving Organization and Gene Expression during Human Prenatal Brain Development. *Cereb. Cortex* **31**, 3610–3621 (2021).
 24. Fjell, A. M. *et al.* Development and aging of cortical thickness correspond to genetic organization patterns. *Proc. Natl. Acad. Sci. U. S. A.* **112**, 15462–15467 (2015).
 25. Sydnor, V. J. *et al.* Neurodevelopment of the association cortices: Patterns, mechanisms, and implications for psychopathology. *Neuron* **109**, 2820–2846 (2021).

26. Holland, D. *et al.* Structural growth trajectories and rates of change in the first 3 months of infant brain development. *JAMA Neurol.* **71**, 1266–1274 (2014).
27. Courchesne, E. *et al.* Normal brain development and aging: quantitative analysis at in vivo MR imaging in healthy volunteers. *Radiology* **216**, 672–682 (2000).
28. Blüml, S. *et al.* Metabolic maturation of the human brain from birth through adolescence: insights from in vivo magnetic resonance spectroscopy. *Cereb. Cortex* **23**, 2944–2955 (2013).
29. Kuzawa, C. W. *et al.* Metabolic costs and evolutionary implications of human brain development. *Proc. Natl. Acad. Sci. U. S. A.* **111**, 13010–13015 (2014).
30. WHO MULTICENTRE GROWTH REFERENCE STUDY GROUP & Onis, M. WHO Motor Development Study: Windows of achievement for six gross motor development milestones. *Acta Paediatr.* **95**, 86–95 (2007).
31. Solmi, M. *et al.* Age at onset of mental disorders worldwide: large-scale meta-analysis of 192 epidemiological studies. *Mol. Psychiatry* 1–15 (2021).
32. Kang, H. J. *et al.* Spatio-temporal transcriptome of the human brain. *Nature* **478**, 483–489 (2011).
33. McCarthy, C. S. *et al.* A comparison of FreeSurfer-generated data with and without manual intervention. *Front. Neurosci.* **9**, 379 (2015).
34. Tamnes, C. K. *et al.* Brain development and aging: overlapping and unique patterns of change. *Neuroimage* **68**, 63–74 (2013).
35. Marquand, A. F., Rezek, I., Buitelaar, J. K. & Beckmann, C. F. Understanding Heterogeneity in Clinical Cohorts Using Normative Models: Beyond Case-Control Studies. *Biol. Psychiatry* **80**, 552–561 (2016).
36. Elliott, L. T. *et al.* Genome-wide association studies of brain imaging phenotypes in UK Biobank. *Nat. Neurosci.* **562**, 210–216 (2018).
37. Cox, S. R. *et al.* Ageing and brain white matter structure in 3,513 UK Biobank participants. *Nat. Commun.* **7**, 13629 (2016).
38. Landman, B. A. *et al.* Multi-parametric neuroimaging reproducibility: a 3-T resource study.

- Neuroimage* **54**, 2854–2866 (2011).
39. Kremen, W. S., Franz, C. E. & Lyons, M. J. VETSA: the Vietnam Era Twin Study of Aging. *Twin Res. Hum. Genet.* **16**, 399–402 (2013).
 40. Koo, T. K. & Li, M. Y. A Guideline of Selecting and Reporting Intraclass Correlation Coefficients for Reliability Research. *J. Chiropr. Med.* **15**, 155–163 (2016).
 41. Narayanan, S. *et al.* Brain volume loss in individuals over time: Source of variance and limits of detectability. *Neuroimage* **214**, 116737 (2020).
 42. Staffaroni, A. M. *et al.* Neuroimaging in Dementia. *Semin. Neurol.* **37**, 510–537 (2017).
 43. Hedges, L. V., Hedges, L. V. & Olkin, I. *Statistical Methods for Meta-Analysis*. (Elsevier Science, 1985).
 44. Stasinopoulos, M. D., Rigby, R. A., Heller, G. Z., Voudouris, V. & De Bastiani, F. *Flexible Regression and Smoothing: Using GAMLSS in R*. (CRC Press, 2017).
 45. Rigby, R. A., Stasinopoulos, M. D., Heller, G. Z. & De Bastiani, F. *Distributions for Modeling Location, Scale, and Shape: Using GAMLSS in R*. (CRC Press, 2019).
 46. Allen, M., Poggiali, D., Whitaker, K., Marshall, T. R. & Kievit, R. A. Raincloud plots: a multi-platform tool for robust data visualization. *Wellcome Open Res* **4**, 63 (2019).
 47. Computing, R. & Others. R: A language and environment for statistical computing. *Vienna: R Core Team* (2013).
 48. Sheather, S. J. & Jones, M. C. A reliable data-based bandwidth selection method for kernel density estimation. *J. R. Stat. Soc. Series B Stat. Methodol.* **53**, 683–690 (1991).
 49. Scott, D. W. *Multivariate Density Estimation: Theory, Practice, and Visualization*. (John Wiley & Sons, 2015).
 50. Silverman, B. W. *Density estimation for statistics and data analysis*. (Routledge, 2018).
 51. Maechler, M. & Ringach, D. Diptest: Hartigan’s dip test Statistic for unimodality-corrected. *R package version 0.75-7*. See <https://CRAN.R-project.org/package=diptest> (2015).
 52. Benjamini, Y. & Hochberg, Y. Controlling the false discovery rate: A practical and powerful approach to multiple testing. *J. R. Stat. Soc.* **57**, 289–300 (1995).

Author Rebuttals to Initial Comments:

Referees' comments:

Referee #1 (Remarks to the Author):

Overall, the authors have done an incredible amount of analysis in response to the first round of review and improved the quality of the work substantially – for which they are to be commended. With that said, this will be a high impact paper and the authors need to be careful about precedents set. In that regard, there are still a number of considerations that should be addressed before potential publication.

General

The authors could improve the scholarship of the intro. A vast literature regarding brain age prediction exists (see <https://www.frontiersin.org/articles/10.3389/fneur.2019.00789/full> for example review), as well as clear demonstrations of the value of relating relative brain age to phenotype (<https://elifesciences.org/articles/54055>). While most do not use gamlss, their goals and base strategy are comparable, and the work should be acknowledged. Prior work also exists that is directly focused on applications of GAMLSS in fMRI (e.g., <https://www.ncbi.nlm.nih.gov/pmc/articles/PMC4387093/>). The key (and meritorious) innovation of the submitted work is that it brings together the datasets needed to sample the near entirety of the lifespan, along with clinical variations.

The discussion acknowledges the work is a proof of principle – why is that not in the intro? And again, looking at the backdrop of the literature, what principle(s) needed to be proven – please be more specific.

Despite the prominence of concerns about data quality and confounds from the reviewers, the authors barely acknowledge them in the main manuscript. This is a concern – especially given some of the concerns raised under the specific comments section below. Data quality and the various confounds would not be expected to preclude meaningful delineation of the mean lifespan trajectory – it is the variance that will be impacted. The greater the representation of noise in the variance, the more limited the data will be in detecting meaningful pathology-related deviations beyond what can be seen by eye (e.g., Alzheimer's patients). It is doubtful the clinically relevant tools pointed at by this work will be meaningfully realized without accounting for data quality – yet, the authors minimize it. I find this highly problematic.

The authors seem to view batch effect correction as an answer to address the many sources of variance in their analyses (e.g., diagnostic protocol, imaging protocol, sampling strategy). When sample composition or procedures are too dramatically different, batch effect correction will not solve the issue. Vogelstein recently tried to draw attention to challenges in batch effect correction in his Causal Combat framework (see [biorxiv](https://www.biorxiv.org/content/10.1101/2019.08.28.279111v1)). At a minimum, greater acknowledgment of differences in diagnostic protocols and sample composition should be highlighted in the main text, as it helps motivate future work toward standardization.

While I respect the value of team science, the authors are missing my point in their response. Generators for open datasets are not recognized on the author line, yet those for more difficult to access datasets are. This is not a push to add the open datasets to the author line; but rather to ask if those who generated the more restricted datasets would consider limiting their acknowledgement

to the text body (at least for data with one or more prior publications).

Specific: Major

Optimal Euler # can vary as a function of site; no general value is established to my knowledge. This is visible in Figure S2.1.2 (note, there are two figures with the same number), where there is large variability in the mean EI across dataset (For example, the Oulu and AOBA have their median EI > 217). Yet, it seems the authors are using a single cutoff based on the initial paper from the PNC to try to fit all; this will likely lead to both inappropriate inclusion and exclusion of data.

Have the authors looked at the relationship of EI and volume within a dataset?

95% data inclusion (ie quality pass) is surprising; some sites may have only shared quality pass data – is this known?

Figure S2.1.2 – the authors are relying on Spearman to state that no association exists between centile and MRI quality. That assumes a linear relationship; this could miss a more complex association, which looking at the data – there is visibly higher EI in the low centiles.

Figure S2.2.1 – a score of 4 seems biased for GMV – why only exclude 5 and 6? Repeating with 4 excluded would make sense.

Comparison of GAMLSS vs. ComBat only carried out in ABCD, which uses V-NAV (thus has minimal motion) and is harmonized; what about other sites.

Specific: Minor

How was the visual inspection performed? Where each image rated by one expert or many? If more than one rater, how was the ICC?

Page 6 – First paragraph. – “The only exception to this generally high ... were the GISTO and ESDS cohorts where excluding scans with EI > 217 substantially reduced the number of scans (by >30%)...” But, by looking at the second Figure S2.1.2, it looks that more than 50% of the data for males in the Oulu, NIH, AOBA, WAYNE, ICBM have a EI > 217. So which one is correct?

Referee #2 (Remarks to the Author):

Overall, the authors have done a commendable job using a very large convenience meta-analytic sample of individuals with MRI data to produce lifespan trajectories of gross brain structure, and variation therein. The key findings are not so much of note in their own right, but rather constitute a series of checks to ensure that the harmonization process has been successful, and that associations produced by the harmonized data are cleaner. The major contribution here is a resource for other researchers seeking to compare their sample to the wider corpus of MRI research, and those seeking to combine MRI data across multiple datasets in a principled way that avoids bias that is typically associated with aggregation across different MRI studies. However, contrary to what has been implied by the direct comparison of the brain charts introduced here to height and weight growth charts that are used routinely in clinical settings, the brain charts cannot be used for individual patient diagnosis or treatment. I elaborate on this concern below. In light of them, I believe that a major revision to the framing of the manuscript is needed.

My concern can be distilled down to what is implied by the title “Brain Charts” and the explicit desire expressed (e.g. in the second sentence of the abstract, the first paragraph of the main text) to

constructed charts for the brain that akin to growth charts that are routinely used in clinical, mostly pediatric, settings for tracking the trajectory of height and weight growth of individuals relative to the population. I need to be clear that when I say “clinical,” I mean in individual patient monitoring, diagnosis and care, not “clinical research,” which refers to research that is on the topic of clinical populations or clinical treatments. This article needs to, in no uncertain terms, remove this inappropriate comparison between brain charts (which cannot be used clinically for individual patients) and height and weight growth charts (which can be used clinically). Statements such as the following need to be removed: “Crucially, for clinical purposes, centile scores provided a standardised and interpretable measure of deviation that revealed new patterns of neuroanatomical differences across neurological and psychiatric disorders.” [These are not “clinical purposes.” I think that the authors mean “for research into the neuroanatomical correlates of clinical disorders.”]

The first reason that the brain charts cannot be used for individual patient diagnostics is that although the total sample size of the analysis is very large, the studies are not representative of the population (and it is not even clear what the “the population” of interest is; growth charts should be normed relative to the population of their intended use). Inasmuch, it would be grossly inappropriate to assign and individual a centile score relative to the brain chart norms, as the centile is unlikely to correspond to centiles within the population. Centiles can of course be used for the purposes of research, so as to reference an individual’s location within the observed (meta-analytic) sample distribution, but they simply cannot be used clinically without the reference norms being representative. The authors have been responsive to my previous comment about issues concerning representativeness, but I continue to worry that the charts are framed as being of use for individual centile scoring in clinical settings.

Second, is that differences in MRI protocol and scanner calibration in real-world clinical settings relative to the studies used to construct the brain charts cannot be corrected for. In their response, the authors go to great lengths to explain how they harmonized the samples and accounted for study-to-study differences, for instance, by obtaining bootstrapped centile scores from leave one sample out analyses. They are correct that these methods strongly mitigate against artifacts in the brain charts that might have otherwise arisen from between-study variation. Indeed, as displayed in Fig S5.2.4, the primary advantage of the centile scoring method is that it allows data from different studies to be combined without associations being attenuated by study-to-study variation. However, this does not mitigate against artifacts that would arise from attempting to centile score an individual from a single outside scan. The authors do appropriately comment that new datasets (with “comparatively smaller samples sizes”) can be integrated with this one, and do not actually indicate that a single scan of an individual in a clinical setting can be centile scored relative to their charts. However, by calling these “brain charts” and comparing them to existing growth charts, the idea that the charts can be used for individuals clinically, rather than simply for individuals within studies, is made implicit.

Referee #3 (Remarks to the Author):

Bethlehem and co-authors have submitted a very impressive revision to their manuscript “Brain charts for the human lifespan”. Extensive additional analyses were performed to address previous

weaknesses and criticisms. I commend the authors for their responsiveness to the previous round of reviews. In particular, I am now convinced that their GAMLSS modeling approach is sufficiently robust to the inclusion of new studies. The Supplement is massive, and thus beyond the capability of any single individual to review thoroughly (in a reasonable amount of time). But it is clear from the ~ 1/3 of the supplement that I did review that the value and overall quality of the Supplemental Material is dramatically improved.

I believe that this manuscript has now (nearly) achieved its potential. It will be a very impactful addition to the understanding and future study of human structural neuroanatomy.

I do have a couple follow-up questions related to two of my initial major concerns. And given the extensive nature of the revisions, there are a number of relatively minor issues that warrant consideration and revision before publication.

Follow-up questions:

While effect sizes are now available in the supplementary material, the reporting of effect sizes remains underemphasized in the main text. In particular, the main text revolving around Figure 4 still comes across as highly p-value centric. I believe that some indication of the Cohen's D effect sizes should be worked into Figure 4, both for the individual MRI phenotypes (4A), as well as the CMD comparison (4C). If that's simply not possible, some (quantitative) discussion of effect sizes for both of those should be included in the main text. Additionally, it should be made clear that effect sizes are available in the ST. (Currently, the main text mentions "effect-size" once in the introduction, but never seems to actually present or discuss effect sizes in the main text).

The authors state in their response that due to the use of a "GAMLSS model that included random effects on three moments of the generalized gamma distribution" that it therefore wasn't possible to "conduct an analysis of the effect of specific technical covariates". I don't follow the logic of this statement. In particular, aren't estimates of the random effects available for each study? (Elsewhere in the manuscript makes the explicit point that it was important that these be available). And if that's the case, then isn't it possible to investigate possible relationships between those random effects estimates and the technical covariates?

Other issues that warrant consideration and clarification:

Fig. 2: There clearly is a discontinuity in the mean cortical thickness values just past age 2. Is this commented on anywhere? Somewhat relatedly, what is the combined N across studies of individuals less than 2 years old? In the interest of transparency, it seems worth pointing this out in the main text, as presumably this N is rather low, yet this data is the basis for a number of the conclusions regarding age at peak (thickness) and age at peak velocity (a number of measures). Also, neither key nor caption indicates what the gray line represents in panel B. Also, the caption says "2.5% and 97% centiles". I presume that the latter value should be "97.5%"?

Figure 3: The colors are not sufficiently distinct to quickly and easily match to the key. Either more unique colors are necessary, or add thin black lines within the color bands, but with different

linestyles (e.g., solid vs. dashed) to help distinguish lines that are the same basic color. Also, the top inset reads “thicknes” rather than “thickness”.

p. 8 (main text): Shouldn't text regarding “mean difference in h^2 ” be referring to Fig.4D (not 4C)? Also, does that “11.8%” refer to an actual *percentage* difference, or a *percentage point* difference? (Figure 4 seems to indicate it is probably the latter, which is more impressive than the former, and thus worth describing precisely).

Figure 4: Caption contains no indication of what 1 vs. 2 vs. 3 asterisks represents.

p. 18 (“Online Methods”): It seems less relevant to me to present the “general” form of the GAMLSS than to make clear how the GAMLSS was modeled in this particular study. In that regard, I find it potentially confusing to readers to include smoothing functions in Eq. (1) when non-parametric smoothing functions were not in fact used. Also, in Eq. (1), what is ‘F’? (Not defined).

S1.7: I still don't understand the justification for computing IQR when you only have two data points. In what sense is IQR “valid” when you have just 2 data points (any more so than a trimmed mean from just 2 data points)? Just because you can select an implementation of IQR that is *defined* for 2 data points doesn't make it a measure with a meaningful interpretation.

S2.1, Euler Index Filtering: Rosen et al. used the strict definition of the Euler Number, which is increasingly negative as the number of surface holes in the non-topology corrected (orig.nofix) surface increases. If you are using $EI > 217$ as a threshold, you are taking the negative of the true Euler Number, which should be stated. Also, please be explicit as to whether you are averaging or summing the EI from the two hemispheres. [Rosen states that they averaged]. Also, it would be beneficial to quantify how many scans were excluded by your filtering operation. Last, it should be noted that $EI > 217$ is a threshold that has only been shown to be an “optimal” threshold in one study, and it remains unknown whether that threshold is optimal in any sense as a QC criterion for data collected on other scanners or using other protocols.

S2.2, Expert visual quality control: Was this a de novo visual quality control performed by the authors, separate from any visual QC and rating that the individual studies might have already performed and provided? Also, it should be made clearer whether the 9704 scans that were rated were drawn from all studies equally, or whether particular studies dominated this endeavor.

S3.2.1: For consistency with changes elsewhere, it should say that quantitative comparison of model *BIC* (rather than AIC) wasn't possible.

Figs. S3.2.1 and S3.2.2: Are these 95% confidence intervals? Relatedly, what exactly do the “uncertainty intervals” in Fig. S4.5 represent? Given the wide variety of different types of “error bars” used in various figures (e.g., SD, standard errors, 95% confidence intervals, box plot whiskers), please make sure that it is clear in each figure caption exactly what the error bars in that particular figure represent.

Fig. S3.2.3: It would be helpful if the caption stated the reason that no estimates for Sigma for

Ventricular volume are available.

S3.2.2: Insufficient detail is provided to understand how the stratified bootstrap sample was implemented. It would be helpful to have a methodical example of how exactly one bootstrap replicate was constructed.

S4.1: Bias cannot be assessed via correlation. Some true measure of bias should be reported (in addition to the correlations).

S4.3: It seems odd to not include the ICC values of the “raw” (uncentiled) volumetric data for the analyses of the reproducibility resource and HBN data, for explicit comparison to the centile ICCs. (They are provided for the VETSA analysis).

Figs. S4.3.[1,2]: What do the p-values in the titles of the plots represent? Are they the p-value of the regression/scatterplot? Or the p-value of the test of the null hypothesis that $ICC = 0$?

Fig. S4.3.2: What does “HCP Session” in the top row of plots refer to?

S4.4: It would be helpful if the “name” of the study used for this analysis was included in the text (for convenient reference against other data that shows results per study).

S4.5: It isn't clear how dataset ‘clones’ were created (e.g., was resampling with replacement used *within study*), nor how many datasets were ‘cloned’. Also, what does NSPN represent in the titles of Fig. S4.5?

Fig. S5.2.5: Seems like the point being made by this figure is incomplete without a panel showing the equivalent analysis applied to the raw data.

Fig S6.2: What were scanners “1,2,3,4” (e.g., vendor and model; e.g., did the vendor and/or field strength change?)

S7 (and S2.1 and S18): It is overly simplistic to say that the cerebrum tissue volumes returned by FS are not contingent on the surface reconstruction – this is certainly not the case for modern versions of FS. Almost every value returned by FS 6.0, including in the aseg.stats, respects the surfaces, and is computed after the surfaces are defined. For example, see <https://surfer.nmr.mgh.harvard.edu/fswiki/MorphometryStats> and <https://surfer.nmr.mgh.harvard.edu/fswiki/ReconAllTableStableV6.0> (noting that the final aseg.stats is derived using the surfaces as inputs to the -segstats stage). It's possible that the situation is more complicated for FS 5.1 and 5.3. An easy way to check for these older versions is to see if the volumes being extracted are the same in the aseg.stats as in the wmparc.stats file. If they are, then they are surface-informed estimates. If not, additional investigation will be necessary to understand (and accurately portray) the differences in this regard between different FS versions.

Fig S7.1: To match the actual figures, the caption should say “baseFO[a][b][c]R[x][y][z]”. (i.e., add “R” to the text string).

Fig. S7.4.1: Captions says that studies are sorted by “median standard deviation”, but that doesn’t appear to be the case.

Fig. S10.1.1 and S10.1.3: Rather than attempting to show all significant pairwise differences via lines above a box plot, it would be easier for readers to simply include a table with p-values. Cells below the diagonal could present the Cohen’s d and p-value for males, and cells above the diagonal could present the values for females. (This way readers would also have access to the Cohen’s d effect sizes of all pairwise comparisons in a convenient “lookup” table in the SI, without needing to download and find the relevant supplementary data table). Same comments apply to Figs. S14.3.[3,4]

Figs. S10.2.[1,2] and S14.3.[1,2,3]: Please match the colors and order of clinical cohorts to those used in Fig. 4 and S10.1.2

S12: No indication of the effect size of the regression lines (e.g., R^2) is provided.

S14.4.1: Unclear whether the LME only included a random subject-level intercept, or if it also included a random subject-level slope. If the latter, what was the correlation between the random intercept and random slope estimates? Also, what do the error bars in panel B represent?

Fig. S15.1: Either some of the measures have very extreme outliers (e.g., GMV/SA, WMV/SA, sGMV/SA), or the allowed jitter along the x-axis is too great, such that values from one Feature Pair are bleeding into the space for an adjacent Feature Pair.

ST 1.1: The information in the “Acquisition.Parameters” column in this table is inconsistent and uneven. For example (1) the very first entry (3R-BRAIN) is listed as having a flip angle of 52, which would be very unusual for an MPRAGE; (2) the manner in which TR/TE/TI/FA is presented across studies is inconsistently formatted; (3) some rows have random special characters (e.g., Calgary, Conte, DLBS, EMBARC, GOSH, LATAM, LIFE; to name just a few based on quick inspection); (4) some don’t report TR/TI/FA at all (e.g., HCP_lifespan). Overall, these imaging parameters would be easier to parse (and easier to check for odd entries) if there was one column for each specific parameter to be reported, rather than just a generic text column of “parameters”.

ST 1.8: It seems like the Figure 1 caption should either refer to ST1.2-1.8 (i.e., all of the “global” measures), or just ST1.2-1.5 (corresponding to the measures shown in Figure 1. Either way, ending at ST1.7 doesn’t make sense (in either the caption, or the text in the “Mapping normative brain growth” paragraph).

ST3: The labels for column E (“Median centile G1”) and column F (“Median centile G2”) are presumably incorrectly swapped.

Author Rebuttals to First Revision:

We would like to thank all the reviewers for their expert reviews of our revised manuscript and supplementary analyses. Their comments have provided extremely helpful guidance in strengthening the paper. Please find a point-by-point response below.

[BLACK] - ORIGINAL COMMENT

[BLUE] - RESPONSE TO COMMENT

[GREEN] - NEW/ALTERED TEXT

Referee #1:

Overall, the authors have done an incredible amount of analysis in response to the first round of review and improved the quality of the work substantially – for which they are to be commended. With that said, this will be a high impact paper and the authors need to be careful about precedents set. In that regard, there are still a number of considerations that should be addressed before potential publication.

We thank the reviewer for their positive appraisal of our original and revised manuscripts, and we are grateful for their thoughtful consideration of issues needing further attention prior to publication of this paper.

General

Ref 1/1: The authors could improve the scholarship of the intro. A vast literature regarding brain age prediction exists (see <https://www.frontiersin.org/articles/10.3389/fneur.2019.00789/full> for example review), as well as clear demonstrations of the value of relating relative brain age to phenotype (<https://elifesciences.org/articles/54055>). While most do not use gamlss, their goals and base strategy are comparable, and the work should be acknowledged. Prior work also exists that is directly focused on applications of GAMLSS in fMRI (e.g., <https://www.ncbi.nlm.nih.gov/pmc/articles/PMC4387093/>). The key (and meritorious) innovation of the submitted work is that it brings together the datasets needed to sample the near entirety of the lifespan, along with clinical variations.

We accept that there was room for improvement in the quality of scholarship in the previous version of the paper. We are also grateful to the editor for encouraging us to include additional content and references in order to situate our study more clearly in the context of relevant prior studies. Overall, we have added 14 new references to the paper to provide a more balanced assessment of the prior literature (note: we also included a further 11 references to acknowledge open MRI datasets more explicitly in the main text).

In revising the main text, we focused on citing prior studies that used similar methods of quantile scoring based on GAMLSS or other growth curve models to benchmark MRI phenotypes against normative age-related trends. There is another body of prior literature which is focused on the distinct objective of predicting “brain age” (or the difference between brain age and chronological age) from MRI phenotypes. In the interests of clarity and concision, we have not referred to such brain age prediction papers in the main text but we have added additional text to Supplemental Information that cites these additional papers and makes the conceptual distinction between age-normed quantification of MRI phenotypes and

MRI-based prediction of brain age: **SI 1 “Modelling lifespan trajectories of brain maturation”**.

<<The following changes were made to the main text>>

In: Introduction

Primary case-control studies are usually focused on a single disorder despite evidence of trans-diagnostically shared risk factors and pathogenic mechanisms, especially in psychiatry^{1,2}. Harmonization of MRI data across primary studies to address these and other deficiencies in the extant literature is challenged by methodological and technical heterogeneity. Compared to relatively simple anthropometric measurements, like height or weight, brain morphometrics are known to be highly sensitive to variation in scanner platforms and sequences, data quality control, pre-processing and statistical analysis³, thus severely limiting the generalisability of trajectories estimated from any individual study⁴. Collaborative initiatives spurring collection of large-scale datasets^{5,6}, recent advances in neuroimaging data processing^{7,8}, and proven statistical frameworks for modelling biological growth curves^{9–11} provide the building blocks for a more comprehensive and generalisable approach to age-normed quantification of MRI phenotypes over the entire lifespan (see **SI 1** for details and consideration of prior work focused on the related but distinct objective of inferring brain age from MRI data). Here, we demonstrate that these convergent advances now enable the generation of brain charts that i) robustly define normative processes of sex-stratified, age-related change in multiple MRI-derived phenotypes; ii) identify previously unreported brain growth milestones; iii) increase sensitivity to detect genetic and early life environmental effects on brain structure; and iv) provide standardised effect sizes to quantify neuroanatomical atypicality of brain scans collected across multiple clinical disorders. We do not claim to have yet reached the ultimate goal of diagnosis of MRI scans from individual patients in clinical practice. However, the present work proves the principle that building normative charts to benchmark individual differences in brain structure is already achievable; and provides a suite of open science resources to accelerate further progress in the direction of standardised quantitative assessment of MRI data by the global neuroimaging research community.

In: Individualised centile scores in clinical samples

This approach is conceptually similar to quantile rank mapping, as previously reported^{12–14}, where the (a)typicality of each phenotype in each scan is quantified by its score on the distribution of phenotypic parameters in the normative or reference sample of scans, with more atypical phenotypes having more extreme centile (or quantile) scores.

<<The following changes were made to the supplementary information>>

In: SI 1. Modelling lifespan trajectories of brain maturation

Finally, it is worth noting that the strategic intent of this study (and some directly relevant prior work) was to quantify brain structural MRI phenotypes relative to age- and sex-specific norms, rather than to predict chronological or biological age of participants from their MRI data^{15,16}. There is a large extant literature on attempts to predict “brain age” and compare brain age to the actual age of study participants^{16–21}. In contrast we do not ask the question: what is a participant’s neurobiological age, or the difference between their neurobiological and chronological ages²², given their brain morphology? Instead we ask: how (a)typical is a participant’s brain structure compared to their demographically matched peer group? More

formally, we assess the vertical deviation of an individual scan from the normative trajectory of the corresponding phenotype in a reference population; whereas brain-age prediction attempts to quantify the horizontal deviation. Brain charting is more analogous than brain age prediction to the ways that traditional growth charts are used in pediatric practice for anthropometric variables. Additionally, normative growth curves allow us to benchmark even a single MRI phenotype – such as one of the global tissue volumes that are abundantly available across primary datasets – as opposed to brain age predictions that typically require a high-dimensional feature space comprising multiple MRI phenotypes^{15,22}. In addition, several methodological critiques of brain age prediction are not relevant to the present approach^{22–24}. Thus, we note that using GAMLSS to quantify centile dispersion of MRI phenotypes on age-normed and sex-stratified distributions shares conceptual goals with, but methodologically entirely distinct from, studies that seek to predict human age (or derive a ‘brain age gap’) from brain imaging data²⁵.

Ref 1/2:The discussion acknowledges the work is a proof of principle – why is that not in the intro? And again, looking at the backdrop of the literature, what principle(s) needed to be proven – please be more specific.

We have added clarifying statements about proof of principle to the abstract, introduction and discussion of the main text.

<<The following changes were made to the main text>>

In Abstract

In sum, brain charts are an essential first step towards robust quantification of individual deviations from normative trajectories in multiple, commonly-used neuroimaging phenotypes. Our collaborative study proves the principle that brain charts are achievable on a global scale over the entire lifespan, and applicable to analysis of diverse developmental and clinical effects on human brain structure. Furthermore, we provide open resources to support future advances towards adoption of brain charts as standards for quantitative benchmarking of typical or atypical brain MRI scans.

In Introduction

We do not claim to have yet reached the ultimate goal of quantitatively precise diagnosis of MRI scans from individual patients in clinical practice. However, the present work proves the principle that building normative charts to benchmark individual differences in brain structure is already achievable at global scale and over the entire life-course; and provides a suite of open science resources for the neuroimaging research community to accelerate further progress in the direction of standardised quantitative assessment of MRI data.

In Discussion

The analogy to paediatric growth charts is not meant to imply that brain charts are immediately suitable for benchmarking or quantitative diagnosis of individual patients in clinical practice. Even for traditional anthropometric growth charts (height, weight and BMI) there are still significant caveats and nuances concerning their diagnostic interpretation in individual children²⁶; and, likewise, it is expected that considerable further research will be required to validate the clinical diagnostic utility of brain charts. However, the current results bode well for future progress towards digital diagnosis of atypical brain structure and development²⁷.

Ref 1/3: Despite the prominence of concerns about data quality and confounds from the reviewers, the authors barely acknowledge them in the main manuscript. This is a concern – especially given some of the concerns raised under the specific comments section below. Data quality and the various confounds would not be expected to preclude meaningful delineation of the mean lifespan trajectory – it is the variance that will be impacted. The greater the representation of noise in the variance, the more limited the data will be in detecting meaningful pathology-related deviations beyond what can be seen by eye (e.g., Alzheimer’s patients). It is doubtful the clinically relevant tools pointed at by this work will be meaningfully realized without accounting for data quality – yet, the authors minimize it. I find this highly problematic.

We have now provided a more extensive and well-referenced discussion of potential confounds and caveats, especially related to MRI data quality and quality control, throughout the revised main text and Online Methods. In response to this comment and subsequent comments specifically about quality control based on the Euler index, we have also added new analyses to the Supplementary Information: see **SI “Euler index and neuroimaging phenotypes”**.

<<The following changes were made to the main text>>

In Introduction

Compared to relatively simple anthropometric measurements, like height or weight, brain morphometrics are known to be highly sensitive to variation in scanner platforms and sequences, data quality control, pre-processing and statistical analysis³, thus severely limiting the generalisability of trajectories estimated from any individual study⁴.

In Discussion

Several important caveats are worth highlighting. The use of brain charts does not circumvent the fundamental requirement for quality control of MRI data. We have shown that GAMLSS modelling of global structural MRI phenotypes is in fact remarkably robust to inclusion of poor quality scans (**SI2**), but it should not be assumed that this level of robustness will apply to future brain charts of regional MRI or functional MRI phenotypes; therefore the importance of quality control remains paramount. It will also be important in future to represent ethnic diversity appropriately in normative brain charts^{28,29}. Even this large MRI dataset was heavily biased towards European and North American populations, as is also common in anthropometric growth charts and existing genetic datasets. Further increasing ethnic and demographic diversity in MRI research will enable more population-representative normative trajectories^{28,29} that can be expected to improve the accuracy and strengthen the interpretation of centile scores in relation to appropriate norms¹⁴. The available reference data were also not equally representative of all ages, e.g., foetal, neonatal and mid-adulthood (30-40y) epochs were under-represented (**SI17-19**). While our statistical modelling approach was designed to mitigate study- or site-specific effects on centile scores, it cannot entirely correct for limitations of primary study design, such as ascertainment bias or variability in diagnostic criteria.

As ongoing and future efforts provide increasing amounts of high quality MRI data, we predict an iterative process of improved brain charts for an increasing number of multimodal¹⁷ neuroimaging phenotypes. Such diversification will require the development, implementation,

and standardisation of additional data quality control procedures³⁰ to underpin robust brain chart modelling.

<<The following changes were made to the online methods>>

The resulting models were evaluated using several sensitivity analyses and validation approaches. See **Supplementary Information [SI]** for further details regarding GAMLSS model specification and estimation (**SI1**), image quality control (**SI2**), model stability and robustness (**SI3-4**), phenotypic validation against non-imaging metrics (**SI3 & SI5.2**), between-study harmonisation (**SI5**) and assessment of cohort effects (**SI6**). While the models of whole brain and regional morphometric development were robust to variations in image quality, and cross-validated by non-imaging metrics, we expect that several sources of variance, including but not limited to MRI data quality and variability of acquisition protocols, may become increasingly important as brain charting methods are applied to more innovative and/or anatomically fine-grained MRI phenotypes. It will be important for future work to remain vigilant about the potential impact of data quality and other sources of noise on robustness and generalisability of both normative trajectories and the centile scores derived from them.

<<The following changes were made to the supplementary information>>

In SI 2.4 “Euler Index and neuroimaging phenotypes”

In short, we have demonstrated by multiple complementary QC studies that our principal results, and additional out-of-sample results for new data not previously analysed, are remarkably robust to image quality across a range of assessments. We conclude that our results are not confounded by uncontrolled image quality issues; but proper QC procedures should, of course, be implemented on all scans before they are submitted for OoS centile scoring on the basis of our model and aggregated reference dataset. In the absence of a single gold standard for automated assessment of imaging data quality, we strongly recommend using a combination of approaches to determine inclusion/exclusion of MRI data for brain charting. In future, as these methods may be extended to more fine-grained structural MRI phenotypes that are likely to be more sensitive to variation in image quality, and/or to benchmark phenotypes measured in fMRI or more innovative modalities of MRI data more likely to be measured in small samples ($N < 100$), we should be prepared for GAMLSS modelling to be significantly less robust to image quality in comparison to the case of global MRI phenotypes, like cerebrum tissue volumes. The importance of rigorous quality control therefore remains paramount.

Ref1/4: The authors seem to view batch effect correction as an answer to address the many sources of variance in their analyses (e.g., diagnostic protocol, imaging protocol, sampling strategy). When sample composition or procedures are too dramatically different, batch effect correction will not solve the issue. Vogelstein recently tried to draw attention to challenges in batch effect correction in his Causal Combat framework (see biorxiv). At a minimum, greater acknowledge of differences in diagnostic protocols and sample composition should be highlighted in the main text, as it helps motivate future work toward standardization.

We agree with the reviewer that batch correction is not a cure-all that addresses all the challenges of data variability and harmonisation, especially in clinical cohorts. As it is not feasible, likely, or perhaps even advisable, to seek harmonisation at the data acquisition stage

(i.e., in the context of specific disease or age groups requiring specialised or bespoke acquisition protocols), we opted for a harmonisation procedure that uses the control data from a given study to determine that study's relative offset to the age- and sex-matched norms defined by the aggregated reference dataset. Thus, our approach provides a way to align diverse datasets to a reference standard (provided sufficient control data is available per primary study), despite technical and biological sources of heterogeneity that are likely to persist in clinical and research imaging practice. We now provide more explicit acknowledgement in the main text that our approach is not a cure-all solution to every challenge related to data harmonisation. We also provide a more extensive discussion of the limitations of GAMLSS, compared to other novel ComBAT-related approaches to data harmonisation in the supplementary materials: see **SI** “Modelling of between-site heterogeneity by GAMLSS: empirical evaluation compared to ComBAT”.

<<The following changes were made to the main text>>

In Discussion:

Several important caveats are worth highlighting. The use of brain charts does not circumvent the fundamental requirement for quality control of MRI data. We have shown that GAMLSS modelling of global structural MRI phenotypes is in fact remarkably robust to inclusion of poor quality scans (**SI2**), but it should not be assumed that this level of robustness will apply to future brain charts of regional MRI or functional MRI phenotypes; therefore the importance of quality control remains paramount. It will also be important in future to represent ethnic diversity appropriately in normative brain charts^{28,29}. Even this large MRI dataset was heavily biased towards European and North American populations, as is also common in anthropometric growth charts and existing genetic datasets. Further increasing ethnic and demographic diversity in MRI research will enable more population-representative normative trajectories^{28,29} that can be expected to improve the accuracy and strengthen the interpretation of centile scores in relation to appropriate norms¹⁴. The available reference data were also not equally representative of all ages, e.g., foetal, neonatal and mid-adulthood (30-40y) epochs were under-represented (**SI17-19**). While our statistical modelling approach was designed to mitigate study- or site-specific effects on centile scores, it cannot entirely correct for limitations of primary study design, such as ascertainment bias or variability in diagnostic criteria.

<<The following changes were made to the supplementary information>>

In SI 5.2 Modelling of between-site heterogeneity by GAMLSS: empirical evaluation compared to ComBAT

In short, there are pros and cons to both harmonisation strategies: ComBAT is better suited for smaller datasets, normalised distributions and multivariate phenotypes; whereas GAMLSS is well suited for large datasets, non-Gaussian distributions and univariate phenotypes. We preferred GAMLSS on the grounds of its greater scalability and flexibility to match the distributional properties of the reference data and the objectives of this project. It is beyond the scope of the present work to provide an exhaustive review on batch correction methods or to evaluate the performance of GAMLSS (or ComBAT) for correction of batch effects under all possibly relevant experimental conditions. We emphasize that our use of GAMLSS for between-site or between-study harmonisation may not be optimal for studies with small ($N < 100$) numbers of healthy control participants per site (**SI 4.5**). In addition, GAMLSS will not mitigate study- or site-specific effects driven by ascertainment bias or variability in diagnostic

criteria between sites. Adaptations of ComBAT have been proposed for batch effect correction of multi-site data where such factors are likely to be problematic³¹. However, these approaches may not be suitable for harmonisation of datasets with partially or totally non-overlapping age-ranges, as required for integration of primary studies to estimate brain charts over the entire lifespan.

Ref 1/5 While I respect the value of team science, the authors are missing my point in their response. Generators for open datasets are not recognized on the author line, yet those for more difficult to access datasets are. This is not a push to add the open datasets to the author line; but rather to ask if those who generated the more restricted datasets would consider limiting their acknowledgement to the text body (at least for data with one or more prior publications).

We acknowledge that we had not fully appreciated the scope of the reviewer's previous comment on this point, and it has been valuable for us to consider these issues in greater depth. First and foremost, we agree completely with the reviewer that provision of truly open data, e.g., data that are accessible to other researchers without requiring any special efforts by the data-generating team or any bespoke legal agreements between teams, is an invaluable service to the neuroimaging community that should be recognised or rewarded, and certainly not disincentivised, by authorship and citation practices. To this end, we have revised the main text in several places, including explicit citations and acknowledgements, to highlight the very important contribution of several open MRI datasets to our work.

We have also carefully considered the reviewer's suggestion that we might ask some people already listed as co-authors to step down from the author line and to have their contributions acknowledged in some other way. However, we would strongly prefer not to take this course of action, for two related reasons.

First, our authorship strategy was planned from the outset to be inclusive of individuals who satisfied the journal's criteria for authorship by making a significant contribution in one or more of the following areas: design of the study, acquisition or analysis of data, provision of software, or drafting or substantive review of the paper. These criteria were satisfied by all the named authors. For example, named authors contributing data to the project made special legal arrangements for sharing data and/or made specific efforts to optimise quality control or pre-processing of their data prior to sharing them with us and/or provided previously unpublished information about how the data had been collected and pre-processed. All authors reviewed and commented on a series of drafts of the paper. We are therefore confident that all the individually named authors merit authorship by the journal's criteria. In cases where we were not satisfied that individuals merited authorship on this basis, we did not offer them a position on the author line, but listed their names as members of the relevant consortium (detailed in supplementary information **SI 22 & 23**) and named the consortium on the author line. All authors understand that if further papers are published on the basis of the reference dataset aggregated for the first time in this paper, they will be acknowledged as members of the Brain Chart Consortium, which will be named as a corporate author; but they will not be individually named as authors of future papers based on this work, unless they have made additional significant contributions to future papers that go beyond their significant contributions to this foundational paper.

Second, it would be very difficult to go back to authors already listed as such and ask them to reconsider their position. Some of the primary datasets were released to us under legal or ethical constraints that mandate recognition of significant contributions in terms of authorship. Preliminary consultation with some authors further indicated that a request to reconsider their position would raise questions about the primacy of the journal's criteria in deciding authorship and would not be interpreted equivalently by all authors. For example, it seemed likely that more junior authors might be more amenable to stepping down from the author line than more senior authors, although it was often the more junior authors that had made the most significant contributions of data or expertise to the project. Thus, it is conceivable that by attempting to make further adjustments to the author line, we would introduce more or different biases in selection, relative to the standard procedure for author selection by journal criteria, which we have carefully adopted since the start of this project.

Finally, this discussion with reviewer 1 and the editor has been very instructive from our point of view, and has raised many general issues for the future of open human brain science. We have added some additional text to Supplementary Material, providing interested readers with more background detail: "**SI 21: A note on data sharing**"

<<the following changes were made to the main text>>

In: Mapping normative brain growth

See **SI19** for details on all primary studies contributing to the reference dataset, including multiple publicly available open MRI datasets^{32–42}.

In: Acknowledgements

We would particularly like to acknowledge the invaluable contribution to this effort made by several openly-shared MRI datasets, specifically: OpenNeuro (<https://openneuro.org/>), the Healthy Brain Network (<https://healthybrainnetwork.org/>), UK BioBank (<https://www.ukbiobank.ac.uk/>), ABCD (<https://abcdstudy.org/>), the Laboratory of NeuroImaging (<https://loni.usc.edu/>), data made available through the Open Science Framework (<https://osf.io/>), COINS; (<http://coins.mrn.org/dx>), the Developing Human Connectome Project (<http://www.developingconnectome.org/>), the Human Connectome Project (<http://www.humanconnectomeproject.org/>), the OpenPain project (<https://www.openpain.org/>), the International Neuroimaging Datasharing Initiative (INDI; http://fcon_1000.projects.nitrc.org/), and the NIMH Data Archive (<https://nda.nih.gov/>); see **SI21** for details on open human MRI science..

<<The following changes were made to the supplementary information>>

The complete dataset aggregated for the purposes of this study contains primary datasets that differ quite widely in terms of their "openness," i.e., their availability for secondary use without restrictions or special efforts by the primary study team. Primary studies ranged from fully open and downloadable datasets in the public domain to more restricted datasets that could only be used for specific purposes, under specific agreements, or after special efforts had been made to provide QC'd data in shareable form. There can be several reasons why data aren't always and immediately shared openly and/or without the active involvement of the researchers who collected the data⁴³. In our experience within the context of this project, the

various factors operating to prevent complete openness can be organised roughly into four categories:

- No informed consent was obtained for the open sharing of data at the time of collection⁴⁴ (or the informed consent does not extend to other uses in general).
- Data protection regulations, either at national or institutional levels, prevent the sharing of more detailed data such as essential demographics.
- The funding agency mandated or encouraged explicit involvement of researchers who collected primary study data in secondary studies where data was shared.
- Primary studies are still ongoing and data cannot be shared openly until the primary study objectives and/or milestones have been achieved.

There are also several reasons for not sharing data openly that cut across these categories such as general concerns about privacy or confidentiality of participants (which may be expressed by researchers, funders or governance bodies), as well as issues of data ownership (which are actively evolving as a result of changing legislation in some jurisdictions, e.g., General Data Protection Regulations [GDPR] in the European Union since 2016).

For these reasons, in practice, data is often shared under individually tailored and specific data usage or material transfer agreements. In the absence of a unified standard academic agreement this means that there is considerable variability in the terms under which data is or can be shared. For the present project, we sometimes had to make the difficult decision not to include potentially relevant datasets because abiding by the terms of the proposed sharing agreements would not have satisfied journal criteria for authorship and/or would have created an unbalanced acknowledgement of individual authors' contributions.

The benefits of truly open data are very clear from a scientific perspective. More open datasets would increase the number and diversity of researchers who are able to conduct secondary or meta-analytic studies without the need to negotiate multiple individual usage agreements. The present project would not have been possible without the availability of several exemplary open datasets³²⁻⁴², which were particularly valuable at the outset of this project, by facilitating pilot studies of brain charting methods. However, journal authorship criteria meant that we could not include members of some of the most open consortia as co-authors because their data were readily available to us without any significant additional contribution meriting authorship. We note that this situation potentially disincentives open science, since the people who do most to make their data openly available could be least likely to merit co-authorship of secondary studies. We therefore consider it is important for all stakeholders (funders, journals, investigators) to continue to think about how open human brain science can be properly recognised and rewarded. Here we have explicitly referenced and acknowledged our debt to the several open MRI datasets without which this study would not have been possible, because and although it has not always been appropriate to list the principal architects of these datasets as co-authors of this paper.

Ref 1/6 Optimal Euler # can vary as a function of site; no general value is established to my knowledge. This is visible in Figure S2.1.2 (note, there are two figures with the same number), where there is large variability in the mean EI across dataset (For example, the Oulu and AOBA have their median EI > 217). Yet, it seems the authors are using a single cutoff based

on the initial paper from the PNC to try to fit all; this will likely lead to both inappropriate inclusion and exclusion of data.

We fully agree with the reviewer that a single cut-off is inevitably arbitrary and this was the primary reason why we did not base our QC solely on this one approach, but have also included a dimensional approach to analysis of the effects of the Euler index, and sensitivity analysis based on in a subset of visually QC-ed images. Dimensional analysis of the correlations between centile scores of various global MRI phenotypes, and EI as a measure of MRI data quality, have consistently demonstrated very limited impact of high or low EI on the normative trajectories of median and variance, or on centile scores benchmarked against these norms. We have provided further evidence for robustness of norms and centile scores on global MRI phenotypes to image quality by reporting highly consistent results from a reference dataset selected by different EI-based criteria from the total available pool of scans: see **SI “2. Quality control”** for details.

In an effort to further clarify that our results are not sensitive to a single arbitrary QC criterion, we have removed the QC analysis using EI = 217 as the quality criterion applied uniformly across all primary studies, and we have reported instead a new analysis using EI > 2 MAD (median absolute deviations above the median) for each primary study. This approach to EI filtering of available data does not impose a uniform prior threshold on EI, but is more adaptive to the variability of image quality within a study or site. Using EI > 2 MAD for QC resulted in exclusion of a larger percentage of available scans (9-10%, depending on MRI phenotype), but there was virtually no difference in resulting norms and centile scores compared to the results reported using the full dataset (or the dataset QC'd by the more inclusive EI > 217 threshold, which only excluded ~5% of available scans). We agree with the reviewer that QC remains of paramount importance for future development of large-scale open neuroimaging science.

<<The following changes were made to the supplementary information>>

In SI 2.1 Euler Index filtering

Although cerebrum tissue volumes are expected to be less sensitive to cortical surface topology, compared to surface-based measures such as indices of cortical folding (see **SI18 “Data processing”**), EI has previously been used as a measure of the quality of “raw”, unprocessed scans⁴⁵. Thus, for the large majority of studies where EI was available (N=101,708 total scans on N=82,023 unique subjects), we assessed the impact on reference models of excluding high-magnitude EI scans. Given that no single EI threshold is expected to be generalizable across studies⁴⁵ (**Fig. S2.1.2**), in this sensitivity analysis we excluded scans that had EI magnitude greater than 2 median absolute deviations from the primary study-specific median EI. This QC threshold, which is adaptive to the variable quality of scans between primary studies, excluded approximately 9-10% of scans from the original dataset. However, as can be seen in **Fig. S2.1.3**, the resulting model parameters were highly correlated with parameters estimated from the reference dataset without applying any EI-based QC threshold. The developmental trajectories estimated for all 4 cerebrum tissue volumes were highly correlated with their trajectories estimated on the basis of the full dataset (all $R^2 > 0.999$ for parametric [Pearson's] and non-parametric [Spearman's] correlations between EI-filtered vs EI-unfiltered median trajectories and lower (2.5%) and upper (97.5%) centiles). Identical parameterisation of fractional polynomials for each random effect was identified by the same

model selection procedure was found in both EI-filtered and EI-unfiltered datasets. Importantly, EI-filtered and unfiltered datasets also showed a high degree of overlap in subsequently estimated model parameters (correlation of study-specific mean (μ) components > 0.99; correlation of study-specific variance (σ) components > 0.93). Model specification thus appeared to be robust to the presence of the poorer quality data.

In addition, we examined the relationships between image quality measured by EI and individual centile scores of each brain phenotype. Both for the full dataset and the EI-filtered subset of higher quality scans, we found no significant associations between EI and individual centile scores (**Fig. S2.1.1**), nor did we find evidence for a non-linear relationship (quadratic, cubic, logarithmic) between EI and centiles.

Fig. S2.1.1 Associations between centile scores and MRI scan quality defined by EI. Panel depicts the relation between Euler indices (EI)⁴⁵ and centile scores for each of 4 cerebrum tissue volumes estimated by GAMLSS. The Spearman correlations between EI and centile scores were negligible (GMV, $p < 0.01$; WMV, $\rho = -0.07$; sGMV, $p < 0.01$; Ventricles, $\rho = 0.05$). All linear mixed effect models examining non-linear (quadratic, cubic or logarithmic) relationships between EI and centile scores for each phenotype were $P > 0.1$.

To assess whether there were any age-related differences that could influence model estimation, we evaluated the linear effect of age (in years) on EI in healthy controls in the reference dataset used to estimate normative lifespan trajectories. Using linear regression stratified by sex and accounting for study-specific random effects, we found no evidence for an age-related bias in image quality as assessed with EI ($t = -1.244$, $P = 0.213$). **Fig. S2.1.2** shows the median and standard deviation of age and EI and highlights the top 10 studies with the highest median EI.

Fig. S2.1.2 Age-related variation in image quality measured by the Euler index in female (left panel) and male (right panel) control subjects. Median age (in years) and median EI are shown per study with cross-hairs indicating the standard deviations for age and EI per study. In red the top ten studies with the highest median EI are highlighted. There is no significant relationship between image quality and age at scanning.

Figure S2.1.3 Robustness of GAMLSS parameters to quality control by exclusion of scans with EI greater than twice the median absolute deviation (MAD) from the median EI in the corresponding primary study. Scatterplots show the relationships between random effects (μ on the top row and σ on the bottom row) estimated for each primary study without exclusion of poor quality scans (y-axis) and for each primary study after exclusion of scans with $EI > 2 \text{ MAD}$, relative to the primary study's median EI. Colored points indicate the relative percentage of primary studies retained after filtering (darker means for subjects were removed) and Rho values in the titles indicate Spearman's correlations between parameters estimated from the unfiltered and EI-filtered datasets. The biggest discrepancy in study-specific random effects as a result of excluding poor quality scans was observed for the variance (σ) parameters, especially those estimated from a subset of datasets, which all included a relatively high proportion of excluded scans. In general, random effect parameter estimation was highly robust to adaptive EI thresholding for quality control.

In SI 2.4 Euler Index and neuroimaging phenotypes

In short, we have demonstrated by multiple complementary QC studies that our principal results, and additional out-of-sample results for new data not previously analysed, are remarkably robust to image quality across a range of assessments. We conclude that our results are not confounded by uncontrolled image quality issues, but proper QC procedures should, of course, be implemented on all scans before they are submitted for OoS centile scoring on the basis of our model and aggregated reference dataset. In the absence of a single gold standard for automated assessment of imaging data quality, we strongly recommend using a combination of approaches to determine inclusion/exclusion of MRI data for brain charting. In future, as these methods may be extended to more fine-grained structural MRI phenotypes that are likely to be more sensitive to variation in image quality and/or to benchmark phenotypes measured in fMRI or more innovative modalities of MRI data, we should be prepared for GAMLSS modelling to be significantly less robust to image quality in comparison to the case of global MRI phenotypes, like cerebrum tissue volumes. The importance of rigorous quality control therefore remains paramount.

Ref 1/7 Have the authors looked at the relationship of EI and volume within a dataset?

This is an interesting question, which we had not previously considered. We now provide a more comprehensive analysis examining the relationship between EI and volume: see SI 2.4 “Euler index and global MRI phenotypes”.

<<The following changes were made to the supplementary information>>

In SI 2.4 Euler Index and global MRI phenotypes

To further examine the potential influence of quality control on the quantification of MRI phenotypes, we evaluated the relationship between Euler index (EI) and the four main global tissue volumes (GMV, WMV, sGMV, CSF) within each study with available EI data. We observed high variability in the range of EI within and between primary studies (**Fig. S2.4.1**). However, using linear models to assess the relationship between EI and non-centiled (“raw”) tissue volumes for the healthy controls within each primary study (controlling for age and sex), we found that the relationship between EI and tissue volume was generally weak, with only a small subset of primary studies showing significant effects of image quality on MRI phenotypes ($P_{\text{Bonferroni}} < 0.05$, corrected for the number of studies of each phenotype). Critically, the sign of this relationship varied across studies and was zero-centred, with the significant effects observed in primary studies with greater sample size (linear mixed effects model with phenotype as a random effect, comparing sample size and $-\log_{10}(P\text{-values})$ for association with EI: $t = 8.77$, $P = 6e^{-16}$; **Fig. S2.4.2**).

Fig. S2.4.1. Relationships between the distributions of non-centiled (“raw”) cerebrum tissue volumes and Euler index within each primary study. Crosshair plots show the range of values (mean \pm 1 standard deviation) for the Euler Index (EI) and cerebrum tissue volumes for each primary study: clockwise from top left, grey matter volume (GMV), white matter volume (WMV), ventricular cerebrospinal fluid volume (CSF) and subcortical grey matter volume (sGMV). The colour scale represents median log age of participants in each primary study.

Fig. S2.4.2. Model statistics examining the relationships between non-centiled (“raw”) cerebrum tissue volumes and the Euler index within each primary study. Volcano plots show the t-statistics (x-axis) versus negative log-scaled Bonferroni corrected P-values (y-axis) estimated from linear models of the relationship between Euler Index (EI) and cerebrum tissue volumes: clockwise from top left, grey matter volume (GMV), white matter volume (WMV), ventricular cerebrospinal fluid volume (CSF) and subcortical grey matter volume (sGMV). Each dot represents a single primary study and is coloured to represent the median log age of participants, and scaled to represent the sample size, in a study where there was a significant relationship between cerebrum tissue volume and EI ($P_{\text{Bonferroni}} < 0.05$). It is clear that the sign of association between EI and volumetrics was inconsistent between primary studies and the association tended to be significant for primary studies with larger sample sizes.

Ref 1/8 95% data inclusion (ie quality pass) is surprising; some sites may have only shared quality pass data – is this known?

This high level of passing was somewhat of a surprise to us as well, based on prior experience. **Supplementary Table 1** lists whether preliminary QC was explicitly conducted prior to sharing data for each of the primary studies. In most cases where data was shared directly, these data had already undergone some QC screening. So preliminary QC of primary data could contribute to the high pass rate when we applied an additional and consistent QC threshold

(EI > 217) to all data. , However, as detailed in response to Ref 1/7, the distribution of EI is variable across studies and when EI > 2 MAD is used as a more adaptive QC threshold the pass rate drops to 90-91%, depending on MRI phenotype: see SI “2.1 Euler Index filtering” including new Fig S2.1.3.

<<The following changes were made to the supplementary information>>

In: “SI 2.1 Euler Index filtering”

Figure S2.1.3 Robustness of GAMLSS parameters to quality control by exclusion of scans with EI greater than twice the median absolute deviation (MAD) from the median EI in the corresponding primary study. Scatterplots show the relationships between random effects (μ on the top row and σ on the bottom row) estimated for each primary study without exclusion of poor quality scans (y-axis) and for each primary study after exclusion of scans with EI > 2 MAD, relative to the primary study’s median EI. Coloured points indicate the relative percentage of primary studies retained after filtering (darker means for subjects were removed) and Rho values in the titles indicate Spearman’s correlations between parameters estimated from the unfiltered and EI-filtered datasets. As with the absolute QC threshold of EI < 217 (SI 2.1), the biggest discrepancy in study-specific random effects as a result of excluding poor quality scans was observed for the variance (Sigma) parameters, especially those estimated from the ICBM, HBN and EDSB datasets, which all included a relatively high proportion of excluded scans. We note that EI > 2 MAD filtering removed a lower proportion of data in primary studies where the distribution of EI was skewed towards higher quality/lower EI across the whole dataset (e.g., HCP, ABCD and UKB all have high data quality with low EI, and 2 MAD filtering in these studies only removed around 6-7% of data). In general, random effect parameter estimation was highly robust to adaptive EI thresholding for quality control.

Ref 1/9 Figure S2.1.2 – the authors are relying on Spearman to state that no association exists between centile and MRI quality. That assumes a linear relationship; this could miss a more complex association, which looking at the data – there is visibly higher EI in the low centiles.

We agree with the reviewer that Spearman’s correlation is a measure of linear association and there could be a more complex relationship between EI and cortical morphology (either quantified as raw phenotypes or as centiles). We therefore additionally explored possible non-

linear associations (quadratic, cubic, logarithmic) between centile scores for global MRI phenotypes and EI, and found no significant non-linear relationships.

<<the following changes were made to the supplementary information>>

In: SI 2.1 Euler Index filtering

...nor did we find evidence for a non-linear relationship (quadratic, cubic, logarithmic) between EI and centiles.

Fig. S2.1.1 Associations between centile scores and MRI scan quality defined by EI. Panel depicts the relation between Euler indices (EI)⁴⁵ and centile scores for each of 4 cerebrum tissue volumes estimated by GAMLSS. The Spearman correlations between EI and centile scores were negligible (GMV, $\rho < 0.01$; WMV, $\rho = -0.07$; sGMV, $\rho < 0.01$; Ventricles, $\rho = 0.05$). All linear mixed effect models examining non-linear (quadratic, cubic or logarithmic) relationships between EI and centile scores for each phenotype were $P > 0.1$.

Ref 1/10 Figure S2.2.1 – a score of 4 seems biased for GMV – why only exclude 5 and 6? Repeating with 4 excluded would make sense.

We agree with the reviewer that the 4th category in GMV may still show some bias. We repeated the analysis, retaining only the top 3 classes of GMV, and again found the same level of stability.

<<the following changes were made to the supplementary information>>

Additionally excluding the 4th category for GMV did also not impact the stability of the resulting trajectories ($R^2 > 0.999$ for both Pearson's and Spearman's correlations).

Ref 1/11 Comparison of GAMLSS vs. ComBat only carried out in ABCD, which uses V-NAV (thus has minimal motion) and is harmonized; what about other sites.

We agree with the reviewer that the comparison of GAMLSS and ComBAT, or any other pair of batch correction methods, is not a trivial business. Specifically, in the absence of a ground truth, it is difficult to determine whether any approach to site-harmonisation over- or under-corrects relative to the true technical nuisance it aims to control for. We chose the ABCD study as an example, not because it used V-NAV, but because it is a harmonised study and designed as a multi-site cohort. In addition, it provides a relatively large per-site sample size that makes it a good candidate for GAMLSS harmonisation. (As explicitly noted in the previous cycle of revision, ComBAT is more likely to be optimal for smaller datasets, with a proportionally higher

dimensional feature space). In the ABCD data, both GAMLSS and ComBAT do an excellent job at removing site-related variation. However, in response to the reviewer's concern that the result of GAMLSS vs ComBAT comparison is conditional on a single dataset, we have now reported a new comparative analysis using a different multi-site dataset (IMAGEN), with very similar results: see SI 5.2 "Modelling of between-site heterogeneity by GAMLSS: empirical evaluation compared to ComBAT" including Figures S5.2.6 and S5.2.7.

<<The following changes were made to the supplementary information>>

In: SI 5.2 Modelling of between-site heterogeneity by GAMLSS: empirical evaluation compared to ComBAT

We specifically chose the ABCD study to test the capacity of GAMLSS and ComBAT to remove between-site noise because it is a demographically harmonised multi-site cohort. In addition, the large sample size of healthy controls per site makes ABCD highly suitable for GAMLSS harmonisation of between-site differences (see also SI 4.5). This means that in the context of ABCD any residual significant differences between sites are less likely to be due to true site variation or recruitment differences and more likely to be due to noise (technical or otherwise), though even in this study recruitment bias can not be fully eliminated. In addition, the ABCD dataset also provided a wide range of non-MRI phenotypes to test any downstream impact of batch-effect correction approaches on analyses of association between MRI centile normalised scores and non-MRI phenotypes. Despite being a technically harmonised cohort, and despite using acquisition protocols that included prospective motion correction, the uncorrected ABCD imaging data still show clear and significant differences between sites across all MRI phenotypes.

Few other datasets fit the selection criteria used for the specific comparison between GAMLSS and ComBAT approaches to normalisation (i.e., $N > 100$ healthy control participants per site, aligned recruitment criteria, and broadly aligned MRI data acquisition protocols). The only other multi-site datasets fitting these criteria in our aggregated dataset were the IMAGEN and UK BioBank cohorts. To explore whether the harmonisation approach worked well in a cohort other than ABCD we chose the IMAGEN cohort as UK BioBank implements an extremely well-harmonised acquisition and recruitment strategy across its 3 sites. While we did not have access to the same extensive set of non-neuroimaging based phenotypes in the IMAGEN dataset as we had for the ABCD dataset, we observed that both GAMLSS and ComBAT were highly effective at removing large site-related variation from raw neuroimaging phenotypes (Fig. S5.2.6-5.2.7).

Fig. S5.2.6. Raw volumetric data and centile scores for female participants from the IMAGEN cohort. Top row shows raw volumetric data across the different sites included in IMAGEN, the middle row shows centile normalised data by GAMLSS, and the bottom row shows data normalised using ComBAT. ANOVA uncorrected P-values refer to one-way analyses of variance across sites for each individual phenotype. ComBAT and GAMLSS are both able to substantially mitigate batch effects in multi-site MRI data from the IMAGEN study (as well as the ABCD study).

Fig. S5.2.7. Raw volumetric data and centile scores for male participants from the IMAGEN cohort. The top row shows raw volumetric data across the different sites included in IMAGEN, the middle row shows centile normalised data by GAMLSS and the bottom row shows data normalised using ComBAT. ANOVA uncorrected P-values refer to one-way analyses of variance across sites for each individual phenotype. Bars are coloured by site. ComBAT and GAMLSS are both able to substantially mitigate batch effects in multi-site MRI data from the IMAGEN study (as well as the ABCD study).

It is beyond the scope of the present work to provide an exhaustive review of batch correction methods or to evaluate the performance of GAMLSS (or ComBAT) for correction of batch effects under all possibly relevant experimental conditions. We emphasise that our use of GAMLSS for between-site or between-study harmonisation may not be optimal for studies with small numbers of controls per site (SI 4.5). In addition, it will not mitigate study- or site-specific effects driven by ascertainment bias or variability in diagnostic criteria across sites that may contribute to site-specific variation. Adaptations of ComBAT have been proposed for batch

effect correction of multi-site data where such factors are likely to be problematic³¹; although these approaches may not be suitable for harmonisation of datasets with partially or totally non-overlapping age-ranges, as required for integration of primary studies to estimate brain charts over the entire lifespan.

Ref 1/12 How was the visual inspection performed? Where each image rated by one expert or many? If more than one rater, how was the ICC?

For each subject a slice stack of images was generated across the three axes, after bias field correction and intensity normalisation, so that they were all easily comparable by visual inspection, and subsequently rated on motion corruption and other failure modes (artefacts, missing brain parts, etc). This was done by a single individual after a consensus labelling schema had been agreed.

<<The following changes were made to the supplementary information>>

In SI 2.2 Expert visual quality control

These scans were provided by openly available datasets and are marked as having “Manual” quality control in the “Extracted QC” column in **ST1.1**. For each subject a slice stack of images was generated across the three axes, after bias field correction and intensity normalization, so that they were all easily comparable by visual inspection, and subsequently rated on motion corruption and other failure modes (artefacts, missing brain parts etc). Visual inspection then rated each image on the following questions: is the brain fully covered by the scan; is there visible noise (due to aliasing, motion etc.), blurriness, or ringing; is there acceptable tissue contrast and image orientation? Based on these criteria, each raw scan was expertly classified on a 6-point scale as perfect (1), very good (2), good (3), bad (4), very bad (5) or unacceptable (6). Only 3% of scans (N=374) were assigned to the two worst quality categories (5 and 6). Each image was rated by a single rater.

Ref 1/13 Page 6 – First paragraph. – “The only exception to this generally high ... were the GISTO and EDSD cohorts where excluding scans with EI > 217 substantially reduced the number of scans (by >30%)...” But, by looking at the second Figure S2.1.2, it looks that more than 50% of the data for males in the Oulu, NIH, AOBA, WAYNE, ICBM have a EI > 217. So which one is correct?

We thank the reviewer for picking this up. The original figure was correct and the percentage mentioned in the text did not convey the variability of excluded percentages across the different studies adequately nor did it stratify male and female percentage (indeed some sites had exclusion in the >50% range). We note that this statement has now been removed from the paper, together with all other text concerning QC with EI > 217, as detailed in response to Ref 1/6 .

Referee #2:

Ref 2/1 Overall, the authors have done a commendable job using a very large convenience meta-analytic sample of individuals with MRI data to produce lifespan trajectories of gross brain structure, and variation therein. The key findings are not so much of note in their own right, but rather constitute a series of checks to ensure that the harmonization process has been successful, and that associations produced by the harmonized data are cleaner. The major contribution here is a resource for other researchers seeking to compare their sample to the wider corpus of MRI research, and those seeking to combine MRI data across multiple datasets in a principled way that avoids bias that is typically associated with aggregation across different MRI studies. However, contrary to what has been implied by the direct comparison of the brain charts introduced here to height and weight growth charts that are used routinely in clinical settings, the brain charts cannot be used for individual patient diagnosis or treatment. I elaborate on this concern below. In light of them, I believe that a major revision to the framing of the manuscript is needed.

My concern can be distilled down to what is implied by the title “Brain Charts” and the explicit desire expressed (e.g. in the second sentence of the abstract, the first paragraph of the main text) to constructed charts for the brain that akin to growth charts that are routinely used in clinical, mostly pediatric, settings for tracking the trajectory of height and weight growth of individuals relative to the population. I need to be clear that when I say “clinical,” I mean in individual patient monitoring, diagnosis and care, not “clinical research,” which refers to research that is on the topic of clinical populations or clinical treatments. This article needs to, in no uncertain terms, remove this inappropriate comparison between brain charts (which cannot be used clinically for individual patients) and height and weight growth charts (which can be used clinically). Statements such as the following need to be removed: “Crucially, for clinical purposes, centile scores provided a standardised and interpretable measure of deviation that revealed new patterns of neuroanatomical differences across neurological and psychiatric disorders.” [These are not “clinical purposes.” I think that the authors mean “for research into the neuroanatomical correlates of clinical disorders.”]. The first reason that the brain charts cannot be used for individual patient diagnostics is that although the total sample size of the analysis is very large, the studies are not representative of the population (and it is not even clear what the “the population” of interest is; growth charts should be normed relative to the population of their intended use). Inasmuch, it would be grossly inappropriate to assign and individual a centile score relative to the brain chart norms, as the centile is unlikely to correspond to centiles within the population. Centiles can of course be used for the purposes of research, so as to reference an individual’s location within the observed (meta-analytic) sample distribution, but they simply cannot be used clinically without the reference norms being representative. The authors have been responsive to my previous comment about issues concerning representativeness, but I continue to worry that the charts are framed as being of use for individual centile scoring in clinical settings.

We thank the reviewer for their positive appraisal of our original and revised manuscripts.

We carefully considered whether to remove the term “brain chart”, and the analogy to paediatric growth charts for height and weight, from the present manuscript. We recognised that there is a substantial prior literature using the terminology of “brain charts” and making

the explicit analogy to paediatric growth charts – see, for example [4,14,46–48] in the revised manuscript – and we thus consider our use of this terminology to be in line with the broader literature. We have therefore retained this phrase, and the analogy it refers to, in the paper.

However, we completely agree with the reviewer that brain charts are not yet suitable for clinical practice and we apologise for any lack of clarity concerning this point in the previous version of the paper. In this revision, we have extensively and explicitly reframed how the growth chart analogy is explained, drawing attention to the limits of the analogy, and repeatedly making it clear in the main text and supplementary information that brain charts are not suitable for quantitative assessment or diagnosis of MRI scan data from individual patients in clinical practice. We continue to think that further progress towards clinical applications of quantitative MRI is to be expected, and may hopefully be accelerated by our provision of open science resources for brain charting. We have signposted this direction of travel towards future clinical utility; but we have also made it very clear that it is not currently possible to use brain charting technology for assessment or diagnosis of an individual patient's brain MRI scan in clinical practice; and we have drawn attention to some of the clinical limitations of existing anthropometric growth charts in paediatric clinical practice. Finally, we updated the disclaimer on www.brainchart.io to state more clearly that the tool is meant for research purposes and is explicitly not intended for clinical use.

<<the following changes were made to the main text>>

In Abstract:

Over the past few decades, neuroimaging has become a ubiquitous tool in basic research and clinical studies of the human brain. However, no reference standards currently exist to quantify individual differences in neuroimaging metrics over time, in contrast to growth charts for anthropometric traits such as height and weight¹. Here, we built an interactive resource to benchmark brain morphology, www.brainchart.io, derived from any current or future sample of magnetic resonance imaging (MRI) data. With the goal of basing these reference charts on the largest and most inclusive dataset available, we aggregated 123,984 MRI scans from 101,457 participants aged from 115 days post-conception through 100 postnatal years, across more than 100 primary research studies. Cerebrum tissue volumes and other global or regional MRI metrics were quantified by centile scores, relative to non-linear trajectories² of brain structural changes, and rates of change, over the lifespan. Brain charts identified previously unreported neurodevelopmental milestones³; showed high stability of individual centile scores over longitudinal assessments; and demonstrated robustness to technical and methodological differences between primary studies. Centile scores showed increased heritability compared to non-centiled MRI phenotypes, and provided a standardised measure of atypical brain structure that revealed patterns of neuroanatomical variation across neurological and psychiatric disorders. In sum, brain charts are an essential first step towards robust quantification of individual deviations from normative trajectories in multiple, commonly-used neuroimaging phenotypes. Our collaborative study proves the principle that brain charts are achievable on a global scale over the entire lifespan, and applicable to analysis of diverse developmental and clinical effects on human brain structure. Furthermore, we provide open resources to support future advances towards adoption of brain charts as standards for quantitative benchmarking of typical or atypical brain MRI scans.

In Introduction:

The simple framework of growth charts to quantify age-related change was first published in the late 18th century⁴⁹ and remains a cornerstone of paediatric healthcare – an enduring example of the utility of standardised norms to benchmark individual trajectories of development. However, growth charts are currently available only for a small set of anthropometric variables, e.g., height, weight and head circumference, and only for the first decade of life. There are no analogous charts available for quantification of age-related changes in the human brain, although it is known to go through a prolonged and complex maturational program from pregnancy to the third decade⁵¹, followed by progressive senescence from the sixth decade⁵², approximately. The lack of tools for standardised assessment of brain development and aging is particularly relevant to research studies of psychiatric disorders, which are increasingly recognised as a consequence of atypical brain development⁵³, and neurodegenerative diseases that cause pathological brain changes in the context of normative senescence⁵⁴. Preterm birth and neurogenetic disorders are also associated with marked abnormalities of brain structure^{55,56} that persist into adult life^{56,57} and are associated with learning disabilities and mental health disorders. Mental illness and dementia collectively represent the single biggest global health burden⁵⁸, highlighting the urgent need for normative brain charts as an anchorpoint for standardised quantification of brain structure over the lifespan⁵⁹.

We do not claim to have yet reached the ultimate goal of standardised, quantitative diagnosis of MRI scans from individual patients in clinical practice. However, the present work proves the principle that building normative charts to benchmark individual differences in brain structure is already achievable; and provides a suite of open science resources to accelerate further progress in the direction of standardised quantitative assessment of MRI data by the global neuroimaging research community.

In Discussion:

The analogy to paediatric growth charts is not meant to imply that brain charts are immediately suitable for benchmarking or quantitative diagnosis of individual patients in clinical practice. Even for traditional anthropometric growth charts (height, weight and BMI) there are still significant caveats and nuances concerning their diagnostic interpretation in individual children²⁶; and, likewise, it is expected that considerable further research will be required to validate the clinical diagnostic utility of brain charts. However, the current results bode well for future progress towards digital diagnosis of atypical brain structure and development²⁷. By providing an age- and sex-normalised metric, centile scores enable trans-diagnostic comparisons between disorders that emerge at different stages of the lifespan (**SI10-11**). The generally high stability of centile scores across longitudinal measurements also enabled assessment of documented changes in diagnosis such as the transition from MCI to AD (**SI14**), which provides one example of how centile scoring could be clinically useful in quantitatively predicting or diagnosing progressive neurodegenerative disorders in the future. Our provision of appropriate normative growth charts and on-line tools also creates an immediate opportunity to quantify atypical brain structure in clinical research samples, to leverage available legacy neuroimaging datasets, and to enhance ongoing studies.

Ref 2/2 Second, is that differences in MRI protocol and scanner calibration in real-world clinical settings relative to the studies used to construct the brain charts cannot be corrected for. In

their response, the authors go to great lengths to explain how they harmonized the samples and accounted for study-to-study differences, for instance, by obtaining bootstrapped centile scores from leave one sample out analyses. They are correct that these methods strongly mitigate against artifacts in the brain charts that might have otherwise arisen from between-study variation. Indeed, as displayed in Fig S5.2.4, the primary advantage of the centile scoring method is that it allows data from different studies to be combined without associations being attenuated by study-to-study variation. However, this does not mitigate against artifacts that would arise from attempting to centile score an individual from a single outside scan. The authors do appropriately comment that new datasets (with “comparatively smaller samples sizes”) can be integrated with this one, and do not actually indicate that a single scan of an individual in a clinical setting can be centile scored relative to their charts. However, by calling these “brain charts” and comparing them to existing growth charts, the idea that the charts can be used for individuals clinically, rather than simply for individuals within studies, is made implicit.

We completely agree with the reviewer that it is critical for readers to understand what the present tools and resources can be used for, and what they cannot be used for. As noted in response to Ref 2/1, we have extensively changed the abstract, introduction and discussion of the main text in order to make this distinction as clear as possible. Here we highlight additional changes that address more specifically the reviewer’s concerns about interpretation of the results on out-of-sample centile scoring.

<<the following changes were made to the main text>>

In: Out-of-sample centile scoring of “new” MRI data

Extensive jack-knife and leave-one-study-out (LOSO) analyses indicated that a study size of $N > 100$ scans was sufficient for stable and unbiased estimation of out-of-sample centile scores (SI4). This study size limit is in line with the size of many contemporary brain MRI research studies. However, these results do not immediately support the use of brain charts to generate centile scores from smaller scale research studies, or from an individual patient’s scan in clinical practice – this remains a goal for future work.

In: Discussion

Even this large MRI dataset was heavily biased towards European and North American populations, as is also common in traditional growth charts and existing genetic datasets. Further increasing ethnic and demographic diversity in MRI research will enable more population-representative normative trajectories^{28,29} that can be expected to improve the accuracy and strengthen the interpretation of centile scores in relation to appropriate norms and allow more detailed cross-cultural comparisons¹⁴. The available reference data were also not equally representative of all ages, e.g., foetal, neonatal and mid-adulthood (30-40y) epochs were under-represented (SI17-19). While our statistical modelling approach was designed to mitigate study- or site-specific effects on centile scores, it cannot entirely correct for limitations of primary study design, such as ascertainment bias or variability in diagnostic criteria.

Referee #3:

Bethlehem and co-authors have submitted a very impressive revision to their manuscript “Brain charts for the human lifespan”. Extensive additional analyses were performed to address previous weaknesses and criticisms. I commend the authors for their responsiveness to the previous round of reviews. In particular, I am now convinced that their GAMLSS modeling approach is sufficiently robust to the inclusion of new studies. The Supplement is massive, and thus beyond the capability of any single individual to review thoroughly (in a reasonable amount of time). But it is clear from the ~ 1/3 of the supplement that I did review that the value and overall quality of the Supplemental Material is dramatically improved.

I believe that this manuscript has now (nearly) achieved its potential. It will be a very impactful addition to the understanding and future study of human structural neuroanatomy.

We thank the reviewer for their positive assessment of the revised manuscript, and we are extremely grateful for their close attention to detail which has continued to improve the quality of our work.

I do have a couple follow-up questions related to two of my initial major concerns. And given the extensive nature of the revisions, there are a number of relatively minor issues that warrant consideration and revision before publication.

Ref 3/1 While effect sizes are now available in the supplementary material, the reporting of effect sizes remains underemphasized in the main text. In particular, the main text revolving around Figure 4 still comes across as highly p-value centric. I believe that some indication of the Cohen’s D effect sizes should be worked into Figure 4, both for the individual MRI phenotypes (4A), as well as the CMD comparison (4C). If that’s simply not possible, some (quantitative) discussion of effect sizes for both of those should be included in the main text. Additionally, it should be made clear that effect sizes are available in the ST. (Currently, the main text mentions “effect-size” once in the introduction, but never seems to actually present or discuss effect sizes in the main text).

We agree with the reviewer that it is important to make the results accessible to readers in terms of effect sizes as well as *P*-values. Unfortunately, we have found that it is not possible to add this level of detail to Figure 4 without compromising its legibility and accessibility. We have therefore instead followed the reviewer’s recommendation to provide quantitative information about the effect sizes in the main text and more explicitly refer to the supplemental tables which provide complete information on effect sizes.

<<The following changes were made to the main text>>

In: Individualised centile scores in clinical samples

...with effect-sizes ranging from medium (Cohen’s *d* ranging from 0.2 to 0.8) to large (Cohen’s *d* > 0.8) (see **ST3-4** for all false discovery rate (FDR)-corrected *P*-values and effect-sizes).

(median centile score = 14%, 36% points difference from CN median, corresponding to Cohen’s *d*=0.88; **Fig.4A**).

Ref 3/2 The authors state in their response that due to the use of a “GAMLSS model that included random effects on three moments of the generalized gamma distribution” that it therefore wasn’t possible to “conduct an analysis of the effect of specific technical covariates”. I don’t follow the logic of this statement. In particular, aren’t estimates of the random effects available for each study? (Elsewhere in the manuscript makes the explicit point that it was important that these be available). And if that’s the case, then isn’t it possible to investigate possible relationships between those random effects estimates and the technical covariates?

The reviewer is correct that the GAMLSS framework provides us with study specific offsets on the mean (μ) and variance (σ) parameters. **Section SI 3.2.3** showed the variability of these estimates across the different studies and noted that we find wider confidence intervals for small or methodologically unique studies. In response to the reviewer’s suggestion, we now report additional new analysis of the relationships between these study-specific offset parameters and 5 demographic or technical covariates that varied between primary studies: median age (in years), age standard deviation, study size, scanner manufacturer, and scanner field strength. We found only limited evidence that any of these covariates significantly influenced random effect parameters. Information on other technical covariates, such as MRI acquisition parameters, is listed in full in **ST 1.1**; but these detailed technical factors were too heterogeneous between studies to be analysed formally in relation to study-specific random effects, i.e., there were almost as many unique combinations of MRI sequences as there were primary studies in the reference dataset. The new results for the effects of analysable covariates on random effect parameters estimated by GAMLSS modelling of cerebrum tissue volumes are detailed in **SI 3.2.3 “Parameter estimates”**, including new **Figs S3.2.3.2 - S3.2.3.6**; analogous results for random effect parameters estimated by GAMLSS modelling of extended global MRI phenotypes are detailed in **SI 7.2 “Normative trajectories of extended global MRI phenotypes”**, including new **Figs S7.2.3-7.21.7**

<<The following changes were made to the supplementary information>>

In SI 3.2.3: Parameter estimates

We further evaluated the potential impact of various technical and demographic covariates on the random effect parameters estimated by GAMLSS as a measure of each primary study’s offset from the normative trajectories of each MRI phenotype. Specifically, we used linear models to estimate the strength of association between random effects (on μ and σ) and median age, standard deviation of age, sample size, scanner manufacturer, and MRI field strength, for each cerebrum tissue volume; see **Figs S3.2.3.2 - S3.2.3.6**. We found only limited evidence for significant effects of any of these covariates on any of these random effect parameters. Other technical covariates, e.g., MRI sequence parameters, were too heterogeneous between primary studies to be assessed for impact on random effects in this way; but full technical specification of all primary studies is detailed in **ST 1.1**.

Fig. S3.2.3.2. Association between median age of participants and random effect parameters estimated by GAMLSS modeling of cerebrum tissue volumes for each primary study. Top row: random effects on Mu (y-axis) are plotted versus median age (x-axis) for each global MRI phenotype, left to right: grey matter volume (GMV), white matter volume (WMV), subcortical grey matter volume (sGMV) and ventricular CSF volume (Ventricles). Fitted lines and confidence intervals indicate the strength of association estimated by linear modeling. Bottom row: random effects on Sigma (y-axis) are plotted versus median age for the same set of global MRI phenotypes (except Ventricular volume for which Sigma was not estimated). There were larger random effects on Mu and Sigma in some of the primary studies of younger participants, as expected by the greater technical and biological variability of studies in childhood. The association between random effects and median age was only significant on the Mu parameter (after FDR correction for multiple comparisons) for ventricular CSF volume ($P_{\text{fdr}} = 0.007$, $R^2 = 0.12$, $F_{(1,82)} = 12.9$).

Fig. S3.2.3.3. Association between the standard deviation of the age of participants and random effect parameters estimated by GAMLSS modeling of cerebrum tissue volumes for each primary study. Top row: random effects on Mu (y-axis) are plotted versus standard deviation of age (x-axis) for

each global MRI phenotype, left to right: grey matter volume (GMV), white matter volume (WMV), subcortical grey matter volume (sGMV) and ventricular CSF volume (Ventricles). Fitted lines and confidence intervals indicate the strength of association estimated by linear modeling. Bottom row: random effects on Sigma (y-axis) are plotted versus standard deviation of age for the same set of global MRI phenotypes (except Ventricles for which Sigma was not estimated). The association between random effects and standard deviation of age was not significant (after FDR correction for multiple comparisons) for any of these global MRI phenotypes.

Fig. S3.2.3.4. Association between sample size and random effect parameters estimated by GAMLSS modelling of cerebrum tissue volumes for each primary study. Top row: random effects on Mu (y-axis) are plotted versus sample size (x-axis) for each global MRI phenotype, left to right: grey matter volume (GMV), white matter volume (WMV), subcortical grey matter volume (sGMV) and ventricular CSF volume (Ventricles). Fitted lines and confidence intervals indicate the strength of association estimated by linear modeling. Bottom row: random effects on Sigma (y-axis) are plotted versus sample size for the same set of global MRI phenotypes (except Ventricles for which Sigma was not estimated). The association between random effects and sample size was not significant (after FDR correction for multiple comparisons) for any of these global MRI phenotypes.

Fig. S3.2.3.5. Association between the scanner manufacturer and random effect parameters estimated by GAMLSS modeling of cerebrum tissue volumes for each primary study. Top row: boxplots of μ (x-axis) are plotted for primary studies using scanners manufactured by General Electric (GE, red), Siemens (purple), Philips (green), or a mixture of different scanners (cyan), for each global MRI phenotype, left to right: grey matter volume (GMV), white matter volume (WMV), subcortical grey matter volume (sGMV) and ventricular CSF volume (Ventricles). Bottom row: boxplots of σ (x-axis) are plotted for primary studies stratified by scanner manufacturer (with the same colour coding) for the same set of global MRI phenotypes (except Ventricles for which σ was not estimated). There was no evidence for a significant difference in mean random effects of primary studies using different scanners (after FDR correction for multiple comparisons) for any of these global MRI phenotypes.

Fig. S3.2.3.6. Association between the scanner field strength and random effect parameters estimated by GAMLSS modeling of cerebrum tissue volumes for each primary study. Top row: boxplots of μ (x-axis) are plotted for primary studies using scanners at different field strengths for each global MRI phenotype, left to right: grey matter volume (GMV), white matter volume (WMV), subcortical grey matter volume (sGMV) and ventricular CSF volume (Ventricles). Bottom row: boxplots of σ (x-axis) are plotted for primary studies stratified by scanner field strength (with the same colour coding) for the same set of global MRI phenotypes (except Ventricles for which σ was not estimated). There was no evidence for a significant difference in mean random effects of primary studies using scanners operating at different field strengths (after FDR correction for multiple comparisons) for any of these global MRI phenotypes. Numbers denote the number of studies included at this field strength.

In SI 7.2: Normative trajectories of extended global MRI phenotypes

Analogous to the analyses reported in SI 3.2.3 for cerebrum tissue volumes, we also examined the linear relationships between study-specific random parameters estimated in the analysis of other global MRI phenotypes and 5 demographic or technical covariates: median age, standard deviation of age, sample size, scanner manufacturer, and scanner field strength; see **Figs. S7.2.3-7.2.7**. We found only limited evidence for significant effects of any of these covariates on any of these random effect parameters.

Fig. S7.2.3. Association between median age of participants and random effect parameters estimated by GAMLSS modeling of extended global MRI phenotypes for each primary study. Top row: random effects on μ (y-axis) are plotted versus median age (x-axis) for each global MRI phenotype, left to right: total cerebrum volume, total surface area, mean cortical thickness. Fitted lines and confidence intervals indicate the strength of association estimated by linear modeling. Bottom row: random effects on σ (y-axis) are plotted versus median age for the same set of global MRI phenotypes. The associations between random effects and median age were not significant for any of these global phenotypes.

Fig. S7.2.4. Association between the standard deviation of the age of participants and random effect parameters estimated by GAMLSS modeling of extended global MRI phenotypes for each primary study. Top row: random effects on μ (y-axis) are plotted versus standard deviation of age (x-axis) for each global MRI phenotype, left to right: total cerebrum volume, total surface area, mean cortical thickness. Fitted lines and confidence intervals indicate the strength of association estimated by linear modeling. Bottom row: random effects on σ (y-axis) are plotted versus standard deviation of age for the same set of global MRI phenotypes. The associations between random effects and standard deviation of age were not significant (after FDR correction for multiple comparisons) for any of these global MRI phenotypes.

Fig. S7.2.5. Association between sample size and random effect parameters estimated by GAMLSS modeling of extended global MRI phenotypes for each primary study. Top row: random effects on μ (y-axis) are plotted versus sample size (x-axis) for each global MRI phenotype, left to right: left to right: total cerebrum volume, total surface area, mean cortical thickness. Fitted lines and confidence intervals indicate the strength of association estimated by linear modeling. Bottom row: random effects on σ (y-axis) are plotted versus sample size for the same set of global MRI phenotypes. The associations between random effects and sample size were not significant (after FDR correction for multiple comparisons) for any of these global MRI phenotypes.

Fig. S7.2.6. Association between the scanner manufacturer and random effect parameters estimated by GAMLSS modeling of extended global MRI phenotypes for each primary study. Top row: boxplots of μ (x-axis) are plotted for primary studies using scanners manufactured by General Electric (GE, red), Siemens (purple), Philips (green), or a mixture of different scanners (cyan), for each global MRI phenotype, left to right: total cerebrum volume, total surface area, mean cortical thickness. Bottom row: boxplots of σ (x-axis) are plotted for primary studies stratified by scanner manufacturer (with the same colour coding) for the same set of global MRI phenotypes. There was no evidence for a significant difference in mean random effects of primary studies using different scanners (after FDR correction for multiple comparisons) for any of these global MRI phenotypes.

Fig. S7.2.7. Association between the scanner field strength and random effect parameters estimated by GAMLSS modeling of extended global MRI phenotypes for each primary study. Top row: boxplots of Mu (x-axis) are plotted for primary studies using scanners at different field strengths (1T, red; 1.5T, purple; 3T, green; or 7T, cyan) for each global MRI phenotype, left to right: total cerebrum volume, total surface area, mean cortical thickness. Bottom row: boxplots of Sigma (x-axis) are plotted for primary studies stratified by scanner field strength (with the same colour coding) for the same set of global MRI phenotypes. There was no evidence for a significant difference in mean random effects of primary studies using scanners operating at different field strengths (after FDR correction for multiple comparisons) for any of these global MRI phenotypes.

Ref 3/3 Fig. 2: There clearly is a discontinuity in the mean cortical thickness values just past age 2. Is this commented on anywhere? Somewhat relatedly, what is the combined N across studies of individuals less than 2 years old? In the interest of transparency, it seems worth pointing this out in the main text, as presumably this N is rather low, yet this data is the basis for a number of the conclusions regarding age at peak (thickness) and age at peak velocity (a number of measures). Also, neither key nor caption indicates what the gray line represents in panel B. Also, the caption says “2.5% and 97% centiles”. I presume that the latter value should be “97.5%”?

We thank the reviewer for picking up these issues in Figure 2 and the related text. We have corrected the legend to Figure 2 accordingly, we have discussed the discontinuity in the cortical thickness trajectory, and we have now clearly stated the number of subjects under 2 years old.

<<The following changes were made to the main text>>

In Developmental milestones

This early peak in cortical thickness velocity has not been reported previously, in part due to obstacles in acquiring adequate and consistent signal from typical MRI sequences in the perinatal period⁶⁰. Similarly, normative trajectories revealed an early period of GMV:WMV differentiation, beginning in the first month after birth with the switch from WMV to GMV as the proportionally dominant tissue compartment, and ending when the absolute difference of GMV and WMV peaked around 3 years (S19). This epoch of GMV:WMV differentiation, which may reflect underlying changes in myelination and synaptic proliferation^{51,61–64}, has not been demarcated by prior studies^{48,65}. It was likely identified in this study due to the substantial amount of early developmental MRI data available for analysis in the aggregated dataset (in total across all primary studies, N=2,571 and N=1,484 participants aged <2 years were available for analysis

of cerebrum tissue volumes and extended global MRI phenotypes, respectively). The period of GMV:WMV differentiation encompasses dynamic changes in brain metabolites⁶⁶ (0-3 months), resting metabolic rate (RMR; minimum=7 months, maximum=4.2 years)⁶⁷, the typical period of acquisition of motor capabilities and other early paediatric milestones⁶⁸, and the most rapid change in TCV (**Fig.3**).

<<The following changes were made to the supplementary material>>

In SI 7.1 Extended neuroimaging phenotypes

We note the discontinuity in the raw, non-centiled CT data that is most likely driven by a combination of brain tissue-related changes in the MRI signal due to the ending of the phase of GMV:WMV differentiation (and peak velocity in WMV) and technical differences in pre-processing of images around this age. Two-three years is often used as the cutoff to determine application of different pre-processing pipelines, e.g., infant FreeSurfer versus adult FreeSurfer. However, multiple studies and associated pre-processing pipelines were extended across this age-range in our aggregated dataset (and we also observed this discontinuity within studies that were processed homogeneously across the 2-3 year span), hence we consider it is unlikely that the transition between pre-processing pipelines can entirely explain the discontinuity in CT. We also emphasise that while there is a relative paucity of data in the <2yr age-range compared to other epochs, and for extended global MRI phenotypes (including CT) compared to cerebrum tissue volumes, the current trajectories are based on the largest early developmental dataset reported to date. Future work will hopefully be able to more precisely disambiguate to what extent this discontinuity is driven by MRI signal changes representative of maturation of brain tissue composition and to what extent the discontinuity represents technical factors such as adoption of infancy-specific pre-processing pipelines.

Ref 3/4 Figure 3: The colors are not sufficiently distinct to quickly and easily match to the key. Either more unique colors are necessary, or add thin black lines within the color bands, but with different linestyles (e.g., solid vs. dashed) to help distinguish lines that are the same basic color. Also, the top inset reads “thicknes” rather than “thickness”.

We thank the reviewer for picking up these issues in Figure 3. The typo in the top inset has been corrected. To maintain alignment with the colour scale used consistently across other figures, we have adopted the reviewer's suggestion of adding a different line-style for the extended global MRI phenotypes in this figure.

<<The following changes were made to the main text>>

Ref 3/5 p. 8 (main text): Shouldn't text regarding "mean difference in h^2 " be referring to Fig.4D (not 4C)? Also, does that "11.8%" refer to an actual *percentage* difference, or a *percentage point* difference? (Figure 4 seems to indicate it is probably the latter, which is more impressive than the former, and thus worth describing precisely).

We thank the reviewer for picking up this point. We have now corrected this error in the reference to Figure 4 in the main text, and we have clarified the reporting of this heritability result. :

<<The following changes were made to the main text>>

In: Individualised centile scores in clinical samples

...(average increase of 11.8% points in h^2 across phenotypes; **Fig.4D, SI13**)...

Ref 3/6 Figure 4: Caption contains no indication of what 1 vs. 2 vs. 3 asterisks represents.

We thank the reviewer for picking up this point. The legend to Figure 4 has been updated accordingly.

<<The following changes were made to the main text>>

In: Figure 4 legend

Asterisks indicate level of FDR-corrected significance: $P < 0.05$, $P < 0.01$ or $P < 0.001$ for *, ** and *** respectively.

Ref 3/7 p. 18 (“Online Methods”): It seems less relevant to me to present the “general” form of the GAMLSS than to make clear how the GAMLSS was modeled in this particular study. In that regard, I find it potentially confusing to readers to include smoothing functions in Eq. (1) when non-parametric smoothing functions were not in fact used. Also, in Eq. (1), what is ‘F’? (Not defined).

We included smoothing terms in description of the model simply to acknowledge the GAM component of GAMLSS. The reviewer is correct in noting that we do not employ any smoothing functions in our implementation of GAMLSS modeling. The online methods and accompanying supplementary methods were intended as a general discussion of centile normalisation. The specifics of our model are now presented in the online methods after this general discussion. F refers to the distribution family, which was incorrectly denoted D in the text; this typo has been corrected.

<<The following changes were made to the online methods>>

The resulting models were evaluated using several sensitivity analyses and validation approaches. See **Supplementary Information [SI]** for further details regarding GAMLSS model specification and estimation (**SI1**), image quality control (**SI2**), model stability and robustness (**SI3-4**), phenotypic validation against non-imaging metrics (**SI3 & SI5.2**), between-study harmonisation (**SI5**) and assessment of cohort effects (**SI6**). While the models of whole brain and regional morphometric development were robust to variations in image quality, and cross-validated by non-imaging metrics, we expect that several sources of variance, including but not limited to MRI data quality and variability of acquisition protocols, may become increasingly important as brain charting methods are applied to more innovative and/or anatomically fine-grained MRI phenotypes. It will be important for future work to remain vigilant about the potential impact of data quality and other sources of noise on robustness and generalisability of both normative trajectories and the centile scores derived from them.

Based on the model selection criteria, outlined in the Supplementary Information (**SI 1**) in detail, the final models for normative trajectories of all MRI phenotypes were specified as illustrated below for GMV:

$$\begin{aligned} & \text{GMV} \sim \text{Generalised Gamma}(\mu, \sigma, \nu) \text{ with} & (2.1) \\ \log(\mu) &= \alpha_{\mu} + \alpha_{\mu,sex}(sex) + \alpha_{\mu,ver}(ver) + \beta_{\mu,1}(age)^{-2} + \beta_{\mu,2}(age)^{-2} \log(age) + \beta_{\mu,3}(age)^{-2} \log(age)^2 \\ & + \gamma_{\mu,study} \\ \log(\sigma) &= \alpha_{\sigma} + \alpha_{\sigma,sex}(sex) + \beta_{\sigma,1}(age)^{-2} + \beta_{\sigma,2}(age)^3 + \gamma_{\sigma,study} \\ \nu &= \alpha_{\nu}. \end{aligned}$$

For each component of the generalised gamma distribution, α terms correspond to fixed effects of the intercept, sex (female/male), and software version (five categories); β terms correspond to the fixed effects of age, modeled as fractional polynomial functions with the number of terms reflecting the order of the fractional polynomials; and γ terms correspond to

the study-level random-effects. Note that we have explicitly included the link-functions for each component of the generalised gamma, namely the natural logarithm for μ and σ (since these parameters must be positive) and the identity for ν .

Similarly for the other phenotypes:

WMV ~ Generalised Gamma(μ, σ, ν) with (2.2)

$$\begin{aligned} \log(\mu) &= \alpha_{\mu} + \alpha_{\mu,sex}(sex) + \alpha_{\mu,ver}(ver) + \beta_{\mu,1}(age)^{-2} + \beta_{\mu,2}(age)^3 + \beta_{\mu,3}(age)^3 \log(age) + \gamma_{\mu,study} \\ \log(\sigma) &= \alpha_{\sigma} + \alpha_{\sigma,sex}(sex) + \beta_{\sigma,1}(age)^{-2} + \beta_{\sigma,2}(age)^3 + \gamma_{\sigma,study} \\ \nu &= \alpha_{\nu}, \end{aligned}$$

sGMV ~ Generalised Gamma(μ, σ, ν) with (2.3)

$$\begin{aligned} \log(\mu) &= \alpha_{\mu} + \alpha_{\mu,sex}(sex) + \alpha_{\mu,ver}(ver) + \beta_{\mu,1}(age)^{-2} + \beta_{\mu,2}(age)^{-2} \log(age) + \beta_{\mu,3}(age)^3 + \gamma_{\mu,study} \\ \log(\sigma) &= \alpha_{\sigma} + \alpha_{\sigma,sex}(sex) + \beta_{\sigma,1}(age)^{-2} + \beta_{\sigma,2}(age)^{-2} \log(age) + \gamma_{\sigma,study} \\ \nu &= \alpha_{\nu}, \end{aligned}$$

Ventricles ~ Generalised Gamma(μ, σ, ν) with (2.4)

$$\begin{aligned} \log(\mu) &= \alpha_{\mu} + \alpha_{\mu,sex}(sex) + \alpha_{\mu,ver}(ver) + \beta_{\mu,1}(age)^3 + \beta_{\mu,2}(age)^3 \log(age) + \beta_{\mu,3}(age)^3 \log(age)^2 \\ &\quad + \gamma_{\mu,study} \\ \log(\sigma) &= \alpha_{\sigma} + \alpha_{\sigma,sex}(sex) + \beta_{\sigma,1}(age)^{-2} + \beta_{\sigma,2}(age)^{-2} \log(age) + \beta_{\sigma,3}(age)^{-2} \log(age)^2 \\ \nu &= \alpha_{\nu}, \end{aligned}$$

TCV ~ Generalised Gamma(μ, σ, ν) with (2.5)

$$\begin{aligned} \log(\mu) &= \alpha_{\mu} + \alpha_{\mu,sex}(sex) + \alpha_{\mu,ver}(ver) + \beta_{\mu,1}(age)^{-2} + \beta_{\mu,2}(age)^{-2} \log(age) + \beta_{\mu,3}(age)^3 + \gamma_{\mu,study} \\ \log(\sigma) &= \alpha_{\sigma} + \alpha_{\sigma,sex}(sex) + \beta_{\sigma,1}(age)^{-2} + \beta_{\sigma,2}(age)^{-2} \log(age) + \beta_{\sigma,3}(age)^{-2} \log(age)^2 + \gamma_{\sigma,study} \\ \nu &= \alpha_{\nu}, \end{aligned}$$

SA ~ Generalised Gamma(μ, σ, ν) with (2.6)

$$\begin{aligned} \log(\mu) &= \alpha_{\mu} + \alpha_{\mu,sex}(sex) + \alpha_{\mu,ver}(ver) + \beta_{\mu,1}(age)^{-2} + \beta_{\mu,2}(age)^{-2} \log(age) + \beta_{\mu,3}(age)^{-2} \log(age)^2 \\ &\quad + \gamma_{\mu,study} \\ \log(\sigma) &= \alpha_{\sigma} + \alpha_{\sigma,sex}(sex) + \beta_{\sigma,1}(age)^{-2} + \beta_{\sigma,2}(age)^{-2} \log(age) + \beta_{\sigma,3}(age)^{-2} \log(age)^2 + \gamma_{\sigma,study} \\ \nu &= \alpha_{\nu}, \end{aligned}$$

CT ~ Generalised Gamma(μ, σ, ν) with (2.7)

$$\begin{aligned} \log(\mu) &= \alpha_{\mu} + \alpha_{\mu,sex}(sex) + \alpha_{\mu,ver}(ver) + \beta_{\mu,1}(age)^{-2} + \beta_{\mu,2}(age)^{-2} \log(age) + \gamma_{\mu,study} \\ \log(\sigma) &= \alpha_{\sigma} + \alpha_{\sigma,sex}(sex) + \beta_{\sigma,1}(age)^{-1} + \beta_{\sigma,2}(age)^{0.5} + \gamma_{\sigma,study} \\ \nu &= \alpha_{\nu}. \end{aligned}$$

No smoothing terms were used in any GAMLSS models implemented in this study, although the fractional polynomials can be regarded as effectively a parametric form of smoothing.

Reliably estimating higher order moments requires increasing amounts of data, hence none of our models specified any fixed- nor random-effects in the ν term. However, α_ν was found to be important in terms of model fit and hence we have used a generalised gamma distribution.

Ref 3/8 S1.7: I still don't understand the justification for computing IQR when you only have two data points. In what sense is IQR "valid" when you have just 2 data points (any more so than a trimmed mean from just 2 data points)? Just because you can select an implementation of IQR that is *defined* for 2 data points doesn't make it a measure with a meaningful interpretation.

The reviewer is correct that any summary statistic of two data points (including IQR) does not provide additional information beyond, for example, a trimmed mean. However, our purpose was to compare the stability of centile scores across individuals with a varying number of longitudinal observations. For this purpose, it was relevant that the IQR is well defined for 2 or more data points, and more importantly coherent across the variable number of longitudinal data points available for different participants. Hence, we considered the IQR an appropriate and consistent metric to explore the stability of centiles across longitudinal measurement with varying numbers of data-points. We note that caveat more explicitly in the description of the IQR.

<<The following changes were made to the supplementary information>>

In SI 1.7 Longitudinal centiles

However, unlike the trimmed mean which requires a "large enough" sample, the IQR is valid for small samples. Given the variable number of longitudinal data-points available for different participants, we chose to use a measure that was consistent for participants that only had 2 observations as well as for participants with more than 2 observations. Unfortunately, there is not a single definition of the IQR (there are 9 different definitions available within GNU R), and some versions are not defined for two observations. We estimated IQR as a continuous value by linear interpolation (within GNU R the default version of IQR, type 7), which is well defined for two (or more) observations.

Ref 3/9 S2.1, Euler Index Filtering: Rosen et al. used the strict definition of the Euler Number, which is increasingly negative as the number of surface holes in the non-topology corrected (orig.nofix) surface increases. If you are using $EI > 217$ as a threshold, you are taking the negative of the true Euler Number, which should be stated. Also, please be explicit as to whether you are averaging or summing the EI from the two hemispheres. [Rosen states that they averaged]. Also, it would be beneficial to quantify how many scans were excluded by your filtering operation. Last, it should be noted that $EI > 217$ is a threshold that has only been shown to be an "optimal" threshold in one study, and it remains unknown whether that threshold is optimal in any sense as a QC criterion for data collected on other scanners or using other protocols.

We thank the reviewer for pointing out the omission of these crucial details concerning quality control by thresholding on the Euler index. As noted in our response to Ref 1/6, we have substantially revised the paper to eliminate the results of QC by a fixed, universal threshold of $EI > 217$; instead we now report the results of QC by an adaptive, study-specific threshold of

EI > 2 MAD (median absolute deviation from the median in each primary study). To avoid possible confusion, we no longer refer to the threshold of EI > 217 in the paper and therefore these specific requests for clarification are no longer relevant. We have additionally added an explicit definition of the EI as used in this study to avoid any confusion with how it may be defined in other literature.

<<The following changes were made to the supplementary information>>

In SI 2.1 Euler Index filtering

The EI metric we used was defined as the sum across hemispheres of the number of surface 'holes' or topological defects in the cortical surface reconstruction prior to a topological correction performed as part of the FreeSurfer pipeline (usually due to errors in white matter segmentation). Although cerebrum tissue volumes are expected to be less sensitive to cortical surface topology, compared to surface-based measures such as indices of cortical folding (see **SI18 "Data processing"**), EI has previously been used as a measure of the quality of "raw", unprocessed scans⁴⁵. Thus for the large majority of studies where EI was available (N=101,708 total scans on N=82,023 unique subjects), we assessed the impact on reference models of excluding high-magnitude EI scans. Given that no single EI threshold is expected to be generalizable across studies⁴⁵ (**Fig. S2.1.2**), in this sensitivity analysis we excluded scans that had EI magnitude greater than 2 median absolute deviations from the primary study-specific median EI. This QC threshold, which is adaptive to the variable quality of scans between primary studies, excluded approximately 9-10% of scans from the original dataset. However, as can be seen in **Fig. S2.1.3**, the resulting model parameters were highly correlated with parameters estimated from the reference dataset without applying any EI-based QC threshold. The developmental trajectories estimated for all 4 cerebrum tissue volumes were highly correlated with their trajectories estimated on the basis of the full dataset (all $R^2 > 0.999$ for parametric [Pearson's] and non-parametric [Spearman's] correlations between EI-filtered versus EI-unfiltered median trajectories and lower (2.5%) and upper (97.5%) centiles). Identical parameterisation of fractional polynomials for each random effect was identified by the same model selection procedure was found in both EI-filtered and EI-unfiltered datasets. Importantly, EI-filtered and unfiltered datasets also showed a high degree of overlap in subsequently estimated model parameters (correlation of study-specific mean (μ) components > 0.99; correlation of study-specific variance (σ) components > 0.93). Model specification thus appeared to be robust to the presence of the poorer quality data.

In addition, we examined the relationships between image quality measured by EI and individual centile scores of each brain phenotype. Both for the full dataset and the EI-filtered subset of higher quality scans, we found no significant associations between EI and individual centile scores (**Fig. S2.1.1**), nor did we find evidence for a non-linear relationship (quadratic, cubic, logarithmic) between EI and centiles.

Fig. S2.1.1 Associations between centile scores and MRI scan quality defined by EI. Panel depicts the relation between Euler indices (EI)⁴⁵ and centile scores for each of 4 cerebrum tissue volumes estimated by GAMLSS. The Spearman correlations between EI and centile scores were negligible (GMV, $\rho < 0.01$; WMV, $\rho = -0.07$; sGMV, $\rho < 0.01$; Ventricles, $\rho = 0.05$). All linear mixed effect models examining non-linear (quadratic, cubic or logarithmic) relationships between EI and centile scores for each phenotype were $P > 0.1$.

To assess whether there were any age-related differences that could influence model estimation, we evaluated the linear effect of age (in years) on EI in healthy controls in the reference dataset used to estimate normative lifespan trajectories. Using linear regression stratified by sex and accounting for study-specific random effects, we found no evidence for an age-related bias in image quality as assessed with EI ($t = -1.244$, $P = 0.213$). **Fig. S2.1.2** shows the median and standard deviation of age and EI and highlights the top 10 studies with the highest median EI.

Fig. S2.1.2 Age-related variation in image quality measured by the Euler index in female (left panel) and male (right panel) control subjects. Median age (in years) and median EI are shown per study with cross-hairs indicating the standard deviations for age and EI per study. In red the top ten studies with the highest median EI are highlighted. There is no significant relationship between image quality and age at scanning.

Figure S2.1.3 Robustness of GAMLSS parameters to quality control by exclusion of scans with EI greater than twice the median absolute deviation (MAD) from the median EI in the corresponding primary study. Scatterplots show the relationships between random effects (μ on the top row and σ on the bottom row) estimated for each primary study without exclusion of poor quality scans (y-axis) and for each primary study after exclusion of scans with $EI > 2 \text{ MAD}$, relative to the primary study's median EI. Coloured points indicate the relative percentage of primary studies retained after filtering (darker means for subjects were removed) and Rho values in the titles indicate Spearman's correlations between parameters estimated from the unfiltered and EI-filtered datasets. As with the absolute QC threshold of $EI < 217$ (SI 2.1), the biggest discrepancy in study-specific random effects as a result of excluding poor quality scans was observed for the variance (Sigma) parameters, especially those estimated from the ICBM, HBN and EDSB datasets, which all included a relatively high proportion of excluded scans. We note that $EI > 2 \text{ MAD}$ filtering removed a lower proportion of data in primary studies where the distribution of EI was skewed towards higher quality/lower EI across the whole dataset (e.g., HCP, ABCD and UKB all have high data quality with low EI, and 2 MAD filtering in these studies only removed around 6-7% of data). In general, random effect parameter estimation was highly robust to adaptive EI thresholding for quality control.

Ref 3/10 S2.2, Expert visual quality control: Was this a de novo visual quality control performed by the authors, separate from any visual QC and rating that the individual studies might have already performed and provided? Also, it should be made clearer whether the 9704 scans that were rated were drawn from all studies equally, or whether particular studies dominated this endeavor.

The expert visual quality control was done by one of the collaborators / co-authors and was not conducted de novo. The images available for visual QC were drawn from open cohorts and are listed as having "Manual" quality control in the "Extracted QC" column in supplementary table ST1.1.

<<The following changes were made to the supplementary information>>

In SI 2.2 Expert visual quality control

These scans were provided by openly available datasets and are marked as having “Manual” quality control in the “Extracted QC” column in **ST1.1**. For each subject a slice stack of images was generated across the three axes, after bias field correction and intensity normalization, so that they were all easily comparable by visual inspection, and subsequently rated on motion corruption and other failure modes (artefacts, missing brain parts etc). Visual inspection then rated each image on the following questions: is the brain fully covered by the scan; is there visible noise (due to aliasing, motion etc.), blurriness, or ringing; is there acceptable tissue contrast and image orientation? Based on these criteria, each raw scan was expertly classified on a 6-point scale as perfect (1), very good (2), good (3), bad (4), very bad (5) or unacceptable (6). Only 3% of scans (N=374) were assigned to the two worst quality categories (5 and 6). Each image was rated by a single rater.

Ref 3/11 S3.2.1: For consistency with changes elsewhere, it should say that quantitative comparison of model *BIC* (rather than AIC) wasn't possible.

We thank the reviewer for pointing out this inconsistency, which has now been corrected.

<<The following changes were made to the supplementary information>>

Given that these models were each derived from different datasets it was not possible to conduct a quantitative comparison of the models in terms of their Bayesian Information Criteria⁶⁹ as we did when evaluating the optimal underlying distribution.

Ref 3/12 Figs. S3.2.1 and S3.2.2: Are these 95% confidence intervals? Relatedly, what exactly do the “uncertainty intervals” in Fis. S4.5 represent? Given the wide variety of different types of “error bars” used in various figures (e.g., SD, standard errors, 95% confidence intervals, box plot whiskers), please make sure that it is clear in each figure caption exactly what the error bars in that particular figure represent.

We can confirm that, in both these figures, the confidence intervals represent the 95% confidence intervals across each iteration of a single left-out study. This information has been clarified in the figure caption.

<<The following changes were made to the supplementary information>>

Fig. S3.2.1. Leave-one-study-out (LOSO) analyses of normative trajectories for cerebrum tissue volumes. A | Confidence intervals (representing the 95% confidence intervals) were computed from the mean and standard deviation of normative trajectories repeatedly estimated after leaving out each primary study in turn: from left to right, grey matter volume (GMV), white matter volume (WMV), subcortical grey matter volume (sGMV) and ventricular CSF volume (Ventricles). B | The same data are shown with the 95% confidence intervals magnified by a factor of 50 to enhance their visibility.

Fig. S3.2.2. Bootstrap resampling of confidence intervals on normative trajectories for cerebrum tissue volumes. A | 95% confidence intervals (estimated across random bootstrap iterations resampling with replacement) were computed from the mean and standard deviation of normative trajectories (with age on log scale, x-axis) after 1000 iterations of a bootstrapping procedure designed to conserve the relative proportion of primary studies, and the sex balance of each primary study, in each resampling with replacement from the representative dataset: from left to right, grey matter volume

(GMV), white matter volume (WMV), subcortical grey matter volume (sGMV) and ventricular CSF volume (Ventricles). B | The same data are shown with age on a natural scale (x-axis).

Ref 3/13 Fig. S3.2.3: It would be helpful if the caption stated the reason that no estimates for Sigma for Ventricular volume are available.

The data-driven process of GAMLSS model specification (detailed in **SI 1**) indicated that the best fitting model for the Ventricular volume phenotype did not include a study random effect on the Sigma term. This has now been added to the caption and included in the full model description in the Online Methods (see response to Ref 3/7).

<<The following changes were made to the supplementary information>>

Fig. S3.2.3.1 Point-range plots of study-specific random effects on the first (Mu) and second (Sigma) moments of the generalised gamma distribution for cerebrum tissue volumes and study-specific random effects on Mu only for ventricular CSF volume. Bootstrapped 95% confidence intervals are shown and point estimates (dots) are coloured by the range of the confidence interval. Where not observable, the confidence intervals are smaller than the size of the dots. There is no Sigma offset for the Ventricular volume as the data-driven process for GAMLSS model specification (SI 1) indicated that the best-fitting model did not include a study-specific random effect on the Sigma term.

Ref 3/14 S3.2.2: Insufficient detail is provided to understand how the stratified bootstrap sample was implemented. It would be helpful to have a methodical example of how exactly one bootstrap replicate was constructed.

The bootstrap stratification is by sex within study, such that each bootstrap replicate maintains the same female/male ratio as per the observed primary study.

<<The following changes were made to the supplementary information>>

In SI 3.2.2 Bootstrap analysis

Specifically, our process of random resampling of aggregated data was constrained by the relative size of each study compared to other primary studies, and by the sex ratio of each primary study, so that the bootstrap replicates conserved the same proportionality and sex balance as the observed primary studies.

Ref 3/15 S4.1: Bias cannot be assessed via correlation. Some true measure of bias should be reported (in addition to the correlations).

We agree with this point. We have now included a more direct measure of bias, namely the difference between the out-of-sample and in-sample estimates of centile scores on global MRI phenotypes.

<<The following changes were made to the supplementary information>>

In SI 4.1 Bias of out-of-sample centile scores: leave-one-study-out analyses for 100 studies

In addition to demonstrating high correlations between OoS and in-sample centile scores, we also evaluated their relative bias, defined as the difference between in-sample estimated

centiles and OoS estimated centiles. The median bias in centile scores was generally low (GMV = $-1.7e-06$; WMV = $1.1e-04$; sGMV = $3.8e-05$; Ventricles = $-7.3e-05$, all with a standard deviation of ~ 0.01 centile). However, it is worth noting that the studies characterised by relatively small sample size, foetal or early postnatal age-range of participants, or idiosyncratic processing pipelines, appeared at the extreme ends of the distributions of the primary studies rank-ordered by the difference between in-sample and OoS centile scores (Fig.S4.1.1), indicating greater bias of OoS centile scoring, as expected, under these conditions.

Fig. S4.1.1. Bias of out-of-sample centile scores for four cerebrum tissue volumes. Each panel shows boxplots of the bias in OoS centile scores (the signed difference between OoS and in-sample centile scores; y-axis) estimated for each primary study when it was excluded from the reference dataset. Studies are ordered on the x-axis from most negatively biased (left) to most positively biased (right) OoS centile scores. Boxplots are colour-coded according to log sample size, indicating that OoS centile scores tend to be most biased for smaller primary studies. From top to bottom, panels represent the bias in OoS centile scores for grey matter volume, white matter volume, subcortical grey matter volume, and ventricular CSF volume.

In SI 7.4 Stability of out of-sample centile scoring for extended global phenotypes: LOSO analyses

Analogous to our assessment of bias in centile scores of cerebrum tissue volumes in **SI 4.1**, we also assessed bias of centile scores of extended global MRI phenotypes, i.e., the difference between OoS-estimated and in-sample estimated centiles. Bias was generally very low except for a few studies (i.e., CHILD, NIHPD, FinnBrain) with smaller sample size or younger participants (**Fig. S7.4.3**).

Fig. S7.4.3. Bias in out-of-sample estimates of centile scores on extended global MRI phenotypes. The distribution of difference scores between in-sample estimated centiles and OoS estimated samples are shown for all included studies and ordered by the median bias.

Ref 3/16 S4.3: It seems odd to not include the ICC values of the “raw” (uncentiled) volumetric data for the analyses of the reproducibility resource and HBN data, for explicit comparison to the centile ICCs. (They are provided for the VETSA analysis).

We have now reported ICCs for the raw cerebral tissue volumes, alongside the parallel analysis of centile scores on these global phenotypes.

<<the following changes have been made to the supplementary information>>

Fig. S4.3.1. Test-retest reliability of out-of-sample centile scores for cerebrum tissue volumes. MRI data were collected in two separate scanning sessions from N=21 participants and each session was analysed as an independent out-of-sample study using GAMLSS. Scatterplots represent OoS centile scores for session 1 (y-axis) versus OoS centile scores for session 2 (x-axis) for each brain tissue volume, from left to right: GMV, WMV, sGMV, Ventricular CSF. Data points represent individual subject centile scores. Test-retest reliability was consistently very high (all ICCs > 0.99) for all cerebrum tissue volumes. The bottom panel shows the identical analysis for non-centiled, “raw” volumetric data. P-values represent the significance of the intraclass correlation coefficient between two sessions.

HCP Acquisition - raw

HCP Acquisition - centile

Fig. S4.3.2. Test-retest reliability of out-of-sample centile scores for cerebrum tissue volumes measured twice in the same N=72 participants using two T1-weighted sequences, MPRAGE and VNaV. For each type of acquisition the top row shows out-of-sample centile scores for session 1 (y-axis) versus out-of-sample centile scores for session 2 (x-axis) for cerebrum tissue volumes. For each type of acquisition the bottom row shows the unprocessed (“raw”) scores for session 1 (y-axis) versus session 2 (x-axis) for cerebrum tissue volumes estimated from VNaV data, from left to right: GMV, WMV, sGMV, Ventricles. In all plots, data points represent individual subject scores. Test-retest reliability was uniformly high (all ICCs > 0.95) and generally somewhat higher for volumetrics derived from prospectively motion-corrected data (VNaV). P-values represent the significance of the intraclass correlation coefficient between two sessions. HCP session refers to the Human Connectome Project MPRAGE acquisition used.

Fig. S4.3.3. Test-retest reliability of out-of-sample centile scores for cerebrum tissue volumes measured twice in the same 1,200 participants (600 twin pairs). The top row shows scatterplots of unprocessed (“raw”) volumes for scan 1 (y-axis) versus scan 2 (x-axis) for cerebrum tissue volumes estimated from MPRAGE data from the same subject, from left to right: GMV, WMV, sGMV, Ventricles. Data points represent individual subject centile scores. The bottom row shows the consistency of centile

scores for the same subjects and same phenotypes. Reliability was uniformly high across all phenotypes (ICCs > 0.95) and comparable to reliability of uncentiled volumetric measurements from the same set of scans. P-values represent the significance of the intraclass correlation coefficient between two sessions.

Ref 3/17 Figs. S4.3.[1,2]: What do the p-values in the titles of the plots represent? Are they the p-value of the regression/scatterplot? Or the p-value of the test of the null hypothesis that ICC = 0 ?

P-values represent the significance of the ICC. This has now been clarified in each figure caption (see response to Ref 3/16).

Ref 3/18 Fig. S4.3.2: What does “HCP Session” in the top row of plots refer to?

HCP session refers to the “human connectome project acquisition” as this was the other type of acquisition used in this test-retest dataset. This is now clarified in the figure caption and description (see response to Ref 3/16).

Ref 3/19 S4.4: It would be helpful if the “name” of the study used for this analysis was included in the text (for convenient reference against other data that shows results per study).

Thank you for pointing out this omission. This is the “NIH” cohort included throughout and has now been cross-referenced explicitly.

<<the following changes were made to the supplementary information>>

To do this we re-analysed a single NIH dataset⁷⁰ (see **SI 19** “NIH” for a fuller description) repeatedly using 4 different versions of FreeSurfer (5.1, 5.3, 6.01, and 7.1).

Ref 3/20 S4.5: It isn't clear how dataset ‘clones’ were created (e.g., was resampling with replacement used *within study*), nor how many datasets were ‘cloned’. Also, what does NSPN represent in the titles of Fig. S4.5?

We apologise for any lack of clarity in this section and have now added more detail to the description of the study cloning procedure. Clones were indeed created within a single study, resampling *without* replacement. We report the results of this procedure for the Neuroscience in Psychiatry Network (NSPN) cohort as an additional sensitivity check for out-of-sample estimation and to estimate at what sample size the out-of-sample estimation approaches the true study offset derived from the original full cohort. Given the computational burden of this process and the complementary nature of this analysis to the already presented out-of-sample stability, we only generated clones for 2 subsets (NSPN and ADNI). The figure legend for S4.5 has been updated to include the full name of the dataset for clarity.

<<the following changes were made to the supplementary information>>

In **SI 4.5:Effects of sample size on reliability of out-of-sample centile scores**

To further assess the validity of the OoS estimates, we generated “clones” of existing datasets. Clones were resampled subsets (without replacement, no duplicate subjects per clone) of

studies included in the reference dataset used to estimate the study specific GAMLSS parameters. Each clone was then treated as if it was a “new” study using the methods for out-of-sample centile scoring.

In: Legend to Fig S5.2.5

NSPN refers to the Neuroscience in Psychiatry Network study included in the reference dataset.

Ref 3/21 Fig. S5.2.5: Seems like the point being made by this figure is incomplete without a panel showing the equivalent analysis applied to the raw data.

We have now included a panel in this figure to show the effects using the non-centiled, “raw”, volumetrics, and modified some text in the SI to reflect the initial motivation of this analysis – which was to replicate previous associations between raw brain volumetrics and either birth weight or fluid intelligence.

In SI 5.2:

To further assess whether batch-corrected MRI data derived from both ComBAT and GAMLSS pipelines would generate convergent results in subsequent analyses, we estimated the correlations between total cerebrum volume (TCV) and fluid intelligence or birth weight, after TCV had been batch-corrected by either GAMLSS or ComBAT. Both these psychological and biological factors have previously been shown to be correlated with similar brain volumetrics^{71–73}. We were able to replicate these significant associations with uncorrected TCV, as well as after both GAMLSS and ComBAT batch correction, all largely showing consistent effects across sites (Fig. S5.4-5.5).

Fig. S5.2.5. Consistency of behavioural (fluid intelligence) and biological (birth weight) associations with total cerebrum volume (TCV) estimated at 22 MRI acquisition sites in the ABCD cohort, either without batch correction (left) or after batch correction for site effects by GAMLSS (middle) or ComBAT (right). Regression coefficients and standard errors from linear regression models of TCV on birth weight or fluid intelligence are plotted using point-ranges for each site. Meta-analytic coefficients and errors, combining all primary study effects, are shown in black at the top of each column. Coefficients (triangles) are scaled based on sample size at each site within the ABCD study.

Ref 3/22 Fig S6.2: What were scanners “1,2,3,4” (e.g., vendor and model; e.g., did the vendor and/or field strength change?)

We have now added more relevant detail in SI 6 “Cohort effects”:

<<the following changes were made to the supplementary information>>

During this time there were multiple upgrades to the hardware and software, but the core system remained a 1.5T GE Signa platform throughout:

Label	Scanner ID	Description	Date of upgrade
1	S1-1	GE Signa 1-1 (Hardware 1)	6/9/90
2	S1-2a	GE Signa 1-2a (Hardware 2--Software a)	3/19/02
3	C1-1	CRADA magnet upgrade (Hardware 1)	12/16/03
4	C1-1b	CRADA magnet (Hardware 1--Software b)	5/15/07

Ref 3/23 S7 (and S2.1 and S18): It is overly simplistic to say that the cerebrum tissue volumes returned by FS are not contingent on the surface reconstruction – this is certainly not the case for modern versions of FS. Almost every value returned by FS 6.0, including in the aseg.stats, respects the surfaces, and is computed after the surfaces are defined. For example, see <https://surfer.nmr.mgh.harvard.edu/fswiki/MorphometryStats> and <https://surfer.nmr.mgh.harvard.edu/fswiki/ReconAllTableStableV6.0> (noting that the final aseg.stats is derived using the surfaces as inputs to the -segstats stage). It’s possible that the situation is more complicated for FS 5.1 and 5.3. An easy way to check for these older versions is to see if the volumes being extracted are the same in the aseg.stats as in the wmparc.stats file. If they are, then they are surface-informed estimates. If not, additional investigation will be necessary to understand (and accurately portray) the differences in this regard between different FS versions.

We thank the reviewer for providing this information. We have amended the section on the description of the FreeSurfer pipeline (SI 18) to remove the claim that tissue volumes are not contingent on the surface reconstruction.

<<the following changes were made to the supplementary information>>

In: SI 18 Data processing

If T1- and T2/FLAIR-weighted raw data were available, as they were for approximately 95% of scans), these data were processed on the same server at the University of Cambridge with FreeSurfer 6.0.1⁷⁴ using the combined T1-T2 recon-all pipeline for improved grey-white matter boundary estimation. If only raw T1-weighted data were available, and subjects were aged over 2 years, data were processed with FreeSurfer 6.0.1 using the standard recon-all pipeline.

If subjects were aged 0–2 years, data were processed with Infant FreeSurfer v1⁷. Briefly, the first processing stage of recon-all includes: non-uniformity correction, projection to Talairach space, intensity normalisation, skull-stripping, automatic tissue and subcortical segmentation. Subsequently, surface interpolation, tessellation and registration are done at the second and third stages of the recon-all pipeline. **ST1.1** lists the number of subjects per site per processing pipeline alongside their respective MRI acquisition and quality control protocols. We noticed that Infant FreeSurfer estimated total subcortical grey matter volume (sGMV) differently from other pipelines included in this dataset, while other cerebrum tissue volumes were estimated consistently across pipelines. We therefore excluded scans processed with Infant FreeSurfer from growth curve estimation for subcortical GMV. All four cerebrum tissue volumes were extracted from the aseg.stats files output by the recon-all process: 'Total cortical gray matter volume' for GMV; 'Total cortical/cerebral (FreeSurfer version dependent) white matter volume' for WMV; 'Subcortical gray matter volume' for sGMV (inclusive of thalamus, caudate nucleus, putamen, pallidum, hippocampus, amygdala, and nucleus accumbens area; <https://freesurfer.net/fswiki/SubcorticalSegmentation>); and the difference between 'BrainSegVol' and 'BrainSegVolNotVent' for Ventricular volume. Regional volume was estimated for each of 34 bilaterally averaged cortical regions defined by the Desikan-Killiany⁷⁵ parcellation template following the final stages of the recon-all pipeline and using the hemisphere-specific aparc.stats files generated by FreeSurfer.

Ref 3/24 Fig S7.1: To match the actual figures, the caption should say “baseFO[a][b][c]R[x][y][z]”. (i.e., add “R” to the text string).

Thank you for picking up this typo, which has been corrected in **Figure S7.1** (and also in **Figure S1.3**).

Ref 3/25 Fig. S7.4.1: Captions says that studies are sorted by “median standard deviation”, but that doesn't appear to be the case.

Thank you for picking up this glitch, which has now been corrected in both **Figure S7.4.1** and **Figure S4.2.1** (they were accidentally ordered by the median across all the panels, as opposed to within each of the panels).

Ref 3/26 Fig. S10.1.1 and S10.1.3: Rather than attempting to show all significant pairwise differences via lines above a box plot, it would be easier for readers to simply include a table with p-values. Cells below the diagonal could present the Cohen's d and p-value for males, and cells above the diagonal could present the values for females. (This way readers would also have access to the Cohen's d effect sizes of all pairwise comparisons in a convenient “lookup” table in the SI, without needing to download and find the relevant supplementary data table). Same comments apply to Figs. S14.3.[3,4]

The full tables for all Cohen's *d* and *P*-values are provided in **Supplementary Tables 3.1-3.28** and are now more clearly referred elsewhere in the supplementary information. As noted in response to a previous comment (Ref 3/1), we chose to use boxplots to give an indication of the centile distribution within the various cohorts and to show the extent of the significant differences. However, we have now also added a matrix of all pairwise Cohen's *d* values to enable a quick look-up of any pairwise effects, and replaced the median difference panels in **Figures 14.3.1 and 14.3.2** with Cohen's *d* instead.

<<the following changes were made to the supplementary information>>

Fig. S10.1.2. Case-control and between-disorder comparisons of centile scores on cerebrum volumes. Matrix plots show the pairwise Cohen's *d* values for every combination. More positive *d* indicates that the centile score on the *x*-axis was higher relative to the corresponding label on the *y*-axis, more negative *d* indicates the opposite effects, i.e., CN > AD in both males and females. Abbreviations: control, CN; Alzheimer's disease, AD; attention deficit hyperactivity disorder, ADHD; autism spectrum disorder, ASD; anxiety/phobia (ANX), mild cognitive impairment, MCI; major depressive disorder, MDD; schizophrenia, SCZ; grey matter volume, GMV; subcortical grey matter volume, sGMV; white matter volume, WMV; ventricular cerebrospinal fluid volume, CSF.

Fig. S10.1.5. Case-control and between-disorder comparisons of centile scores on extended global MRI phenotypes. Matrix plots show the pairwise Cohen's *d* values for every combination. More positive *d* indicates that the centile score on the *x*-axis was higher relative to the corresponding label on the *y*-axis, more negative *d* indicates the opposite effects, i.e., CN > AD in both males and females. Abbreviations; control, CN; Alzheimer's disease, AD; attention deficit hyperactivity disorder, ADHD;

autism spectrum disorder, ASD; anxiety/phobia (ANX), mild cognitive impairment, MCI; major depressive disorder, MDD; schizophrenia, SCZ.

Ref 3/27 Figs. S10.2.[1,2] and S14.3.[1,2,3]: Please match the colors and order of clinical cohorts to those used in Fig. 4 and S10.1.2

These figures have been edited to have the same colour scale and ordering of clinical cohorts.

Ref 3/28 S12: No indication of the effect size of the regression lines (e.g., R^2) is provided.

As requested, we have now added the R-squared values to **Figure S12**:

Fig. S12. Relationships between centile scores on cerebrum tissue volumes and birth weight (left panel) and gestational age at birth (right panel) for each of 5 primary studies with relevant data available. Centile-normalised z-scores were computed for each global phenotype in each individual study and then averaged across phenotypes to compute a mean centile z-score for each subject. The black dashed lines represent the relationships between mean centile scores and birth weight or gestational age at birth estimated by a linear mixed-effects model: for gestational age at birth, $t = 12.624$, $P < 2e-16$; for birth weight, $t = 34.945$, $P < 2e-16$. The black dotted line in the right panel denotes the commonly-used threshold for defining premature birth at 37 weeks post-conception. Conditional R-squared in each panel represents the variance explained by the entire model (black dashed lines).

Ref 3/29 S14.4.1: Unclear whether the LME only included a random subject-level intercept, or if it also included a random subject-level slope. If the latter, what was the correlation between

the random intercept and random slope estimates? Also, what do the error bars in panel B represent?

This analysis only included a random intercept, this has been clarified in the description. Higher order random-effects require more longitudinal observations to be identifiable. In this case, adding a random-effect on slope leads to convergence issues (since this excludes individuals with only two observations). The error bars in this figure represent the confidence interval on the beta coefficient (as a measure of change relative to remaining cognitively normal); this has now been clarified in the caption to **Figure S14.4.1**:

...but not a random intercept as some individuals only had 2 observations and including random slopes would cause convergence issues. The error bars in panel B depict the confidence intervals around the beta coefficients.

Ref 3/30 Fig. S15.1: Either some of the measures have very extreme outliers (e.g., GMV/SA, WMV/SA, sGMV/SA), or the allowed jitter along the x-axis is too great, such that values from one Feature Pair are bleeding into the space for an adjacent Feature Pair.

We thank the reviewer for pointing this out. We have now corrected this figure to more clearly delineate the correlations for each feature pair by colouring them. There is indeed some between-study variation in the correlations between MRI phenotypes. This level of variability is likely dominated by the age range or developmental epoch of specific studies, as is also highlighted in the amount of variability in these associations across the lifespan in **Fig S15.2**.

Fig. S15.1. Box-plots of Pearson correlations between each pair of global neuroimaging metrics in each of the primary studies in the reference dataset. Each datapoint represents a single primary study; boxes highlight the median and interquartile range of between-study variation in correlations of “raw”, non-centiled volumetrics for all possible pairs of global MRI phenotypes. Alternating colours are for visualisation purposes only.

Ref 3/31 ST 1.1: The information in the “Acquisition.Parameters” column in this table is inconsistent and uneven. For example (1) the very first entry (3R-BRAIN) is listed as having a

flip angle of 52, which would be very unusual for an MPRAGE; (2) the manner in which TR/TE/TI/FA is presented across studies is inconsistently formatted; (3) some rows have random special characters (e.g., Calgary, Conte, DLBS, EMBARC, GOSH, LATAM, LIFE; to name just a few based on quick inspection); (4) some don't report TR/TI/FA at all (e.g., HCP_lifespan). Overall, these imaging parameters would be easier to parse (and easier to check for odd entries) if there was one column for each specific parameter to be reported, rather than just a generic text column of "parameters".

The suggested columns have been added to Supplementary Table 1.1, which has been thoroughly edited for any remaining inconsistencies. We note that some parameterization of acquisition isn't easily captured in a standard column (i.e., multi-echo versus single echo, "shortest" as a setting on Philips platforms versus numerical entries on Siemens, different acquisition protocols across platforms that make repetition times generally not directly comparable, etc).

Ref 3/32 ST 1.8: It seems like the Figure 1 caption should either refer to ST1.2-1.8 (i.e., all of the "global" measures), or just ST1.2-1.5 (corresponding to the measures shown in Figure 1. Either way, ending at ST1.7 doesn't make sense (in either the caption, or the text in the "Mapping normative brain growth" paragraph).

Thank you for picking this up. The Figure 1 legend should indeed have referred to all tables, and the typo has been corrected. We thank the reviewer again for the level of attention to detail, and we note that we have also performed additional internal proofreading of the entire manuscript for the present resubmission.

Ref 3/33 ST3: The labels for column E ("Median centile G1") and column F ("Median centile G2") are presumably incorrectly swapped.

This typo has been corrected. In addition, we cross-checked all other tables for similar typographic errors (as well as the code that produced them) and found no other instances where labels had been swapped.

References

*note that these may not be with the same numbering as the main text and supplements as they are ordered differently here and combine the references of multiple documents.

1. Gandal, M. J., Leppa, V., Won, H., Parikshak, N. N. & Geschwind, D. H. The road to precision psychiatry: translating genetics into disease mechanisms. *Nat. Neurosci.* **19**, 1397–1407 (2016).
2. Opel, N. *et al.* Cross-Disorder Analysis of Brain Structural Abnormalities in Six Major Psychiatric Disorders: A Secondary Analysis of Mega- and Meta-analytical Findings From the ENIGMA Consortium. *Biol. Psychiatry* **88**, 678–686 (2020).

3. Li, X. *et al.* Moving Beyond Processing and Analysis-Related Variation in Neuroscience. *bioRxiv* 2021.12.01.470790 (2021) doi:10.1101/2021.12.01.470790.
4. Peterson, M. R. *et al.* Normal childhood brain growth and a universal sex and anthropomorphic relationship to cerebrospinal fluid. *J. Neurosurg. Pediatr.* 1–11 (2021).
5. Bycroft, C. *et al.* The UK Biobank resource with deep phenotyping and genomic data. *Nature* **562**, 203–209 (2018).
6. Casey, B. J. *et al.* The Adolescent Brain Cognitive Development (ABCD) study: Imaging acquisition across 21 sites. *Dev. Cogn. Neurosci.* **32**, 43–54 (2018).
7. Zöllei, L., Iglesias, J. E., Ou, Y., Grant, P. E. & Fischl, B. Infant FreeSurfer: An automated segmentation and surface extraction pipeline for T1-weighted neuroimaging data of infants 0-2 years. *Neuroimage* **218**, 116946 (2020).
8. Kim, H. *et al.* NEOCIVET: Towards accurate morphometry of neonatal gyrification and clinical applications in preterm newborns. *Neuroimage* **138**, 28–42 (2016).
9. Stasinopoulos, D. & Rigby, R. Generalized Additive Models for Location Scale and Shape (GAMLSS) in R. *Journal of Statistical Software, Articles* **23**, 1–46 (2007).
10. Borghi, E. *et al.* Construction of the World Health Organization child growth standards: selection of methods for attained growth curves. *Stat. Med.* **25**, 247–265 (2006).
11. Pomponio, R. *et al.* Harmonization of large MRI datasets for the analysis of brain imaging patterns throughout the lifespan. *Neuroimage* **208**, 116450 (2020).
12. Chen, H. *et al.* Quantile rank maps: a new tool for understanding individual brain development. *Neuroimage* **111**, 454–463 (2015).
13. Frangou, S. *et al.* Cortical thickness across the lifespan: Data from 17,075 healthy individuals aged 3-90 years. *Hum. Brain Mapp.* (2021) doi:10.1002/hbm.25364.
14. Dong, H.-M. *et al.* Charting brain growth in tandem with brain templates at school age. *Sci Bull. Fac. Agric. Kyushu Univ.* **65**, 1924–1934 (2020).
15. Cole, J. H. *et al.* Predicting brain age with deep learning from raw imaging data results in a reliable and heritable biomarker. *Neuroimage* **163**, 115–124 (2017).
16. Franke, K. & Gaser, C. Ten Years of BrainAGE as a Neuroimaging Biomarker of Brain

- Aging: What Insights Have We Gained? *Front. Neurol.* **10**, 789 (2019).
17. Engemann, D. A. *et al.* Combining magnetoencephalography with magnetic resonance imaging enhances learning of surrogate-biomarkers. *Elife* **9**, (2020).
 18. Cole, J. H. & Franke, K. Predicting Age Using Neuroimaging: Innovative Brain Ageing Biomarkers. *Trends Neurosci.* **40**, 681–690 (2017).
 19. Franke, K., Ziegler, G., Klöppel, S., Gaser, C. & Alzheimer's Disease Neuroimaging Initiative. Estimating the age of healthy subjects from T1-weighted MRI scans using kernel methods: exploring the influence of various parameters. *Neuroimage* **50**, 883–892 (2010).
 20. Liem, F. *et al.* Predicting brain-age from multimodal imaging data captures cognitive impairment. *Neuroimage* **148**, 179–188 (2017).
 21. Valizadeh, S. A., Hänggi, J., Mérillat, S. & Jäncke, L. Age prediction on the basis of brain anatomical measures. *Hum. Brain Mapp.* **38**, 997–1008 (2017).
 22. Butler, E. R. *et al.* Pitfalls in brain age analyses. *Hum. Brain Mapp.* **42**, 4092–4101 (2021).
 23. Smith, S. M., Vidaurre, D., Alfaro-Almagro, F., Nichols, T. E. & Miller, K. L. Estimation of brain age delta from brain imaging. *Neuroimage* **200**, 528–539 (2019).
 24. Le, T. T. *et al.* A Nonlinear Simulation Framework Supports Adjusting for Age When Analyzing BrainAGE. *Front. Aging Neurosci.* **10**, 317 (2018).
 25. Butler, E. R. *et al.* Statistical Pitfalls in Brain Age Analyses. *bioRxiv* 2020.06.21.163741 (2020) doi:10.1101/2020.06.21.163741.
 26. Hendrickson, M. A. & Pitt, M. B. Three Areas Where Our Growth Chart Conversations Fall Short-Room to Grow. *JAMA Pediatr.* (2021) doi:10.1001/jamapediatrics.2021.4330.
 27. Marquand, A. F., Rezek, I., Buitelaar, J. K. & Beckmann, C. F. Understanding Heterogeneity in Clinical Cohorts Using Normative Models: Beyond Case-Control Studies. *Biol. Psychiatry* **80**, 552–561 (2016).
 28. Sharma, E. *et al.* Consortium on Vulnerability to Externalizing Disorders and Addictions (cVEDA): A developmental cohort study protocol. *BMC Psychiatry* **20**, 2 (2020).

29. Liu, S. *et al.* Chinese Color Nest Project : An accelerated longitudinal brain-mind cohort. *Dev. Cogn. Neurosci.* **52**, 101020 (2021).
30. Zuo, X.-N. *et al.* Human Connectomics across the Life Span. *Trends Cogn. Sci.* **21**, 32–45 (2017).
31. Bridgeford, E. W. *et al.* Batch Effects are Causal Effects: Applications in Human Connectomics. *bioRxiv* 2021.09.03.458920 (2021) doi:10.1101/2021.09.03.458920.
32. Milham, M., Fair, D., Mennes, M. & Mostofsky, S. The adhd-200 consortium: a model to advance the translational potential of neuroimaging in clinical neuroscience. *Front. Syst. Neurosci.* **6**, 62 (2012).
33. Di Martino, A. *et al.* The autism brain imaging data exchange: towards a large-scale evaluation of the intrinsic brain architecture in autism. *Mol. Psychiatry* **19**, 659–667 (2014).
34. Snoek, L. *et al.* AOMIC-PIOP1. (2020) doi:10.18112/OPENNEURO.DS002785.V2.0.0.
35. Bilder, R. *et al.* UCLA Consortium for Neuropsychiatric Phenomics LA5c Study. (2020) doi:10.18112/OPENNEURO.DS000030.V1.0.0.
36. Nastase, S. A. *et al.* Narratives. (2020) doi:10.18112/OPENNEURO.DS002345.V1.1.4.
37. Alexander, L. M. *et al.* An open resource for transdiagnostic research in pediatric mental health and learning disorders. *Scientific Data* **4**, 1–26 (2017).
38. Snoek, L. *et al.* AOMIC-PIOP2. (2020) doi:10.18112/OPENNEURO.DS002790.V2.0.0.
39. Richardson, H., Lisandrelli, G., Riobueno-Naylor, A. & Saxe, R. Development of the social brain from age three to twelve years. *Nat. Commun.* **9**, 1027 (2018).
40. Kuklisova-Murgasova, M. *et al.* A dynamic 4D probabilistic atlas of the developing brain. *Neuroimage* **54**, 2750–2763 (2011).
41. Snoek, L. *et al.* AOMIC-ID1000. (2021) doi:10.18112/OPENNEURO.DS003097.V1.2.1.
42. Reynolds, J. E., Long, X., Paniukov, D., Bagshawe, M. & Lebel, C. Calgary Preschool magnetic resonance imaging (MRI) dataset. *Data Brief* **29**, 105224 (2020).
43. White, T., Blok, E. & Calhoun, V. D. Data sharing and privacy issues in neuroimaging research: Opportunities, obstacles, challenges, and monsters under the bed. *Hum.*

Brain Mapp. (2020) doi:10.1002/hbm.25120.

44. Bannier, E. *et al.* The Open Brain Consent: Informing research participants and obtaining consent to share brain imaging data. *Hum. Brain Mapp.* **42**, 1945–1951 (2021).
45. Rosen, A. F. G. *et al.* Quantitative assessment of structural image quality. *Neuroimage* **169**, 407–418 (2018).
46. Habes, M. *et al.* The Brain Chart of Aging: Machine-learning analytics reveals links between brain aging, white matter disease, amyloid burden, and cognition in the iSTAGING consortium of 10,216 harmonized MR scans. *Alzheimers. Dement.* **17**, 89–102 (2021).
47. Zhang, H. *et al.* Growth charts for individualized evaluation of brain morphometry for preschool children. *bioRxiv* (2021) doi:10.1101/2021.04.08.21255068.
48. Holland, D. *et al.* Structural growth trajectories and rates of change in the first 3 months of infant brain development. *JAMA Neurol.* **71**, 1266–1274 (2014).
49. Cole, T. J. The development of growth references and growth charts. *Ann. Hum. Biol.* **39**, 382–394 (2012).
50. Gilmore, J. H., Knickmeyer, R. C. & Gao, W. Imaging structural and functional brain development in early childhood. *Nat. Rev. Neurosci.* **19**, 123–137 (2018).
51. Tau, G. Z. & Peterson, B. S. Normal development of brain circuits. *Neuropsychopharmacology* **35**, 147–168 (2010).
52. Grydeland, H. *et al.* Waves of Maturation and Senescence in Micro-structural MRI Markers of Human Cortical Myelination over the Lifespan. *Cereb. Cortex* **29**, 1369–1381 (2019).
53. Paus, T., Keshavan, M. & Giedd, J. N. Why do many psychiatric disorders emerge during adolescence? *Nat. Rev. Neurosci.* **9**, 947–957 (2008).
54. Jack, C. R., Jr *et al.* Tracking pathophysiological processes in Alzheimer's disease: an updated hypothetical model of dynamic biomarkers. *Lancet Neurol.* **12**, 207–216 (2013).
55. Volpe, J. J. Brain injury in premature infants: a complex amalgam of destructive and

- developmental disturbances. *Lancet Neurol.* **8**, 110–124 (2009).
56. Nosarti, C. *et al.* Preterm birth and psychiatric disorders in young adult life. *Arch. Gen. Psychiatry* **69**, E1–8 (2012).
 57. Wheeler, E. N. W. *et al.* Birth weight is associated with brain tissue volumes seven decades later, but not with age-associated changes to brain structure. *bioRxiv* 2020.08.27.270033 (2020) doi:10.1101/2020.08.27.270033.
 58. Vigo, D., Thornicroft, G. & Atun, R. Estimating the true global burden of mental illness. *Lancet Psychiatry* **3**, 171–178 (2016).
 59. Marquand, A. F. *et al.* Conceptualizing mental disorders as deviations from normative functioning. *Mol. Psychiatry* **24**, 1415–1424 (2019).
 60. Dubois, J. *et al.* The early development of brain white matter: a review of imaging studies in fetuses, newborns and infants. *Neuroscience* **276**, 48–71 (2014).
 61. Huttenlocher, P. R. & Dabholkar, A. S. Regional differences in synaptogenesis in human cerebral cortex. *J. Comp. Neurol.* **387**, 167–178 (1997).
 62. Mountcastle, V. B. The columnar organization of the neocortex. *Brain* **120**, 4, 701–722 (1997).
 63. Petanjek, Z. *et al.* Extraordinary neoteny of synaptic spines in the human prefrontal cortex. *Proc. Natl. Acad. Sci. U. S. A.* **108**, 13281–13286 (2011).
 64. Miller, D. J. *et al.* Prolonged myelination in human neocortical evolution. *Proc. Natl. Acad. Sci. U. S. A.* **109**, 16480–16485 (2012).
 65. Courchesne, E. *et al.* Normal brain development and aging: quantitative analysis at in vivo MR imaging in healthy volunteers. *Radiology* **216**, 672–682 (2000).
 66. Blüml, S. *et al.* Metabolic maturation of the human brain from birth through adolescence: insights from in vivo magnetic resonance spectroscopy. *Cereb. Cortex* **23**, 2944–2955 (2013).
 67. Kuzawa, C. W. *et al.* Metabolic costs and evolutionary implications of human brain development. *Proc. Natl. Acad. Sci. U. S. A.* **111**, 13010–13015 (2014).
 68. WHO MULTICENTRE GROWTH REFERENCE STUDY GROUP & Onis, M. WHO

- Motor Development Study: Windows of achievement for six gross motor development milestones. *Acta Paediatr.* **95**, 86–95 (2007).
69. Wit, E., van den Heuvel, E. & Romeijn, J.-W. ‘All models are wrong...’: an introduction to model uncertainty. *Stat. Neerl.* **66**, 217–236 (2012).
70. Giedd, J. N. *et al.* Child psychiatry branch of the National Institute of Mental Health longitudinal structural magnetic resonance imaging study of human brain development. *Neuropsychopharmacology* **40**, 43–49 (2015).
71. Pietschnig, J., Penke, L., Wicherts, J. M., Zeiler, M. & Voracek, M. Meta-analysis of associations between human brain volume and intelligence differences: How strong are they and what do they mean? *Neurosci. Biobehav. Rev.* **57**, 411–432 (2015).
72. Wheeler, E. *et al.* Birth weight is associated with brain tissue volumes seven decades later but not with MRI markers of brain ageing. *Neuroimage Clin* **31**, 102776 (2021).
73. Walhovd, K. B. *et al.* Long-term influence of normal variation in neonatal characteristics on human brain development. *Proc. Natl. Acad. Sci. U. S. A.* **109**, 20089–20094 (2012).
74. Fischl, B. FreeSurfer. *Neuroimage* **62**, 774–781 (2012).
75. Desikan, R. S. *et al.* An automated labeling system for subdividing the human cerebral cortex on MRI scans into gyral based regions of interest. *Neuroimage* **31**, 968–980 (2006).

Reviewer Reports on the Second Revision:

Referees' comments:

Referee #1 (Remarks to the Author):

The authors should be commended for their persistence in the review process. This effort will seed a significant body of work in years to come. At this point, I believe the manuscript is suitable for publication.

Referee #2 (Remarks to the Author):

The authors have made important edits to the language regarding the clinical relevance of their work. The overall paper is very strong and is sure to be impactful on the field.

Referee #3 (Remarks to the Author):

Bethlehem and co-authors are again to be commended for their responsiveness to the previous round of reviews. The manuscript is further improved.

A few items related to the additions need some cleanup and revision:

Online Methods: Nothing in the framing leading up to the explicit models (Eq's 2.1 – 2.7) appears to explain how one can get terms of the form, $\beta * (\text{age}^p) * (\log(\text{age}^p))$. i.e., Where is the interaction of age (to a power) with $\log(\text{age})$ (to a power) coming from? Intuitively, what is such a term supposed to represent? This also raises the question, since 8 powers were investigated, if each model explored a space of $8 * 8 = 64$ total $(\text{age}^p) * (\log(\text{age}^p))$ interaction terms. If so, wouldn't that lead to an excessive amount of model flexibility prone to overfitting (and instability)? Also, having presented a general GAMLSS formulation that includes a "kurtosis"-related term in the probability density function, it seems like the Online Methods, as part of presentation of the explicit models, should explain why none of them include such a term.

Relatedly, I'm trying to reconcile the explicit models added to the Online Methods with the text in SI1.3, which now seems underspecified relative to the details provided in Eq's 2.1 – 2.7. SI1.3 also seems inconsistent with those equations. For example, it says that the model selection suggested a 3rd order polynomial fit for the mean for all 4 phenotypes – but I don't see a cubic term for the mean in Eq. 2.1. The text of SI1.3 also implies that the model for the Ventricles (Eq. 2.4) should have a cubic age term for sigma, but it does not. It further says that sGMV should only have a study random effect for sigma, whereas Eq. 2.3 includes a study random effect for both the mean and sigma for sGMV. It also says that there should be 3 polynomials except for sigma of sGMV. But sigma for both GMV and WMV also show only 2 age-related polynomial terms (Eq's. 2.1 and 2.2). [At this point, I need to comment: I appreciate that this is a complicated paper with many analyses. And that my "attention to detail" is more tuned than most. But, the number of inconsistencies and errors that

have been present through the rounds of this manuscript is disappointing, not consistent with a Nature publication, and burdensome on the review process].

SI 3.2.3: Please make explicit the number of comparisons that you computed the FDR correction over. E.g., Was it 4 for Mu and 3 for Sigma? Was it 7 (across Mu and Sigma together)? Was it even higher than that (e.g., across all investigated technical covariates simultaneously)? This matters for understanding the severity of the correction applied. Similar comment applies for the text in SI 7.2.

Figure 1, Fig S3.2.3.4, S7.2.5, S4.1.1 (and probably others): I didn't realize this previously, but presumably all the figures are using natural log, rather than log₁₀, for the sample size? This seems a rather odd ('unnatural') base to use for quantifying sample size. At this late stage, rather than regenerating all these figures to use log₁₀, perhaps the caption for Figure 1 could simply make explicit for readers that you used natural log as the base for sample size?

Fig. S3.2.3.6: I believe that this figure is intended to include just the same 4 volume measures as Figs S.3.2.3.2-5, especially since the values for the 3 'extended' phenotypes are shown in Fig. S.7.2.7.

SI 7.1: I appreciate that the authors added a brief discussion regarding the clearly visible discontinuity in the raw, non-centiled thickness data (Figure 2). However, it defies credulity to postulate that this discontinuity may reflect some real biological phenomenon. I think this discussion needs to be revised to reflect the common sense interpretation that this discontinuity must (solely) reflect some combination of sample selection bias and/or the impact of different pre-processing pipelines.

p. 9 ("Individualised centile scores in clinical samples"): The phrasing "XX% points" is odd, and one I've never seen before. I suggest (e.g.) "36 percentage point difference"; "increase of 11.8 percentage points".

S1.7: Please either cite a reference for the statistical properties of IQR computed from just 2 data points, preferably explaining why IQR is considered preferable to standard deviation for quantifying the range/variability in the situation of a small number of observations. Or, alternatively, drop the language implying that IQR is somehow a more "valid" measure for a small number of observations (to me, "valid" and "measure is defined" are not the same thing).

S2.2: I was curious to see which datasets contributed to the "Manual" QC review, so I opened ST1.1. Filtering on "Extracted QC metrics" = 'Manual', and then summing on 'Sample size' yields n = 384. Filtering on 'Euler, Manual' yields n = 24138. Either way, neither match (or even come close to) the n = 9704 that SI2.2 indicates were manually reviewed by a single rater. How does one identify the scans/datasets that were manually reviewed as part of the analysis of S2.2?

Fig. S4.3.1: Please swap the rows so that the raw data is the top row, and the centiled data is the bottom row, consistent with the layout of Figs. S4.3.2-4.

Fig. S4.3.2: The usage of "HCP" in the figure titles is potentially confusing, given that the data was collected by HBN, not HCP. If you wish to make the point that the HBN MPRAGE was modelled after

the HCP acquisition, that seems an appropriate point to make in the SI text. But I'd suggest avoiding labelling the figures themselves as "HCP Acquisition".

Fig. S4.5: It would be nice to actually include the ADNI data, to support the NSPN results. Could you just include a 2nd row to the figure with the ADNI-derived results?

S5.2: In the analyses of the association between total cerebral volume and either fluid intelligence or birth weight, how was sex handled? Was it included as a covariate? Also, the added text refers to "Fig S5.4-5.5", which don't exist.

Fig. S5.2.5: It remains unclear to me what point this figure is intended to convey. It is very difficult to compare betas across panels (raw vs. GAMLSS vs ComBAT), but the overall impression is that the estimated betas are very similar across panels for each site, even compared to the Raw data. But that interpretation seems inconsistent with the point of Fig S5.2.4. Please make clearer what S5.2.5 is supposed to demonstrate, esp. in relation to S5.2.4.

S6: I'm having trouble figuring out how to interpret "Hardware 1" vs "Hardware 2". In particular, for "Label 3" (Scanner ID C1-1), did the scanner actually revert to the same hardware in place for the S1-1 scanner? (That's the implication of labeling them both as "Hardware 1").

Fig. S14.4.1: I believe the authors meant to say, "..., but not a random *slope* as some individuals..."

Author Rebuttals to Second Revision:

We would like to thank Reviewer 3 and the editor for their expert reviews of our revised manuscript and supplementary analyses. Their comments have provided extremely helpful guidance in strengthening the paper. Please find a point-by-point response below.

[BLACK] - ORIGINAL COMMENT

[BLUE] - RESPONSE TO COMMENT

[GREEN] - NEW/ALTERED TEXT

Referee #3

Bethlehem and co-authors are again to be commended for their responsiveness to the previous round of reviews. The manuscript is further improved.

We sincerely thank the reviewer for this positive appraisal and, more generally, for their sustained and diligent attention to critiquing our work. Their input has materially improved the quality and precision of the paper in many ways, over the course of 3 review cycles, and we are very appreciative of their expertise and commitment to the highest scientific and presentational standards.

A few items related to the additions need some cleanup and revision:

R3/1: Online Methods: Nothing in the framing leading up to the explicit models (Eq's 2.1 – 2.7) appears to explain how one can get terms of the form, $\beta_p \cdot (\text{age}^p) \cdot (\log(\text{age}^p))$. i.e., Where is the interaction of age (to a power) with $\log(\text{age})$ (to a power) coming from? Intuitively, what is such a term supposed to represent? This also raises the question, since 8 powers were investigated, if each model explored a space of $8 \cdot 8 = 64$ total $(\text{age}^p) \cdot (\log(\text{age}^p))$ interaction terms. If so, wouldn't that lead to an excessive amount of model flexibility prone to overfitting (and instability)? Also, having presented a general GAMLSS formulation that includes a "kurtosis"-related term in the probability density function, it seems like the Online Methods, as part of presentation of the explicit models, should explain why none of them include such a term.

We apologise for any lack of clarity in formal presentation of the models in Eqs 2.1-2.7. As defined by Royston and Altman (1994), a power of zero in fractional polynomials is $\log(x)$ (rather than what might be expected, since $x^0 = 1$), and furthermore a repeated power is $\beta_p x^p + \beta_{p,r} x^p \log(x)$. These definitions follow by taking limits and ensure that the fractional

polynomials are well defined, as now referenced in SI1.3. We consider polynomials of the first order, $\beta_p x^p$, second order, $\beta_p x^p + \beta_q x^q$, and third order, $\beta_p x^p + \beta_q x^q + \beta_r x^r$, using Royston and Altman's proposed eight powers: $\{-2, -1, -0.5, 0, 0.5, 1, 2, 3\}$. As the reviewer correctly noted, this results in a potentially large number of parameters with attendant risks of overfitting and instability. However, we performed data-driven model selection using the Bayesian Information Criterion (BIC) to manage this risk by selecting only the subset of polynomial parameters that materially improved goodness of fit. By this empirical process of model selection, we found no evidence to support a fractional polynomial within the ν -component; however, there was evidence to support the need for an intercept, implying a generalised gamma distribution. This process resulted in the optimally specified models presented in Eqs 2.1-2.7 as described in the Online Methods. As the reviewer also noted, our models do include a kurtosis-related ν -component. However, the data-driven process of model selection showed that only an intercept term in the ν -component was supported by the data. This term was included in the optimal GAMLSS models and as is now clearly indicated in revised Eqs 2.1-2.7 and clarified in the Online Methods.

<<the following changes were made to the Supplementary Information>>

In SI1.3

As noted above, fractional polynomials can be viewed as a simpler form of spline modelling using a fixed set of polynomials (GAMLSS uses the standard set of polynomial powers: -2, -1, -0.5, 0, 0.5, 1, 2, 3, see Royston & Altman (1994)¹²). Some standard definitional issues should be noted. First, the term “order” is used to refer to the number of terms in the fractional polynomial model rather than the power, e.g., a third order fractional polynomial does not necessarily contain x^3 . We consider polynomials of the first order, $\beta_p x^p$; second order, $\beta_p x^p + \beta_q x^q$; and third order, $\beta_p x^p + \beta_q x^q + \beta_r x^r$. Second, as conventionally defined by Royston and Altman, a power of zero in fractional polynomials is $\log(x)$ rather than x^0 (since $x^0 = 1$ for all x). Third, “repeated powers” are evaluated: a second order fractional polynomial where power p is repeated is defined as $\beta_p x^p + \beta_{p'} x^p \log(x)$, while a third order fractional polynomial where power p is repeated is defined as $\beta_p x^p + \beta_{p'} x^p \log(x) + \beta_{p''} x^p \log(x)^2$.

<<the following changes were made to the Online Methods>>

In Online Methods

Reliably estimating higher order moments requires increasing amounts of data, hence none of our models specified any age-related fixed-effects or random-effects in the ν term. However, α_ν was found to be important in terms of model fit and hence we have used a generalised gamma distribution (SI1).

R3/2: Relatedly, I'm trying to reconcile the explicit models added to the Online Methods with the text in SI1.3, which now seems underspecified relative to the details provided in Eq's 2.1 – 2.7. SI1.3 also seems inconsistent with those equations. For example, it says that the model selection suggested a 3rd order polynomial fit for the mean for all 4 phenotypes – but I don't see a cubic term for the mean in Eq. 2.1. The text of SI1.3 also implies that the model for the Ventricles (Eq. 2.4) should have a cubic age term for sigma, but it does not. It further says that sGMV should only have a study random effect for sigma, whereas Eq. 2.3 includes a study random effect for both the mean and sigma for sGMV. It also says that there should be 3

polynomials except for sigma of sGMV. But sigma for both GMV and WMV also show only 2 age-related polynomial terms (Eq's. 2.1 and 2.2). [At this point, I need to comment: I appreciate that this is a complicated paper with many analyses. And that my "attention to detail" is more tuned than most. But, the number of inconsistencies and errors that have been present through the rounds of this manuscript is disappointing, not consistent with a Nature publication, and burdensome on the review process].

We apologise for any lack of clarity or consistency in our formal and written descriptions of the models. We consider that at least some of the apparent confusion is attributable to standard differences in the terminology for fractional polynomial models compared to the more familiar case of (integer) polynomial models, and we have explicitly noted these definitional issues in revised **SI1.3** as excerpted above in response to R3/2. "Third order" in the context of fractional polynomials refers to the number of terms not the power, e.g., a third order fractional polynomial does not necessarily contain a cubic term. For example, in Eq 2.1, there are three fractional polynomial terms for the μ -component: $\beta_{\mu_1}(age)^{-2}$, $\beta_{\mu_2}(age)^{-2} \log(age)$, and $\beta_{\mu_3}(age)^{-2} \log(age)^2$: this would conventionally be described as a third-order fractional polynomial where the power -2 is repeated. With regard to the specific model descriptions in **SI1.3**, our earlier use of the phrase "third order polynomial" has been referred to more explicitly as a "third order fractional polynomial" to clarify our usage.

We thank the reviewer for noticing a discrepancy in the discussion of the random effects in **SI1.3** compared to the models presented in the **Online Methods** ("sGMV should only have a study random effect for sigma"). We have corrected the description in the Supplementary Information. Specifically, modelling of the Ventricles included a study random-effect in the μ -component only. Modelling of sGMV, GMV and WMV included a study random effect in both the μ -component and σ -component. The **Online Methods** are correct and remain unchanged (in terms of the order of fractional polynomials and random-effects); the edited **SI1.3** text is shown below.

<<the following changes were made to Supplementary Information>>

In SI1.3:

The GAMLSS framework includes a fractional polynomial function that automatically performs the model selection step within the fitting process. In addition to this standard estimation, we chose to evaluate model permutations of all possible combinations of the number of modelled polynomials in each of the terms (μ, σ, ν) of the generalised gamma distribution (between 1–3 terms for each of the three parameters). Across all four main global tissue volumes (GMV, sGMV, WMV, ventricles) this approach suggested 3rd order fractional polynomial fits for the μ -component. For the σ -component, modelling indicated 2nd order fractional polynomial fits for GMV, sGMV, and WMV, but a 3rd order fractional polynomial for ventricles (**Fig. S1.3**). For GMV, sGMV, and WMV, the model evaluation procedure also suggested including a study random effect in both μ and σ , whereas for ventricles it indicated inclusion of a study random effect only for μ .

R3/3: SI 3.2.3: Please make explicit the number of comparisons that you computed the FDR correction over. E.g., Was it 4 for Mu and 3 for Sigma? Was it 7 (across Mu and Sigma together)? Was it even higher than that (e.g., across all investigated technical covariates

simultaneously)? This matters for understanding the severity of the correction applied. Similar comment applies for the text in SI 7.2.

We have now made explicit the number of comparisons corrected by the FDR in **SI 3.2.3**. Corrections for multiple testing were applied within each parameter across phenotypes, i.e., 4 tests for Mu and 3 tests for Sigma per reported phenotype.

<<the following changes were made to the Supplementary Information>>

In SI 3.2.3:

For each of these models, we corrected for multiple comparisons within each parameter, i.e., correcting for 4 tests on the Mu term and 3 tests on the Sigma term.

In SI 7.2:

For each of these models, we corrected for multiple comparisons within each parameter, i.e., correcting for 3 tests on the Mu term and 3 tests on the Sigma term.

Ref 3/4: Figure 1, Fig S3.2.3.4, S7.2.5, S4.1.1 (and probably others): I didn't realize this previously, but presumably all the figures are using natural log, rather than log10, for the sample size? This seems a rather odd ('unnatural') base to use for quantifying sample size. At this late stage, rather than regenerating all these figures to use log10, perhaps the caption for Figure 1 could simply make explicit for readers that you used natural log as the base for sample size?

The reviewer is correct that we used the natural log transform to represent sample sizes of the primary studies in **Figure 1** (and related figures in **Supplementary Information**). We did this because we found the natural log transform provided a smooth colour range that made it easier to infer the differences in primary study sample size from the visible differences in colouring of each box plot. This has been made explicit in all the relevant figure legends reporting sample size..

Ref 3/5: Fig. S3.2.3.6: I believe that this figure is intended to include just the same 4 volume measures as Figs S.3.2.3.2-5, especially since the values for the 3 'extended' phenotypes are shown in Fig. S.7.2.7.

We thank the reviewer for catching this point and the figure has now been corrected accordingly.

<<the following changes have been made to the Supplementary Information 3.2.3.6>>

Ref 3/6: SI 7.1: I appreciate that the authors added a brief discussion regarding the clearly visible discontinuity in the raw, non-centiled thickness data (Figure 2). However, it defies credulity to postulate that this discontinuity may reflect some real biological phenomenon. I think this discussion needs to be revised to reflect the common sense interpretation that this discontinuity must (solely) reflect some combination of sample selection bias and/or the impact of different pre-processing pipelines.

We appreciate the reviewer's strong opinion concerning interpretation of the discontinuity in the non-centiled cortical thickness data in **Figure 2**. The discontinuity arises between data measured before and after 2 years, approximately. We agree that the most obvious interpretation is that this reflects methodological differences, including differences in image analysis and pre-processing pipelines, between MRI studies of early childhood and later life. However, we have found this discontinuity to be robust to sensitivity analyses of the effects of image processing pipelines and we have found similar discontinuities in primary studies that have used identical methods to measure cortical thickness in participants ranging in age across the 2 year transition point. These observations are difficult to reconcile with the obvious methodological interpretation and suggest instead the alternative explanation that this discontinuity may reflect a critical nonlinearity in brain development, perhaps related to the process of grey/white matter differentiation that is active during the first two years of life. We have revised the relevant discussion of this issue to give greater emphasis to the methodological interpretation, as requested by the reviewer, while retaining the neurodevelopmental interpretation as a hypothesis for further investigation in future: see **SI7.1, "Extended neuroimaging types"**.

<<the following changes have been made to the Supplementary Information>>

In SI 7.1 Extended neuroimaging phenotypes

We note the discontinuity between the raw, non-centiled CT data for participants younger versus older than 2 years (approximately) that is evident by inspection of **Fig. 2**. The common sense interpretation of this discontinuity must be some combination of sample selection bias

and/or the impact of different preprocessing pipelines in the primary studies of early childhood (<2 y) compared to studies of later childhood and adults (> 2y). It is consistent with this interpretation that participant age of 2-3 years is often used as the cutoff to decide application of different, specialised preprocessing pipelines, e.g., infant FreeSurfer versus adult FreeSurfer. However, we note that this discontinuity was evident also in data from a number of primary studies that applied identical sampling criteria and image processing methods to measure cortical thickness in participants younger and older than the ~2 year transition point. Thus it remains conceivable, in our view, that this discontinuity may partially reflect a neurodevelopmental nonlinearity occurring in the context of the process of grey/white matter differentiation that is actively ongoing throughout the first 2-3 years of postnatal life. Definitive resolution of this issue is currently hampered by the relative lack of primary MRI studies of early childhood development; but it is expected that the correct interpretation of the discontinuity apparent in the existing data will become clearer in future as studies apply more consistent methods to analysis of larger samples of participants recruited from either side of the ~2y transition point.

Ref 3/7: p. 9 (“Individualised centile scores in clinical samples”): The phrasing “XX% points” is odd, and one I’ve never seen before. I suggest (e.g.) “36 percentage point difference”; “increase of 11.8 percentage points”.

We have revised this phrasing according to the reviewer’s suggestions.

Ref 3/8: S1.7: Please either cite a reference for the statistical properties of IQR computed from just 2 data points, preferably explaining why IQR is considered preferable to standard deviation for quantifying the range/variability in the situation of a small number of observations. Or, alternatively, drop the language implying that IQR is somehow a more “valid” measure for a small number of observations (to me, “valid” and “measure is defined” are not the same thing).

We accept the reviewer’s point. We have edited the text accordingly to remove the implication that IQR is a more “valid” measure. In the revised manuscript, we simply note that the IQR is well defined for 2 data points.

<<the following changes have been made to the Supplementary Information>>

In SI 1.7

A univariate summary for variation across observations will assess the stability of the centiles within an individual. The summary must be defined for two or more observations, the minimal longitudinal follow-up period, and be comparable across individuals. The range, i.e., $\max(q_{ij1}, q_{ij2}, \dots, q_{ijm}) - \min(q_{ij1}, q_{ij2}, \dots, q_{ijm})$, would be well defined for two or more observations; however, the range is susceptible to outliers and statistically unstable under small samples. Instead, the interquartile range (IQR) acts as a robust equivalent of the range (in the same way that the trimmed mean is a robust version of the mean). Given the variable number of longitudinal data-points available for different participants, we chose to use a measure that was consistent for participants that only had 2 observations as well as for participants with more than 2 observations. Unfortunately, there is not a single definition of the IQR (there are 9 different definitions available within GNU R), and some versions are not defined for two observations. We estimated IQR as a continuous value by linear interpolation

(within GNU R the default version of IQR, type 7), which is well defined for two (or more) observations.

Ref 3/9: S2.2: I was curious to see which datasets contributed to the “Manual” QC review, so I opened ST1.1. Filtering on “Extracted QC metrics” = ‘Manual’, and then summing on ‘Sample size’ yields n = 384. Filtering on ‘Euler, Manual’ yields n = 24138. Either way, neither match (or even come close to) the n = 9704 that SI2.2 indicates were manually reviewed by a single rater. How does one identify the scans/datasets that were manually reviewed as part of the analysis of S2.2?

We apologise for any confusion in the labelling of columns in the **Supplementary Table**. The “manual” label in the column titled “Extracted QC metric” was designated for datasets that were either pre-filtered manually or for datasets where manual segmentation was used. We have now added an additional column (“QC Rating Included”) to the spreadsheet to explicitly indicate which studies had QC ratings included in the analysis in **SI 2.2**. This yields a total N=11498, which, after excluding longitudinal follow-ups and removing NA values, corresponds exactly to the numbers listed in **S2.2**. This is now clarified in **SI2.2**.

<<the following changes have been made to the Supplementary information >>

In SI2.2:

These scans were provided by openly available datasets and are marked as having “QC Rating Included” in **ST1.1** (note that the total number of scans with QC rating designated in the table is larger due to the fact that the table also includes longitudinal data, which were not included in this assessment).

Ref 3/10: Fig. S4.3.1: Please swap the rows so that the raw data is the top row, and the centiled data is the bottom row, consistent with the layout of Figs. S4.3.2-4.

We have updated the figure and legend according to the reviewer's suggestion.

<<the following changes have been made to the Supplementary information>>

Fig. S4.3.1. Test-retest reliability of out-of-sample centile scores for cerebrum tissue volumes. MRI data were collected in two separate scanning sessions from N=21 participants and each session was analysed as an independent out-of-sample study using GAMLSS. The top panel shows the analysis for non-centiled, “raw” volumetric data. Bottom scatterplots represent OoS centile scores for session 1 (y-axis) versus OoS centile scores for session 2 (x-axis) for each brain tissue volume, from left to right: GMV, WMV, sGMV, Ventricular CSF. Data points represent individual subject centile scores. Test-retest reliability was consistently very high (all intra-class correlation coefficients > 0.99) for all cerebrum tissue volumes. Uncorrected (for multiple comparisons) P-values represent the significance of the intraclass correlation coefficient between two sessions. Shaded regions indicate the 95% confidence intervals of the linear association.

Ref 3/11: Fig. S4.3.2: The usage of “HCP” in the figure titles is potentially confusing, given that the data was collected by HBN, not HCP. If you wish to make the point that the HBN MPRAGE was modelled after the HCP acquisition, that seems an appropriate point to make in the SI text. But I’d suggest avoiding labelling the figures themselves as “HCP Acquisition”.

The figure titles have been amended according to the reviewer’s suggestion and the caption has been updated to clarify that the MPRAGE acquisition mimicked that of the HCP study.

<<the following changes have been made to the Supplementary Information>>

Fig. S4.3.2. Test-retest reliability of out-of-sample centile scores for cerebrum tissue volumes measured twice in the same $N=72$ participants using two T1-weighted sequences, MPRAGE and VNaV. For each type of acquisition, the top row shows out-of-sample centile scores for session 1 (y-axis) versus out-of-sample centile scores for session 2 (x-axis) for cerebrum tissue volumes. For each type of acquisition, the bottom row shows the unprocessed (“raw”) scores for session 1 (y-axis) versus session 2 (x-axis) for cerebrum tissue volumes estimated from VNaV data, from left to right: GMV, WMV, sGMV, Ventricles. In all plots, data points represent individual subject scores. Test-retest reliability was uniformly high (all ICCs > 0.95) and generally somewhat higher for volumetrics derived from prospectively motion-corrected data (VNaV). P-values represent the significance of the intraclass correlation coefficient between two sessions. *MPRAGE acquisition refers to the T1-weighted MPRAGE sequence used in the Human Connectome Project.*

Ref 3/12: Fig. S4.5: It would be nice to actually include the ADNI data, to support the NSPN results. Could you just include a 2nd row to the figure with the ADNI-derived results?

We now include the ADNI figures as an extra row in Figure S4.5.

<<the following changes have been made to the Supplementary Information>>

Fig. S4.5. Out-of-sample estimates of cloned study random-effect parameters compared to in-sample estimates of random-effect parameters in the original or non-cloned study. The plot shows random-effects estimated using the out-of-sample approach across a range of possible sample sizes for a “new” study, generated by taking subsets of the same cloned study with uncertainty intervals derived from the bootstrap replicates. The purple horizontal lines are the equivalent in-sample estimates of the random-effects parameters. We see that the out of sample estimates are somewhat unreliable below $N=100$ subjects, but with larger samples the out-of-sample estimates from the cloned data converge with the in-sample estimates from the original data for both μ -component and σ -component random effects. Top row, cloned NSPN refers to the Neuroscience in Psychiatry Network study; bottom row, cloned ADNI refers to the Alzheimer’s Disease Neuroimaging Initiative. Error bars indicate the standard deviation of the parameter estimates at each sample size. Error bars indicate the standard deviation of the parameter estimates at each sample size.

Ref 3/13: S5.2: In the analyses of the association between total cerebral volume and either fluid intelligence or birth weight, how was sex handled? Was it included as a covariate? Also, the added text refers to “Fig S5.4-5.5”, which don’t exist.

We thank the reviewer for catching this error in the numbering of supplementary figures, which we have now corrected in the text. We have also updated the figure legend to include a description of the models for comparing birth weight and fluid intelligence to total cerebrum volume that clarifies how sex was handled in these analyses.

<<the following changes have been made to the Supplementary Information>>

To further assess whether batch-corrected MRI data derived from both ComBAT and GAMLSS pipelines would generate convergent results in subsequent analyses, we estimated the correlations between total cerebrum volume (TCV) and fluid intelligence or birth weight, after TCV had been batch-corrected by either GAMLSS or ComBAT. Both these psychological

and biological factors have previously been shown to be correlated with similar brain volumetrics^{51–53}. We were able to replicate these significant associations with uncorrected TCV, as well as after both GAMLSS and ComBAT batch correction, all largely showing consistent effects across sites (**Fig. S5.2.4-5.2.5**).

Fig. S5.2.4. Associations between total cerebrum volume (TCV) and birth weight (top) or fluid intelligence (bottom) after batch correction by GAMLSS (left), by ComBAT (middle), or without batch correction (raw, right). Linear relationships for each of the 22 sites in the ABCD study are in coloured solid lines; dashed lines signify overall linear mixed-effect model fit across sites; fluid intelligence was assessed using the NIH Toolbox⁵⁴. These results show that predicted relationships between TCV and both birth weight and fluid intelligence are more convincingly replicated in these $N=10,583$ scans from the ABCD multi-site study when the MRI data have been batch-corrected by either GAMLSS or ComBAT compared to when the MRI data have been analysed without correction of between-site differences. Linear mixed-effect models, with either birth weight or fluid intelligence as independent variables, included fixed effects for TCV, binary sex, and age (in days); and a random effect of site.

Ref 3/14: Fig. S5.2.5: It remains unclear to me what point this figure is intended to convey. It is very difficult to compare betas across panels (raw vs. GAMLSS vs ComBAT), but the overall impression is that the estimated betas are very similar across panels for each site, even compared to the Raw data. But that interpretation seems inconsistent with the point of Fig S5.2.4. Please make clearer what S5.2.5 is supposed to demonstrate, esp. in relation to S5.2.4.

We thank the reviewer for bringing up this point. We agree that following changes to **Fig S5.2.4** as a result of the review process, **Fig S5.2.5** is now redundant and we have therefore deleted this figure from the revised Supplementary Information.

Ref 3/15: S6: I'm having trouble figuring out how to interpret "Hardware 1" vs "Hardware 2". In particular, for "Label 3" (Scanner ID C1-1), did the scanner actually revert to the same

hardware in place for the S1-1 scanner? (That's the implication of labeling them both as "Hardware 1").

We have now updated the inline table in SI6 to clarify the date and type of upgrades implemented on each of the scanners.

<<the following changes have been made to the Supplementary Information>>

Label	Scanner ID	Description	Date of upgrade
1	S1-1	GE Signa 1-1	6/9/90
2	S1-2a	GE Signa 1-2a (Hardware + Software upgrade)	3/19/02
3	C1-1	CRADA magnet (Hardware upgrade)	12/16/03
4	C1-1b	CRADA magnet (Software upgrade)	5/15/07

Ref 3/16: Fig. S14.4.1: I believe the authors meant to say, "..., but not a random *slope* as some individuals..."

The reviewer is correct and this phrasing has been corrected accordingly.